**Investigation**

# Clade distillation for genome-wide association studies

Ryan Christ ⓘ ,[1,*] Xinxin Wang,[1,2] Louis J.M. Aslett,[3] David Steinsaltz,[4] Ira Hall[1,*]

[1]Department of Genetics, Yale University School of Medicine, New Haven, CT 06510, United States
[2]Department of Genetics, Washington University School of Medicine, Saint Louis, MO 63110, United States
[3]Department of Mathematical Sciences, Durham University, Durham DH1 3LE, United Kingdom
[4]Department of Statistics, University of Oxford, Oxford OX1 3LB, United Kingdom

*Corresponding authors: Ryan Christ, Department of Genetics, Yale University School of Medicine, New Haven, CT 06510, United States. Email: ryan.christ@yale.edu;
Ira Hall, Department of Genetics, Yale University School of Medicine, New Haven, CT 06510, United States. Email: ira.hall@yale.edu

Testing inferred haplotype genealogies for association with phenotypes has been a longstanding goal in human genetics given their potential to detect association signals driven by allelic heterogeneity—when multiple causal variants modulate a phenotype—in both coding and noncoding regions. Recent scalable methods for inferring locus-specific genealogical trees along the genome, or representations thereof, have made substantial progress towards this goal; however, the problem of testing these trees for association with phenotypes has remained unsolved due to the growth in the number of clades with increasing sample size. To address this issue, we introduce several practical improvements to the kalis ancestry inference engine, including a general optimal checkpointing algorithm for decoding hidden Markov models, thereby enabling efficient genome-wide analyses. We then propose LOCATER, a powerful new procedure based on the recently proposed Stable Distillation framework, to test local tree representations for trait association. Although LOCATER is demonstrated here in conjunction with kalis, it may be used for testing output from any ancestry inference engine, regardless of whether such engines return discrete tree structures, relatedness matrices, or some combination of the two at each locus. Using simulated quantitative phenotypes, our results indicate that LOCATER achieves substantial power gains over traditional single marker testing, ARG-Needle, and window-based testing in cases of allelic heterogeneity, while also improving causal region localization. These findings suggest that genealogy-based association testing will be a fruitful approach for gene discovery, especially for signals driven by multiple ultra-rare variants.

Keywords: ancestral recombination graph; stable distillation; checkpointing; quadratic form

## Introduction

Recent heritability estimates predict that rare variants in regions with low linkage disequilibrium account for a substantial fraction of the unexplained (missing) heritability of common traits and diseases (Wang et al. 2021). Since the statistical power to detect effects driven by rare variants is inherently limited by their low frequency, methods for identifying rare variant associations leverage allelic heterogeneity: the presence of multiple independent causal mutations affecting the trait of interest. These methods merge association signals from nearby rare variants under the premise that rare causal variants may be proximal to other causal variants (Wu et al. 2011; Lee et al. 2014).

Mounting evidence suggests that allelic heterogeneity is quite common for human traits (Hormozdiari et al. 2017; GTEx Consortium 2020). Notably, a large-scale in vitro study following up on identified associations estimated that between 10% and 20% of expression quantitative trait loci (eQTLs) have multiple causal regulatory variants circulating in human populations (Abell et al. 2022). Such results underscore the importance of these methods for defining new alleles and genes contributing to disease risk (Cirulli et al. 2020; Momozawa and Mizukami 2021; Wang et al. 2021). The opportunity for methods that can leverage allelic heterogeneity to identify overlooked associations will only increase with the size, population diversity, and sequencing depth of emerging genomic datasets.

Early groundbreaking association methods designed to harness allelic heterogeneity focused on testing inferred locus-specific genealogies, which provide a natural way of collecting independent association signals driven by nearby variants and imputing any unobserved variants (Zöllner and Pritchard 2005; Minichiello and Durbin 2006; Thompson and Kubatko 2013). These approaches have the added benefit of implicitly imputing unobserved variants, making it particularly advantageous for analyzing datasets with partial variant calling: SNP array data, low-coverage sequencing data, or data from understudied species. The uncertainty and poor scalability of early local genealogy inference methods hamstrung the adoption of these early testing approaches. To partly address these challenges, Browning and Thompson (2012) proposed identity-by-descent mapping to test only very recent, locus-specific relationships, which could be rapidly and confidently inferred from the observed haplotypes. While very elegant, all of these early methods suffer from technical limitations related to statistical testing, such as requiring resampling or permutation to generate P-values, and ultimately have not been used much.

Recent advances in ancestry inference algorithms have made it possible to revisit genealogy-based trait association. Algorithms

such as ARG-Needle (Zhang et al. 2023), Relate (Speidel et al. 2019), tsinfer (Kelleher et al. 2019), and kalis (Aslett and Christ 2024) have made it possible to perform local ancestry inference across the entire genome in modern datasets with hundreds of thousands or millions of samples. Given their accuracy in resolving recent genealogical relationships, inferred local ancestries are expected to be especially useful for detecting loci with multiple causal ultra-rare variants, which are signals that standard single marker testing (SMT) will struggle to identify. This may be a particularly effective strategy for traits under strong purifying selection and cases where some of these ultra-rare causal variants correspond to complex hidden structural variations that are only observed in a given sequencing dataset via ultra-rare tagging variants that are far upstream or downstream. However, recent work aimed at applying these algorithms to improve disease mapping, most notably ARG-Needle (Zhang et al. 2023), has focused on imputing hidden variants not explicitly observed in the original dataset and testing the inferred genotypes via SMT. Given the plummeting cost of high-coverage sequencing data and recent initiatives to improve structural variant detection and imputation (Liao et al. 2023), the number of missing variants is shrinking in modern datasets, limiting the gains available from testing hidden variation via local ancestries.

Link et al. (2023) evidenced a renewed interest in using genealogies to map genes by leveraging allelic heterogeneity. Building on earlier efforts like that of Zöllner and Pritchard (2005), their approach targets loci with allelic heterogeneity by using local ancestry inference methods to build a local genetic relatedness matrix for prespecified windows along the genome or gene regions. These matrices are then tested for association with the phenotype of interest using a quadratic form test statistic. Using effectively the same default test statistic, recent work by Zhu et al. (2024) and Gunnarsson et al. (2024) provides a much more scalable implementation of this approach. Building on the idea of identity-by-descent mapping, Cai and Browning (2025) very recently proposed a distinct, scalable approach that also relies on a quadratic form test statistic. All of these methods mirror SKAT and more recent approaches in rare-variant gene-based testing that also aim to harness allelic heterogeneity to gain statistical power (Wu et al. 2011; Li et al. 2020). However, due to the inherent sensitivity of quadratic form test statistics to the presence of many noncausal (null) variants (Christ et al. 2024a), these approaches struggle to maintain statistical power under the enormous multiple testing burden incurred when testing many components of the local tree structure, which will often reflect but may not always directly correspond to inferred clades in that tree structure, for trait association.

Existing rare variant association methods that aim to leverage allelic heterogeneity by testing collections of variants, such as STAAR, limit their multiple testing burden by using functional information, such as gene coding sequences, to define restricted sets of variants or more flexibly down-weight certain variants (Wu et al. 2011; Li et al. 2020). This approach has been applied to many different sequencing based studies, and has proven to be a fruitful approach for identifying new gene–phenotype associations across a multitude of traits (Wainschtein et al. 2022). Despite their success in coding regions, it has proven difficult to extend rare variant association tests beyond coding regions where the majority of biologically critical signals are found. Ninety percent of genome-wide association study (GWAS) hits for common diseases lie in noncoding regions, at a median distance of 36 kilobase pair (kb) from the nearest transcription start site (Maurano et al. 2012; Mostafavi et al. 2023). It is unclear how one should

define collections of variants in noncoding regions; sliding windows are the standard approach (Li et al. 2020).

Outside of a gene's coding region, which in humans has a median length $\approx 3$ kb, there is a much larger regulatory region over which causal variants may be dispersed, complicating the use of sliding windows. Ideally, one would try to incorporate many variants over a genomic region in order to maximize the chance of aggregating signals from more than one causal variant. However, including too many noncausal (null) variants diminishes statistical power, and variant impact prediction, which could be used to narrow down variants to test in a given sliding window, remains extremely challenging in noncoding regions.

There are "sparse-signal" statistical methods, often deployed alongside quadratic forms, that aim to improve power in the presence of many null variants. Notable examples include the Cauchy Combination Test (CCT), which underlies the Aggregated Cauchy Association Test (ACAT) routine in STAAR (Li et al. 2020), and Generalized Higher Criticism (Barnett et al. 2017). However, these sparse-signal methods do not distinguish between a variant set where two highly-linked variants are observed to be associated with the phenotype and a variant set where two unlinked variants are observed to be associated with the phenotype. In the highly-linked case, one variant is essentially a proxy for the other and we have only one association signal. In the unlinked case, the signals coming from the two independent variants serve as independent pieces of evidence against the null hypothesis and should be combined. In order to control the type-I error in the highly-linked case, the CCT underlying ACAT cannot combine signals across variants with high efficiency, which yields a loss of power in the unlinked case. This simple two-variant argument extends to the case where we may be attempting to combine association signals across several variants. Section 1 of Christ et al. (2024a) provides further discussion of this point.

We recently proposed a general statistical approach, Stable Distillation (SD), which can distinguish between the highly-linked and unlinked case (Christ et al. 2024a). There, in a gene-testing example using simulated data, we used SD to explicitly model the dependence structure between variants and achieved increased power over ACAT and related methods as a result. Building on SD, we present a general framework, LOCATER, for trait association based on inferred local genealogies that works in both coding and noncoding regions. We focus exclusively on testing quantitative traits, although we plan to extend LOCATER to binary traits in future work.

Modern ancestry inference methods typically represent local ancestries as discrete trees (perhaps with probabilistic weights on the edges), local relatedness matrices, or some combination of the two. Examples of discrete tree inference methods include tsinfer. These clades may have probabilistic weights, as provided by recent probabilistic ARG inference methods such as SINGER (Deng et al. 2024). On the other hand, ancestry may also be represented in terms of local pairwise relatedness, typically summarized as a local relatedness matrix, as produced by kalis, Relate, and Gamma-SMC (Schweiger and Durbin 2023). A set of observed haplotypes can typically be explained by an enormous number of underlying tree topologies, especially once we look beyond the recent past; pairwise methods account for this topological uncertainty. LOCATER provides a framework for boosting SMT results with independent association signals based on local ancestry represented in either, or both, of these two forms produced by any ancestry inference engine (see Results). In order to highlight this feature, in this paper we apply LOCATER to complementary discrete clade and matrix-based representations of local ancestry obtained via the local ancestry inference engine kalis.

LOCATER is designed to work in conjunction with any ancestry inference engine of the user's choosing with an easy-to-use API available through our locater package for the R language (R Core Team 2024). Since our locater package exposes all of our testing subroutines as documented R functions, if a set of clade calls or a local relatedness matrix produced by some ancestry inference engine can be coerced into base or sparse matrices in R, then locater can be directly used to test those structures for association with a given phenotype. Please see the Data and Code Availability section for details.

Although kalis does not scale as well as alternatives like tsinfer, a probabilistic model allows us to limit statistical testing to clades that have substantial evidence of existing at a locus of interest, thereby conserving statistical power. SINGER may provide a strong probabilistic alternative in future studies. The algorithmic improvements to kalis that we present in this paper, including an optimal checkpointing routine for discrete-time hidden Markov models (HMMs) and a linear-time clustering algorithm, may be useful for accelerating alternative models.

Our focus on using genealogies to boost SMT signals rather than testing gene windows or sliding windows along the genome is another key point of departure of LOCATER from existing work. This approach leverages the statistical efficiency of SMT against sparse signals. Testing the inferred genealogy at a locus also removes questions of window size and step length. By returning "genealogy-boosted" SMT signals, we find LOCATER generally improves the localization of causal variants relative to SMT in the presence of allelic heterogeneity (see Results). The precise variants aggregated at a given locus will depend on the structure of the local ancestral recombination graph (ARG) and the parameters of the ancestry inference engine used (Li and Stephens 2003; Hein et al. 2004; Speidel et al. 2019). At a high level, older edges in a local genealogy tend to persist over much shorter stretches of the genome than recent edges due to recombination (Hein et al. 2004). Accordingly, any procedure that tests a local genealogy for association with a phenotype will tend to aggregate association signals from rare variants over a wider region than common variants.

In this paper we characterize LOCATER's performance in simulated datasets. Dealing with the challenges of trait mapping in real datasets, such as rigorously adjusting for population structure and cryptic relatedness, is a difficult and open problem for genealogy-based trait mapping methods. We provide our perspective on these challenges in the Discussion and take them on in a subsequent work (Wang et al. 2024). There we demonstrate the ability of LOCATER to substantially increase statistical power at loci with allelic heterogeneity and identify loci missed by SMT in a dataset of 6,795 Finnish genomes with extensive quantitative trait data.

## Results

LOCATER assumes that genome-wide SMT has already been performed. This is done simply to avoid the computational burden of inferring ancestries and running LOCATER at every locus. We focus on the subset of variants with putatively significant SMT results (e.g. $P < 10^{-4}$) and compute the local ancestry at each of those variants. At each target variant, LOCATER then takes the residuals from the SMT and tests any inferred discrete clade structure with Stable Distillation (SD) (Christ et al. 2024a). SD returns a new set of residuals which are guaranteed to be independent of the original SMT $P$-value and the $P$-value returned by SD under the null hypothesis (see Methods). We then pass this set of residuals to a quadratic-form based method that tests the pairwise-relatedness structure inferred at the variant of interest. The resulting three independent $P$-values

may then be combined to obtain a potentially boosted signal at a locus with allelic heterogeneity.

This approach makes it straightforward to integrate LOCATER into the analysis of genome-wide association results. Of course, the resulting $P$-values must still be compared against a genome-wide multiple testing threshold as if LOCATER was run at every candidate variant. LOCATER can easily be applied in special cases where the inferred ancestry at a locus only comes in the form of discrete clades (eg: a tree) or pairwise-relatedness (eg: a local relatedness matrix). In both our SD procedure and quadratic form testing procedure, we have developed scalable methods to adjust for population structure and background covariates (see Methods).

Below we introduce the LOCATER model. We then proceed to describe our methodological contributions in two parts. The first describes the new routines we have introduced in an update to kalis to generate the ancestry representations required by LOCATER (Aslett and Christ 2024). The second focuses on making the association testing procedures in LOCATER fast and robust. Finally, we demonstrate the calibration and power of LOCATER via simulation.

## The LOCATER model

Consider a genomic dataset with $n$ participants phased in segments along the genome, each segment consisting of $N = 2n$ phased haplotypes along an entire or subsection of a chromosome with $V$ variants. Although we only address the diploid case in this paper, our approach may be readily extended to nondiploid organisms. Below we will consider testing each variant within a given segment for association with some quantitative phenotype of interest $Y \in \mathbb{R}^n$. When determining genome-wide significance thresholds, the total number of candidate variants across all genomic segments must be accounted for. However, in order to conserve computational resources, as depicted in Fig. 1, we may only be interested in a subset of candidate variants within each segment based on preliminary SMT results or other genomic annotations. We call this subset of variants our target loci $\mathcal{L} \subseteq [V]$ and index them by their position along a given segment sequentially from $\ell = 1, \ldots, L = |\mathcal{L}|$.

Let $A \in \mathbb{R}^{n \times q}$ be a matrix of background covariates and $G^{(\ell)} \in \{0, 1, 2\}^n$ be the genotype vector observed at locus $\ell$. Depending on the type of inference engine used to infer the local ancestry structure at locus $\ell$, we may have inferred clade genotypes $X^{(\ell)} \in \{0, 1, 2\}^{n \times p}$ corresponding to edges in a tree inferred at $\ell$, a local relatedness matrix $\Omega^{(l)} \in \mathbb{R}^{n \times n}$ inferred at $\ell$, or both. In other words, for each of $p$ inferred clades at locus $\ell$, $X^{(\ell)}_{ij}$ is the number of haplotypes in sample $i$ that have been assigned to an inferred clade $j$ at locus $\ell$. We tackle the general case assuming that our ancestry inference engine has returned both $X^{(\ell)}$ and $\Omega^{(\ell)}$, each capturing different parts of the ancestral structure at locus $\ell$. Our approach can easily be applied to the special cases where only $X^{(\ell)}$ or $\Omega^{(\ell)}$ are available.

By allowing genealogical relationships to be expressed in terms of pairwise similarity rather than explicitly called clades, $\Omega^{(\ell)}$ accommodates more uncertainty about the precise topology of the underlying tree than $X^{(\ell)}$ (see Methods). However, this flexibility comes with a cost to power: as our power simulations below demonstrate, it is generally preferable to encode clades in $X^{(\ell)}$ rather than $\Omega^{(\ell)}$, at least when their membership is known with high confidence. Due to recombination, more distant genealogical relationships, corresponding to larger clades, at a given locus $\ell$ are more difficult to accurately estimate than more recent genealogical relationships, corresponding to small clades in a local genealogy. Thus, in this paper, we will demonstrate LOCATER by encoding small clades (typically each with at most 10 haplotypes under them), which we will refer to as "sprigs," in $X^{(\ell)}$ and encode larger clades in $\Omega^{(\ell)}$.

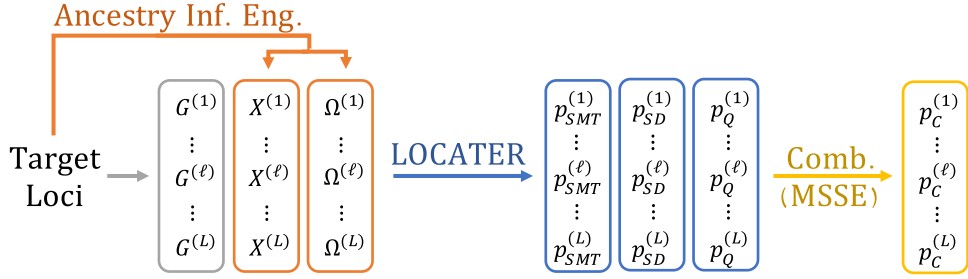

**Fig. 1.** The LOCATER Pipeline. We begin ancestry-based association testing with a set of putatively interesting target loci, typically identified via single marker testing, indexed $\{1, \ldots, L\}$. At each target locus $\ell$, we extract the genotype vector $G^{(\ell)} \in \{0, 1, 2\}^n$ and use an ancestry inference engine to infer local clade genotypes $X^{(\ell)} \in \{0, 1, 2\}^{n \times p}$ and/or a local relatedness matrix $\Omega^{(l)} \in \mathbb{R}^{n \times n}$. We then use LOCATER to calculate three $P$-values testing whether $G^{(\ell)}$, $X^{(\ell)}$, or $\Omega^{(l)}$ predict the phenotype respectively. These three $P$-values are guaranteed to be independent under the null hypothesis, so they may be easily combined with many methods, in this paper we propose and use MSSE (see Methods), to obtain a combined ancestry-association $P$-value $p_C^{(\ell)}$ at each target locus.

We describe how we call sprigs in the Calculating Inferred Clade Genotypes from the LS Model subsection below.

For a set of fixed effects $\alpha \in \mathbb{R}^q$, $\gamma \in \mathbb{R}$, and $\beta \in \mathbb{R}^p$, and a variance component parameter $\tau \in \mathbb{R}_{\geq 0}$, LOCATER assumes the following model for a quantitative phenotype vector $Y \in \mathbb{R}^n$.

$$Y = A\alpha + G^{(\ell)}\gamma + X^{(\ell)}\beta + \epsilon \quad \text{where} \quad \epsilon \sim N\left(0, \ \exp\left(\tau\Omega^{(\ell)}\right)\right). \quad (1)$$

Here, exp denotes the matrix exponential. Under this model we test whether genetic variation at locus $\ell$ affects phenotype $Y$ by testing the null hypothesis $H_0^C : \{\gamma = 0, \beta = 0, \tau = 0\}$. To be clear, $H_0^C$ represents the union, not the intersection, of these statements about the parameters. In other words, in what follows, we will reject $H_0^C$ if there is evidence that $\gamma \neq 0$, $\beta \neq 0$, or $\tau \neq 0$. We will assume that $Y$ has been obtained using our rank-matching procedure (Supplementary Section 2.11). This normalization ensures that the residuals of $Y$ have unit variance under $H_0^C$, justifying the absence of a variance scale parameter (typically denoted as $\sigma^2$) in Equation (1).

LOCATER tests $H_0^C$ by decomposing it into three sub-hypotheses. First, we use the standard SMT to test $H_0^{SMT} : \{\gamma = 0 \mid \beta = 0, \tau = 0\}$, yielding a $P$-value $p_{SMT}$. Then we test whether any of the locally inferred clades predict the phenotype $H_0^{SD} : \{\beta = 0 \mid \tau = 0\}$ using SD, yielding a $P$-value $p_{SD}$. Finally, we test whether any remaining local ancestry structure encoded in $\Omega^{(\ell)}$ affects the phenotype by testing $H_0^Q : \{\tau = 0\}$ with a quadratic form test statistic, yielding a $P$-value $p_Q$. See Methods for the explicit routines LOCATER uses to test $H_0^{SD}$ and $H_0^Q$. As further explained in Methods, when these three sub-hypotheses are tested in this order, the independence guarantees of SD ensure that the resulting $P$-values ($p_{SMT}$, $p_{SD}$, $p_Q$) are mutually independent under the null hypothesis $H_0^C$. Thus, after running LOCATER, the user may combine these three $P$-values using any valid method for aggregating independent $P$-values. We propose a variant of Fisher's method that we call Maximizing over Subsets of Summed Exponentials (MSSE) which yields more power in this setting (see Methods). Figure 1 overviews the role of LOCATER in the context of an ancestry-based association testing pipeline. Next we delineate how the algorithms we have introduced in our new release of kalis, kalis 2.0, allow us to rapidly obtain $X^{(\ell)}$ and $\Omega^{(\ell)}$ across target loci in our present study.

## Algorithmic advances in kalis 2.0
### Local genealogy inference with kalis

kalis (Aslett and Christ 2024) provides a high-performance implementation of various versions of the Li & Stephens (LS) haplotype copying model which have become ubiquitous in modern genomic analysis (Li and Stephens 2003; Song 2016). Our novel algorithmic contributions to kalis 2.0 allow us to efficiently calculate $X^{(\ell)}$ and $\Omega^{(\ell)}$ sequentially at a given set of target loci $\ell = 1, \ldots, L$ so that they can be tested downstream using LOCATER. In order to explain these contributions, we begin with a brief overview of local ancestry inference using kalis.

As with all ancestry inference engines, the ancestry at a given target locus $\ell$ is learned based on the observed genomic variation upstream and downstream of $\ell$. Since the LS model is a special case of an HMM, ancestry information provided by variants upstream of $\ell$ can be summarized by the forward probabilities at $\ell$; and variants downstream by the backward probabilities at $\ell$ (Rabiner 1989). Given a set of $N$ haplotypes and a single target locus $\ell$, kalis implements the forward algorithm to iterate over variants upstream of $\ell$, starting at the left end of the genomic segment, to obtain a matrix of forward probabilities $f^{(\ell)} \in \mathbb{R}^{N \times N}$ at $\ell$. Similarly, kalis implements the backward algorithm to iterate over variants downstream of $\ell$, starting at the right end of the genomic segment, to obtain a matrix of backward probabilities $b^{(\ell)} \in \mathbb{R}^{N \times N}$ at $\ell$. Each column $f_{\cdot j}^{(\ell)}$ and column $b_{\cdot j}^{(\ell)}$ corresponds to a separate LS HMM where we model recipient haplotype $j$ as a mosaic of the other $N - 1$ haplotypes in the sample. This separation allows kalis to compute the columns of $f^{(\ell)}$ and $b^{(\ell)}$ in parallel and exploit modern compute architectures. See Aslett and Christ (2024) for further details. The product $f_{ij}^{(\ell)} b_{ij}^{(\ell)}$ can be interpreted as proportional to the probability that recipient haplotype $j$ "copies" from donor haplotype $i$ at locus $\ell$ under the LS model. By definition, $f_{ii}^{(\ell)} = b_{ii}^{(\ell)} = 0$ for all haplotypes $i$. kalis makes $f^{(\ell)}$ and $b^{(\ell)}$ easily and rapidly accessible in the R language (R Core Team 2024) for downstream computation, with all time-critical code written in high performance C.

Along the lines of Speidel et al. (2019) and Aslett and Christ (2024), we define the distance from haplotype $j$ to haplotype $i$ as

$$d_{ij}^{(\ell)} = -\log\left(\max\left(\frac{f_{ij}^{(\ell)} b_{ij}^{(\ell)}}{\sum_{k=1}^N f_{kj}^{(\ell)} b_{kj}^{(\ell)}}, \ v\right)\right) \quad (2)$$

where $v \approx 4.94 \times 10^{-324}$ to guard against underflow to zero with double precision floating point arithmetic. For efficiency the distance matrix $d^{(\ell)} = (d_{ij}^{(\ell)}) \in \mathbb{R}_{\geq 0}^{N \times N}$ is never explicitly constructed, but it is implicitly used to construct $X^{(\ell)}$ and $\Omega^{(\ell)}$ for testing with LOCATER, as further delineated below.

Throughout this paper, we use kalis to run the modified LS model used in Relate (Speidel et al. 2019). See Supplementary

Section 2.1 for further details. This modified model leverages ancestral allele information to improve local genealogy inference. In this paper we only simulate phased genomic datasets where the ancestral allele of each variant is known; this is a feature of our chosen ancestry inference engine and not a general requirement of LOCATER. Under this modified LS model, Speidel et al. (2019) showed that the distance $d_{ij}^{(\ell)}$ will be proportional to the number of proximal variants that differ between haplotype $i$ and haplotype $j$ in nonrecombining segments and that the full matrix $d^{(\ell)}$ yields consistent local ancestry inference.

### Optimal checkpointing

Especially when processing many phenotypes in parallel, the number of target variants along a given genomic segment, $L$, may be very large. Since the amount of memory required to store a local relatedness matrix $\Omega^{(\ell)}$ at a given target locus scales $\mathcal{O}(n^2)$, storing these matrices at any appreciable number of variants quickly becomes untenable: even in the case where $n = 30,000$ samples (a scale we will consider in our simulations), 28.8 GB of memory is required to store a single $\Omega^{(\ell)}$. Offloading from memory also incurs a considerable time cost from writing and reading $\Omega^{(\ell)}$ to and from disk. To avoid writing any $\Omega^{(\ell)}$ to disk, we will take a "test-it-and-forget-it" approach: we will obtain $X^{(\ell)}$ and $\Omega^{(\ell)}$ at one target variant at a time and test both $X^{(\ell)}$ and $\Omega^{(\ell)}$ with LOCATER before moving on to the next target variant in $\mathcal{L}$.

This "test-it-and-forget-it" approach is only computationally tractable due to the checkpointing algorithm for discrete-time HMMs that we have introduced in kalis 2.0. Checkpointing involves repeatedly updating a cache of forward matrices $f^{(\ell)}$ that are used to seed subsequent iterations of the forward algorithm. Each stored forward matrix is a "checkpoint," and we assume that the user has a fixed memory budget sufficient to store $C$ checkpoints. Our checkpointing algorithm schedules where and when to overwrite each of the $C$ checkpoints in order minimize the computational cost required to sequentially propagate the forward algorithm to consecutive target loci $\ell = L, L-1, L-2, \dots, 1$. While maintaining a cache of checkpoints is far from a new idea in HMM inference —the idea is used in several implementations of the LS model (Speidel et al. 2019)—our checkpointing algorithm achieves the lower bound on computational cost given memory for $C$ checkpoints and can be applied to any discrete time HMM where sequential posterior decodings are required at consecutive times. Our approach yields massive reductions in computational cost compared to more naïve checkpointing approaches (see Methods).

### Calculating inferred clade genotypes from the LS model

The LOCATER model (Equation (1)) admits a matrix of genotypes $X^{(\ell)}$ encoding any clades (marginal tree edges) inferred at locus $\ell$. In principle, given the distance matrix $d^{(\ell)}$ obtained via kalis at locus $\ell$ (Equation (2)), any number of clustering algorithms could be used to infer a marginal tree topology from $d^{(\ell)}$. For example, Relate clusters a normalized version of $d^{(\ell)}$ with average linkage (UPGMA) (Speidel et al. 2019). From the resulting tree topology, one could then encode each of the inferred clades (or some subset of them) via $X^{(\ell)}$. While this is a promising approach for future work, in this paper, we only stored clade genotypes corresponding to very small inferred clades (each typically including 2 to 10 haplotypes) in $X^{(\ell)}$ and encode all larger-scale relatedness structure in $\Omega^{(\ell)}$. Focusing on just these rare clades, which we will refer to as "sprigs," rather than all of the clades in the tree, allows us to showcase the flexibility of LOCATER—the ability of LOCATER to incorporate hard-called clades via $X^{(\ell)}$ and remaining relatedness structure via $\Omega^{(\ell)}$.

We identify sprigs at a given locus $\ell$ based on the neighborhood —i.e. the set of tied nearest-neighbors—of each haplotype $j$:

$$\eta_j^\ell = \left\{ i \in [N] : d_{ij}^{(\ell)} \leq \min_{i \neq j} d_{ij}^{(\ell)} \right\}. \tag{3}$$

Note that, by this definition, haplotype $j$ is always a nearest neighbor of itself. In practice we obtain these neighborhoods as a by-product of the clustering algorithm we use to construct $\Omega^{(\ell)}$ (see Methods). We implicitly use the collection of neighborhoods $\{\eta_j\}_{j=1}^N$ to construct an undirected graph where each haplotype is a vertex, and edges connect haplotypes that agree on being in each other's neighborhood. We use a greedy clique-finding procedure over the nearest neighborhoods to rapidly identify maximal cliques within this implicit graph. Haplotypes within each clique are assumed to belong to the same sprig, yielding sprig genotypes that we encode in $X^{(\ell)}$. Having encoded the locally inferred sprig genotypes in $X^{(\ell)}$, we summarize all of the remaining genealogical structure in $d^{(\ell)}$ via $\Omega^{(\ell)}$.

### Calculating relatedness matrices from the LS model

While LOCATER can accept any real symmetric matrix $\Omega^{(\ell)}$, in order to optimize power we model our choice of $\Omega^{(\ell)}$ in this paper after the expected genetic relatedness matrix (eGRM) proposed by Fan et al. (2022). We cannot directly use their definition of the eGRM because the construction there requires a set of discrete clade calls. Constructing a local relatedness matrix from the distances $d^{(\ell)}$ is more complicated because, as described above, each column is calculated using an independent LS HMM. Thus, different columns of $d^{(\ell)}$ may disagree on the exact boundaries of particular clades in the underlying genealogy. This is a general feature of ancestry inference methods that work in a parallel or pairwise fashion across haplotypes. Rather than overriding the LS model and using hierarchical clustering or some other approach to try to align these clade calls, we generalize the eGRM to allow for this asynchrony. Our generalization, Equation (6) in Methods, expresses an eGRM in terms of asymmetric distances like those provided by $d^{(\ell)}$ while allowing for such unaligned probabilistic clade calls.

This generalization of the eGRM requires us to use the distances within each column of $d^{(\ell)}$ to call a set of nested neighborhoods around the corresponding haplotype. Calling these nested neighborhoods amounts to clustering the distances in each column of $d^{(\ell)}$. In order to do this efficiently for large $n$ datasets, we developed a general, multithreaded, single-pass algorithm based on doubly linked lists to cluster real numbers on a closed interval when clusters must be separated by some fixed minimum distance. This approach allows us to cluster each column of $d^{(\ell)}$ in $\mathcal{O}(N)$ time. In experiments on simulated haplotype data, we achieve roughly an order of magnitude speedup over merge sort. In order to conserve memory, our implementation does not explicitly store the clustering results for each column of $d^{(\ell)}$. Rather, we use these clusters to directly construct columns of an asymmetric version of $\Omega^{(\ell)}$ on the fly, directly collapsing haplotype level relatedness down to sample level relatedness as we go. Taking the symmetric part of the resulting matrix gives us $\Omega^{(\ell)}$ (see Methods).

As a by-product of the clustering used to construct $\Omega^{(\ell)}$, we also return the nearest-neighbor set of each haplotype, which is then used to call sprigs and construct $X^{(\ell)}$ as described above. After calling sprigs using these neighborhoods, in order to avoid testing these sprigs in both $X^{(\ell)}$ and $\Omega^{(\ell)}$, we efficiently remove the structure associated with those sprigs from $\Omega^{(\ell)}$ using some additional

statistics reported by our clustering algorithm before passing $X^{(\ell)}$ and $\Omega^{(\ell)}$ on to LOCATER for testing. All of these methods are available in kalis 2.0.

## LOCATER testing routines

All of LOCATER's routines have been written in terms of matrix operations, allowing multiple quantitative traits to be tested in parallel with minimal additional computational cost. This includes the first implementation of a parallelized SD algorithm. For a given phenotype, this SD algorithm yields decoupled estimators for the effect $\beta_j$ of each inferred clade genotype $X_j^{(\ell)}$. We then combine the independent two-sided *P*-values corresponding to these independent estimators via the Rényi Outlier Test (Christ et al. 2024b) to obtain $p_{SD}^{(\ell)}$ at each locus. As demonstrated in Christ et al. (2024a), this approach yields considerable gains in power over alternative methods when very few (but more than one) of the $\beta_j$ are nonzero; in other words, when more than one of the inferred genotype clades is associated with the phenotype. See Methods for details about the specific SD procedure we use.

As explained in Section 3 and shown in Fig. 6 of Christ et al. (2024a), SD returns an updated version of the data, there denoted as $Y^{(L+1)}$, which is independent of the information extracted to calculate $p_{SD}^{(\ell)}$. It is this $Y^{(L+1)}$ that LOCATER passes on to calculate the quadratic form testing procedure to calculate $p_Q^{(\ell)}$, guaranteeing the independence of $p_{SD}^{(\ell)}$ and $p_Q^{(\ell)}$.

LOCATER also deploys several statistical and algorithmic innovations to efficiently calculate $p_Q^{(\ell)}$. Under the LOCATER model (Equation (1)), the score statistic against the null $\tau = 0$ is a quadratic form,

$$Y^\top P \Omega^{(\ell)} P Y \qquad (4)$$

where $P = I - QQ^\top$ and $(A, G^{(\ell)}) = QR$ is the QR decomposition adjusting for the background covariates and the tested genotype $G^{(\ell)}$. In order to avoid launching unnecessary and expensive partial eigendecomposition routines at every target variant, we use a series of approximations to first assess whether the combined LOCATER *P*-value $p_C^{(\ell)}$ is sufficiently small to be interesting across any of the phenotypes. When it is, further eigendecomposition of $P\Omega^{(\ell)}P$ is deployed in order to obtain precise estimates.

We found that the Satterthwaite approximation (Satterthwaite 1946), which is commonly used for testing quadratic forms (Lumley et al. 2018), did not yield robust tail probability estimates for LOCATER. This may be because in this setting the matrices $P\Omega^{(\ell)}P$ are typically close to, but not quite, positive semi-definite—a key assumption of the Satterthwaite approximation. We overcame this obstacle with a new, robust tail approximation method for quadratic forms based on a shifted difference of chi-square random variables (see Methods). In combination with our approximation stopping criteria, this tail approximation provides a basis for emerging genealogy-based association methods to reliably test local pairwise relatedness matrices, which may often not be positive semi-definite.

Our tail approximation method has the added advantage that it admits three parameters—$v$, $\delta_\star^2$, and $\delta_\dagger^2$—to help control inflation of the null distribution due to population structure and polygenicity. In effect, these parameters generalize genomic control to quadratic forms (Devlin and Roeder 1999). Importantly, these three parameters were chosen to be orthogonal to the spectral parameters governing the distribution of Equation (4). If any inflation is observed in the Q–Q plot of $p_Q^{(\ell)}$ *P*-values after running LOCATER, this orthogonal parameterization allows us to adjust $(v, \delta_\star^2, \delta_\dagger^2)$ and rapidly calculate new $p_Q^{(\ell)}$ *P*-values without requiring

the re-estimation of local ancestries at any target locus (see Methods). Supplementary Section 2.5 provides an interpretation of our parameters $(v, \delta_\star^2, \delta_\dagger^2)$.

We use a novel multi-threaded algorithm for efficiently projecting out background covariates when calculating the matrix traces needed for these tail approximations (Supplementary Section 2.6). All final *P*-values $p_Q$ involving eigenvalue terms are calculated using the fast Fourier transform implemented in the R package QForm (Christ and Aslett 2024). Finally, we combine our three *P*-values, $(p_{SMT}^{(\ell)}, p_{SD}^{(\ell)}, p_Q^{(\ell)})$, using a modified version of Fisher's combination test we call MSSE (see Methods).

## Type-I error control

In order to confirm the calibration of LOCATER empirically, we simulated 1,000 independent genomic datasets, each consisting of 30,000 samples of a 1 Mb chromosome (see Methods). For each dataset, we simulated 1,000 independent phenotype vectors assuming no causal variants (see Methods). This yielded a total of 1 million independent phenotype vectors. We tested each phenotype vector for association with the ancestry inferred at the mid-point of the corresponding chromosome using LOCATER. We display a Q–Q plot for $-\log_{10}$ of those *P*-values, as well as for each LOCATER sub-test—SMT, SD, and QForm—in Supplementary Section 1.2 (Supplementary Figs. 2-5). These Q–Q plots confirm that the *P*-values returned by each sub-test and the combined LOCATER *P*-value are all well calibrated under the null hypothesis.

## Power

We compared LOCATER to ARG-Needle and standard SMT across a variety of genetic architectures. Following Zhang et al. (2023), we ran ARG-Needle with mutation rates $\mu = 10^{-3}$ and $\mu = 10^{-5}$. For an additional comparison, we also ran ARG-Needle with $\mu = 10^{-7}$ (Supplementary Section 2.8). We refer to these three variations of ARG-Needle as AN3, AN5, and AN7 respectively. In order to give ARG-Needle the best possible advantage, in each simulation, we gave ARG-Needle the true underlying ARG generated by msprime (Kelleher et al. 2016), rather than asking ARG-Needle to infer that ARG from the observed haplotypes. Thus our results represent an upper bound on the performance of ARG-Needle.

We assessed every possible combination of the following causal variant assumptions. We considered 3, 9, or 15 causal variants based on the number of independent causal alleles that were observed in the large follow-up study of GWAS hits (Abell et al. 2022 Fig. 4b). We also considered causal variants with any derived allele count, derived allele count of 2 (doubletons only), or intermediate variants with derived allele count in [150,750]. That is equivalent to a derived allele frequency (DAF) in [0.0025,0.0125]. Lastly, we considered the case where all causal variants are observed or all causal variants are hidden. This yielded a total of 18 genetic architectures. Note that by "observed" we mean that the causal variants were included in the dataset passed to each association method; by "hidden," that they were not included in the dataset passed to each association method and thus could only be inferred via LD.

In each simulation, causal variants were randomly assigned from among those fulfilling the required allele count requirements within a 10 kb window in the center of each simulated 1 Mb segment. Under each genetic architecture, we estimated power as a function of the underlying total association signal strength: the $-\log_{10}$ *P*-value that one would obtain by testing the simulated phenotype Y with an oracle ANOVA model that "knows" the causal variants and targets only those for testing. See Methods for a more precise definition. To improve the interpretability of our power curves, following Christ et al. (2024a), we used the

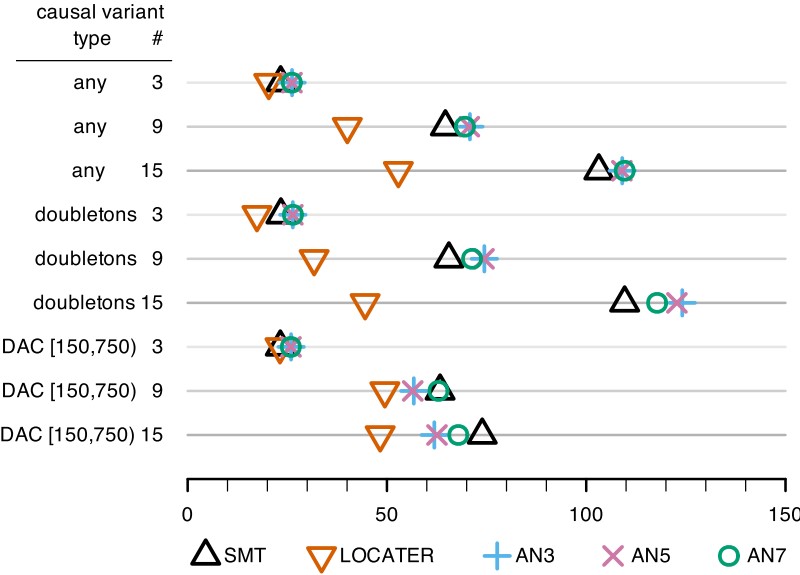

**Fig. 2.** Dotplot of total association signal strength required to achieve 80% power (lower is better) under various simulation conditions where all causal variants were observed. Total association signal strength is the $-\log_{10}$ $P$-value that one would obtain by testing the simulated phenotype Y with an oracle ANOVA model that "knows" the causal variants and targets only those for testing. Causal variant # denotes the number of simulated causal variants. Causal variant type "any" means any variant could be causal; "doubletons" means only doubletons could be causal; "DAC [150,750]" means only variants with a derived allele count in [150, 750], corresponding to a derived allele frequency in [0.0025, 0.0125), could be causal.

QR-decomposition to ensure that the total association signal was evenly split among the causal variants in every simulation. In other words, we ensured that the *observed* contribution of each causal variant to the total association signal was essentially equal for each simulated Y (see Methods).

For each method, we used 9,000 independent null simulations to estimate a genome-wide discovery threshold to maintain a family-wise error rate (FWER) below 0.05 (Supplementary Section 2.10). This yielded a $-\log_{10}$ genome-wide discovery threshold 8.40 for LOCATER, 8.79 for SMT, 9.36 for AN7, 9.73 for AN5, and 9.78 for AN3. In calculating power, we count our causal region as "discovered" if a testing method has a $-\log_{10}$ $P$-value greater than their discovery threshold *anywhere* along the entire 1 Mb region. This definition reflects how new associations are discovered in practice and provides a relatively strict benchmark. Each point of the resulting power curves was estimated via 1,000 independent samples: we simulated 10 independent phenotype vectors for each of 100 independent genomic datasets, each consisting of a 1 Mb chromosome sampled for 30,000 individuals. These power curves are available in Supplementary Section 1.4 (Supplementary Figs. 8, 11, 14, 17, 20, & 23). We summarize each of these curves with the estimated minimum signal strength required to achieve 80% power (lower is better). Figure 2 displays those estimates for LOCATER and SMT across all simulations where the underlying causal variants were observed; Fig. 3, for those hidden.

From both Figs. 2 and 3 we see that LOCATER ties or improves upon the statistical power of SMT and ARG-Needle across all settings. LOCATER can detect substantially weaker association signals than SMT and ARG-Needle when there are 9 or 15 causal variants. The power gains achieved by LOCATER over SMT in the observed causal variants case (Fig. 2) are impressive given the fact that SMT has been shown to be surprisingly powerful this context. Across all analogous power simulations with full variant ascertainment and allelic heterogeneity, Link et al. (2023) found that SMT (which they refer to as "GWAS") had the same or more power than their ancestry-based quadratic form (eGRM) approach (Link et al. 2023 Supplementary Fig. S2). To be clear, Link et al. (2023) did

observe that the eGRM had more power than SMT in simulated array data with very incomplete variant ascertainment.

Comparing Figs. 3 to 2, we see that the relative power gains available from LOCATER are typically less in the case of hidden causal variants compared to the case of observed causal variants, but still substantial, across settings. The performance of ARG-Needle is the same in both figures because we provided ARG-Needle with the true underlying ARG in each simulation, making its performance unaffected by whether the simulated causal variants were observed or hidden. Except for the case of 3 doubletons, the power results reported in Fig. 3 for SMT are remarkably similar to those reported in Fig. 2 despite all of the causal variants being hidden. For the case of 3 doubletons, we see that the power of SMT is markedly reduced when the causal variants are hidden, making the relative power gain from LOCATER markedly large.

In order to confirm that these power results are robust to our choice of 10 kb as the size of the causal region, we replicated all of our experiments involving 9 causal variants assuming a 100 kb causal region. The resulting power curves are very similar (Supplementary Figs. 26 & 27).

In order to compare LOCATER to the results one might obtain using sliding windows, we ran ACAT-O (STAAR without variant annotations) on our observed variant simulations from Fig. 2, where any variant could be causal. Rather than testing all sliding windows for every simulation and effect size, we gave ACAT-O the precise location and width of the 10 kb causal window for each simulated dataset. This is an upper bound on the performance of ACAT-O in real-world settings where the location and size of the causal window are unknown. We ran ACAT-O in two different ways: one in which we restricted the variants considered to rare variants (MAF < 0.01) and another where all variants are tested regardless of frequency (Supplementary Section 2.9). Similar to the other methods, the genome-wide discovery threshold for ACAT-O was determined via null simulations (Supplementary Section 2.10). As shown in Fig. 4, the performance of both ACAT-O approaches is roughly the same as SMT. LOCATER maintains its power advantage.

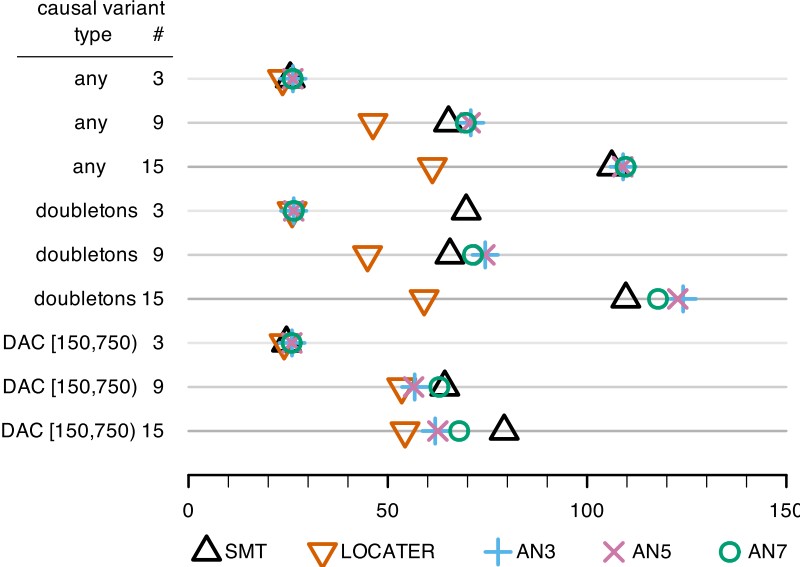

**Fig. 3.** Dotplot of total association signal strength required to achieve 80% power (lower is better) under various simulation conditions where all causal variants were hidden. Total association signal strength is the $-\log_{10}$ P-value that one would obtain by testing the simulated phenotype Y with an oracle ANOVA model that "knows" the causal variants and targets only those for testing. Causal variant # denotes the number of simulated causal variants. Causal variant type "any" means any variant could be causal; "doubletons" means only doubletons could be causal; "DAC [150,750]" means only variants with a derived allele count in [150, 750], corresponding to a derived allele frequency in [0.0025, 0.0125), could be causal.

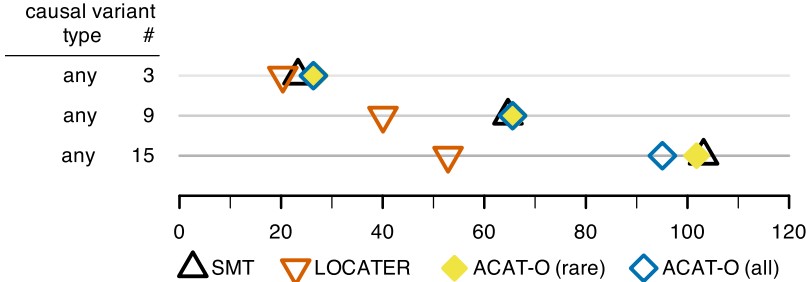

**Fig. 4.** Dotplot of total association signal strength required to achieve 80% power (lower is better) under various simulation conditions where all causal variants were observed, including comparison to oracle ACAT-O methods that are given the causal variant window. ACAT-O (rare) only tests variants with MAF < 0.01 whereas ACAT-O (all) tests all variants within the causal window. Total association signal strength is the $-\log_{10}$ P-value that one would obtain by testing the simulated phenotype Y with an oracle ANOVA model that "knows" the causal variants and targets only those for testing.

In Supplementary Section 1.4, we pair each plot of power curves with a companion plot showing which LOCATER sub-test is driving the gain in power (Supplementary Figs. 9, 12, 15, 18, 21, & 24). These results show that SD is typically the source of LOCATER power gains, not QForm, reflecting the statistical advantage of SD-based methods over quadratic form based procedures in the case of sparse signals (Christ et al. 2024a). While these simulations appear to imply that SD is only effective at capturing signals driven by very rare variants, this is expected since we only encoded very rare variants in the clade genotype matrix $X^{(\ell)}$ passed to SD. If SD was used to test the entire local genealogy at every locus, we may see increased statistical efficiency in incorporating common variant associations (see Discussion). Substantial power gains are possible via LOCATER in the case of multiple rare causal variants.

## Localization

Alongside power, an important factor for real-world utility is the ability of an association method to accurately localize causal variant(s) within a relatively narrow genomic interval. In Supplementary Section 1.4, we also pair each plot of power curves with a companion localization plot (Supplementary Figs. 10, 13, 16, 19, 22, & 25). To measure the ability of a given method to

localize the causal region, we calculated the distance between the most significant marker (lead variant) reported by a method and the midpoint of the causal region in every simulation, taking the average distance in the case of tied lead variants. We used these distances to estimate the width of an 80% confidence interval. This width represents the answer to the question, "How large of a search window centered on the lead variant would an investigator need in order for that window to capture the midpoint of the causal region 80% of the time?" In our plots reporting these confidence interval widths (smaller is better), we only report confidence interval widths at signal strengths where the corresponding method had at least 80% power to detect the causal region. As in our power curves, every point is estimated based on 1,000 independent simulated samples.

For intuition, in this setting, 50% confidence intervals would be equivalent to 2 times the median distance from the lead variant to the midpoint of the causal region; 100% confidence intervals, 2 times the max distance from the lead variant to the midpoint of the causal region. Our 80% confidence intervals are a compromise between these two extremes, providing a estimate of how far out from a lead variant an investigator would have to look before feeling reasonably confident that they've reached the causal region.

As expected, both LOCATER and SMT struggled to localize the causal region more when the causal variants were hidden. The localization performance of ARG-Needle was unaffected by whether the simulated causal variants were observed or hidden because we provided ARG-Needle with the true underlying ARG in every simulation.

Across simulations where all variants in the causal region were potential causal variants, LOCATER was substantially more accurate in localizing the causal region than SMT or ARG-Needle, regardless of whether the causal variants were observed or hidden. In simulations with intermediate causal variants (derived allele count in [150, 750)), ARG-Needle and SMT slightly outperformed LOCATER for some signal strengths. Overall, LOCATER improved or effectively tied the localization accuracy of SMT and ARG-Needle in the intermediate causal variant case. With the exception of a few signal strengths when there were 9 causal variants, LOCATER also outperformed SMT and ARG-Needle in localizing the causal region when doubleton causal variants were observed. In all simulations where the doubleton causal variants were hidden, regardless of the number of causal variants, all methods performed very poorly in localization: all methods reported confidence intervals wider than 500 kb across all signal strengths where they achieved at least 80% power.

Overall, our results suggest that LOCATER can leverage allelic heterogeneity to improve the localization of trait mapping compared to standard SMT and ARG-Needle.

## Scalability

The ability to scale to large modern datasets with hundreds of thousands of samples is essential for the success of any trait mapping approach and the size of local genealogies presents a significant challenge. Based on our simulations involving 30,000 samples (60,000 haplotypes), LOCATER took an average of 19.14 s (sd: 2.88 seconds) to perform sprig testing and an average of 3.07 min (sd: 0.98 minutes) to perform quadratic form testing at each variant. These simulations were run on a shared-time university HPC cluster with heterogeneous nodes hosting a mix of CPU architectures. All jobs requested 8 cores and 160 GB of memory. When combined with the computational overhead required to form the clade genotype matrix $X^{(\ell)}$ and $\Omega^{(\ell)}$ provided to LOCATER, our simulations required an average of 4.00 min (SD: 0.99 minutes) to test each target locus. When this is combined with the additional cost of performing ancestry inference with kalis, our simulations required an overall average of 6.42 minutes per target locus. These results suggest that future applications of LOCATER that only use inferred clades $X^{(\ell)}$ and avoid use of $\Omega^{(\ell)}$ will achieve substantial computational savings.

In parallel work, we ran LOCATER in combination with kalis on a real sequencing dataset including 6,795 individuals and 101 correlated quantitative traits (Wang et al. 2024). We divided the genome into 4,580 (partially overlapping) segments; each segment had an average of 13,000 variants. We allocated 12 cores and 60 GB of memory per segment. This allowed kalis to store two checkpoints in memory. The average time required for kalis and LOCATER to screen each segment was 32 minutes. That is equivalent to 3.35 yr of single-core compute time. While substantial, this equates to 1.22 days using a cluster of 1,000 CPUs. In other preliminary work, we have run LOCATER on 12,964 genomes using commodity hardware. This experiment required 8,257 CPU-days to screen 4 quantitative traits for associations genome-wide (unpublished data). These results show that it is feasible to run LOCATER on a moderately large genomic dataset using an academic compute cluster.

## Discussion

We have presented a general framework for using inferred local ancestries to boost SMT association signals in the presence of allelic heterogeneity. To our knowledge, this is the first demonstration of any ancestry-testing approach that yields significant power gains over SMT in a genome-wide screen that includes noncoding regions. More importantly, our approach can be applied in conjunction with any ancestry inference engine, thus providing a flexible association testing framework that can adapt to rapidly improving ancestry inference methods.

As mentioned in the Introduction and the Results, we have demonstrated the real-world power gains attainable via LOCATER in a dataset of 6,795 Finnish genomes with extensive quantitative trait measurements (Wang et al. 2024). There, as in our simulations (see Results), we find that LOCATER requires a less stringent genome wide discovery threshold than SMT. We believe this is due to the increased dependence between proximal tests induced by the dependence between proximal local genealogies, which suggests that it will be safe for investigators to apply their SMT discovery threshold for a given dataset (e.g. $10^{-8.5}$) to their LOCATER results. However, these results also suggest that running null simulations, analogous to those described in Supplementary Section 2.10, to estimate a LOCATER-specific genome-wide discovery threshold for a given dataset will improve statistical power. Since LOCATER is parallelized across phenotypes, performing these null simulations is straightforward once a user can run LOCATER on a given dataset, and in our experience, requires computational resources comparable to a genome-wide association screen.

Our power simulations demonstrate that SMT typically outperforms ARG-Needle in the context of high variant ascertainment (e.g. high coverage sequencing data). The fact that the performance of SMT is essentially unchanged when the causal variants are hidden (Fig. 3) rather than observed (Fig. 2), besides the case of 3 causal doubletons, reflects the fact that SMT typically detects associated loci by finding a lead variant that serves as a proxy for one or more of the causal variants. This is precisely the same strategy used by ARG-Needle: it infers an ARG from observed variants, discards the observed variants, and then places new hypothetical mutations on edges of the inferred ARG for testing. In short, in the case of high variant ascertainment, ARG-Needle effectively is just exchanging one set of potential proxy variants for another. Thus, it is not surprising that the set of observed variants does about as well as a resampled set of hypothetical variants in most cases.

Our hidden causal variant simulations expose the multiple testing tradeoff inherent to the ARG-Needle approach. In our simulations, running ARG-Needle with a mutation rate of $10^{-3}$ (AN3) or $10^{-5}$ (AN5) places a hypothetical variant on effectively every edge in the ARG. Since we provide ARG-Needle with the true ARG, the question becomes whether imputing the hidden causal variants was worth the additional testing burden of testing all of the clades in the ARG. It turns out this is true in the case of 3 hidden doubletons. There's about a 1/3 chance that there will be no strong proxy among the observed variants for any of the 3 causal doubletons when they are hidden, as evidenced by the abrupt kink in the SMT power curve where it reaches Power $\approx 0.66$ (Supplementary Fig. 11). However, this turns out to be false in all of our other simulations: there is virtually always a strong proxy variant somewhere along each simulated 1 Mb chromosome (Fig. 3).

The largest power gains demonstrated in this paper were seen in the case of multiple rare causal variants evaluated using the SD sub-test. This suggests that if we had tested more of the underlying tree structure at each locus with SD we may have achieved

even greater power. In other words, as mentioned above, we may have clustered each distance matrix and tested more common clades than simply the sprigs with SD. This approach would present more of the underlying ancestral tree to LOCATER via $X^{(\ell)}$ rather than via $\Omega^{(\ell)}$. Exploring the power of LOCATER at different points along the continuum between testing all of the ancestral structure with $X^{(\ell)}$ and testing all of the ancestral structure with $\Omega^{(\ell)}$, will be a focus of future research.

In conjunction with the new features added to kalis (Aslett and Christ 2024), LOCATER provides an efficient method for genome-wide testing that is ready for use on real-world datasets now. These new features involve several algorithms—a general HMM checkpointing algorithm, a fast clustering algorithm, and a fast trace calculation method—that will likely prove helpful for the acceleration of other ancestry inference and association methods (see Methods). Our quadratic form tail approximation approach, based on a shifted difference of chi-square random variables (Equation (11)), provides a basis for emerging association methods to reliably test local relatedness matrices that may not be positive semi-definite.

Adequately adjusting for population structure when testing inferred local ancestries is an open and challenging problem. In this initial version of LOCATER, we allow principal components (PCs) to be included in A. As mentioned in the Results, we also parameterized our quadratic form tail approximation in a way that accommodates genomic-control-like inflation adjustments without requiring recalculating the genealogy at any LOCATER target loci (Supplementary Section 2.5).

While future analyses applying LOCATER or any local ancestry testing method to real genomic data will need to take special care when examining Q–Q plots for inflation, we believe that these methods have promise to avoid false discoveries due to residual population structure that would be mistakenly found by classic SMT. In parallel work applying LOCATER and SMT to real quantitative traits, adjusting both models for the same principal components, we observed that LOCATER P-values typically exhibit substantial genome-wide inflation even when SMT P-values appear well-calibrated (Wang et al. 2024). This can be explained by the presence of cryptic confounding population structure: residual medium-to-fine scale population structure that is correlated with our trait(s) of interest but orthogonal to our PCs. Current SMT methods struggle to correct for this cryptic structure (Blanc and Berg 2025). Since the vast majority of individual variants genome-wide are not correlated with these cryptic population features, their presence is hidden in SMT Q–Q plots.

However, by testing large sets of variants (clades) in every test rather than individual variants, LOCATER makes it much more likely that a given test statistic will be affected by some confounding aspect of residual population structure. Hence the inflation in the body of the unadjusted LOCATER Q–Q plots. We expect this sensitivity is not unique to LOCATER or even ARG-focused methods, but rather a feature of any method targeting allelic heterogeneity: testing sliding windows with ACAT-O.

We expect that the observed inflation in LOCATER will prove to be a feature rather than a bug. First, this inflation reveals the presence of residual population structure that is likely to still drive false positive discoveries in SMT results that appear to be calibrated. Since ARG-Needle is effectively performing SMT on inferred clades, such signals are also likely to beguile ARG-Needle results. Second, the observed inflation provides a means to adjust for the cryptic structure. As we show in Wang et al. (2024), the inflation in LOCATER can be removed via a generalized version of genomic control to obtain calibrated P-values. A similar solution

is not available to SMT. Since the vast majority of variants are not correlated with the confounding process, the SMT P-values already appear calibrated. In other words, genomic control implicitly assumes that the confounding effects impacting the P-values in the body of the Q–Q plot are exchangeable with those impacting the P-values in the tail of the Q–Q plot. By incorporating residual confounding signals into the majority of test statistics genome-wide, LOCATER makes this crucial assumption safer. In contrast, this assumption fails for SMT when relatively few of the individual variant test statistics are confounded. We leave further discussion and investigation of population structure adjustment for ancestry-based methods to Wang et al. (2024) and future work.

LOCATER makes a number of critical methodological advances towards powerful ancestry-based association testing. We expect that further work building on these advances alongside the application of LOCATER to more diverse datasets will yield new functional discoveries.

## Methods
### Haplotype data simulation
In order to assess the calibration and power of LOCATER, we simulated 100 genomic datasets, each consisting of a 1 Mb chromosome for 30,000 human samples (60,000 haplotypes). Each dataset was simulated using msprime (Kelleher et al. 2016). In order to model the diversity of arising genomic datasets, 10,000 samples in each dataset were drawn from each of three 1,000 Genomes populations—Yoruba, Han Chinese, and Central European. See Supplementary Section 2.7 for further details.

### Phenotype simulation
To simulate each phenotype vector, following Li et al. (2020), we first simulated two background covariate vectors: $a_1$ a vector of independent standard Gaussian random variables and $a_2$ a vector of independent Rademacher random variables. We tested each phenotype vector assuming that these two background covariates were observed and included in A from Equation (1). For our null simulations—without any causal variants—this amounted to sampling each phenotype vector from $Y \sim N(a_1 + a_2, I)$. For our power simulations, in the middle of each 1 Mb region, we selected causal variants within a 10 kb causal window. We fully replicated these simulations under all 18 possible combinations of 3 parameters: the number of causal variants, the allele frequency constraint imposed on those causal variants, and whether the causal variants were assumed to have been observed (called during sequencing) or hidden. More explicitly, we considered the case of 3, 9, or 15 causal variants. These causal variants were selected uniformly at random from among variants within the 10 kb causal window meeting the given allele frequency constraint. As our primary focus, we considered the case of no allele frequency restraint, in which case every variant in the 10 kb causal window had an equal chance of being selected as a causal variant. We also considered the case where all causal variants were constrained to be doubletons (present in two copies) and the case where all of the causal variants were constrained to have derived allele frequency in the half open interval [0.0025, 0.0125). If the simulated chromosomes did not include the requisite number of causal variants within the 10 kb causal window, we rejected that simulated dataset and simulated a new set of chromosomes.

Given a set of causal variants (variants with nonzero effects), we simulated Y while distributing the observed effects across the causal variants as evenly as possible by manipulating the QR-decomposition as done by (Christ et al. 2024a). Following their

approach, let $\mathcal{A}$ denote this selected set of causal variants and $X_{\mathcal{A}}$ denote the genotype matrix encoding those causal variants. We define the total association signal strength as the $-\log_{10}$ P-value that one would obtain by testing the resulting Y with an oracle ANOVA model that "knows" the causal variants. More explicitly, it is the $-\log_{10}$ P-value that one would obtain from the likelihood ratio test comparing the oracle regression model $Y \sim A\alpha + X_{\mathcal{A}}\beta + \epsilon$ to its nested null model $Y \sim A\alpha + \epsilon$ where the variance of the noise term is known to be one: $\epsilon \sim N(0, I_n)$.

Consider the QR-decomposition $QR = \widetilde{X}_{\mathcal{A}}$, which we define as $P_A^\perp X_{\mathcal{A}}$ with length-normalized columns. The sufficient statistic for testing oracle model against its nested null model is $\|Q^\top Y\|_2^2$ with expected value $\|R\beta\|_2^2$. For a desired total association signal strength s, we solve $R\beta = \sqrt{F_{\chi_d^2}^{-1}(1 - 10^{-s})}\,\mathbf{1}$ to make the magnitude of each entry of $\beta$ as similar as possible. Then, we simulate $Y = A\alpha + (\widetilde{X}_{\mathcal{A}}\beta + P_{X_{\mathcal{A}}}^\perp \epsilon)$ where $\epsilon \sim N(0, I_n)$. This ensures that the observed $\hat{\beta} = R^{-1}Q^\top Y = \beta$ is stable across simulations and that the total association signal strength will be approximately s in every simulation. Supplementary Section 1.3 delineates the effect sizes, $\beta$s, induced by this procedure as a function of s and minor allele count (Supplementary Figs. 6 & 7, Supplementary Table 1).

## Checkpointing approach

To understand the need for checkpointing, as mentioned above and further detailed below, recall that we implicitly need the pairwise distance matrix $d^{(\ell)}$ from Equation (2) returned by kalis in order to obtain $X^{(\ell)}$ and $\Omega^{(\ell)}$ at each target locus $\ell$. While the original kalis 1.0 release can efficiently propagate the forward and backward recursions to obtain the forward probability matrix $f^{(\ell)}$ and backward probability matrix $b^{(\ell)}$ needed to calculate $d^{(\ell)}$ at a single locus $\ell$, obtaining the pair $(f^{(\ell)}, b^{(\ell)})$ at sequential positions $\ell$ is challenging for HMMs due to the uni-directionality of the forward and backward recursions. The compute-minimizing approach would involve running a single pass of the forward algorithm—iterating the forward recursion to target locus 1, then locus 2, and so on until locus L—and a single pass of the backward recursion from target locus L to target locus 1 while storing $f^{(\ell)}$ and $b^{(\ell)}$ at every $\ell = 1, \ldots, L$. Since each $f^{(\ell)}$ and $b^{(\ell)}$ consumes $8N^2$ bytes of memory (e.g.: 80 GB for $n = N/2 = 50{,}000$ haplotypes), this approach requires far too much storage for most genomic datasets. On the other hand, we have the memory-minimizing approach, where we restart the forward and backward recursions from the respective ends of the genomic segment for every target locus. While this approach only requires storing a single $f^{(\ell)}$ and a single $b^{(\ell)}$ at any given time, it demands far too much compute time for most genomic datasets, requiring $\mathcal{O}(L^2N^2)$ floating point operations (FLOPs)—a prohibitive cost. An attempt to rescue this approach by splitting the genome into smaller segments (running in chunks) would still require $\mathcal{O}(L^2N^2)$ compute time.

We provide a checkpointing algorithm that finds an optimal balance in this memory-compute trade-off, minimizing the compute time required given a fixed memory budget. The overall idea is to occasionally stop the forward recursion and store $f^{(\ell)}$ at its current position as a checkpoint (typically overwriting an old checkpoint) in order to avoid repeatedly restarting the forward recursion from the beginning of the genomic segment. We start with a user-specified memory budget capable of holding C checkpoints, each storing a $N \times N$ matrix of forward probabilities $f^{(\ell)}$. We run the backward recursion once across the entire chromosome or genomic segment, stopping at each consecutive target locus sequentially from the target locus with the largest position ($\ell = L$) to the target locus with the smallest position ($\ell = 1$). When the backward recursion stops at a given target locus, we run the

forward recursion from the nearest checkpoint to meet the backward recursion and so obtain $X^{(\ell)}$ and $\Omega^{(\ell)}$ at that target locus. Note it is natural for us to perform this backwards along the genomic segment, since there is a slightly higher computational cost for the backward recursion and hence we favor repetitive restarts of the forward recursion.

Iterating from locus L down to locus 1 makes minimizing the compute required for the backward recursion trivial: we simply visit each locus sequentially in a single pass. The challenge is determining where and when to overwrite existing checkpoints to minimize the total distance (number of variants) that the forward algorithm needs to iterate over in order to provide forward matrices in reverse order $f^{(L)} \to f^{(L-1)} \to \cdots$. In Supplementary Section 2.3 we show how to solve for a schedule of checkpoints that achieves this minimum for any discrete time HMM, given storage for a fixed number of checkpoints C. We call this solution the optimal checkpointing schedule. After a forward matrix $f^{(\ell)}$ is obtained at a given target locus $\ell$, this schedule instructs kalis which checkpoint to use to restart the forward recursion to obtain the next forward matrix at locus $\ell - 1$, and where to lay down new checkpoints (if any) as the forward recursion proceeds to that locus. The checkpointing schedule also dictates where to initialize the C checkpoints as we iterate the forward recursion to the first target locus L.

Supplementary Fig. 1 shows how the efficiency of our checkpointing algorithm scales in both L and C. In the case where $L = 10^5$ equally-spaced target variants, the figure shows that the forward algorithm would be required to propagate over a total distance $D > 10^4 L$ target variants without checkpoints. Sufficient memory to store $C = 2$ checkpoints with our approach brings D under 100L target variants; $C = 8$ checkpoints brings D under 10L target variants. Even in the $L = 10^6$ case, which is larger than we would expect over any phased genomic segments since we only test target variants with moderately significant SMT P-values, $C = 10$ checkpoints brings D nearly down to 10L target variants.

Solving for the optimal checkpointing schedule can be computationally intensive for any given set of target loci. The version of the checkpointing schedule solver currently implemented in LOCATER assumes that target loci are evenly spaced. This simplification does not qualitatively change performance but allows us to solve for the optimal checkpointing strategy for a given L via a dynamic program, making the solution readily available (Supplementary Section 2.3). Our checkpointing implementation is available via the ForwardIterator function and associated helper functions now provided in kalis 2.0.

Of course, for datasets with a large number of samples, there may not be sufficient capacity to store many checkpoints in memory. At a minimum, running kalis on n samples (2n phased haplotypes) to obtain each $\Omega^{(\ell)}$ requires $32n^2$ bytes to store the forward and backward probabilities and another $8n^2$ bytes to store $\Omega^{(\ell)}$. Storing each additional checkpoint of forward probabilities requires $16n^2$ bytes. Given the nested nature of our checkpointing algorithm, most checkpoints can be stored on disk rather than memory, which comes at minimal computational cost as long as one or two of the checkpoints (the ones that are closest to the current target loci) are always kept in memory. We plan to add native support for storing file-backed checkpoints to kalis in the near future. Looking further ahead, kalis can already be distributed across machines, each running the LS model on a different subset of recipient haplotypes (Aslett and Christ 2024), but running LOCATER across distributed machines would require substantial network communication. Reducing this communication is a direction of future work.

## Defining our generalized relatedness matrix

Here we build up to our definition of a generalized eGRM, which we will pass to LOCATER as $\Omega^{(\ell)}$. We will construct $\Omega^{(\ell)}$ based on an asymmetric genetic distance matrix $d^{(\ell)} \in \mathbb{R}_{\geq 0}^{N \times N}$, such as the one provided by kalis (Equation (2)), and a set of monotonic regularization functions $g_1, \ldots, g_N$ which we will introduce shortly. Recall that $d_{ij}^{(\ell)}$ measures the distance to haplotype $i$ from haplotype $j$. The distance from any given haplotype to itself $d_{ij}^{(\ell)} = 0$. Let $\pi_j : [N] \to [N]$ be the permutation that sorts $d_{\cdot j}^{(\ell)}$ such that $d_{\pi_j(1)j}^{(\ell)} \leq d_{\pi_j(2)j}^{(\ell)} \leq \ldots \leq d_{\pi_j(N)j}^{(\ell)}$. By convention, $\pi_j(1) = j$ and $\pi_j(N+1) = \pi_j(N)$. Using $d^{(\ell)}$, we define a local haplotype relatedness matrix $\Psi^{(\ell)} \in \mathbb{R}_{\geq 0}^{N \times N}$ with elements

$$\Psi_{ij}^{(\ell)} = \sum_{k = \pi_j^{-1}(i)}^{N} \psi_j(k) \quad \text{where} \quad \psi_j(k) = \frac{1}{k} g_k\left(d_{(k+1)j}^{(\ell)} - d_{kj}^{(\ell)}\right) \tag{5}$$

and each $g_k : \mathbb{R}_{\geq 0} \to \mathbb{R}_{\geq 0}$ is a monotonic function of $x$ such that $g_k(0) = 0$. Note, given these definitions, $\psi_j(N) = 0$.

Our construction does not require the assumption of diploid samples but we will assume that here for ease of exposition. We will assume that the rows and columns of $\Psi^{(\ell)}$ are permuted such that haplotypes from the same sample are grouped together. This allows us to succinctly write our generalized eGRM in terms of Equation (5) as

$$\Omega^{(\ell)} = \text{sym}\left(B_{n,2}^{\mathsf{T}} \Psi^{(\ell)} B_{n,2}\right) \tag{6}$$

where $\text{sym}(M) = \frac{1}{2}(M + M^{\mathsf{T}})$ is the symmetric part of a square matrix $M$, $B_{n,2} = I_{n \times n} \otimes \mathbf{1}_2$, and $\otimes$ is the Kronecker product.

In Supplementary Section 2.2 we explicitly show how this construction of $\Omega^{(\ell)}$ generalizes the standard eGRM. In short, there we show that the eGRM can be expressed in terms of a haplotype similarity matrix assuming Hardy–Weinberg equilibrium and specific choices for the background covariates and allele frequency weights. Then we connect that haplotype similarity matrix representation to our choice of $\Psi^{(\ell)}$ in Equation (5).

## Efficiently constructing clade genotypes and our generalized eGRM in LOCATER

Building on the notation used in Equation (5), currently in LOCATER and this paper, we set

$$g_k(x) = \begin{cases} 0 & x < c \\ \min(x, 1) & x \geq c \end{cases} \tag{7}$$

for all $k$ where our threshold $c = -0.2 \log(\mu)$ and $\mu$ is the mutation probability parameter provided to the LS model. This choice of regularization function(s) tends to filter out many low evidence clades. This function also restricts the clade matrix representation so that at most one mutation can be present on a given branch.

Given such a regularization, $\Psi_{\cdot j}$ has a series of nested neighborhoods of donor haplotypes along $i = \pi_j(1), \pi_j(2), \ldots, \pi_j(N)$ where there are distances of at least $c$ between adjacent level sets of distances. This allows us to represent the level sets of $\Psi_{ij}$ as the solution to a clustering problem on the real interval from 0 to the maximum possible distance $(D)$ where we require unique clusters to be at least distance $c$ apart. Each cluster corresponds to a level set of donor haplotypes. We use a single-pass partial sorting algorithm based on doubly-linked-lists to solve this clustering problem in $\mathcal{O}(N)$ time. In our experiments on simulated haplotypes, our partial

sorting algorithm achieves roughly an order of magnitude speed up over merge sort. Given the definition of $v$ in Equation (2), the maximum possible distance is $D = -\log(v) \approx 744.44$. This maximum is helpful in accelerating our implementation because the number of possible level sets (clusters) $d$ is bounded above $d \leq \lceil D/c + 1 \rceil$, allowing us to efficiently preallocate sufficient memory to store the clustering solution. Since this partial sorting algorithm can be run in parallel for each recipient haplotype (for each column $\Psi_{\cdot j}$), we use a multi-threaded implementation in kalis 2.0 that processes columns of $\Psi$ in pairs. This allows us to directly compute our sample by sample matrix $B_{n,d}^{\mathsf{T}} \Psi B_{n,d}$ in a single pass. Symmetrizing this matrix yields $\Omega^{(\ell)}$ as defined in Equation (6). Future work will focus on reducing the computational and memory requirements of this symmetrization step.

## Parallelized stable distillation procedure

Given a matrix of inferred, clade-based genotypes $X^{(\ell)}$, we use the one-predictor-at-a-time SD procedure described in Equation 4 of (Christ et al. 2024a) equipped with the simple quantile filter presented in Algorithm 1 of (Christ et al. 2024a). In this SD procedure, we take $(A, G^{(\ell)})$ as the background covariates when testing $H_0^{\text{SD}}$ at a particular target locus $\ell$. In short, using this approach, we "distill" one $\beta_j$ for $j = 1, \ldots, p$ at a time, obtaining an independent $P$-value for each. These $P$-values are then tested using the Rényi Outlier Test (Christ et al. 2024b). To run this procedure, LOCATER requires an estimated upper bound $c$ on the number of independent causal clades: $c \geq |\{j : \beta_j \neq 0\}|$. By default and throughout this paper, we set $c = 16$. This bound $c$ is used to set the simple quantile filtering threshold used during distillation. Explicitly, LOCATER sets the quantile filtering threshold $t = F_{c,p-c+1}^{-1}(0.01)$ where $F_{a,b}$ is the CDF of the Beta distribution with expectation $\frac{a}{a+b}$. By default, the maximum number of outliers considered by the Rényi Outlier Test is set to $c$.

## Quadratic form testing & tail approximation

Let $QR = (A, G_j)$ be the QR-decomposition of the $n \times (q + 1)$ matrix $(A, G_j)$ and let $P = I - QQ^{\mathsf{T}}$ project onto the subspace orthogonal to the columns of $(A, G_j)$. Differentiating the likelihood corresponding to Equation (1) with respect to $\tau$ yields $Y^{\mathsf{T}} P \Omega^{(\ell)} P Y$ as the score statistic. Under $H_0^Q$,

$$Y^{\mathsf{T}} P \Omega^{(\ell)} P Y \sim \sum_{j=1}^{n} \lambda_j Z_j^2 \tag{8}$$

where each $Z_j \overset{\text{iid}}{\sim} N(0, 1)$. Following the approach of FastSKAT (Lumley et al. 2018), we use partial eigendecomposition to obtain a computationally tractable approximation to this null distribution. Given, a top-$k$ eigendecomposition in which we explicitly calculate the leading $k$ eigenvalues, we have

$$Y^{\mathsf{T}} P \Omega^{(\ell)} P Y \sim T_k + R_k \tag{9}$$

where $T_k = \sum_{j=1}^{k} \lambda_j Z_j^2$ and $R_k = \sum_{j=k+1}^{n} \lambda_j Z_j^2$. Since $\lambda_{k+1}, \ldots, \lambda_n$ are unknown, FastSKAT proposes approximating the distribution of $R_k$ using a single chi-square random variable $\widetilde{R}_k \sim \alpha \chi_\nu^2$. The scale parameter $\alpha$ and degrees of freedom parameter $\nu$ are set to match the mean and variance of $R_k$, as initially proposed by Satterthwaite (1946):

$$\alpha = \frac{\eta_2}{\eta_1} \quad \nu = \frac{\eta_1^2}{\eta_2} \tag{10}$$

where $\eta_1 = \mathrm{tr}(P\Omega^{(\ell)}P) - \sum_{j=1}^{k} \lambda_j$ and $\eta_2 = \|P\Omega^{(\ell)}P\|_{HS}^2 - \sum_{j=1}^{k} \lambda_j^2$. Substituting $\widetilde{R}_k$ in for $R_k$, FastSKAT uses $T_k + \widetilde{R}_k$ as an approximate null distribution. This is still a linear combination of chi-square random variables, but since all of its parameters are now known, its distribution is readily available via the fast Fourier transform (FFT).

Unfortunately, in the context of LOCATER, $\widetilde{R}_k$ does not provide an adequate approximation to the distribution of $R_k$. First, the Satterthwaite approximation assumes that $P\Omega^{(\ell)}P$ is positive semi-definite (PSD). This is not at all guaranteed in our application. The closeness of any $P\Omega^{(\ell)}P$ to PSD will depend on the ancestry at $\ell$ as well as the user's choice of ancestry inference engine and clade-encoding method. The $P\Omega^{(\ell)}P$ matrices we have observed in the development of LOCATER are typically close to, but not exactly, PSD.

Second, we found that the tails of $\widetilde{R}_k$ tended to decay much faster than those of $R_k$, yielding anti-conservative estimates of more extreme $P$-values. This is a consequence of the fact that, $\widetilde{R}_k$, as a gamma random variable, has an exponential right tail with a rate of decay $(2\alpha)^{-1}$. Since $R_k$ is itself a linear combination of gamma random variables, the term with the largest coefficient governs the behavior in the right tail. More precisely, as long as some $\lambda_j > 0$ for $j > k$, $R_k$ has an exponential right tail with rate $(2\max_{j>k} \lambda_j)^{-1}$. Intuitively, this comes from the fact that extreme observations of $R_k$ in the far right tail are much more likely to arise from the term with the largest scale. We have analogous behavior in the left tail: as long as some $\lambda_j < 0$ for $j > k$, $R_k$ has an exponential left tail with rate $(2\min_{j>k} \lambda_j)^{-1}$. We found that the Satterthwaite approximation consistently over-estimates the rate of decay because $(2\alpha)^{-1}$ tends to be much larger than $(2\max_{j>k} \lambda_j)^{-1}$. In fact, in the PSD case, $(2\alpha)^{-1} \geq (2\max_{j>k} \lambda_j)^{-1}$ (Supplementary Section 2.4).

In order to make the tails of our approximating distribution more robust (heavier) and obviate the PSD requirement, we propose approximating $R_k$ with a difference of independent chi-square random variables:

$$\ddot{R}_k \sim |\lambda_k|(W' - W'' + \mu) \tag{11}$$

where $W' \sim \chi_a^2$ is independent of $W'' \sim \chi_b^2$. A derivation and explicit equations for the three scalar parameters—$a$, $b$, and $\mu$—are given in Supplementary Section 2.5 in terms of $\eta_1$ and $\eta_2$. In short, these parameters are set so that our approximation matches the mean and variance of $R_k$ while simultaneously minimizing $|\mu|$. Like $T_k + \widetilde{R}_k$, $T_k + \ddot{R}_k$ is also a convolution of weighted chi-square random variables, making it just as easy to obtain $P$-values for using the FFT.

Since the left and right exponential tails of $\ddot{R}_k$ decay with rate $|2\lambda_k|^{-1} \leq (2\max_{j>k} |\lambda_j|)^{-1}$, the tails of $\ddot{R}_k$ are at least as heavy as those of $R_k$. Of course, this alone does not guarantee accurate $P$-value estimation across the range of quantiles required for LOCATER. We must also accurately estimate of the body of the distribution across the full range of $P\Omega^{(\ell)}P$ we observe. The stopping criteria we use for top-$k$ eigendecomposition in LOCATER addresses this issue and combines with $\ddot{R}_k$ to yield reliable $p_Q$ $P$-values (Supplementary Section 2.4). A key advantage of our specification of $\ddot{R}_k$ is that it can be easily generalized to accommodate three inflation parameters—$v$, $\delta_\star^2$, and $\delta_\dagger^2$—that are orthogonal to $a$, $b$, and $\mu$ (Supplementary Section 2.5). There we show how these parameters arise when considering the presence of some unobserved confounder although, in practice, inflation of the null distribution may also be driven by polygenicity. The orthogonality

allows us to tune $v$, $\delta_\star^2$, and $\delta_\dagger^2$ to help reduce any inflation observed in an initial Q–Q plot of $P$-values $p_Q^{(\ell)}$ without having to recalculate $a$, $b$, or $\mu$. In other words, we can adjust $v$, $\delta_\star^2$, and $\delta_\dagger^2$ without having to re-calculate the genealogy at any target variant. While we do not make use of that capability in this paper, we expect it will be important in real data analyses.

Critical to the scalability of this approach is the availability of $\eta_1$ and $\eta_2$ via fast trace calculations. FastSKAT (Lumley et al. 2018) uses stochastic estimator of $\eta_2$. Starting with a symmetric $\Omega^{(\ell)}$, LOCATER employs a fully parallelizable and distributable trace calculation routine that obtains both $\eta_1$ and $\eta_2$ with fewer than $7(q+1)n^2 + 3n(q+1)^2$ FLOPs (Supplementary Section 2.6). Recall that $q$ is the number of columns in our background covariate matrix $A$.

## Combining P-values: MSSE

The classic Fisher method (Mosteller and Fisher 1948) of combining a vector of $P$-values $U = (U_1, \ldots, U_p)^\top$ is based on comparing the test statistic

$$\mathcal{F}(U) = \sum_{i=1}^{p} -\log(U_i) \tag{12}$$

to its null distribution: a gamma distribution with scale parameter 1 and shape parameter $p$, whose survival function we denote by $G_p$. Since it takes the form of simple sum, $\mathcal{F}$ implicitly considers all possible alternatives: any combination of the $P$-values may be non-null. However, in applications such as LOCATER, where we are only interested in certain subsets of alternatives, this can lead to an unnecessary loss in power.

The aim of LOCATER is to enhance SMT signals by leveraging allelic heterogeneity. In a genomic region with allelic heterogeneity, we expect the most significant LOCATER signal to arise at a target locus where the core marker at that locus tags one of the causal alleles. Thus, we are only interested in testing combinations of $P$-values where $p_{SMT}$ is non-null. This in effect reduces the number of combinations of non-null $P$-value alternatives we want to test, which we can achieve by using

$$\begin{aligned} \min\big(&G_1^{-1}(\mathcal{F}(p_{SMT})), \\ &G_2^{-1}(\mathcal{F}(p_{SMT}, p_{SD})), G_2^{-1}(\mathcal{F}(p_{SMT}, p_Q)), \\ &G_1^{-3}(\mathcal{F}(p_{SMT}, p_{SD}, p_Q))\big) \end{aligned} \tag{13}$$

as our test statistic. The null distribution of this test statistic is easily precalculated via simulation. Based on 100 million samples, we verified that the null distribution was very smooth and rapidly converged to an exponential tail (Balkema and De Haan 1974). Hence we summarized the body of the null distribution via a monotone cubic spline and fit the exponential tail to obtain a rapidly computable test.

## Data availability

Our locater R package, alongside installation instructions and the source code, is available on GitHub at https://github.com/ryanchrist/locater. Package documentation and a simple example vignette/article is available on the package website https://ryanchrist.github.io/locater/. Functions documented in the locater package provide an API exposing all of our testing subroutines. These functions accept generic matrices $Y$, $A$, $G^{(\ell)}$, $X^{(\ell)}$, and $\Omega^{(\ell)}$. Porting these matrices into R, as base matrices or sparse matrices, is sufficient to run the testing routines implemented in locater.

While this requires some work, it allows users hoping to deploy locater in conjunction with other ancestry inference engines the ability to do so without re-implementing our testing procedures. The Simple LOCATER Example vignette on the locater package website provides further guidance on how to do this: https://ryanchrist.github.io/locater/articles/simple_gwas_example.html. Code and instructions for replicating our figures are available to browse here: https://github.com/ryanchrist/locater_paper_scripts. At snapshot of that repository is stored here: www.doi.org/10.5281/zenodo.16543317. As detailed in those replication instructions, the simulation data tables underlying our figures are available here: www.doi.org/10.5281/zenodo.16423204. Our public Docker image can be found here https://hub.docker.com/r/rchrist7/mini-shark. This image contains an installation of msprime and other python utilities used to run our haplotype simulations.

Supplemental material available at GENETICS online.

# Acknowledgments

The authors would like to thank Nathan Stitziel, Chris Holmes, and Chris Spencer for helpful advice and discussions.

# Funding

R.C. and X.W. were supported by National Institutes of Health grants R01HG013371-01 and UM1HG008853 to I.H. L.J.M.A. was partially supported by the Engineering and Physical Sciences Research Council research grant "PINCODE," reference EP/X028100/1, and United Kingdom Research and Innovation grant, "OCEAN," reference EP/Y014650/1. D.S. was supported by Biotechnology and Biological Sciences Research Council research grant BB/S001824/1. For the purpose of Open Access, the authors have applied a CC BY public copyright license to any Author Accepted Manuscript version arising from this submission.

# Conflicts of interest

None declared.

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

*Editor: P. Ralph*