## [Peer Review File · Genetics]

Clade Distillation for Genome-wide Association Studies

Ryan Christ, Xinxin Wang, Louis Aslett, David Steinsaltz, and Ira Hall

NOTE: The reviews and decision letters are unedited and appear as submitted by the reviewers.

In extremely rare instances and as determined by a Senior Editor or the EIC, portions of a review may be redacted. If a review is signed, the reviewer has agreed to no longer remain anonymous.

The review history appears in chronological order.

Review Timeline:

Submission Date:	2025-01-04
Editorial Decision:	2025-02-18
Resubmission Received:	2025-04-25
Editorial Decision:	2025-06-14
Resubmission Received:	2025-07-08
Accepted:	2025-07-17

February 18, 2025

GENETICS-2024-307747
Clade Distillation for Genome-wide Association Studies

Dear Dr. Christ:

Two experts in the field have reviewed your manuscript, and I have read it as well. The paper clearly represents an impressive amount of highly technical work on an important subject, and I think furthermore contributes to understanding of how to work on this problem generally. While your manuscript is not currently acceptable for publication in GENETICS, we would welcome a substantially revised manuscript. Both reviewers have comments and concerns to be addressed in a revised manuscript. You can read their reviews at the end of this email.

As you can see below, the reviewers had a number of suggestions to both clarify the manuscript as well as make its key takeaways more salient. Please consider them carefully. In particular: currently, the paper is very heavy on many technical points, which can make it hard to extract the takeaways - consider carefully whether Reviewer 2's suggestions of additional comparisons or simulations (and, tying in the associated medRxiv paper) would help make the paper more readable (and hence impactful) for GENETICS. We look forward to receiving your revised manuscript. Please let the editorial office know approximately how long you expect to need for revisions.

Upon resubmission, please include:

1. A clean version of your manuscript;
2. A marked version of your manuscript in which you highlight significant revisions carried out in response to the major points raised by the editor/reviewers (track changes is acceptable if preferred);
3. A detailed response to the editor's/reviewers' feedback and to the concerns listed above. Please reference line numbers in this response to aid the editor and reviewers.

Your paper will likely be sent back out for review.

Additionally, please ensure that your resubmission is formatted for GENETICS
<https://academic.oup.com/genetics/pages/general-instructions>

Follow this link to submit the revised manuscript: Link Not Available

Sincerely,

Peter Ralph
Associate Editor
GENETICS

Approved by:
Konrad Lohse
Senior Editor
GENETICS

AE comment:

- "the expected genetic relatedness matrix (eGRM) as presented in Equation 1 of [17]":
this is incorrect; equation 1 of Link et al defines the usual GRM, not the eGRM.

Reviewer #1 :

This paper present LOCATOR, a method for jointly testing signals from a single nucleotide polymorphism and its local genealogical tree using a recently proposed stable distillation (SD) framework. The paper demonstrates the advantage of LOCATOR over single marker testing (SMT), which is a standard approach in the literature.

Major comments:

- The correctness of the method hinges on the fact that the three P-values from the observed variant, inferred clade membership encoded in X , and the local genetic relatedness matrix are mutually independent. The authors suggest that SD guarantees this, but I was not able to follow the argument and felt the writeup should be clearer. Here is my understanding of how the procedure works: $p_{\{SMT\}}$ is independent from the others because the other statistics are computed from the trait vector after projecting out the first two terms, A and $G^{\{l\}}$, in equation (1). This looks like a standard result from regression theory. However, it is unclear to me why $p_{\{SD\}}$ and $p_{\{Q\}}$ are independent. This is not a result of SD since, as far as I can tell, SD is only employed when combining p-values that correspond to the columns of $X^{\{l\}}$. Since both $p_{\{SD\}}$ and $p_{\{Q\}}$ use PY (trait after projecting out A and $G^{\{l\}}$), they should be dependent in general. For $p_{\{Q\}}$ to be independent from $p_{\{SD\}}$, shouldn't you additionally project out $X^{\{l\}}$ from Y ? Please clarify the argument and make it more precise.
- How exactly are the branches assigned to $X^{\{l\}}$ and $\Omega^{\{l\}}$? I did not have any hints until I read line 516 where the authors write "tested more common clades than simply the sprigs with SD". If the authors used the term "sprig" in a similar fashion to how it's used commonly, is LOCATOR testing tip clades to $X^{\{l\}}$ and older branches to $\Omega^{\{l\}}$? If there were any heuristics to separate branches into $X^{\{l\}}$ and $\Omega^{\{l\}}$, please describe in the paper.
- Related, I think it's clearer to define $X^{\{l\}}$ as clade membership/assignment in line 159 than calling it inferred genotypes.
- The authors should elaborate on how the null hypotheses in line 171-175 is related to the combining procedure described in section 4.8. From what is written in line 719-724, the combined hypothesis LOCATOR is trying to test is certainly not the intersection of the three hypothesis, which should have low power to test against as the authors pointed out.
- The scope of the local test seems to be the span of the local tree that contains the observed SNP. If so, as adjacent trees will have a similar topology with the one that's being tested and they are omitted in the regression, this local test will inadvertently draw signals from those adjacent regions. I think this is why TRACTOR fails to localize the signal like SMT in a number of scenarios that authors have presented. Nevertheless, the scope of the test is not explicitly stated and only stated as "target loci". Please elaborate on how big the region captured by the proposed local test is.
- When combining signals from the columns of $X^{\{l\}}$ with SD, the order of the columns being tested affects the result according to the SD paper (Christ, Hall, and Steinsaltz). Was there a default setting that is used for LOCATOR?

Minor comments:

- Localization according to the distance between the mid-point and the lead variant looks like an important result presented in the paper. However, all the related figures are in the supplementary material. I guess that the choice was because a series of supplementary figures are in groups as they correspond to a particular experiment, so it is hard to separate just one figure and move it to the main writings. I suggest moving at least one of the experiments to the main section together with the figure that describes localization.
- The claim that the procedure can handle unmeasured confounding looks slippery. It's not exactly correcting confounding but removing test statistics deviating from the $y=x$ line in the QQ-plot. Of course, such an inflation could have been a result of confounding, but it also could've been the extreme polygenicity (or similar genetic architecture) that is found in traits like height and BMI. In such a case, the large number of rejections would be a true signal and not false positives. The authors should phrase their procedure more carefully.

Reviewer #2 :

Please see the attached feedback document.

Associate Editor Comments:

GENETICS-2024-307747

Reviewer feedback

Summary:

The authors have developed a revised version of their ancestry inference methodology (kalis) and implemented a series of tests to detect phenotype-genotype associations when there are multiple causal variants at the same locus. Their kalis v2 update includes numerous modifications that help the method scale up to enormous biobanks, where memory and runtime are concerns. Some of these improvements, like checkpointing or only rigorously testing the largest signals, are standard techniques and incremental. They show in simulations that kalis v2 can detect some associations with hidden variants that single marker tests cannot.

The description of the methods and the experiments is mostly sufficient (see comments). In particular, if their tail approximation is to be a major statistical contribution, the authors should explain the intuition or mathematical justification for their approximation in more detail. If this paper is to stand on its own, without the real data analysis that the authors perform in a separate paper, I recommend **further simulation studies including comparisons to other methods**. I view the **genealogy-based association tests** in Threads or ARG-Needle as the most natural methods to compare to.

The authors allude to three parameters that could be used in their methodology to address confounding. This paper could be enhanced by showing how these parameters are important in simulations with population structure. **The ability of their statistical test to address some confounding would be a more than incremental advance if demonstrated.**

Major comments:

- The authors only show simulation results, as their real data analysis is in a separate medRxiv paper. Given this paper's detailed description of the methods, I respect this decision. At the same time, **I recommend some discussion of the medRxiv paper in the Introduction so as to motivate this work.**
- All Figures: the authors should explicitly **say what "total association signal strength is" in the captions**. For someone skim reading (abstract, Figures, conclusion), it is difficult to understand the x-axis label context.

- Line 748 – 752: I did not click the link to the private Drive folder because I was concerned about disclosing my reviewer identity. I do not see why the software cannot be shared, given that the authors have already written a pre-print.
- Lines 49 – 54: These sentences do not negate that the GWAS methodology in ARG-Needle and Threads is a direct competitor to your method. Given that your results come exclusively from a simulation study and that you are presenting a new method, I believe ARG-Needle (Zhang et al. 2023; Gunnarsson et al. 2024), or something equivalent, **ought to be compared/benchmarked to**. Another clear competitor, w.r.t. detecting a hidden signal of multiple causal variants, is IBD mapping (Cai and Browning 2025; Browning and Thompson 2012), which is not discussed nor compared to.
- Line 159: what is the variable p in X (n by p) matrix? How can an edge be encoded as 0, 1, 2? Please clarify. Are these the clades detected as “sprigs”, which is defined many pages after this sentence?
- Figures 2 and 3: I would replace “intermediate” with the exact allele frequency bin. I would combine these figures into one figure.
- Line 364: Please **provide an interpretation of the parameters**.
- Line 381 – 383: should this be $a_1 * a_2$? I imagine the authors want to consider +- quantitative phenotypes, not mean a_1 plus 1 or -1.
- Many of the Supplementary Figures **could/should be combined into panel figures**: (6-9), combining Figures based on 3, 9, and 15 causal variants and observed versus hidden.

Minor comments:

- Line 46: what is SMT? The acronym is not defined prior. My guess is single marker test.
- Line 73: spell out the acronym TSS. My guess is transcription start site.
- Line 73: spell out kilobase pair (kb)
- Line 82: consider spelling out “Aggregated Cauchy Association Test”. This one is less essential an acronym though.
- Line 89: do you have a citation to back up this claim?
- Line 97: your binary traits plan sounds like a discussion point
- If the authors care to show some simulation results with SINGER tree inference, it could improve claims. Not essential. It could require substantial work.
- Lines 108 – 110: while your math may apply to “any ancestry engine”, unless you have written code to accept any ancestry engine, I would acknowledge in Data and

Code Availability that your package currently only accepts that. It would likely take some time to create code that can reformat the outputs of another ancestry inference engine.

- Line 121: I don't think $5e-8$ is the same as $10^{-8.5}$. Or where does $10^{-8.5}$ come from? Is this the Bonferroni correction with the user-defined number of tests you did in your simulation study? If so, why even give ($\approx 10^{-8.5}$) in the sentence?
- Line 128: I would be cautious about the term "local ancestry", as the term is often used in admixture mapping studies with reference panels
- Line 131: "guaranteed to be independent". You should refer to your Materials and Methods or Appendix section when making such a claim.
- Line 164: define the Greek letter parameters as "fixed effects" w.r.t. [insert type of data].
- Line 200, 201: maybe say "separate LS HMM" and "This separation", or some other term, to reserve the term "independence" in a statistical sense.
- Line 207: is this a distance in the mathematical sense? That is, does it satisfy the triangle inequality, etc.?
- Line 227: remove "new". In general, I would remove terms "new", "novel", and "first".
- Line 299: your idea of "sprigs" feels similar to "twigs" in twigstats • twigstats . It may be worth distinguishing between them.
- Line 409: if you are doing a Bonferroni correction, shouldn't this definition reflect how discoveries are made "in humans"?
- Scalability Section: consider making a figure with increasing sample sizes, to demonstrate how k scales.
- Equation (15): I recommend against reusing the notation X
- Line 701: how exactly does your tail approximation make it heavier? Please provide some motivation; otherwise, the choice comes off as *ad hoc*.
- Lines 731 – 732: Please provide motivation for why this is a good choice.

Typos/writing:

- Citations:
 - Some of the journal names are not correctly capitalized [7,8,11,13,20,33,36,39,40].
 - [27] Capitalize Bayesian
 - [32] Is there a proceeding of the journal attached to this self-citation of the authors?
 - [35] Is this something that should be cited? It feels like a web resource.

- [38] Is this something that should be cited? It feels like a web resource.
- [39] Capitalize SNP
-
- Line 56: like [9]. Spell out who [9] is/are.
- Line 72: should 90% be spelled out as “Ninety percent”
- Line 75: “3 kb” not “3Kb”
- Lines 95, 96: remove “here”
- Line 111: “complementary”
- Line 123: remove “below”
- Line 130: remove “naturally”. Search for terms like “clearly” or “obviously” or “naturally” to remove.
- Line 167: can remove “Here and for the rest of the paper”
- Line 179: can remove “here”. Search the document for “here”. It isn’t necessary in many places.
- Line 187: the verb “overview” is an awkward choice
- Line 202: say the authors of [15]
- Line 207: say the authors. Please look for all instances where the authors are not parenthetically cited.
- Line 372: what is FFT? My guess is fast Fourier transform.
- Line 379: say Section. Please look closely at all instances where you cite or refer to a section.
- Line 407: remove “Please” throughout. Refer to a later section: “this is a sentence about something (Section 6)”.

References

- Browning, Sharon R., and Elizabeth A. Thompson. 2012. “Detecting Rare Variant Associations by Identity-by-Descent Mapping in Case-Control Studies.” *Genetics* 190 (4): 1521–31.
- Cai, Ruoyi, and Sharon Browning. 2025. “Identity-by-Descent Mapping Using Multi-Individual IBD with Genome-Wide Multiple Testing Adjustment.” *BioRxiv*org. <https://doi.org/10.1101/2025.01.28.635369>.
- Gunnarsson, Árni Freyr, Jiazheng Zhu, Brian C. Zhang, Zoi Tsangalidou, Alex Allmont, and Pier Francesco Palamara. 2024. “A Scalable Approach for Genome-Wide Inference of Ancestral Recombination Graphs.” *BioRxiv*. <https://doi.org/10.1101/2024.08.31.610248>.

- Zhang, Brian C., Arjun Biddanda, Árni Freyr Gunnarsson, Fergus Cooper, and Pier Francesco Palamara. 2023. "Biobank-Scale Inference of Ancestral Recombination Graphs Enables Genealogical Analysis of Complex Traits." *Nature Genetics* 55 (5): 768–76.

Author Replies to Reviewer Comments for Manuscript

GENETICS-2024-307747

The authors would like to sincerely thank the editor and reviewers for their very helpful comments and suggestions regarding our manuscript titled *Clade Distillation for Genome-wide Association Studies*. We believe we have fulfilled all changes requested by the the editor and reviewers and that the paper is substantially improved from those revisions. Please see our replies below.

For the reviewers' convenience, the line numbers at the end of each of our replies are links that jump to the corresponding edits in the main text. They may not appear highlighted as links after this pdf has been processed by the journal platform but they will hopefully still work if you try clicking on them. At the end of each section of edits in the main text, we have also included reverse links which appear as Reply, that should allow reviewers to easily jump back to the corresponding reply. Figure and Section numbers also link to their locations in the text.

Associate Editor Comments

Associate Editor Comment — As you can see below, the reviewers had a number of suggestions to both clarify the manuscript as well as make its key takeaways more salient. Please consider them carefully. In particular: currently, the paper is very heavy on many technical points, which can make it hard to extract the takeaways - consider carefully whether Reviewer 2's suggestions of additional comparisons or simulations (and, tying in the associated medRxiv paper) would help make the paper more readable (and hence impactful) for GENETICS.

Reply: Thank you for these helpful and constructive comments. In response we have made substantial revisions:

1. We moved an entire page of technical material describing our checkpointing algorithm from the Results section to the Methods and Supplement. See **E32** on Line 272 This comes alongside several smaller demotions of technical points, e.g. See **E33** on Line 449.
2. We explicitly relate this manuscript to the real data application of LOCATER in our medRxiv pre-print. See **E44** on Line 150
3. We have added careful comparisons to ARG-Needle with method-specific genome-wide discovery thresholds. Please see our response to Reviewer Comment 2.3 below.

4. We have added mathematical justification and intuition for our robust generalization of the Satterthwaite approximation, which we use to help us evaluate the quadratic form null distribution. Please see our response to Reviewer Comment 2.2 below.

Associate Editor Comment — "the expected genetic relatedness matrix (eGRM) as presented in Equation 1 of [17]": this is incorrect; equation 1 of Link et al defines the usual GRM, not the eGRM.

Reply: This has been corrected. See **E1** on Line 373

Reviewer 1

Reviewer Comment 1.1 — The correctness of the method hinges on the fact that the three P-values from the observed variant, inferred clade membership encoded in X , and the local genetic relatedness matrix are mutually independent. The authors suggest that SD guarantees this, but I was not able to follow the argument and felt the writeup should be clearer. Here is my understanding of how the procedure works: p_{SMT} is independent from the others because the other statistics are computed from the trait vector after projecting out the first two terms, A and $G^{(l)}$, in equation (1). This looks like a standard result from regression theory. However, it is unclear to me why p_{SD} and p_Q are independent. This is **not** a result of SD since, as far as I can tell, SD is only employed when combining p-values that correspond to the columns of $X^{(l)}$. Since both p_{SD} and p_Q use PY (trait after projecting out A and $G^{(l)}$), they should be dependent in general. For p_Q to be independent from p_{SD} , shouldn't you additionally project out $X^{(l)}$ from Y ? Please clarify the argument and make it more precise.

Reply: We should have made this point clearer. In short, SD provides us with an updated version of Y , referred to as $Y^{(L+1)}$, that is independent of p_{SD} . On page 12 of the SD pre-print [1], Christ et al. write

"As shown in Figure 6, after iterating EFR-SD L times, we can also obtain a final $Y^{(L+1)}$ that is independent of the collection $(U^{(l)})_{l=1}^L$. The by-product $Y^{(L+1)}$ may be used for further inference, though we will not specifically make use of it in this paper."

In other words, $Y^{(L+1)}$, which is used to calculate p_Q in LOCATER, is independent of the collection $(U^{(l)})_{l=1}^L$, which is used to calculate p_{SD} . We have introduced language clarifying this important point in the both Results See **E2** on Line 221 and Methods See **E3** on Line 412. Intuitively, the $Y^{(L+1)}$ returned by SD can be thought of as a version of Y where only the significant columns of $X^{(l)}$ and a small set of non-significant columns of $X^{(l)}$ have been projected out. Thus, $Y^{(L+1)}$ retains much more information about the original Y than if we were to attempt to project out all of the columns of $X^{(l)}$.

1st Revision - Authors' Response to Reviewers: April 25, 2025

Reviewer Comment 1.2 — How exactly are the branches assigned to $X^{(l)}$ and $\Omega^{(l)}$? I did not have any hints until I read line 516 where the authors write "tested more common clades than simply the sprigs with SD". If the authors used the term "sprig" in a similar fashion to how it's used commonly, is LOCATOR testing tip clades to $X^{(l)}$ and older branches to $\Omega^{(l)}$? If there were any heuristics to separate branches into $X^{(l)}$ and $\Omega^{(l)}$, please describe in the paper.

Reply: We've added new content to our Results to try to better explain how we define sprigs and the role of $X^{(\ell)}$ versus the role of $\Omega^{(\ell)}$. Please see See **E23** on Line 189. The reviewer is correct that, in essence, we aim to test tip clades $X^{(\ell)}$ and all remaining, more common clade structure via $\Omega^{(\ell)}$. We describe how we define/call sprigs in Section 2.2.3 and there we've added some content to try to clarify this section See **E35** on Line 349. As described there, Relate uses hierarchical clustering to build entire trees at each position and then identifies clades with strong support by hanging variants on the tree edges [2]. Following the argument in that Relate paper, in a world without recombination these clades should correspond to actual clades (i.e., if there is no noise in the ancestry inference). Our clique-calling approach to identifying sprigs can be thought of as a faster and more direct way of getting tip clades than following the full Relate approach. Here we do not want to make any strong claims that these neighborhoods are precise, unbiased genealogical tree calls; however they should be close to this, and they work well in practice.

Reviewer Comment 1.3 — Related, I think it's clearer to define $X^{(l)}$ as clade membership/assignment in line 159 than calling it inferred genotypes.

Reply: This has been corrected. See **E13** on Line 192

Reviewer Comment 1.4 — The authors should elaborate on how the null hypotheses in line 171-175 is related to the combining procedure described in section 4.8. From what is written in line 719-724, the combined hypothesis LOCATOR is trying to test is certainly not the intersection of the three hypothesis, which should have low power to test against as the authors pointed out.

Reply: We have added a brief statement explaining that we are testing the union rather than the intersection of three hypotheses when we introduce H_0^C . See **E14** on Line 211

Reviewer Comment 1.5 — The scope of the local test seems to be the span of the local tree that contains the observed SNP. If so, as adjacent trees will have a similar topology with the one that's being tested and they are omitted in the regression, this local test will inadvertently draw signals from those adjacent regions. I think this is why TRACTOR fails to localize the signal like SMT in a number of scenarios that authors have presented. Nevertheless, the scope of the test is not explicitly stated and only stated as "target loci". Please elaborate on how big the region captured by the proposed local test is.

Reply: Unfortunately, we do not believe it is possible to provide a more concrete definition of the “region” captured. We’ve added a small paragraph attempting to address this point in the Introduction. See **E36** on Line 142

LD complicates signal localization even when testing a single variant with SMT – the lead variant need not be causal. When we test more than one variant together, grouping them together either by some sliding window or placing them on a local genealogical tree, localization can become even more challenging because we don’t just have to consider pairwise LD: variants in adjacent regions that are in strong LD with combinations of our grouped variants may achieve a smaller overall p-value. In other words, SMT and tests that aggregate variants both can struggle with localization. Like SMT, the primary goal of LOCATER is to identify regions of the genome as associated, leaving localization to more specialized fine-mapping software downstream.

That said, as described in Section 2.6, overall our simulations demonstrate that LOCATER improves signal localization relative to SMT. We hypothesize that this is because, when there are many independent causal variants of weak effect in a causal region, it is unlikely that there will be a distant tagging variant that will capture their signal better than a test focused on the causal locus. We hope to further examine this hypothesis in future work.

Reviewer Comment 1.6 — When combining signals from the columns of $X^{(l)}$ with SD, the order of the columns being tested affects the result according to the SD paper (Christ, Hall, and Steinsaltz). Was there a default setting that is used for LOCATOR?

Reply: By default, LOCATER distills the columns of $X^{(\ell)}$ in the order in which they are provided: $X_{.1}^{(\ell)}, X_{.2}^{(\ell)}, \dots, X_{.p}^{(\ell)}$. Thus, if the user has some prior information that allows some columns to be prioritized over others, they may choose to place the high priority columns ahead of others.

Reviewer Comment 1.7 — Localization according to the distance between the mid-point and the lead variant looks like an important result presented in the paper. However, all the related figures are in the supplementary material. I guess that the choice was because a series of supplementary figures are in groups as they correspond to a particular experiment, so it is hard to separate just one figure and move it to the main writings. I suggest moving at least one of the experiments to the main section together with the figure that describes localization.

Reply: This is a good point and we considered different options for doing this, but felt that none of our summary plots could accurately represent the complex localization curves in the supplement in a condensed form, especially after the addition of comparisons to ARG-Needle. Thus, we have left the presentation as is, but are happy to reconsider this point if you or the editor have a suggestion.

Reviewer Comment 1.8 — The claim that the procedure can handle unmeasured confounding looks slippery. It’s not exactly correcting confounding but removing test statistics deviating from the $y=x$ line in

1st Revision - Authors' Response to Reviewers: April 25, 2025

the QQ-plot. Of course, such an inflation could have been a result of confounding, but it also could've been the extreme polygenicity (or similar genetic architecture) that is found in traits like height and BMI. In such a case, the large number of rejections would be a true signal and not false positives. The authors should phrase their procedure more carefully.

Reply: That's certainly true and we've provided that context in the relevant in both the Results See **E16** on Line 430 and Methods See **E17** on Line 898.

Reviewer 2

Reviewer Comment 2.1 — The authors have developed a revised version of their ancestry inference methodology (kalis) and implemented a series of tests to detect phenotype-genotype associations when there are multiple causal variants at the same locus. Their kalis v2 update includes numerous modifications that help the method scale up to enormous biobanks, where memory and runtime are concerns. Some of these improvements, like checkpointing or only rigorously testing the largest signals, are standard techniques and incremental. They show in simulations that kalis v2 can detect some associations with hidden variants that single marker tests cannot.

Reply: Thank you very much for the thorough and helpful comments. We would like to gently push back on two points here. First, while the overall idea of checkpointing and caching is standard, obtaining posterior decodings at L sequential loci presents a formidable computational challenge that would be completely intractable to classic checkpointing approaches. Our checkpointing scheduler achieves the lower bound on computational cost subject to a fixed memory constraint. As shown by Figure 5, in practice the checkpointing solutions obtained yield marked reductions to computational cost. This result can be used to obtain sequential decodings for any discrete HMM. We've tried to make this point more clear in the text (Section 2.2.2).

Second, it is important to note that LOCATER achieves power gains over SMT even when the causal variants are observed. As our power simulations show, this is a much harder problem than trying to improve on SMT when the causal variants are unobserved, which has been the focus of prior methods like ARG-Needle (Section 2.5).

Reviewer Comment 2.2 — The description of the methods and the experiments is mostly sufficient (see comments). In particular, if their tail approximation is to be a major statistical contribution, the authors should explain the intuition or mathematical justification for their approximation in more detail.

Reply: Thank you. The Reviewer is astute in noticing our contribution to quadratic form tail approximation. Our approach generalizes the Satterthwaite approximation, a common tool in the field, to indefinite matrices while providing robust tails. Although this is not something that we elected to focus on in our initial manuscript, it has a

1st Revision - Authors' Response to Reviewers: April 25, 2025

much firmer mathematical foundation than we originally presented. We have summarized this, alongside intuition for a more general audience, in the Methods section See **E31** on Line 867. The supplement now provides a much more detailed justification See **E18** on Line 1226.

Reviewer Comment 2.3 — If this paper is to stand on its own, without the real data analysis that the authors perform in a separate paper, I recommend further simulation studies including comparisons to other methods. I view the genealogy-based association tests in Threads or ARG-Needle as the most natural methods to compare to.

Reply: We have added comparisons to ARG-Needle [3] (with $\mu \in \{10^{-3}, 10^{-5}, 10^{-7}\}$) across all of our power simulations. Please see the updated text and figures in the Results section See **E11** on Line 460 and the supplement See **E12** on Line 1373. We used our power calculations to use an empirical genome-wide discovery threshold for ARG-Needle, LOCATER, and SMT. See **E45** on Line 484 We also added a new Section to the supplement describing how these thresholds were estimated. (Section 6.10).

In order to give ARG-Needle the best possible advantage in our power simulations, we gave ARG-Needle the true ARG. In other words, we skipped the ARG inference step of ARG-Needle and provided the testing module of ARG-Needle the true ARG generated by `msprime` that yielded the observed haplotypes. This provides an upper bound on the performance of ARG-Needle given the difficulty of ARG inference. As shown in Figure 2 LOCATER outperforms ARG-Needle in 9 out of 9 simulations involving observed causal variants (in some cases by a very large margin). In the case of hidden causal variants, which is the use case targeted by ARG-Needle, LOCATER outperforms ARG-Needle in 8 out of 9 simulations and the two methods are tied in 1 out of 9 simulations. Thus, even under conditions that benefit ARG-Needle, LOCATER is the superior method.

We also tried to compare to the quadratic-form-based testing procedure proposed by Gunnarsson et al. (2024) in their recent pre-print [4]. We could not find the relevant functions in the latest public `arg-needle` package. We wrote to Pier Palamara who confirmed that the methods implemented in the Gunnarsson et al. (2024) pre-print are not public yet. We asked that he notify us once that code is available. As of the writing of this response, the code was still not available online and he has not reached back out to us.

Reviewer Comment 2.4 — The authors allude to three parameters that could be used in their methodology to address confounding. This paper could be enhanced by showing how these parameters are important in simulations with population structure. The ability of their statistical test to address some confounding would be a more than incremental advance if demonstrated.

Reply: This is an excellent idea and one that we hope to explore in future work, but we believe it is outside the scope of our current manuscript. In our preliminary work we have found that adjusting for population structure is extremely complicated for genealogy-based methods, and we do not believe we can do the topic justice here.

1st Revision - Authors' Response to Reviewers: April 25, 2025

In our medRxiv pre-print applying LOCATER to a real-world GWAS dataset, we developed a more simple yet reasonably effective strategy to control for confounding [5], however, we believe that the ultimate solution will indeed entail statistically rigorous use of these three parameters.

Reviewer Comment 2.5 — The authors only show simulation results, as their real data analysis is in a separate medRxiv paper. Given this paper's detailed description of the methods, I respect this decision. At the same time, I recommend some discussion of the medRxiv paper in the Introduction so as to motivate this work.

Reply: As mentioned in our response to the Associate Editor above, we have added a paragraph at the end of the Introduction providing this connection and motivating this work. See **E44** on Line 150

Reviewer Comment 2.6 — All Figures: the authors should explicitly say what “total association signal strength is” in the captions. For someone skim reading (abstract, Figures, conclusion), it is difficult to understand the x-axis label context.

Reply: Corrected. See **E19** on Line 476 See **E20** on Line 496 See **E21** on Line 522 See **E22** on Line 537

Reviewer Comment 2.7 — Line 748 – 752: I did not click the link to the private Drive folder because I was concerned about disclosing my reviewer identity. I do not see why the software cannot be shared, given that the authors have already written a pre-print.

Reply: The software alongside installation instructions are now available on GitHub <https://github.com/ryanchrist/locater>. The package ships with an introductory vignette which can be found under the Articles tab on the package website: <https://ryanchrist.github.io/locater/>. The updates in kalis v2 can be found on the GitHub page: <https://github.com/louisaslett/kalis/>. Please see our updated Data and Code Availability section 4.8 for more details.

Reviewer Comment 2.8 — Lines 49 – 54: These sentences do not negate that the GWAS methodology in ARG- Needle and Threads is a direct competitor to your method. Given that your results come exclusively from a simulation study and that you are presenting a new method, I believe ARG-Needle (Zhang et al. 2023; Gunnarsson et al. 2024), or something equivalent, ought to be compared/benchmarked to. Another clear competitor, w.r.t. detecting a hidden signal of multiple causal variants, is IBD mapping (Cai and Browning 2025; Browning and Thompson 2012), which is not discussed nor compared to.

Reply: As discussed in our reply to Reviewer Comment 2.3, we have added extensive comparisons to ARG-Needle. We have also added discussion of Browning and Thompson 2012 See **E37** on Line 38 and Cai and Browning 2025 See **E38** on Line 66.

1st Revision - Authors' Response to Reviewers: April 25, 2025

Reviewer Comment 2.9 — Line 159: what is the variable p in X (n by p) matrix? How can an edge be encoded as 0, 1, 2? Please clarify. Are these the clades detected as “sprigs”, which is defined many pages after this sentence?

Reply: We apologize for the confusion. We've tried to clarify these points which Reviewer 1 also raised. Please see our response to Reviewer Comment 1.2 and Reviewer Comment 1.3 above for more details and links to the corresponding edits.

Reviewer Comment 2.10 — Figures 2 and 3: I would replace “intermediate” with the exact allele frequency bin. I would combine these figures into one figure.

Reply: We have added information about the allele frequency bins to Figures 2 and 3. See **E39** on Line 496 See **E40** on Line 522 We appreciate the suggestion on combining these figures but we prefer to keep them separate given the amount of information being presented, especially with the addition of ARG-Needle results.

Reviewer Comment 2.11 — Line 364: Please provide an interpretation of the parameters.

Reply: Corrected. See **E25** on Line 437 See **E26** on Line 1315

Reviewer Comment 2.12 — Line 381 – 383: should this be $a_1 * a_2$? I imagine the authors want to consider +- quantitative phenotypes, not mean a_1 plus 1 or -1.

Reply: We purposefully used $a_1 + a_2$ here. The goal is to show that implementation is numerically robust to the inclusion of both continuous and binary background covariates. That is why we sample background covariates, $a_1 \sim N(0, I)$ and each component of a_2 is an independent Rademacher random variable. This idea appears in various association test simulation studies, such as the one performed in the STAAR paper [6]. We've added [6] as a citation for the idea and moved this information to the supplement in order to comply with the editor's request that more technical details be relegated to the Methods. See **E27** on Line 655

Reviewer Comment 2.13 — Many of the Supplementary Figures could/should be combined into panel figures: (6-9), combining Figures based on 3, 9, and 15 causal variants and observed versus hidden.

Reply: We respectfully disagree and would prefer to keep the current format.

Reviewer Comment 2.14 — Line 46: what is SMT? The acronym is not defined prior. My guess is single marker test.

Reply: Corrected. See **E4** on Line 50

1st Revision - Authors' Response to Reviewers: April 25, 2025

Reviewer Comment 2.15 — Line 73: spell out the acronym TSS. My guess is transcription start site.
Line 73: spell out kilobase pair (kb)

Reply: Corrected. See **E5** on Line 81

Reviewer Comment 2.16 — Line 82: consider spelling out “Aggregated Cauchy Association Test”. This one is less essential an acronym though.

Reply: Corrected. See **E6** on Line 91

Reviewer Comment 2.17 — Line 89: do you have a citation to back up this claim?

Reply: We added a citation pointing readers back to discussion of the Cauchy Combination Test and Generalized Higher Criticism in the original SD pre-print. See **E7** on Line 98

Reviewer Comment 2.18 — Line 97: your binary traits plan sounds like a discussion point

Reply: It is, but we felt it was important to leave this clause in the introduction as a flag to readers who may be questioning whether they could eventually apply this approach to binary traits.

Reviewer Comment 2.19 — If the authors care to show some simulation results with SINGER tree inference, it could improve claims. Not essential. It could require substantial work.

Reply: This is a good idea but will indeed require substantial work. We plan to deploy LOCATER in combination with other ancestry inference methods in future work.

Reviewer Comment 2.20 — Lines 108 – 110: while your math may apply to “any ancestry engine”, unless you have written code to accept any ancestry engine, I would acknowledge in Data and Code Availability that your package currently only accepts that. It would likely take some time to create code that can reformat the outputs of another ancestry inference engine.

Reply: We have added language contextualizing this claim in the main text. See **E41** on Line 124 and the Data and Code Availability section. See **E42** on Line 931. As explained there, our R package exposes all of our testing sub-routines to the user. The main impediment to a user deploying `locator` in conjunction with any ancestry inference engine is simply porting the matrix of clade calls $X^{(\ell)}$ and/or local relatedness matrix $\Omega^{(\ell)}$ returned by that engine into R as base or sparse matrix objects. Once this is done, users do not need to re-implement any of our math to run our testing procedures. Given the broad popularity of R and these matrix objects, we believe this provides sufficient generality to justify our “any ancestry engine” claim. While we aim to provide command line utilities and explicit functions in Python that interface directly between our LOCATER testing routines and popular

1st Revision - Authors' Response to Reviewers: April 25, 2025

ancestry inference engines like `tsinfer`, this approach can yield bridge interfaces since many of these engines are still in active development. From a software development and maintenance standpoint, our API approach is more robust to changes in existing engines and the emergence of new engines.

Reviewer Comment 2.21 — Line 121: I don't think $5e - 8$ is the same as $10^{-8.5}$. Or where does $10^{-8.5}$ come from? Is this the Bonferroni correction with the user-defined number of tests you did in your simulation study? If so, why even give ($\approx 10^{-8.5}$) in the sentence?

Reply: Thank you, we have removed this statement and introduced a few sentences to clarify our discussion of genome-wide discovery thresholds in the introduction. See **E43** on Line 136 and the discussion. See **E46** on Line 609. We also updated our power calculations to use empirically estimated genome-wide discovery thresholds for each method – please see our reply to Reviewer Comment 2.3.

Reviewer Comment 2.22 — Line 128: I would be cautious about the term “local ancestry”, as the term is often used in admixture mapping studies with reference panels

Reply: The reviewer is correct that the term "local ancestry" is often used in a more general way than we use it here. Although we still feel that the term is appropriate, we have gone through the text and tried to make sure that the context is clear. We prefer "local ancestry" because the terms "local genealogy" or "haplotype tree" or "marginal tree" all refer to a discrete tree structure, however, we would like to use a term that refers to this discrete tree structure or other possible representations of it, e.g., a local relatedness matrix.

Reviewer Comment 2.23 — Line 131: “guaranteed to be independent”. You should refer to your Materials and Methods or Appendix section when making such a claim.

Reply: This has been corrected. See **E8** on Line 161

Reviewer Comment 2.24 — Line 164: define the Greek letter parameters as “fixed effects” w.r.t. [insert type of data].

Reply: This has been corrected. See **E9** on Line 207

Reviewer Comment 2.25 — Line 200, 201: maybe say “separate LS HMM” and “This separation”, or some other term, to reserve the term “independence” in a statistical sense.

Reply: This has been corrected. See **E10** on Line 245

Reviewer Comment 2.26 — Line 207: is this a distance in the mathematical sense? That is, does it satisfy the triangle inequality, etc.?

1st Revision - Authors' Response to Reviewers: April 25, 2025

Reply: The pairwise distances between haplotypes we obtain under the LS model do not correspond to a metric over haplotypes. For instance, there is no guarantee of a triangle inequality. However, our method does not require these distances to be a metric.

Reviewer Comment 2.27 — Line 227: remove “new”. In general, I would remove terms “new”, “novel”, and “first”.

Reply: We've gone through the text and minimized our use of these terms.

Reviewer Comment 2.28 — Line 299: your idea of “sprigs” feels similar to “twigs” in twigstats • twigstats . It may be worth distinguishing between them.

Reply: We use “sprig” to describe very small clades in a local phylogeny that can be confidently called using an ancestry inference engine. When applied to a large GWAS dataset, we expect these sprigs to correspond to individuals who share a long haplotype at a given locus, which may correspond to common ancestor within a handful of generations. In contrast, the concept of “twigs” in twigstats refers to coalescence events that occurred around the same time as a demographic event of interest. While these coalescence events might be relatively recent – for example, in their paper, Speidel et al. [7] use twigs to date recent admixture events – demography-informative twigs do not necessarily correspond to recent events. Furthermore, by averaging over the genome, the twigs in twigstats do not need to correspond confidently called clades, whereas that's a requirement for our “sprigs”. Given the significant differences noted above, we feel that it would be confusing to explicitly introduce “twigs” in our manuscript.

Reviewer Comment 2.29 — Line 409: if you are doing a Bonferroni correction, shouldn't this definition reflect how discoveries are made “in humans”?

Reply: This has been corrected. We updated our power calculations to use empirically estimated genome-wide discovery thresholds for each method – please see our replies to Reivewer Comment 2.3 and Reivewer Comment 2.21.

Reviewer Comment 2.30 — Scalability Section: consider making a figure with increasing sample sizes, to demonstrate how kalis v2 scales.

Reply: The scalability of ancestry inference using kalis v1 is documented in detail in the original kalis paper [8]. As described there, kalis v1 obtains the expected $\mathcal{O}(N^2L)$ cost in compute and $\mathcal{O}(N^2)$ cost in memory. As described in Section 4.3, due to the efficiency of our checkpointing approach, as long as at least a few checkpoints are used, in practice we achieve the same scalability. For example, sequentially iterating the forward algorithm to $L = 10^5$ target loci from locus $L \rightarrow L - 1 \rightarrow \dots \rightarrow 2 \rightarrow 1$ using $C = 8$ checkpoints requires less than 10

1st Revision - Authors' Response to Reviewers: April 25, 2025

times the computational cost required to directly iterate the forward algorithm across those loci in a single pass $1 \rightarrow 2 \rightarrow \dots \rightarrow L - 1 \rightarrow L$. Measuring the scalability of trait association with LOCATER is a complex and time consuming task that depends on the number of variants in a given simulation that are associated with a trait of interest. Given that we reference it's application to a large real-world dataset [5], we feel the current discussion is adequate. See **E49** on Line 592

Reviewer Comment 2.31 — Equation (15): I recommend against reusing the notation X

Reply: This has been corrected. Please see reply to Reviewer Comment 2.2.

Reviewer Comment 2.32 — Line 701: how exactly does your tail approximation make it heavier? Please provide some motivation; otherwise, the choice comes off as ad hoc.

Reply: This has been corrected. Please see reply to Reviewer Comment 2.2.

Reviewer Comment 2.33 — Lines 731 – 732: Please provide motivation for why this is a good choice.

Reply: This has been corrected. See **E30** on Line 924

Reviewer Comment 2.34 — Typos/writing:

Reply: Thank you very much for noting these errors. We've resolved them. From among these points, note that we've corrected the citation for the Rényi Outlier Test [9].

Clade Distillation for Genome-wide Association Studies

Ryan Christ ^{1*}, Xinxin Wang ^{1,2}, Louis J.M. Aslett ³, David Steinsaltz ⁴, Ira Hall ^{1*},

1 Department of Genetics, Yale University School of Medicine, New Haven, CT, USA

2 Department of Genetics, Washington University School of Medicine, Saint Louis, MO, USA

3 Department of Mathematical Sciences, Durham University, Durham, UK

4 Department of Statistics, University of Oxford, Oxford, UK

✉ Current Address: Department of Genetics, Yale University School of Medicine, New Haven, CT, USA

* ryanchrist@yale.edu

Abstract

Testing inferred haplotype genealogies for association with phenotypes has been a longstanding goal in human genetics given their potential to detect association signals ~~caused~~driven by allelic heterogeneity — when multiple causal variants modulate a phenotype — in both coding and noncoding regions. Recent scalable methods for inferring locus-specific genealogical trees along the genome, or representations thereof, have made substantial progress towards this goal; however, the problem of testing these trees for association with phenotypes has remained unsolved due to the growth in the number of clades with increasing sample size. To address this issue, we introduce several practical improvements to the kalis ancestry inference engine, including a general optimal checkpointing algorithm for decoding hidden Markov models, thereby enabling efficient genome-wide analyses. We then propose ‘LOCATER’, a powerful new procedure based on the recently proposed Stable Distillation framework, to test local tree representations for trait association. Although LOCATER is demonstrated here in conjunction with kalis, it may be used for testing output from any ancestry inference engine, regardless of whether such engines return discrete tree structures, relatedness matrices, or some combination of the two at each locus. Using simulated quantitative phenotypes, our results indicate that LOCATER achieves substantial power gains over traditional single marker testing, ARG-Needle, and window-based testing in cases of allelic heterogeneity, while also improving causal region localization~~relative to single marker tests~~. These findings suggest that genealogy-based association testing will be a fruitful approach for gene discovery, especially for signals driven by multiple ultra-rare variants.

1 Introduction

Recent heritability estimates predict that rare variants in regions with low linkage disequilibrium account for a substantial fraction of the unexplained (missing) heritability of common traits and diseases [10]. Since the statistical power to detect effects driven by rare variants is inherently limited by their low frequency, methods for identifying rare variant associations leverage allelic heterogeneity: the presence of multiple independent causal mutations affecting the trait of interest. These methods merge association signals from nearby rare variants under the premise that rare causal variants may be proximal to other causal variants ~~[11, 12]~~[11, 12].

Mounting evidence suggests that allelic heterogeneity is quite common for human traits ~~[13, 14]~~[13, 14]. Notably, a large-scale *in vitro* study following up on identified associations estimated that between 10% and 20% of expression quantitative trait loci (eQTLs) have multiple causal regulatory variants circulating in human populations [15]. Such results underscore the importance of these methods for defining new alleles and genes contributing to disease risk [16, 17, 10]. The opportunity for methods that can leverage allelic heterogeneity to identify overlooked associations will only increase with the size, population diversity, and

sequencing depth of emerging genomic datasets.

Early groundbreaking association methods designed to harness allelic heterogeneity focused on testing inferred locus-specific genealogies, which provide a natural way of collecting independent association signals driven by nearby variants and imputing any unobserved variants [?, 18, 19]. ~~This approach has [20, 18, 19].~~ These approaches have the added benefit of implicitly imputing unobserved variants, making it particularly advantageous for analyzing datasets with partial variant calling: SNP array data, low-coverage sequencing data, or data from understudied species. **E37**► The uncertainty and poor scalability of early local genealogy inference methods hamstrung the adoption of these early testing approaches. To partly address these challenges, Browning and Thompson proposed identity-by-descent mapping to test only very recent, locus-specific relationships, which could be rapidly and confidently inferred from the observed haplotypes [21]. While very elegant, all of these early methods suffer from ~~several~~ technical limitations related to ~~tree construction and~~ statistical testing, such as requiring resampling or permutation to generate p-values, and ultimately have not been used much. ◀**E37 Reply**

Recent advances in ancestry inference algorithms have made it possible to revisit genealogy-based trait association. Algorithms such as ARG-Needle [3], Relate [2], tsinfer [22], and kalis [8] have made it possible to perform local ancestry inference across the entire genome in modern datasets with hundreds of thousands or millions of samples. Given their accuracy in resolving recent genealogical relationships, inferred local ancestries are expected to be especially useful for detecting loci with multiple causal ultra-rare variants, **E4**► which are signals that standard ~~SMT~~ single marker testing (SMT) will struggle to identify ◀**E4 Reply**. This may be a particularly effective strategy for traits under strong purifying selection and cases where some of these ultra-rare causal variants correspond to complex hidden structural variations that are only observed in a given sequencing dataset via ultra-rare tagging variants that are far upstream or downstream. However, recent work aimed at applying these algorithms to improve disease mapping, most notably ARG-Needle [3], has focused on imputing hidden variants not explicitly observed in the original dataset and testing the inferred genotypes via ~~single marker testing (SMT)~~ SMT. Given the plummeting cost of high-coverage sequencing data and recent initiatives to improve structural variant detection and imputation [23], the number of missing variants is shrinking in modern datasets, limiting the gains available ~~by using local ancestries to infer hidden variation from testing hidden variation via local ancestries.~~

Link et al. ~~[24] is a notable exception, showing a~~ evidenced a renewed interest in using genealogies to map genes by leveraging allelic heterogeneity [24]. Building on earlier efforts like [?], ~~this that of Zöllner & Pritchard [20], their~~ approach targets loci with allelic heterogeneity by using local ancestry inference methods to build a local genetic relatedness matrix for pre-specified windows along the genome or gene regions. These matrices are then tested for association with the phenotype of interest using a quadratic form test statistic. Using effectively the same default test statistic, very recent work by Zhu et al. [25] and

Gunnarsson et al. [4] provides a much more scalable implementation of this approach. **E38**► Building on the idea of identity-by-descent mapping, Cai and Browning [26] very recently proposed a distinct scalable approach that also relies on a quadratic form test statistic. ◀**E38 Reply** All of these methods mirror SKAT and more recent approaches in rare-variant gene-based testing that also aim to harness allelic heterogeneity to gain statistical power [11, 6][11, 6]. However, due to the inherent sensitivity of quadratic form test statistics to the presence of many non-causal (null) variants [1], these approaches struggle to maintain statistical power under the enormous multiple testing burden incurred when testing all of the clades in a local tree for trait association.

Existing rare variant association methods that aim to leverage allelic heterogeneity by testing collections of variants, such as STAAR, limit their multiple testing burden by using functional information, such as gene coding sequences, to define restricted sets of variants or more flexibly down-weight certain variants [11, 6]. This approach has been applied to many different sequencing based studies, and has proven to be a fruitful approach for identifying new gene-phenotype associations across a multitude of traits [27]. Despite their success in coding regions, it has proven difficult to extend rare variant association tests beyond coding regions where the majority of biologically critical signals are found. ~~90%~~ Ninety percent of GWAS hits for common diseases lie in non-coding regions, **E5**► at a median distance of 36 ~~kb~~ kilobase pair (kb) from the nearest ~~TSS~~ transcription start site [28, 29] ◀**E5 Reply**. It is unclear how one should define collections of variants in non-coding regions; sliding windows are the standard approach [6].

Outside of a gene's coding region, which in humans has a median length \approx ~~3Kb~~ 3 kb, there is a much larger regulatory region over which causal variants may be dispersed, complicating the use of sliding windows. Ideally, one would try to incorporate many variants over a genomic region in order to maximize the chance of aggregating signals from more than one causal variant. However, including too many non-causal (null) variants diminishes statistical power, and variant impact prediction, which could be used to narrow down variants to test in a given sliding window, remains extremely challenging in noncoding regions.

There are “sparse-signal” statistical methods, often deployed alongside quadratic forms, that aim to improve power in the presence of many null variants. **E6**► Notable examples include the ~~ACAT routine built into STAAR [6]~~ Cauchy Combination Test (CCT), which underlies the Aggregated Cauchy Association Test (ACAT) routine in STAAR [6], and Generalized Higher Criticism [30] ◀**E6 Reply**. However, these sparse-signal methods do not distinguish between a variant set where two highly-linked variants are observed to be associated with the phenotype and a variant set where two unlinked variants are observed to be associated with the phenotype. In the highly-linked case, one variant is essentially a proxy for the other and we have only one association signal. In the unlinked case, the signals coming from the two independent variants serve as independent pieces of evidence against the null hypothesis and should be combined. **E7**► In order to control the type-I error in the highly-linked case, the CCT underlying ACAT cannot combine signals

across variants with high efficiency, which yields a loss of power in the unlinked case. This simple two-variant argument extends to the case where we may be attempting to combine association signals across several variants. Section 1 of [1] provides further discussion of this point ◀E7 Reply.

We recently proposed a general statistical approach, Stable Distillation (SD), which can distinguish between the highly-linked and unlinked case [1]. There, in a gene-testing example using simulated data, we used SD to explicitly model the dependence structure between variants and achieved increased power over ACAT and related methods as a result. Building on SD, ~~here~~ we present a general framework, LOCATER, for trait association based on inferred local genealogies in both coding and non-coding regions. ~~Here we~~ We focus exclusively on testing quantitative traits, although we plan to extend LOCATER to binary traits in future work.

~~LOCATER is designed to work in conjunction with any ancestry inference engine of the user's choosing with an easy-to-use API available through our LOCATER package for the R language [31].~~ Modern ancestry inference methods typically represent local ancestries as discrete trees (perhaps with probabilistic weights on the edges), local relatedness matrices, or some combination of the two. Examples of discrete tree inference methods include `tsinfer`. These clades may have probabilistic weights, as provided by recent probabilistic ARG inference methods such as SINGER [32]. On the other hand, ancestry may also be represented in terms of local pairwise relatedness, typically summarized as a local relatedness matrix, as produced by `kalis`, `Relate`, and *Gamma-SMC* [33]. A set of observed haplotypes can typically be explained by an enormous number of underlying tree topologies, especially once we look beyond the recent past; pairwise methods account ~~naturally~~ for this topological uncertainty. As described in Section 2.1, LOCATER provides a framework for boosting SMT results with independent association signals based on local ancestry represented in either, or both, of these two forms produced by any ancestry inference engine. In order to highlight this feature, in this paper we apply LOCATER to ~~eomplimentary~~ complementary discrete clade and matrix-based representations of local ancestry obtained via the local ancestry inference engine `kalis`.

E41▶LOCATER is designed to work in conjunction with any ancestry inference engine of the user's choosing with an easy-to-use API available through our `locater` package for the R language [31]. Since our `locater` package exposes all of our testing subroutines as documented R functions, if a set of clade calls or a local relatedness matrix produced by some ancestry inference engine can be coerced into base or sparse matrices in R, then `locater` can be directly used to test those structures for association with a given phenotype. Please see the Data and Code Availability section for details. ◀E41 Reply

Although `kalis` does not scale as well as alternatives like `tsinfer`, a probabilistic model allows us to limit statistical testing to clades that have substantial evidence of existing at a locus of interest, thereby conserving statistical power. SINGER may provide a strong probabilistic alternative in future studies. The algorithmic improvements to `kalis` that we present in this paper, including an optimal checkpointing routine

for discrete-time hidden Markov models (HMMs) and linear-time clustering algorithm, may be useful for accelerating alternative models.

E43► Our focus on using genealogies to boost SMT signals rather than testing gene windows or sliding windows along the genome is another key point of departure of LOCATER from existing work. ~~While our approach may appear ill-advised at first, given that it requires LOCATER to clear the standard SMT Bonferroni threshold ($\approx 10^{-8.5}$), it~~ This approach leverages the statistical efficiency of SMT against sparse signals. ~~This approach~~ Testing the inferred genealogy at a locus also removes questions of window size and step length, ~~which is a natural advantage of using genealogies in the first place. As we will demonstrate below, by~~. ◀**E43 Reply** **E36**► By returning “genealogy-boosted” SMT signals, ~~LOCATER also aids we find~~ LOCATER generally improves the localization of causal variants ~~relative to SMT in the presence of allelic heterogeneity (Section 2.6). The precise variants aggregated at a given locus will depend on the structure of the local ancestral recombination graph (ARG) and the parameters of the ancestry inference engine used [34, 35, 2]. At a high level, older edges in a local genealogy tend to persist over much shorter stretches of the genome than recent edges due to recombination [34]. Accordingly, any procedure that tests a local genealogy for association with a phenotype will tend to aggregate association signals from rare variants over a wider region than common variants.~~ ◀**E36 Reply**

E44► In this paper we characterize LOCATER’s performance in simulated datasets. Dealing with the challenges of trait mapping in real datasets, such as rigorously adjusting for population structure and cryptic relatedness, is a difficult and open problem for genealogy-based trait mapping methods. We take on these problems in a subsequent work [5]. There we demonstrate the ability of LOCATER to substantially increase statistical power at loci with allelic heterogeneity and identify loci missed by SMT in a dataset of 6,795 Finnish genomes with extensive quantitative trait data. ◀**E44 Reply**

2 Results

LOCATER assumes that genome-wide SMT has already been performed. This is done simply to avoid the computational burden of inferring ancestries and running LOCATER at every locus. We focus on the subset of variants with putatively significant SMT results (e.g., $p < 10^{-4}$) and compute the local ancestry at each of those variants. At each target variant, LOCATER then takes the residuals from the SMT and tests any inferred discrete clade structure with Stable Distillation (SD) [1]. ~~SD naturally~~ **E8**► As discussed in Section 2.3, SD returns a new set of residuals which are guaranteed to be independent of the original SMT p-value and the p-value returned by SD under the null hypothesis. ◀**E8 Reply** We then pass this set of residuals to any quadratic-form based method that tests the pairwise-relatedness structure inferred at the variant of interest. The resulting three independent p-values may then be combined to obtain a potentially boosted signal at a

locus with allelic heterogeneity. This approach makes it straightforward to integrate LOCATER into the analysis of genome-wide association results. Of course, the resulting p-values must still be compared against a genome-wide multiple testing threshold as if LOCATER was run at every candidate variant. LOCATER can easily be applied in special cases where the inferred ancestry at a locus only comes in the form of discrete clades (eg: a tree) or pairwise-relatedness (eg: a local relatedness matrix). In both our SD procedure and quadratic form testing procedure, we have developed scalable methods to adjust for population structure and background covariates (see-Methods).

Below we introduce the LOCATER model. We then proceed to describe our methodological contributions in three parts: generating the ancestry representations required by LOCATER using the new routines we have introduced in an update to `kalis` [8], making the association testing procedures in LOCATER fast and robust to population structure, and efficiently combining the p-values returned by LOCATER. Finally, we demonstrate the calibration and power of LOCATER via simulation.

2.1 The LOCATER Model

Consider a genomic dataset with n participants phased in segments along the genome, each segment consisting of $N = 2n$ phased haplotypes along an entire or subsection of a chromosome with a total of V variants. Although we only address the diploid case in this paper, our approach may be readily extended to non-diploid organisms. Below we will consider testing each variant within a given segment for association with some quantitative phenotype of interest $Y \in \mathbb{R}^n$. When determining genome-wide significance thresholds, the total number of candidate variants across all genomic segments must be accounted for. However, in order to conserve computational resources, as depicted in Figure 1, we may only be interested in a subset of candidate variants within each segment based on preliminary SMT results or other genomic annotations. We call this set of target loci $\mathcal{L} \subseteq [V]$ and index them by their position along a given segment sequentially from $\ell = 1, \dots, L = |\mathcal{L}|$.

E23► Let $A \in \mathbb{R}^{n \times q}$ be a matrix of background covariates and $G^{(\ell)} \in \{0, 1, 2\}^n$ be the genotype vector observed at locus ℓ . Depending on the type of inference engine used to infer the local ancestry structure at locus ℓ , we may have ~~a set of inferred~~ inferred clade genotypes $X^{(\ell)} \in \{0, 1, 2\}^{n \times p}$ corresponding to edges in a tree inferred at ℓ , a local relatedness matrix $\Omega^{(\ell)} \in \mathbb{R}^{n \times n}$ inferred at ℓ , or both. ~~Here we~~ **E13**► In other words, for each of \$p\$ inferred clades at locus \$\ell\$, \$X_{ij}^{(\ell)}\$ is the number of haplotypes in sample \$i\$ that have been assigned to an inferred clade \$j\$ at locus \$\ell\$ ◀ **E13 Reply**. We tackle the general case assuming that our ancestry inference engine has returned both $X^{(\ell)}$ and $\Omega^{(\ell)}$, each capturing different parts of the ancestral structure at locus ℓ . Our approach can easily be applied to the special cases where only $X^{(\ell)}$ or $\Omega^{(\ell)}$ are available.

As further described in Section 2.2.4, by allowing genealogical relationships to be expressed in terms of pairwise similarity rather than explicitly called clades, \$\Omega^{(\ell)}\$ accommodates more uncertainty about the

precise topology of the underlying tree than $X^{(\ell)}$. However, this flexibility comes with a cost to power: as our simulations demonstrate, it is generally preferable to encode clades in $X^{(\ell)}$ rather than $\Omega^{(\ell)}$, at least when their membership is known with high confidence (end of Section 2.5). Due to recombination, more distant genealogical relationships, corresponding to larger clades, at a given locus ℓ are more difficult to accurately estimate than more recent genealogical relationships, corresponding to small clades in a local genealogy. Thus, in this paper, we will demonstrate LOCATER by encoding small clades (typically each with at most 10 haplotypes under them), which we will refer to as “sprigs,” in $X^{(\ell)}$ and encoding larger clades in $\Omega^{(\ell)}$. Section 2.2.3 further details how we call sprigs. ◀E23 Reply

E9► For a set of ~~parameters~~ fixed effects $\alpha \in \mathbb{R}^q$, $\gamma \in \mathbb{R}$, and $\beta \in \mathbb{R}^p$, and a variance component parameter $\tau \in \mathbb{R}_{\geq 0}$, LOCATER assumes the following model for a quantitative phenotype vector $Y \in \mathbb{R}^n$. ◀E9 Reply

$$Y = A\alpha + G^{(\ell)}\gamma + X^{(\ell)}\beta + \epsilon \quad \text{where} \quad \epsilon \sim N\left(0, \exp\left(\tau\Omega^{(\ell)}\right)\right) \quad (1)$$

Here, \exp denotes the matrix exponential. Under this model we test whether genetic variation at locus ℓ affects phenotype Y by testing the null hypothesis $H_0^C : \{\gamma = 0, \beta = 0, \tau = 0\}$. ~~Here and for the rest of the paper, we will~~ **E14**► To be clear, H_0^C represents the union, not the intersection, of these statements about the parameters. In other words, in what follows, we will reject H_0^C if there is evidence that $\gamma \neq 0$, $\beta \neq 0$, or $\tau \neq 0$. ◀E14 Reply We will assume that Y has been obtained using the rank-matching procedure described in Section 6.11. This normalization ensures that the residuals of Y have unit variance under H_0^C , justifying the absence of a variance scale parameter (typically denoted as σ^2) in Equation (1).

LOCATER tests H_0^C by decomposing it into three sub-hypotheses. First, we use the standard SMT to test $H_0^{\text{SMT}} : \{\gamma = 0 | \beta = 0, \tau = 0\}$, yielding a p-value p_{SMT} . Then we test whether any of the locally inferred clades predict the phenotype $H_0^{\text{SD}} : \{\beta = 0 | \tau = 0\}$ using SD, yielding a p-value p_{SD} . Finally, we test whether any remaining local ancestry structure encoded in $\Omega^{(\ell)}$ affects the phenotype by testing $H_0^Q : \{\tau = 0\}$ with a quadratic form test statistic, yielding a p-value p_Q . We further describe the explicit routines LOCATER uses to test H_0^{SD} and H_0^Q in Section 2.3. ~~When~~ **E2**► As further explained in Section 2.3, when these three sub-hypotheses are tested in this order, the independence guarantees of SD ensure that the resulting p-values ($p_{\text{SMT}}, p_{\text{SD}}, p_Q$) are mutually independent under the null hypothesis H_0^C . ◀E2 Reply Thus, after running LOCATER, the user may combine these three p-values using any valid method for aggregating independent p-values. ~~Here we~~ We propose a variant of Fisher’s method that we call Maximizing over Subsets of Summed Exponentials (MSSE) which yields more power in this setting (see Section 4.8). Figure 1 overviews the role of LOCATER in the context of an ancestry-based association testing pipeline. Next we delineate how the algorithms we have implemented in `kalis v2` allow us to rapidly obtain $X^{(\ell)}$ and $\Omega^{(\ell)}$ across target loci in our present study.

Figure 1: **The LOCATER Pipeline.** We begin ancestry-based association testing with a set of putatively interesting target loci, typically identified via single marker testing, indexed $\{1, \dots, L\}$. At each target locus ℓ , we extract the genotype vector $G^{(\ell)} \in \{0, 1, 2\}^n$ and use an ancestry inference engine to infer local clade genotypes $X^{(\ell)} \in \{0, 1, 2\}^{n \times p}$ and/or a local relatedness matrix $\Omega^{(\ell)} \in \mathbb{R}^{n \times n}$. We then use LOCATER to calculate three p-values testing whether $G^{(\ell)}$, $X^{(\ell)}$, or $\Omega^{(\ell)}$ predict the phenotype respectively. These three p-values are guaranteed to be independent under the null hypothesis, so they may be easily combined with many methods, in this paper we propose and use MSSE 4.8, to obtain a combined ancestry-association p-value $p_C^{(\ell)}$ at each target locus.

2.2 Algorithmic Advances in kalis v2

230

2.2.1 Local Genealogy Inference with kalis

231

`kalis` [8] provides a high-performance implementation of various versions of the Li & Stephens (LS) haplotype copying model which have become ubiquitous in modern genomic analysis [35, 36]. ~~Here we overview~~ We outline our novel algorithmic contributions to a new release, `kalis v2`, which together allow us to efficiently calculate $X^{(\ell)}$ and $\Omega^{(\ell)}$ sequentially at a given set of target loci $\ell = 1, \dots, L$ so that they can be tested downstream using LOCATER. In order to explain these contributions, we begin with a brief overview of local ancestry inference using `kalis`.

232
233
234
235
236
237

As with all ancestry inference engines, the ancestry at a given target locus ℓ is learned based on the observed genomic variation upstream and downstream of ℓ . Since the LS model is a special case of an HMM, ancestry information provided by variants upstream of ℓ can be summarized by the forward probabilities at ℓ ; and variants downstream by the backward probabilities at ℓ [37]. Given a set of N haplotypes and a single target locus ℓ , `kalis` implements the forward algorithm to iterate over variants upstream of ℓ , starting at the left end of the genomic segment, to obtain a matrix of forward probabilities $f^{(\ell)} \in \mathbb{R}^{N \times N}$ at ℓ . Similarly, `kalis` implements the backward algorithm to iterate over variants downstream of ℓ , starting at the right end of the genomic segment, to obtain a matrix of backward probabilities $b^{(\ell)} \in \mathbb{R}^{N \times N}$ at ℓ . **E10**► Each column $f_{.j}^{(\ell)}$ and column $b_{.j}^{(\ell)}$ corresponds to ~~an independent a separate~~ LS HMM where we model recipient haplotype

238
239
240
241
242
243
244
245
246

j as a mosaic of the other $N - 1$ haplotypes in the sample. This ~~independence separation~~ allows `kalis` to compute the columns of $f^{(\ell)}$ and $b^{(\ell)}$ in parallel and exploit modern compute architectures. ~~See E10 Reply.~~ See Aslett & Christ [8] for further details. The product $f_{ij}^{(\ell)} b_{ij}^{(\ell)}$ can be interpreted as proportional to the probability that recipient haplotype j “copies” from donor haplotype i at locus ℓ under the LS model. By definition, $f_{ii}^{(\ell)} = b_{ii}^{(\ell)} = 0$ for all haplotypes i . `kalis` makes $f^{(\ell)}$ and $b^{(\ell)}$ easily and rapidly accessible in the R language [31] for downstream computation, with all time-critical code written in high performance C.

Along the lines of ~~[2] and Speidel et al. [2] and Aslett & Christ [8]~~, we define the distance from haplotype j to haplotype i as

$$d_{ij}^{(\ell)} = -\log \left(\max \left(\frac{f_{ij}^{(\ell)} b_{ij}^{(\ell)}}{\sum_{k=1}^N f_{kj}^{(\ell)} b_{kj}^{(\ell)}}, v \right) \right) \quad (2)$$

where $v \approx 4.94 \times 10^{-324}$ to guard against underflow to zero with double precision floating point arithmetic. For efficiency the distance matrix $d^{(\ell)} = (d_{ij}^{(\ell)}) \in \mathbb{R}_{\geq 0}^{N \times N}$ is never explicitly constructed, but it is implicitly used to construct $X^{(\ell)}$ and $\Omega^{(\ell)}$ for testing with LOCATER, as further delineated below.

Throughout this paper, we use `kalis` to run the modified LS model used in `Relate` [2]. See Section 6.1 for further details. This modified model leverages ancestral allele information to improve local genealogy inference. In this paper we only simulate phased genomic datasets where the ancestral allele of each variant is known; this is a feature of our chosen ancestry inference engine and not a general requirement of LOCATER. Under this modified LS model, Speidel et al. showed that the distance $d_{ij}^{(\ell)}$ will be proportional to the number of proximal variants that differ between haplotype i and haplotype j in non-recombining segments and that the full matrix $d^{(\ell)}$ yields consistent local ancestry inference.

2.2.2 Optimal Checkpointing

Especially when processing many phenotypes in parallel, the number of target variants along a given genomic segment, L , may be very large. Since the amount of memory required to store a local relatedness matrix $\Omega^{(\ell)}$ at a given target locus scales $\mathcal{O}(n^2)$, storing these matrices at any appreciable number of variants quickly becomes untenable: even in the case where $n = 30,000$ samples (a scale we will consider in our simulations), 28.8 GB of memory is required to store a single $\Omega^{(\ell)}$. ~~In particular, offloading from memory may be impractical as there would be~~ Offloading from memory also incurs a considerable time cost ~~to from~~ writing and reading $\Omega^{(\ell)}$ to and from disk. **E32**► To avoid storing any $\Omega^{(\ell)}$, we will take a “test-it-and-forget-it” approach: we will obtain $X^{(\ell)}$ and $\Omega^{(\ell)}$ at one target variant at a time and test both $X^{(\ell)}$ and $\Omega^{(\ell)}$ with LOCATER before moving on to the next target variant in \mathcal{L} . This “test-it-and-forget-it” approach is only computationally tractable due to ~~a new, general~~ the checkpointing algorithm for discrete-time HMMs that we have introduced in `kalis` v2. ~~In short, our checkpointing approach allows us to infer local relatedness~~

matrices sequentially across consecutive target loci while minimizing the computation required subject to a fixed memory budget.

To understand the need for checkpointing, as mentioned above and further detailed below, recall that we implicitly need the pairwise distance matrix $d^{(\ell)}$ from Equation (2) returned by `kalis` in order to obtain $X^{(\ell)}$ and $\Omega^{(\ell)}$ at each target locus ℓ . While `kalis v1` can efficiently propagate the forward and backward recursions to obtain the forward probability matrix Checkpointing involves repeatedly updating a cache of forward matrices $f^{(\ell)}$ and backward probability matrix $b^{(\ell)}$ needed to calculate $d^{(\ell)}$ at a single locus ℓ , obtaining the pair $(f^{(\ell)}, b^{(\ell)})$ at sequential positions ℓ is challenging for HMMs due to the uni-directionality that are used to seed subsequent iterations of the forward and backward recursions. The compute-minimizing approach would involve running a single pass of the forward algorithm—iterating the forward recursion to target locus 1, then locus 2, and so on until locus L —and a single pass of the backward recursion from target locus L to target locus 1 while storing $f^{(\ell)}$ and $b^{(\ell)}$ at every $\ell = 1, \dots, L$. Since each $f^{(\ell)}$ and $b^{(\ell)}$ consumes $8N^2$ bytes of memory (e.g.: 80 GB for $n = N/2 = 50,000$ haplotypes), this approach requires far too much storage for most genomic datasets. On the other hand, we have the memory-minimizing approach, where we restart the forward and backward recursions from the respective ends of the genomic segment for every target locus. While this approach only requires storing a single $f^{(\ell)}$ and a single $b^{(\ell)}$ at any given time, it demands far too much compute time for most genomic datasets, requiring $\mathcal{O}(L^2N^2)$ floating point operations (FLOPs)—a prohibitive cost. An attempt to rescue this approach by splitting the genome into smaller segments (running in chunks) would still require $\mathcal{O}(L^2N^2)$ compute time.

We provide a checkpointing algorithm that finds an optimal balance in this memory-compute trade-off, minimizing the compute time required given algorithm. Each stored forward matrix is a “checkpoint,” and we assume that the user has a fixed memory budget. The overall idea is to occasionally stop the forward recursion and store $f^{(\ell)}$ at its current position as a checkpoint (typically overwriting an old checkpoint) in order to avoid repeatedly restarting the forward recursion from the beginning of the genomic segment. Here we provide a broad overview of our checkpointing approach. We start with a user-specified memory budget capable of holding sufficient to store C checkpoints, each storing a $N \times N$ matrix of forward probabilities $f^{(\ell)}$. We run the backward recursion once across the entire chromosome or genomic segment, stopping at each consecutive target locus sequentially from the target locus with the largest position ($\ell = L$) to the target locus with the smallest position ($\ell = 1$). When the backward recursion stops at a given target locus, we run the forward recursion from the nearest checkpoint to meet the backward recursion and so obtain $X^{(\ell)}$ and $\Omega^{(\ell)}$ at that target locus. Note it is natural for us to perform this backwards along the genomic segment, since there is a slightly higher computational cost for the backward recursion and hence we favor repetitive restarts of the forward recursion.

Iterating from locus L down to locus 1 makes minimizing the compute required for the backward recursion trivial: we simply visit each locus sequentially in a single pass. The challenge is determining Our checkpointing algorithm schedules where and when to overwrite existing checkpoints to minimize the total distance (number of variants) that each of the C checkpoints in order minimize the computational cost required to sequentially propagate the forward algorithm needs to iterate over in order to provide forward matrices in reverse order $f^{(L)} \rightarrow f^{(L-1)} \rightarrow \dots$. In Section 4.3 we show how to solve for a schedule of checkpoints that achieves this minimum for any discrete time HMM, given storage for a fixed number of checkpoints C . We call this solution the “optimal checkpointing schedule.” After a forward matrix $f^{(\ell)}$ is obtained at a given target locus ℓ , this schedule instructs `kalis` which checkpoint to use to restart the forward recursion to obtain the next forward matrix at locus $\ell - 1$, and where to lay down new checkpoints (if any) as the forward recursion proceeds to that locus. The checkpointing schedule also dictates where to initialize the C checkpoints as we iterate the forward recursion to the first target locus L . As shown in Supplementary Figure 5, the use of only 5 to 10 checkpoints already reduces the required number of FLOPs orders of magnitude closer to the best possible complexity, which would be $\mathcal{O}(N^2L)$ given substantial memory consecutive target loci $\ell = L, L - 1, L - 2, \dots, 1$. While maintaining a cache of checkpoints is far from a new idea in HMM inference – the idea is used in several implementations of the LS model [2] – our checkpointing algorithm achieves the lower bound on computational cost given memory for C checkpoints and can be applied to any discrete time HMM where sequential posterior decodings are required at consecutive times. Our approach yields massive reductions in computational cost compared to more naive checkpointing approaches (Section 4.3).

As detailed in Section 4.3, solving for the optimal checkpointing schedule can be computationally intensive for any given set of target loci. The version of the checkpointing schedule solver currently implemented in `LOCATER` assumes that target loci are evenly spaced. This simplification does not qualitatively change performance but allows us to solve for the optimal checkpointing strategy for a given L via a dynamic program, making the solution readily available. Our checkpointing implementation is available via the `ForwardIterator` function and associated helper functions now provided in `kalis v2`.

Of course, for datasets with a large number of samples, there may not be sufficient capacity to store many checkpoints in memory. At a minimum, running `kalis` on n samples ($2n$ phased haplotypes) to obtain each $\Omega^{(\ell)}$ requires $32n^2$ bytes to store the forward and backward probabilities and another $8n^2$ bytes to store $\Omega^{(\ell)}$. Storing each additional checkpoint of forward probabilities requires $16n^2$ bytes. Given the nested nature of our checkpointing algorithm, most checkpoints can be stored on disk rather than memory, which comes at minimal computational cost as long as one or two of the checkpoints (the ones that are closest to the current target loci) are always kept in memory. We plan to add native support for storing file-backed checkpoints to `kalis` in the near future. Looking further ahead, `kalis` can already be distributed across machines,

~~each running the LS model on a different subset of recipient haplotypes [8], but running LOCATER across distributed machines would require substantial network communication. Reducing this communication is a direction of future work.~~

◀E32 Reply

2.2.3 Calculating Inferred Clade Genotypes from the LS Model

E35► The LOCATER model (Equation (1)) admits a matrix of genotypes $X^{(\ell)}$ encoding any clades (marginal tree edges) inferred at locus ℓ . In principle, given the distance matrix $d^{(\ell)}$ obtained via `kalis` at locus ℓ (Equation (2)), any number of clustering algorithms could be used to infer a marginal tree topology from $d^{(\ell)}$. For example, `Relate` clusters a normalized version of $d^{(\ell)}$ with average linkage (UPGMA) [2]. From the resulting tree topology, one could then encode each of the inferred clades (or some subset of them) via $X^{(\ell)}$. While this is a promising approach for future work, in this paper, we only stored clade genotypes corresponding to very small inferred clades (each typically including 2 to 10 haplotypes) in $X^{(\ell)}$ and encode all larger-scale relatedness structure in $\Omega^{(\ell)}$. Focusing on just these rare clades, which we will refer to as “sprigs,” rather than all of the clades in the tree, allows us to showcase the flexibility of LOCATER — the ability of LOCATER to incorporate hard-called clades via $X^{(\ell)}$ and remaining relatedness structure via $\Omega^{(\ell)}$.

◀E35 Reply

We identify sprigs at a given locus ℓ based on the neighborhood — i.e., the set of tied nearest-neighbors — of each haplotype j :

$$\eta_j^\ell = \left\{ i \in [N] : d_{ij}^{(\ell)} \leq \min_{i \neq j} d_{ij}^{(\ell)} \right\}. \quad (3)$$

Note that, by this definition, haplotype j is always a nearest neighbor of itself. In practice we obtain these neighborhoods as a by-product of the clustering algorithm we use to construct $\Omega^{(\ell)}$ (see Section 2.2.4). We implicitly use the collection of neighborhoods $\{\eta_j\}_{j=1}^N$ to construct an undirected graph where each haplotype is a vertex, and edges connect haplotypes that agree on being in each other’s neighborhood. We use a greedy clique-finding procedure over the nearest neighborhoods to rapidly identify maximal cliques within this implicit graph. Haplotypes within each clique are assumed to belong to the same sprig, yielding sprig genotypes that we encode in $X^{(\ell)}$.

2.2.4 ~~Calculating Relatedness Matrices from the LS Model~~

Having encoded the locally inferred sprig genotypes in $X^{(\ell)}$, we summarize all of the remaining genealogical structure in $d^{(\ell)}$ via $\Omega^{(\ell)}$.

2.2.4 Calculating Relatedness Matrices from the LS Model

E1► While LOCATER can accept any real symmetric matrix $\Omega^{(\ell)}$, in order to optimize power we model our choice of $\Omega^{(\ell)}$ in this paper after the expected genetic relatedness matrix (eGRM) ~~as presented in Equation 1 of [24]~~ proposed by Fan, Mancuso, and Chiang [38]. We cannot directly use ~~the their~~ definition of the eGRM ~~given in [24]~~ because the construction there requires a set of discrete clade calls. **◀E1 Reply** Constructing a local relatedness matrix from the distances $d^{(\ell)}$ is more complicated because, as described in Section 2.2.1, each column is calculated using an independent LS HMM. Thus, different columns of $d^{(\ell)}$ may disagree on the exact boundaries of particular clades in the underlying genealogy. This is a general feature of ancestry inference methods that work in a parallel or pairwise fashion across haplotypes. Rather than overriding the LS model and using hierarchical clustering or some other approach to try to align these clade calls, ~~here~~ we generalize the eGRM to allow for this asynchrony. Our generalization, presented in Section 4.4, expresses an eGRM in terms of asymmetric distances like those provided by $d^{(\ell)}$ while ~~naturally~~ allowing for such unaligned probabilistic clade calls.

This generalization of the eGRM requires us to use the distances within each column of $d^{(\ell)}$ to call a set of nested neighborhoods around the corresponding haplotype. Calling these nested neighborhoods amounts to clustering the distances in each column of $d^{(\ell)}$. In order to do this efficiently for large n datasets, we developed a general, multithreaded, single-pass algorithm based on doubly linked lists to cluster real numbers on a closed interval when clusters must be separated by some fixed minimum distance. This approach allows us to cluster each column of $d^{(\ell)}$ in $\mathcal{O}(N)$ time. In experiments on simulated haplotype data, we achieve roughly an order of magnitude speedup over merge sort. In order to conserve memory, our implementation does not explicitly store the clustering results for each column of $d^{(\ell)}$. Rather, we use these clusters to directly construct columns of an asymmetric version of $\Omega^{(\ell)}$ on the fly, directly collapsing haplotype level relatedness down to sample level relatedness as we go. Taking the symmetric part of the resulting matrix gives us $\Omega^{(\ell)}$. ~~See Section 4.5 for details and the specific~~ Section 4.5 further details our construction of $\Omega^{(\ell)}$ ~~we use in this paper~~.

As a by-product of the clustering used to construct $\Omega^{(\ell)}$, we also return the nearest-neighbor set of each haplotype, which is then used to call sprigs and construct $X^{(\ell)}$ (~~see~~ Section 2.2.3). After calling sprigs using these neighborhoods, in order to avoid testing these sprigs in both $X^{(\ell)}$ and $\Omega^{(\ell)}$, we efficiently remove the structure associated with those sprigs from $\Omega^{(\ell)}$ using some additional statistics reported by our clustering algorithm before passing $X^{(\ell)}$ and $\Omega^{(\ell)}$ on to LOCATER for testing. All of these methods are available in `kalis v2`.

2.3 LOCATER Testing Routines

All of LOCATER's routines have been written in terms of matrix operations, allowing multiple quantitative traits to be tested in parallel with minimal additional computational cost. This includes the first implementation of a parallelized SD algorithm. For a given phenotype, this SD algorithm yields decoupled estimators for the effect β_j of each inferred clade genotype $X_j^{(\ell)}$. We then combine the independent two-sided p-values corresponding to these independent estimators via the Rényi Outlier Test [9] to obtain $p_{SD}^{(\ell)}$ at each locus. As demonstrated in [1], this approach yields considerable gains in power over alternative methods when very few (but more than one) of the β_j are non-zero; in other words, when more than one of the inferred genotype clades is associated with the phenotype. See Section 4.6 for details about the specific SD procedure we use.

E3► As explained in Section 3 and shown in Figure 6 of [1], SD returns an updated version of the data, there denoted as $Y^{(L+1)}$, which is independent of the information extracted to calculate $p_{SD}^{(\ell)}$. It is this $Y^{(L+1)}$ that LOCATER passes on to calculate the quadratic form testing procedure to calculate $p_Q^{(\ell)}$, guaranteeing the independence of $p_{SD}^{(\ell)}$ and $p_Q^{(\ell)}$. ◀**E3 Reply**

LOCATER also deploys several statistical and algorithmic innovations to efficiently calculate $p_Q^{(\ell)}$. Under the LOCATER model (Equation (1)), the score statistic against the null $\tau = 0$ is a quadratic form,

$$Y^\top P \Omega^{(\ell)} P Y \quad (4)$$

where $P = I - QQ^\top$ and $(A, G^{(\ell)}) = QR$ is the QR decomposition adjusting for the background covariates and the tested genotype $G^{(\ell)}$. In order to avoid launching unnecessary and expensive partial eigendecomposition routines at every target variant, we use a series of approximations to first assess whether the combined LOCATER p-value $p_C^{(\ell)}$ is sufficiently small to be interesting across any of the phenotypes. When it is, further eigendecomposition of $P \Omega^{(\ell)} P$ is deployed in order to obtain precise estimates.

We found that the Satterthwaite approximation [39], which is commonly used for testing quadratic forms [40], did not yield robust tail probability estimates for LOCATER. This may be because in this setting the matrices $P \Omega^{(\ell)} P$ are typically close to, but not quite, positive semi-definite — a key assumption of the Satterthwaite approximation. We overcame this obstacle with a new, robust tail approximation method for quadratic forms based on a shifted difference of chi-square random variables. In combination with our approximation stopping criteria, this tail approximation provides a basis for emerging genealogy-based association methods to reliably test local pairwise relatedness matrices, which may often not be positive semi-definite. **E16**► Our tail approximation method has the added advantage that it naturally admits three parameters — ν , δ_x^2 , and δ_\dagger^2 — to help control for population structure (in a way that generalizes inflation of the null distribution due to population structure and polygenicity. In effect, these parameters generalize genomic control to quadratic forms). [41]. ◀**E16 Reply** Importantly, these three parameters were chosen to be

orthogonal to the spectral parameters governing the distribution of Equation (4). If any inflation is observed in the Q-Q plot of $p_Q^{(\ell)}$ p-values after running LOCATER, this orthogonal parameterization allows us to adjust $(\nu, \delta_{\star}^2, \delta_{\dagger}^2)$ and rapidly calculate new $p_Q^{(\ell)}$ p-values without requiring the re-estimation of local ancestries at any target locus. ~~Please see Section 4.7 for further details.~~ (Section 4.7). **E25**► An interpretation of our parameters \$(\nu, \delta_{\star}^2, \delta_{\dagger}^2)\$ is provided in Section 6.5. ◀**E25** Reply

We ~~also~~ use a novel multi-threaded algorithm for efficiently projecting out background covariates when calculating the matrix traces needed for these tail approximations, ~~which is further described in Section 6.6~~ (Section 6.6). All final p-values p_Q involving eigenvalue terms are calculated using the ~~FFT-based approach~~ fast Fourier transform implemented in the R package **QForm** [42].

Finally, we combine our three p-values, $(p_{\text{SMT}}^{(\ell)}, p_{\text{SD}}^{(\ell)}, p_Q^{(\ell)})$, using a modified version of Fisher's combination test we call MSSE, ~~as described in Section 4.8~~ (Section 4.8).

2.4 Type-I Error Control

In order to confirm the calibration of LOCATER empirically, we simulated ~~4k~~ 1,000 independent genomic datasets, each consisting of ~~30k~~ 30,000 samples of a ~~1Mb~~ 1 Mb chromosome. See Section 4.1 for more details. For each dataset, we simulated ~~4k~~ 1,000 independent phenotype vectors assuming no causal variants. See Section 4.2 for more details of our phenotype simulation approach. **E33**► This yielded a total of 1 million independent phenotype vectors. ~~To simulate each phenotype, we first sampled two background covariate vectors: a_1 from independent standard Gaussian random variables and a_2 from independent Rademacher random variables. We then sampled the phenotype vector from $Y \sim N(a_1 + a_2, I)$.~~ We ~~We~~ tested each phenotype vector for association with the ancestry inferred at the mid-point of the corresponding chromosome using LOCATER. ◀**E33** Reply We display a Q-Q plot for $-\log_{10}$ of those p-values, as well as for each LOCATER sub-test — SMT, SD, and QForm — in Section 5.2 (Supplementary Figures 6,7,8,9). These Q-Q plots confirm that the p-values returned by each sub-test and the combined LOCATER p-value are all well calibrated under the null hypothesis.

2.5 Power

We compared LOCATER to ARG-Needle and standard SMT across a variety of genetic architectures. **E11**► Following Zhang et al. [3], we ran ARG-Needle with mutation rates \$\mu = 10^{-3}\$ and \$\mu = 10^{-5}\$. For an additional comparison, we also ran ARG-Needle with \$\mu = 10^{-7}\$ (Section 6.8). We refer to these three variations of ARG-Needle as AN3, AN5, and AN7 respectively. In order to give ARG-Needle the best possible advantage, in each simulation, we gave ARG-Needle the true underlying ARG generated by **msprime** rather than asking ARG-Needle to infer that ARG from the observed haplotypes. Thus our results represent an upper bound on the performance of ARG-Needle. ◀**E11** Reply We assessed every possible combination of

the following causal variant assumptions. We considered 3, 9, or 15 causal variants based on the number of independent causal alleles that were observed in the large follow-up study of GWAS hits by Abell et al. (see Figure 4b of [15]). We also considered causal variants with any derived allele count, derived allele count of 2 (doubletons only), or intermediate variants with derived allele count in [150,750]. That is equivalent to a derived allele frequency (DAF) in [0.0025,0.0125]. Lastly, we considered the case where all causal variants are observed or all causal variants are hidden. This yielded a total of 18 genetic architectures. Note that by 'observed', we mean that the causal variants were included in the dataset passed to each association method; by 'hidden', that they were not included in the dataset passed to each association method and thus could only be inferred via LD. In each simulation, causal variants were randomly assigned from among those fulfilling the required allele count requirements within a 10 kb window in the center of each simulated 1 Mb segment.

E19► Under each genetic architecture, we estimated power as a function of the underlying total association signal strength: the $-\log_{10}$ p-value that one would obtain by testing the simulated phenotype Y with an oracle ANOVA model that "knows" which are the active predictors and targets only those for testing. See Section 4.2 for a more precise definition. ◀**E19 Reply** To improve the interpretability of our power curves, following [1], we used the QR -decomposition to ensure that the total association signal was evenly split among the causal variants in every simulation. In other words, we ensured that the *observed* contribution of each causal variant to the total association signal was essentially equal for each simulated Y . ~~Please see~~ See Section 4.2 for more details of our phenotype simulation approach.

E45► For each method, we used 9,000 independent null simulations to estimate a genome-wide discovery threshold to maintain a family-wise error rate (FWER) below 0.05 (Section 6.10). This yielded a \$-\log_{10}\$ genome-wide discovery threshold 8.40 for LOCATER, 8.79 for SMT, 9.36 for AN7, 9.73 for AN5, and 9.78 for AN3. ◀**E45 Reply** In calculating power, we count our causal region as "discovered" if a testing method has a $-\log_{10}$ p-value ~~less than the $10^{-8.5}$~~ greater than their discovery threshold *anywhere* along the entire 1Mb region. This definition reflects how new associations are discovered in practice and provides a relatively strict benchmark. Each point of the resulting power curves was estimated via 1k independent samples: we simulated 10 independent phenotype vectors for each of 100 independent genomic datasets, each consisting of a 1 Mb chromosome sampled for 30k individuals. These power curves are available in Section 5.3 (Supplementary Figures 10,16,22,13,19,25). We summarize each of these curves with the estimated minimum signal strength required to achieve 80% power (lower is better). Figure 2 displays those estimates for LOCATER and SMT across all simulations where the underlying causal variants were observed; Figure 3, for those hidden.

E20► **E39**►

Figure 2: Dotplot of total association signal strength required to achieve 80% power (lower is better) under various simulation conditions where all causal variants were observed. Total association signal strength is the $-\log_{10}$ p-value that one would obtain by testing the simulated phenotype Y with an oracle ANOVA model that “knows” which are the active predictors and targets only those for testing. Causal variant # denotes the number of simulated causal variants. Causal variant type “any” means any variant could be causal; “doubletons” means only doubletons could be causal; “intermediateDAC [150,750]” means only variants with DAF a derived allele count in [150, 750], corresponding to a derived allele frequency in [0.0025, 0.0125], could be causal.

Dotplot of total association signal strength required

to achieve 80% power (lower is better) under various simulation conditions where all causal variants were hidden.

◀E39 Reply ▶E20 Reply From both Figure 2 and Figure 3 we see that LOCATER can reliably ties or improves upon the statistical power of SMT and ARG-Needle across all settings. LOCATER can detect substantially weaker association signals than SMT across a variety of causal variant settings, especially and ARG-Needle when there are 9 or 15 causal variants. The only exception we see to this pattern is the case of 3 intermediate variants (derived allele count in [150, 750]), where SMT has a very slight advantage over LOCATER. The causal variants in the intermediate case are far too common to be picked up by SD, so LOCATER must rely on quadratic form testing to gain any advantage over SMT. This highlights the fact that quadratic forms struggle to have power against very sparse signals (e.g., only 3 causal variants).

The power The power gains achieved by LOCATER over SMT in the observed causal variants case (Figure 2) are impressive given the fact that SMT has been shown to be surprisingly powerful this context. Across all analogous power simulations with full variant ascertainment and allelic heterogeneity, Link et al. found that SMT (which they refer to as "GWAS") had the same or more power than their ancestry-based quadratic form (eGRM) approach (see [24] Figure S2). To be clear, Link et al. did observe that the eGRM had more power than SMT in simulated array data with very incomplete variant ascertainment.

Comparing Figure 3 to Figure 2, we see that the relative power gains available from LOCATER are typically less in the case of hidden causal variants compared to the case of observed causal variants, but still substantial, across settings. The performance of ARG-Needle is the same in both figures because we provided ARG-Needle with the true underlying ARG in each simulation, making its performance unaffected by whether the simulated causal variants were observed or hidden. Except for the case of 3 doubletons, the power results reported in Figure 3 for SMT are remarkably similar to those reported in Figure 2 despite all of the causal variants being hidden. For the case of 3 doubletons, we see that the power of SMT is markedly reduced when the causal variants are hidden, making the relative power gain from LOCATER markedly large.

E21 ▶ E40 ▶

Figure 3: Dotplot of total association signal strength required to achieve 80% power (lower is better) under various simulation conditions where all causal variants were hidden. Total association signal strength is the $-\log_{10}$ p-value that one would obtain by testing the simulated phenotype Y with an oracle ANOVA model that “knows” which are the active predictors and targets only those for testing. Causal variant # denotes the number of simulated causal variants. Causal variant type “any” means any variant could be causal; “doubletons” means only doubletons could be causal; “DAC [150,750]” means only variants with a derived allele count in [150, 750], corresponding to a derived allele frequency in [0.0025, 0.0125], could be causal.

◀E40 Reply ▶E21 Reply

523

In order to confirm that these power results are robust to our choice of 10 kb as the size of the causal region, we replicated all of our experiments involving 9 causal variants assuming a 100 kb causal region. As can be seen in Section 5.4 (Supplementary Figures 28,29), the resulting power curves are very similar.

524

525

526

In order to compare LOCATER to the results one might obtain using sliding windows, we ran ACAT-O (STAAR without variant annotations) on our observed variant simulations from Figure 2, where any variant could be causal. Rather than testing all sliding windows for every simulation and effect size, we gave ACAT-O the precise location and width of the 10 kb causal window for each simulated dataset. This is an upper bound on the performance of ACAT-O in real-world settings where the location and size of the causal window

527

528

529

530

531

are unknown. We ran ACAT-O in two different ways: one in which we restricted the variants considered to rare variants ($MAF < 0.01$) and another where all variants are tested regardless of frequency (Section 6.9). Similar to the other methods, the genome-wide discovery threshold for ACAT-O was determined via null simulations (Section 6.10). As can be seen from Figure 4, the performance of both oracle ACAT-O approaches is roughly the same as SMT. LOCATER maintains its power advantage.

E22

Figure 4: Dotplot of total association signal strength required to achieve 80% power (lower is better) under various simulation conditions where all causal variants were observed, including comparison to oracle ACAT-O methods that are given the causal variant window. ACAT-O (rare) only tests variants with $MAF < 0.01$ whereas ACAT-O (all) tests all variants within the causal window. Total association signal strength is the $-\log_{10}$ p-value that one would obtain by testing the simulated phenotype Y with an oracle ANOVA model that “knows” which are the active predictors and targets only those for testing.

E22 Reply

In Section 5.3, we pair each plot of power curves with a companion plot showing which LOCATER sub-test is driving the gain in power (Supplementary Figures 11,17,23,14,20,26). These results show that SD is typically the source of LOCATER power gains, not QForm, reflecting the statistical advantage of SD-based methods over quadratic form based procedures in the case of sparse signals [1]. While these simulations appear to imply that SD is only effective at capturing signals driven by very rare variants, this is expected since we only encoded very rare variants in the clade genotype matrix $X^{(\ell)}$ passed to SD. If SD was used to test the entire local genealogy at every locus, we may see increased statistical efficiency in incorporating

common variant associations. We return to this point in the Discussion section. Substantial power gains are possible via LOCATER in the case of multiple rare causal variants.

2.6 Localization

Alongside power, an important factor for real-world utility is the ability of an association method to accurately localize causal variant(s) within a relatively narrow genomic interval. In Section 5.3, we also pair each plot of power curves with a companion localization plot (Supplementary Figures 12,18,24,15,21,27). To measure the ability of a given method to localize the causal region, we calculated the distance between the most significant marker (lead variant) reported by a method and the midpoint of the causal region in every simulation, taking the average distance in the case of tied lead variants. We used these distances to estimate the width of an 80% confidence interval. This width represents the answer to the question: how large of a search window centered on the lead variant would an investigator need in order for that window to capture the midpoint of the causal region 80% of the time? In our plots reporting these confidence interval widths, we only report confidence interval widths at signal strengths where the corresponding method had at least 80% power to detect the causal region. As expected, both LOCATER and SMT struggled to localize the causal region more when the causal variants were hidden. The localization performance of ARG-Needle was unaffected by whether the simulated causal variants were observed or hidden because we provided ARG-Needle with the true underlying ARG in every simulation.

Across simulations where all variants in the causal region were potential causal variants, LOCATER ~~more accurately localized~~ was substantially more accurate in localizing the causal region than SMT or ARG-Needle, regardless of whether the causal variants were observed or hidden. ~~This pattern also held when all of the causal variants were rare, the sole exception being the case of 3 hidden casual variants where SMT and LOCATER were effectively tied. Simulations where all of the causal variants were doubletons yielded somewhat more mixed results. In simulations with intermediate causal variants (derived allele count in [150, 750]), ARG-Needle and SMT slightly outperformed LOCATER for some signal strengths. Overall, LOCATER improved or effectively tied the localization accuracy of SMT and ARG-Needle in the intermediate causal variant case.~~ With the exception of a few signal strengths when there were 9 causal variants, LOCATER ~~still outperformed SMT~~ also outperformed SMT and ARG-Needle in localizing the causal region when doubleton causal variants were observed. In all simulations where the doubleton causal variants were hidden, regardless of the number of causal variants, ~~both SMT and LOCATER~~ all methods performed very poorly in localization, ~~with both methods reporting~~ all methods reported confidence intervals wider than ~~600~~ 500 kb across all signal strengths where they achieved at least 80% power. Overall our results suggest that LOCATER can leverage allelic heterogeneity to improve the localization of trait mapping compared to standard SMT and ARG-Needle.

2.7 Scalability

The ability to scale to large modern datasets with hundreds of thousands of samples is essential for the success of any trait mapping approach and the size of local genealogies presents a significant challenge. Based on our simulations involving 30k samples (60k haplotypes), LOCATER took an average of 19.14 seconds (sd: 2.88 seconds) to perform sprig testing and an average of 3.07 minutes (sd: 0.98 minutes) to perform quadratic form testing at each variant. These simulations were run on a shared-time university HPC cluster with heterogeneous nodes hosting a mix of CPU architectures. All jobs requested 8 cores and 160 GB of memory. When combined with the computational overhead required to form the clade genotype matrix $X^{(\ell)}$ and $\Omega^{(\ell)}$ provided to LOCATER, our simulations required an average of 4.00 minutes (SD: 0.99 minutes) to test each target locus. When this is combined with the additional cost of performing ancestry inference with kalis, our simulations required an overall average of 6.42 minutes per target locus. These results suggest that future applications of LOCATER that only use inferred clades $X^{(\ell)}$ and avoid use of $\Omega^{(\ell)}$ will achieve substantial computational savings.

~~In separate~~ E49 ► In parallel work, we ~~also~~ ran LOCATER in combination with kalis v2 on a real sequencing dataset including 6795 individuals and 101 correlated quantitative traits (~~manuscript in prep~~)—[5]. We divided the genome into 4580 (partially overlapping) segments; each segment had an average of 13,000 variants. We allocated 12 cores and 60 GB of memory per segment. This allowed kalis v2 to store two checkpoints in memory. The average time required for kalis v2 and LOCATER to screen each segment was 32 minutes. That is equivalent to 3.35 years of single-core compute time. While substantial, this equates to 1.22 days using a cluster of 1000 CPUs. ~~This result shows~~ In other preliminary work we have run LOCATER on as many as 12,964 genomes on commodity hardware, which required 8,257 CPU-days to analyze 4 traits (unpublished data). These results show that it is feasible to run LOCATER on a moderately large genomic dataset using an academic compute cluster. ◀E49 Reply

3 Discussion

We have presented a general framework for using inferred local ancestries to boost SMT association signals in the presence of allelic heterogeneity. To our knowledge, this is the first demonstration of any ancestry-testing approach that yields significant power gains over SMT in a genome-wide screen that includes non-coding regions. More importantly, our approach can be applied in conjunction with any ancestry inference engine, thus providing a flexible association testing framework that can adapt to rapidly improving ancestry inference methods.

~~In parallel work~~ E46 ► As mentioned in the Introduction, we have demonstrated the ~~efficacy of real-world~~ power gains attainable via LOCATER in a dataset of 6,795 Finnish genomes with extensive quantitative

trait data (manuscript in prep.), where we observed a significant power boost at several loci marked by allelic heterogeneity, and in our preliminary work we have run LOCATER on as many as 12,964 genomes on commodity hardware, which required 8,257 CPU days to analyze 4 traits (unpublished data) measurements [5]. There, as in our simulations in Section 2.5, we find that LOCATER requires a less stringent genome wide discovery threshold than SMT. We believe this is due to the increased dependence between proximal tests induced by the dependence between proximal local genealogies, which suggests that it will be safe for investigators to apply their SMT discovery threshold for a given dataset to their LOCATER results. However, these results also suggest that running null simulations, analogous to those described in Section 6.10, to estimate a LOCATER-specific genome-wide discovery threshold for a given dataset will improve statistical power. ◀E46 Reply

The largest power gains demonstrated in this paper were seen in the case of multiple rare causal variants evaluated using the SD ~~subtest~~sub-test. This suggests that if we had tested more of the underlying tree structure at each locus with SD we may have achieved even greater power. In other words, as mentioned above, we may have clustered each distance matrix and tested more common clades than simply the sprigs with SD. This approach would present more of the underlying ancestral tree to LOCATER via $X^{(\ell)}$ rather than via $\Omega^{(\ell)}$. Exploring the power of LOCATER at different points along the continuum between testing all of the ancestral structure with $X^{(\ell)}$ and testing all of the ancestral structure with $\Omega^{(\ell)}$, will be a focus of future research.

In conjunction with the new features added to `kalis` [8], LOCATER provides an efficient method for genome-wide testing that is ready for use on real-world datasets now. These new features involve several algorithms — a general HMM checkpointing algorithm, a fast clustering algorithm, and a fast trace calculation method — that will likely prove helpful for the acceleration of other ancestry inference and association methods. Our ~~novel~~ quadratic form tail approximation approach, based on a shifted difference of chi-square random variables, see Equation (11), provides a basis for emerging association methods to reliably test local relatedness matrices that may not be positive semi-definite.

Adequately adjusting for population structure when testing inferred local ancestries is an open and challenging problem. In this initial version of LOCATER, we allow principal components (PCs) to be included in A . As mentioned in Section 2.3, we also parameterized our ~~novel~~ quadratic form tail approximation in a way that ~~naturally~~ accommodates genomic-control-like inflation adjustments without requiring recalculating the genealogy at any LOCATER target loci. See Section ~~??~~6.5 for details and theoretical motivation. However, future work applying LOCATER or any ancestry testing method to real genomic data will need to take special care when examining Q-Q plots for inflation. We further address adjusting for population and cryptic structure in parallel work [5].

LOCATER makes a number of critical methodological advances towards powerful ancestry-based associa-

tion testing. We expect that further work building on these advances alongside the application of LOCATER to more diverse datasets will yield new functional discoveries.

4 Methods

4.1 Haplotype Data Simulation

In order to assess the calibration and power of LOCATER, we simulated 100 genomic datasets, each consisting of a 1 Mb chromosome for 30k human samples (60k haplotypes). Each dataset was simulated using `msprime` [43]. In order to model the diversity of arising genomic datasets, 10k samples in each dataset were drawn from each of three 1000 Genomes populations – Yoruba, Han Chinese, and Central European. See Section 6.7 for further details.

4.2 Phenotype Simulation

~~In~~ **E27** ▶ To simulate each phenotype vector, following [6], we first simulated two background covariate vectors: a_1 a vector of independent standard Gaussian random variables and a_2 a vector of independent Rademacher random variables. We tested each phenotype vector assuming that these two background covariates were observed and included in A from Equation (1). For our null simulations – without any causal variants – this amounted to sampling each phenotype vector from $Y \sim N(a_1 + a_2, I)$. **E27 Reply** For our power simulations, in the middle of each 1 Mb region, we selected causal variants within a 10 kb causal window. We fully replicated these simulations under all 18 possible combinations of 3 parameters: the number of causal variants, the allele frequency constraint imposed on those causal variants, and whether the causal variants were assumed to have been observed (called during sequencing) or hidden. More explicitly, we considered the case of 3, 9, or 15 causal variants. These causal variants were selected uniformly at random from among variants within the 10 kb causal window meeting the given allele frequency constraint. As our primary focus, we considered the case of no allele frequency restraint, in which case every variant in the 10 kb causal window had an equal chance of being selected as a causal variant. We also considered the case where all causal variants were constrained to be doubletons (present in two copies) and the case where all of the causal variants were constrained to have derived allele frequency in the half open interval $[0.0025, 0.0125)$. If the simulated chromosomes did not include the requisite number of causal variants within the 10 kb causal window, we rejected that simulated dataset and simulated a new set of chromosomes.

Given an active set of causal variants, we simulated Y while distributing the observed effects across the causal variants as evenly as possible by manipulating the QR -decomposition as done by [1]. Following their approach, let \mathcal{A} denote this selected set of causal variants and $X_{\mathcal{A}}$ denote the genotype matrix encoding those causal variants. Consider the QR -decomposition $QR = \tilde{X}_{\mathcal{A}}$, which we define as $P_{\mathcal{A}}^{\perp} X_{\mathcal{A}}$ with length-normalized

columns. The sufficient statistic for the oracle model is $\|Q^\top Y\|_2^2$ with expected value $\|R\beta\|_2^2$. For a desired total association signal strength s , we solve $R\beta = \sqrt{F_{\chi_a^2}^{-1}(1 - 10^{-s})} \mathbf{1}$ to make the magnitude of each entry of β as similar as possible. Then, we simulate $Y = A\alpha + (\tilde{X}_A\beta + P_{\tilde{X}_A}^\perp \epsilon)$ where $\epsilon \sim N(0, I_n)$. This ensures that the observed $\hat{\beta} = R^{-1}Q^\top Y = \beta$ is stable across simulations and that the $-\log_{10}$ p-value that one would obtain by testing the resulting Y with an oracle ANOVA model that “knows” the active predictors will be approximately s in every simulation.

4.3 Checkpointing Approach

~~Let j index the positions of the~~ To understand the need for checkpointing, as mentioned above and further detailed below, recall that we implicitly need the pairwise distance matrix $d^{(\ell)}$ from Equation (2) returned by `kalis` in order to obtain $X^{(\ell)}$ and $\Omega^{(\ell)}$ at each target locus ℓ . While `kalis v1` can efficiently propagate the forward and backward recursions to obtain the forward probability matrix $f^{(\ell)}$ and backward probability matrix $b^{(\ell)}$ needed to calculate $d^{(\ell)}$ at a single locus ℓ , obtaining the pair $(f^{(\ell)}, b^{(\ell)})$ at sequential positions ℓ is challenging for HMMs due to the uni-directionality of the forward and backward recursions. The compute-minimizing approach would involve running a single pass of the forward algorithm – iterating the forward recursion to target locus 1, then locus 2, and so on until locus L – ~~target loci from smallest to largest.~~ We will solve for an optimal checkpointing schedule for the forward algorithm to propagate to target loci sequentially in reverse order, from $j=L$ to $j=1$. We will use $j=0$ to index the HMM prior hidden state probabilities used to initialize the forward algorithm. Informally, for each subsequent target locus, a checkpointing schedule states the checkpoint at which the forward algorithm should start and the intervening target loci where it should stop to store any new checkpoints on its way to reach that target locus – and a single pass of the backward recursion from target locus L to target locus 1 while storing $f^{(\ell)}$ and $b^{(\ell)}$ at every $\ell = 1, \dots, L$. Since each $f^{(\ell)}$ and $b^{(\ell)}$ consumes $8N^2$ bytes of memory (e.g.: 80 GB for $n = N/2 = 50,000$ haplotypes), this approach requires far too much storage for most genomic datasets. On the other hand, we have the memory-minimizing approach, where we restart the forward and backward recursions from the respective ends of the genomic segment for every target locus. While this approach only requires storing a single $f^{(\ell)}$ and a single $b^{(\ell)}$ at any given time, it demands far too much compute time for most genomic datasets, requiring $\mathcal{O}(L^2N^2)$ floating point operations (FLOPs) – a prohibitive cost. An attempt to rescue this approach by splitting the genome into smaller segments (running in chunks) would still require $\mathcal{O}(L^2N^2)$ compute time. ~~To optimize this schedule, we do not need to consider storing checkpoints at variants that are not target loci since any such checkpointing strategy could be improved by moving those checkpoints to target loci. Thus, our checkpointing schedule will only need to render integers in $1, \dots, L$.~~

~~Optimizing the checkpointing schedule is based on recycling the memory used to store checkpoints that are no longer being used. Here, we solve for an optimal checkpointing schedule where the computational cost~~

~~to propagate the forward algorithm from any position to any other may be arbitrary. The checkpointing routine that minimizes the total computational cost in this setting may be obtained by solving nested optimization problems. While tractable, solving this system of nested optimization problems can be rather time consuming to solve. Therefore, we subsequently consider the special case where the cost of propagating the forward algorithm between consecutive target positions is constant. For example, in a genomics context this may be true if we have equally spaced target loci. In that special case, we explain how the optimal checkpointing schedule can be solved rapidly via dynamic programming.~~

~~We start with the general case. Define $g: \mathbb{Z}_{\geq 0}^2 \rightarrow \mathbb{R}_{\geq 0}$ such that $g(s, t)$ is~~ We provide a checkpointing algorithm that finds an optimal balance in this memory-compute trade-off, minimizing the compute time required given a fixed memory budget. The overall idea is to occasionally stop the forward recursion and store $f^{(\ell)}$ at its current position as a checkpoint (typically overwriting an old checkpoint) in order to avoid repeatedly restarting the forward recursion from the beginning of the genomic segment. We start with a user-specified memory budget capable of holding C checkpoints, each storing a $N \times N$ matrix of forward probabilities $f^{(\ell)}$. We run the backward recursion once across the entire chromosome or genomic segment, stopping at each consecutive target locus sequentially from the target locus with the largest position ($\ell = L$) to the target locus with the smallest position ($\ell = 1$). When the backward recursion stops at a given target locus, we run the forward recursion from the nearest checkpoint to meet the backward recursion and so obtain $X^{(\ell)}$ and $\Omega^{(\ell)}$ at that target locus. Note it is natural for us to perform this backwards along the genomic segment, since there is a slightly higher computational cost for the backward recursion and hence we favor repetitive restarts of ~~the cost of propagating the forward algorithm directly from target locus s to target locus t , defined to be zero whenever $s = t$. Define $f: \mathbb{Z}_{\geq 0}^3 \rightarrow \mathbb{R}_{\geq 0}$ such that $f(s, t, c)$ is the minimal cost forward recursion.~~

Iterating from locus L down to locus 1 makes minimizing the compute required for the forward algorithm to sequentially visit target loci $t, t-1, t-2, \dots, s$ given c available checkpoints and a known set of forward probabilities (i.e. a checkpoint, or the HMM prior) at s . That is, $f(s, t, c)$ corresponds to backward recursion trivial: we simply visit each locus sequentially in a single pass. The challenge is determining where and when to overwrite existing checkpoints to minimize the total distance (number of variants) that the forward algorithm needs to iterate over in order to provide forward matrices in reverse order $f^{(L)} \rightarrow f^{(L-1)} \rightarrow \dots$. In Section 6.3 we show how to solve for a schedule of checkpoints that achieves this minimum for any discrete time HMM, given storage for a fixed number of checkpoints C . We call this solution the optimal checkpointing schedule. After a forward matrix $f^{(\ell)}$ is obtained at a given target locus ℓ , this schedule instructs `kalis` which checkpoint to use to restart the forward recursion to obtain the next forward matrix at locus $\ell-1$, and where to lay down new checkpoints (if any) as the forward recursion proceeds to that locus. The checkpointing schedule also dictates where to initialize the C checkpoints as we iterate the forward

recursion to the cost of the (unknown) optimal schedule given memory for c checkpoints first target locus L .
 Supplementary Figure 5 shows how the efficiency of our checkpointing algorithm scales in both L and C .

We aim to obtain a checkpoint schedule which enables us to achieve the cost $f(0, T, C)$. This objective is defined by the three following equations. First, if there are no checkpoints available, simply propagating to each target locus sequentially from the initial position is our only option, which comes with cost $f(x, y, 0) = \sum_{j=x}^y g(x, j)$. Second, if we already know the forward probabilities at a given position, there is no computational cost to obtaining them, so $f(x, x, c) = 0$ for any c . Third, we have the following recursive relationship

$$f(s, t, c) = \min_{j \in \mathbb{Z}_{\geq 0}: j \in (s, t]} g(s, j) + f(s, j - 1, c) + f(j, t, c - 1).$$

Intuitively, this recursion expresses the idea that solving for an optimal checkpointing schedule with c checkpoints over an interval can be thought of as placing one of the checkpoints at some optimal index to divide the interval so that the upper part is solved with $c - 1$ checkpoints; the lower part, c checkpoints. We will write that optimal index, the argument that achieves the minimum of Equation (19) as

$$h(s, t, c) = \operatorname{argmin}_{j \in \mathbb{Z}_{\geq 0}: j \in (s, t]} g(s, j) + f(s, j - 1, c) + f(j, t, c - 1).$$

Together, these three equations allow us to solve for an optimal checkpoint schedule by transversing a bifurcating tree from right to left, from the leaves toward the root, where each node can be identified with an optimal index h . In the case where $L = 10^5$ equally-spaced target variants, Supplementary Figure 5 shows that the forward algorithm would be required to propagate over a total distance $D > 10^4 L$ target variants without checkpoints. Sufficient memory to store $C = 2$ checkpoints with our approach brings D under $100L$ target variants; $C = 8$ checkpoints brings D under $10L$ target variants. Even in the $L = 10^6$ case, which is larger than we would expect over any phased genomic segments since we only test target variants with moderately significant SMT p-values, $C = 10$ checkpoints brings D nearly down to $10L$ target variants.

While solving this recursion is tractable, it can be rapidly accelerated if we assume that the computational cost of propagating the forward probabilities between adjacent target loci is fixed. In other words, we assume $g(x - 1, x)$ is constant for all target indices $x \in \{1, 2, \dots, L\}$. This makes the computational cost a simple linear function of the distance between indices, yielding the following simplified system of equations.

Analogous to above, we have $\tilde{f}(i, 0) = i(i + 1)/2$ and $\tilde{f}(0, c) = 0$ for all c . Our recursions simplify to

$$\tilde{f}(i, c) = \min_{j \in \mathbb{Z}_{\geq 0}: j \leq i} j + \tilde{f}(j-1, c) + \tilde{f}(i-j, c-1),$$

$$\tilde{h}(i, c) = \operatorname{argmin}_{j \in \mathbb{Z}_{\geq 0}: j \leq i} j + \tilde{f}(j-1, c) + \tilde{f}(i-j, c-1).$$

For some pre-specified maximum number of target loci Solving for the optimal checkpointing schedule can be computationally intensive for any given set of target loci. The version of the checkpointing schedule solver currently implemented in LOCATER assumes that target loci are evenly spaced. This simplification does not qualitatively change performance but allows us to solve for the optimal checkpointing strategy for a given L and maximum number of available checkpoints C , let us define two matrices that we will use as look-up tables for our dynamic program. Let $F \in \mathbb{R}_{\geq 0}^{(L+1) \times (C+1)}$ such that $F_{ij} = \tilde{f}(i-1, j-1)$ and $H \in \mathbb{Z}_{\geq 0}^{(L+1) \times (C+1)}$ such that $H_{ij} = \tilde{h}(i-1, j-1)$. via a dynamic program, making the solution readily available (Section 6.3). Our checkpointing implementation is available via the `ForwardIterator` function and associated helper functions now provided in `kalis v2`.

Of course, for datasets with a large number of samples, there may not be sufficient capacity to store many checkpoints in memory. At a minimum, running `kalis` on n samples ($2n$ phased haplotypes) to obtain each $\Omega^{(\ell)}$ requires $32n^2$ bytes to store the forward and backward probabilities and another $8n^2$ bytes to store $\Omega^{(\ell)}$. Storing each additional checkpoint of forward probabilities requires $16n^2$ bytes. Given the nested nature of 21 and 22, we can rapidly solve for each entry of F and H by synchronously iterating over the entries of both matrices in column-major order. Once we have a completed F and H , we can use 22 to read off the appropriate entries in table H to obtain the optimal checkpointing schedule for any problem with up to L target variants and C available checkpoints. Methods for constructing H and reading it to construct a checkpoint schedule are available in `kalis v2`. This implementation covers the slightly more general case than (21) where the cost depends on distance but is independent of locus. This corresponds to a translation invariance assumption that does not cover the most general case in (19). our checkpointing algorithm, most checkpoints can be stored on disk rather than memory, which comes at minimal computational cost as long as one or two of the checkpoints (the ones that are closest to the current target loci) are always kept in memory. We plan to add native support for storing file-backed checkpoints to `kalis` in the near future. Looking further ahead, `kalis` can already be distributed across machines, each running the LS model on a different subset of recipient haplotypes [8], but running LOCATER across distributed machines would require substantial network communication. Reducing this communication is a direction of future work.

4.4 Defining our Generalized Relatedness Matrix

Here we build up to our definition of a generalized eGRM, which we will pass to LOCATER as $\Omega^{(\ell)}$. We will construct $\Omega^{(\ell)}$ based on an asymmetric genetic distance matrix $d^{(\ell)} \in \mathbb{R}_{\geq 0}^{N \times N}$, such as the one provided by `kalis` (see Equation (2)), and a set of monotonic regularization functions g_1, \dots, g_N which we will introduce shortly. Recall that $d_{ij}^{(\ell)}$ measures the distance to haplotype i from haplotype j . The distance from any given haplotype to itself $d_{ij}^{(\ell)} = 0$. Let $\pi_j : [N] \rightarrow [N]$ be the permutation that sorts $d_{\cdot j}^{(\ell)}$ such that $d_{\pi_j(1)j}^{(\ell)} \leq d_{\pi_j(2)j}^{(\ell)} \leq \dots \leq d_{\pi_j(N)j}^{(\ell)}$. By convention, $\pi_j(1) = j$ and $\pi_j(N+1) = \pi_j(N)$. Using $d^{(\ell)}$, we define a local haplotype relatedness matrix $\Psi^{(\ell)} \in \mathbb{R}_{\geq 0}^{N \times N}$ with elements

$$\Psi_{ij}^{(\ell)} = \sum_{k=\pi_j^{-1}(i)}^N \psi_j(k) \quad \text{where} \quad \psi_j(k) = \frac{1}{k} g_k \left(d_{(k+1)j}^{(\ell)} - d_{kj}^{(\ell)} \right) \quad (5)$$

and each $g_k : \mathbb{R}_{\geq 0} \rightarrow \mathbb{R}_{\geq 0}$ is a monotonic function of x such that $g_k(0) = 0$. Note, given these definitions, $\psi_j(N) = 0$.

Our construction does not require the assumption of diploid samples but we will assume that here for ease of exposition. We will assume that the rows and columns of $\Psi^{(\ell)}$ are permuted such that haplotypes from the same sample are grouped together. This allows us to succinctly write our generalized eGRM in terms of Equation (5) as

$$\Omega^{(\ell)} = \text{sym} \left(B_{n,2}^\top \Psi B_{n,2} \right) = \frac{1}{2} \left(B_{n,2}^\top \Psi B_{n,2} + \left(B_{n,2}^\top \Psi B_{n,2} \right)^\top \right), \quad (6)$$

where $B_{n,2} = I_{n \times n} \otimes \mathbf{1}_2$ and \otimes is the Kronecker product.

In Section 6.2 we explicitly show how this construction of $\Omega^{(\ell)}$ generalizes the standard eGRM. In short, there we show that the eGRM can be expressed in terms of a haplotype similarity matrix assuming Hardy–Weinberg equilibrium and specific choices for the background covariates and allele frequency weights. Then we connect that haplotype similarity matrix representation to our choice of $\Psi^{(\ell)}$ in Equation (5) above.

4.5 Efficiently Constructing Clade Genotypes and our Generalized eGRM in LOCATER

Building on the notation above in Section 4.4, currently in LOCATER and this paper, we set

$$g_k(x) = \begin{cases} 0 & x < c \\ \min(x, 1) & x \geq c \end{cases} \quad (7)$$

for all k where our threshold $c = -0.2 \log(\mu)$ and μ is the mutation probability parameter provided to the LS model. This choice of regularization function(s) tends to filter out many low evidence clades. This function also restricts the clade matrix representation so that at most one mutation can be present on a given branch.

Given such a regularization, $\Psi_{.j}$ has a series of nested neighborhoods of donor haplotypes along $i = \pi_j(1), \pi_j(2), \dots, \pi_j(N)$ where there are distances of at least c between adjacent level sets of distances. This allows us to represent the level sets of Ψ_{ij} as the solution to a clustering problem on the real interval from 0 to the maximum possible distance (D) where we require unique clusters to be at least distance c apart. ~~Here,~~ each-Each cluster corresponds to a level set of donor haplotypes. We use a single-pass partial sorting algorithm based on doubly-linked-lists to solve this clustering problem in $\mathcal{O}(N)$ time. In our experiments on simulated haplotype data, our partial sorting algorithm achieves roughly an order of magnitude speed up over merge sort. Given the definition of v in Equation (2), the maximum possible distance is $D = -\log(v) \approx 744.44$. This maximum is helpful in accelerating our implementation because the number of possible level sets (clusters) d is bounded above $d \leq \lceil D/c + 1 \rceil$, allowing us to efficiently pre-allocate sufficient memory to store the clustering solution. Since this partial sorting algorithm can be run in parallel for each recipient haplotype (for each column $\Psi_{.j}$), we use a multi-threaded implementation in `kalis v2`.

~~Let $\nu_j^1, \nu_j^2, \dots, \nu_j^d$ denote the nested neighborhoods corresponding to recipient haplotype j ordered from the nearest neighborhood to the furthest possible neighborhood. Some of the more distant neighborhoods may be empty if fewer than d clusters (level sets) are observed among the distances $d_{.j}^{(\ell)}$. From these neighborhoods, we call a set of clades for each layer 1 through d by iteratively identifying fully connected cliques within the directed graph implied by the neighborhoods as follows. In this single-pass clustering algorithm, we also compute the level sets of Ψ_{ij} , the nearest neighbors of j and various other quantities so that the downstream modifications we make to Ψ related to sprig testing may be done in place (without recalculating Ψ). Many pairs of that processes columns of Ψ are then calculated in parallel. By processing columns of Ψ in pairs, we can in pairs. This allows us to directly compute our sample by sample matrix $B_{n,d}^\top \Psi B_{n,d}$ in a single pass. Symmetrizing this matrix yields $\Omega^{(\ell)}$ as defined in Equation (6). Future work will focus on reducing the computational and memory requirements of this symmetrization step.~~

4.6 Parallelized Stable Distillation Procedure

Given a matrix of inferred, clade-based genotypes $X^{(\ell)}$, we use the one-predictor-at-a-time SD procedure described in Equation 4 of [1] equipped with the simple quantile filter presented in Algorithm 1 of [1]. In this SD procedure, we take $(A, G^{(\ell)})$ as the background covariates when testing H_0^{SD} at a particular target locus ℓ . In short, using this approach, we “distill” one β_j for $j = 1, \dots, p$ at a time, obtaining an independent p-value for each. These p-values are then tested using the Rényi Outlier Test [?] [9]. To run this procedure, LOCATER requires an estimated upper bound c on the number of independent causal clades:

$c \geq |\{j : \beta_j \neq 0\}|$. By default and throughout this paper, we set $c = 16$. This bound c is used to set the simple quantile filtering threshold used during distillation. Explicitly, LOCATER sets the quantile filtering threshold $t = F_{c,p-c+1}^{-1}(0.01)$ where $F_{a,b}$ is the CDF of the Beta distribution with expectation $\frac{a}{a+b}$. By default, the maximum number of outliers considered by the Rényi Outlier Test is set to c .

4.7 Quadratic Form Testing & Tail Approximation

Let $QR = (A, G_j)$ be the QR-decomposition of the $n \times (q + 1)$ matrix (A, G_j) and let $P = I - QQ^\top$ project onto the subspace orthogonal to the columns of (A, G_j) . Differentiating the Gaussian likelihood corresponding to Equation (1) with respect to τ yields $Y^\top P\Omega^{(\ell)}PY$ as the score statistic. Under H_0^Q ,

$$Y^\top P\Omega^{(\ell)}PY \sim \sum_{j=1}^n \lambda_j Z_j^2 \quad (8)$$

where each $Z_j \stackrel{iid}{\sim} N(0, 1)$. Following the approach of FastSKAT [40], we use partial eigendecomposition to obtain a computationally tractable approximation to this null distribution. Given, a top- k eigendecomposition in which we explicitly calculate the leading k eigenvalues, we have

$$Y^\top P\Omega^{(\ell)}PY \sim T_k + R_k \quad (9)$$

where $T_k = \sum_{j=1}^k \lambda_j Z_j^2$ and $R_k = \sum_{j=k+1}^n \lambda_j Z_j^2$. Since $\lambda_{k+1}, \dots, \lambda_n$ are unknown, FastSKAT proposes approximating the distribution of R_k using a single chi-square random variable $\tilde{R}_k \sim \alpha\chi_\nu^2$. The scale parameter α and degrees of freedom parameter ν are set to match the mean and variance of R_k , as initially proposed by Satterthwaite [39]:

$$\alpha = \frac{\eta_2}{\eta_1} \quad \nu = \frac{\eta_1^2}{\eta_2} \quad (10)$$

where $\eta_1 = \text{tr}(P\Omega^{(\ell)}P) - \sum_{j=1}^k \lambda_j$ and $\eta_2 = \|P\Omega^{(\ell)}P\|_{HS}^2 - \sum_{j=1}^k \lambda_j^2$. Substituting \tilde{R}_k in for R_k , FastSKAT uses $T_k + \tilde{R}_k$ as an approximate null distribution. This is still a linear combination of chi-square random variables, but since all of its parameters are now known, its distribution is readily available via the fast Fourier transform (FFT).

E31► Unfortunately, in the context of LOCATER, \tilde{R}_k does not provide an adequate approximation to the distribution of R_k . ~~We found that the tails of \tilde{R}_k tended to decay faster than those of R_k , yielding anti-conservative estimates of more extreme p-values. This is because the scale α in Equation (10) can over-estimate the rate of decay in the tail of R_k . On a technical note, the Satterthwaite approximation also~~ First, the Satterthwaite approximation assumes that $P\Omega^{(\ell)}P$ is positive semi-definite (PSD). This is not at all guaranteed in our application. The closeness of any $P\Omega^{(\ell)}P$ to PSD will depend on the ancestry at ℓ as well as the user's choice of ancestry inference engine and clade-encoding method. The $P\Omega^{(\ell)}P$ matrices we

have observed in the development of LOCATER are typically close to, but not exactly, PSD.

Second, we found that the tails of \tilde{R}_k tended to decay much faster than those of R_k , yielding anti-conservative estimates of more extreme p-values. This is a consequence of the fact that, \tilde{R}_k , as a gamma random variable, has an exponential right tail with a rate of decay $(2\alpha)^{-1}$. Since R_k is itself a linear combination of gamma random variables, the term with the largest coefficient governs the behavior in the right tail. More precisely, as long as some $\lambda_j > 0$ for $j > k$, R_k has an exponential right tail with rate $\left(2 \max_{j>k} \lambda_j\right)^{-1}$. Intuitively, this is a consequence of the fact that extreme observations of R_k in the far right tail are much more likely to arise from the term with the largest scale. We have analogous behavior in the left tail: as long as some $\lambda_j < 0$ for $j > k$, R_k has an exponential left tail with rate $\left(2 \min_{j>k} \lambda_j\right)^{-1}$. We found that the Satterthwaite approximation consistently over-estimates the rate of decay because $(2\alpha)^{-1}$ tends to be much larger than $\left(2 \max_{j>k} \lambda_j\right)^{-1}$. In fact, in the PSD case, $(2\alpha)^{-1} > \left(2 \max_{j>k} \lambda_j\right)^{-1}$ (Section 6.4).

In order to make the tails of our approximating distribution more robust (heavier) and obviate the PSD requirement, we propose approximating R_k with a difference of independent chi-square random variables:

$$\begin{aligned} \ddot{R}_k &\sim |\lambda_k| (X - X' + \mu) \\ \ddot{R}_k &\sim |\lambda_k| (W' - W'' + \mu) \end{aligned} \tag{11}$$

where $X \sim \chi_a^2$, $W' \sim \chi_b^2$ is independent of $X' \sim \chi_b^2$, $W'' \sim \chi_b^2$. A derivation and explicit equations for the three scalar parameters – a , b , and μ – are given in Section 6.5 in terms of η_1 and η_2 . In short, these parameters are set so that our approximation matches the mean and variance of R_k while simultaneously minimizing $|\mu|$. Like $T_k + \tilde{R}_k$, $T_k + \ddot{R}_k$ is also a convolution of weighted chi-square random variables, making it just as easy to obtain p-values for using the FFT.

Since the left and right exponential tails of \ddot{R}_k decay with rate $|2\lambda_k|^{-1} \leq \left(2 \max_{j>k} |\lambda_j|\right)^{-1}$, the tails of \ddot{R}_k at least as heavy as those of R_k . Of course, this alone does not guarantee accurate p-value estimation across the range of quantiles required for LOCATER. We must also accurately estimate of the body of the distribution across the full range of $P\Omega^{(\ell)}P$ we observe. Section 6.4 explains how the stopping criteria we use for top- k eigendecomposition in LOCATER addresses this issue and combines with \ddot{R}_k to yield reliable p_Q p-values. **E31 Reply E17** As shown in Section 6.5, a key advantage of our specification of \ddot{R}_k is that it can be easily generalized to accommodate three inflation parameters — ν , δ_\star^2 , and δ_\dagger^2 — that are orthogonal to a , b , and μ . There we show how these parameters naturally arise when considering the presence of some unobserved confounder although, in practice, inflation of the null distribution may also be driven by polygenicity. The orthogonality allows us to tune ν , δ_\star^2 , and δ_\dagger^2 to adjust for population confounding after observing help reduce any inflation observed in an initial Q-Q plot of p-values $p_Q^{(\ell)}$ without having to recalculate a , b , or μ . In other words, we can adjust ν , δ_\star^2 , and δ_\dagger^2 without having to re-calculate the genealogy

at any target variant. **E17 Reply** While we do not make use of that capability in this paper, we expect it will play an important role in future analyses.

Critical to the scalability of this approach is the availability of η_1 and η_2 via fast trace calculations. FastSKAT [40] uses stochastic estimator of η_2 . Starting with a symmetric $\Omega^{(\ell)}$, LOCATER employs a fully parallelizable and distributable trace calculation routine that obtains both η_1 and η_2 with fewer than $7(q+1)n^2 + 3n(q+1)^2$ FLOPs (Section 6.6). Recall that q is the number of columns in our background covariate matrix A . ~~See Section 6.6 for further details.~~

4.8 Combining p-values: MSSE

The classic Fisher method [44] of combining a vector of p-values $U = (U_1, \dots, U_p)^\top$ is based on comparing the test statistic

$$\mathcal{F}(U) = \sum_{i=1}^p -\log(U_i) \tag{12}$$

to its null distribution: a gamma distribution with scale parameter 1 and shape parameter p , whose survival function we denote by G_p . Since it takes the form of simple sum, \mathcal{F} implicitly considers all possible alternatives: any combination of the p-values may be non-null. However, in applications such as LOCATER, where we are only interested in certain subsets of alternatives, this can lead to an unnecessary loss in power.

The aim of LOCATER is to enhance SMT signals by leveraging allelic heterogeneity. In a genomic region with allelic heterogeneity, we expect the most significant LOCATER signal to arise at a target locus where the core marker at that locus tags one of the causal alleles. Thus, we are only interested in testing combinations of p-values where p_{SMT} is non-null. This in effect reduces the number of combinations of non-null p-value alternatives we want to test, which we can achieve by using

$$\min(G_1^{-1}(\mathcal{F}(p_{\text{SMT}})), G_2^{-1}(\mathcal{F}(p_{\text{SMT}}, p_{\text{SD}})), G_2^{-1}(\mathcal{F}(p_{\text{SMT}}, p_{\text{Q}})), G_1^{-3}(\mathcal{F}(p_{\text{SMT}}, p_{\text{SD}}, p_{\text{Q}}))) \tag{13}$$

as our test statistic. **E30**► The null distribution of this test statistic ~~can be is~~ easily pre-calculated via simulation. ~~Using these simulations, we summarize the~~ Based on 100 million samples, we verified that the null distribution was very smooth and rapidly converged to an exponential tail [45]. Hence we summarized the body of the null distribution via a monotone cubic spline and fit the exponential tail ~~of the null distribution~~ to obtain a rapidly computable test. **E30 Reply**

~~Declaration of Interests~~The authors declare no competing interests.

Data and Code Availability

E42► Our `locator` R package, alongside installation instructions and the source code, is available on GitHub at <https://github.com/ryanchrist/locator>. Package documentation and a simple example vignette/article is available on the package website <https://ryanchrist.github.io/locator/>. Functions documented in the `locator` package provide an API exposing all of our testing subroutines. These functions accept generic matrices Y , A , $G^{(\ell)}$, $X^{(\ell)}$ and $\Omega^{(\ell)}$. Porting these matrices into R, as base matrices or sparse matrices, is sufficient to run the testing routines implemented in `locator`. While this requires some work, it allows users hoping to deploy `locator` in conjunction with other ancestry inference engines the ability to do so without re-implementing our testing procedures. The Simple LOCATER Example vignette on the `locator` package website provides further guidance on how to do this: https://ryanchrist.github.io/locator/articles/simple_gwas_example.html. Scripts required for replicating our results are available here: https://github.com/ryanchrist/locator_paper_scripts. They may be run under our public Docker image, which can be found here <https://hub.docker.com/r/rchrist7/mini-shark>. This image contains an installation of `msprime` and other python utilities used to run our haplotype simulations. ◀**E42**

Reply

Acknowledgments

The authors would like to thank Nathan Stitzel, Chris Holmes, and Chris Spencer for helpful advice and discussions.

Funding

RC and XW were supported by NIH grants R01HG013371-01 and UM1HG008853 to IH. LJMA was partially supported by the EPSRC research grant "PINCODE", reference EP/X028100/1, and UKRI grant, "OCEAN", reference EP/Y014650/1. DS was supported by BBSRC research grant BB/S001824/1. For the purpose of Open Access, the authors have applied a CC BY public copyright license to any Author Accepted Manuscript version arising from this submission. ~~The authors would like to thank Nathan Stitzel, Chris Holmes, and Chris Spencer for helpful advice and discussions.~~

Author Contributions

~~RC conceived the methodology with advice from DS, LJMA and IMH and iterative testing and experimentation
by XW. RC and LJMA implemented the core software. RC and XW designed the user interface. RC and
IMH conceived the experiments. RC performed the analysis and visualization. RC, XW, and IMH wrote
the paper with input from DS and LJMA.~~

~~Data and Code Availability~~

~~During peer review, our LOCATER R package is available at private link alongside source code, installation
instructions, a simple vignette, and instructions with scripts for replicating our power simulations. Upon
acceptance all of this content (source code, instructions, and scripts) will be made public in-~~

Declaration of Interests

The authors declare no competing interests.

References

- [1] Ryan Christ, Ira Hall, and David Steinsaltz. Stable distillation and high-dimensional hypothesis testing. *arXiv:2212.12539v3*, 2024.
- [2] Leo Speidel, Marie Forest, Sinan Shi, and Simon R Myers. A method for genome-wide genealogy estimation for thousands of samples. *Nature Genetics*, 51(9):1321–1329, 2019.
- [3] Brian C Zhang, Arjun Biddanda, Árni Freyr Gunnarsson, Fergus Cooper, and Pier Francesco Palamara. Biobank-scale inference of ancestral recombination graphs enables genealogical analysis of complex traits. *Nature Genetics*, 55(5):768–776, 2023.
- [4] Árni Freyr Gunnarsson, Jiazheng Zhu, Brian C Zhang, Zoi Tsangalidou, Alex Allmont, and Pier Francesco Palamara. A scalable approach for genome-wide inference of ancestral recombination graphs. *bioRxiv*, pages 2024–08, 2024. <https://www.biorxiv.org/content/10.1101/2024.08.31.610248>.
- [5] Xinxin Wang, Ryan Christ, Erica Young, Chul Joo Kang, Indrani Das, Edward A Belter Jr, Markku Laakso, Louis JM Aslett, David Steinsaltz, Nathan O Stitzel, et al. Genealogy based trait association with locater boosts power at loci with allelic heterogeneity. *medRxiv*, pages 2024–11, 2024.
- [6] Xihao Li, Zilin Li, Hufeng Zhou, Sheila M Gaynor, Yaowu Liu, Han Chen, Ryan Sun, Rounak Dey, Donna K Arnett, Stella Aslibekyan, et al. Dynamic incorporation of multiple in silico functional

1st Revision - Authors' Response to Reviewers: April 25, 2025

- annotations empowers rare variant association analysis of large whole-genome sequencing studies at scale. *Nature Genetics*, 52(9):969–983, 2020. 982
- [7] Leo Speidel, Marina Silva, Thomas Booth, Ben Raffield, Kyriaki Anastasiadou, Christopher Barrington, Anders Götherström, Peter Heather, and Pontus Skoglund. High-resolution genomic history of early medieval europe. *Nature*, 637(8044):118–126, 2025. 983
- [8] Louis J.M. Aslett and Ryan R. Christ. kalis: a modern implementation of the Li & Stephens model for local ancestry inference in R. *BMC Bioinformatics*, 25(86), 2024. 984
- [9] Ryan Christ, Ira Hall, and David Steinsaltz. The Rényi outlier test, 2024. <https://arxiv.org/abs/2411.13542>. 985
- [10] Quanli Wang, Ryan S Dhindsa, Keren Carss, Andrew R Harper, Abhishek Nag, Ioanna Tachmazidou, Dimitrios Vitsios, Sri VV Deevi, Alex Mackay, Daniel Muthas, et al. Rare variant contribution to human disease in 281,104 UK Biobank exomes. *Nature*, 597(7877):527–532, 2021. 986
- [11] Michael C Wu, Seunggeun Lee, Tianxi Cai, Yun Li, Michael Boehnke, and Xihong Lin. Rare-variant association testing for sequencing data with the sequence kernel association test. *The American Journal of Human Genetics*, 89(1):82–93, 2011. 987
- [12] Seunggeun Lee, Gonçalo R Abecasis, Michael Boehnke, and Xihong Lin. Rare-variant association analysis: study designs and statistical tests. *The American Journal of Human Genetics*, 95(1):5–23, 2014. 988
- [13] Farhad Hormozdiari, Anthony Zhu, Gleb Kichaev, Chelsea J-T Ju, Ayellet V Segrè, Jong Wha Joo, Hyejung Won, Sriram Sankararaman, Bogdan Pasaniuc, Sagiv Shifman, et al. Widespread allelic heterogeneity in complex traits. *The American Journal of Human Genetics*, 100(5):789–802, 2017. 989
- [14] GTEx Consortium. The GTEx consortium atlas of genetic regulatory effects across human tissues. *Science*, 369(6509):1318–1330, 2020. 1000
- [15] Nathan S Abell, Marianne K DeGorter, Michael J Gloudemans, Emily Greenwald, Kevin S Smith, Zihuai He, and Stephen B Montgomery. Multiple causal variants underlie genetic associations in humans. *Science*, 375(6586):1247–1254, 2022. 1001
- [16] Yukihide Momozawa and Keijiro Mizukami. Unique roles of rare variants in the genetics of complex diseases in humans. *Journal of Human Genetics*, 66(1):11–23, 2021. 1002
- [17] Elizabeth T Cirulli, Simon White, Robert W Read, Gai Elhanan, William J Metcalf, Francisco Tanudjaja, Donna M Fath, Efren Sandoval, Magnus Isaksson, Karen A Schlauch, et al. Genome-wide rare variant 1003

1st Revision - Authors' Response to Reviewers: April 25, 2025

- analysis for thousands of phenotypes in over 70,000 exomes from two cohorts. *Nature Communications*, 11(1):542, 2020. 1012
- [18] Mark J Minichiello and Richard Durbin. Mapping trait loci by use of inferred ancestral recombination graphs. *The American Journal of Human Genetics*, 79(5):910–922, 2006. 1013
- [19] Katherine L Thompson and Laura S Kubatko. Using ancestral information to detect and localize quantitative trait loci in genome-wide association studies. *BMC Bioinformatics*, 14:1–10, 2013. 1016
- [20] Sebastian Zöllner and Jonathan K Pritchard. Coalescent-based association mapping and fine mapping of complex trait loci. *Genetics*, 169(2):1071–1092, 2005. 1018
- [21] Sharon R Browning and Elizabeth A Thompson. Detecting rare variant associations by identity-by-descent mapping in case-control studies. *Genetics*, 190(4):1521–1531, 2012. 1020
- [22] Jerome Kelleher, Yan Wong, Anthony W. Wohns, Chaimaa Fadil, Patrick K. Albers, and Gil McVean. Inferring whole-genome histories in large population datasets. *Nature Genetics*, 51(9):1330–1338, September 2019. 1022
- [23] Wen-Wei Liao, Mobin Asri, Jana Ebler, Daniel Doerr, Marina Haukness, Glenn Hickey, Shuangjia Lu, Julian K. Lucas, Jean Monlong, Haley J. Abel, Silvia Buonaiuto, Xian H. Chang, Haoyu Cheng, Justin Chu, Vincenza Colonna, Jordan M. Eizenga, Xiaowen Feng, Christian Fischer, Robert S. Fulton, Shilpa Garg, Cristian Groza, Andrea Guarracino, William T. Harvey, Simon Heumos, Kerstin Howe, Miten Jain, Tsung-Yu Lu, Charles Markello, Fergal J. Martin, Matthew W. Mitchell, Katherine M. Munson, Moses Njagi Mwaniki, Adam M. Novak, Hugh E. Olsen, Trevor Pesout, David Porubsky, Pjotr Prins, Jonas A. Sibbesen, Jouni Sirén, Chad Tomlinson, Flavia Villani, Mitchell R. Vollger, Lucinda L. Antonacci-Fulton, Gunjan Baid, Carl A. Baker, Anastasiya Belyaeva, Konstantinos Billis, Andrew Carroll, Pi-Chuan Chang, Sarah Cody, Daniel E. Cook, Robert M. Cook-Deegan, Omar E. Cornejo, Mark Diekhans, Peter Ebert, Susan Fairley, Olivier Fedrigo, Adam L. Felsenfeld, Giulio Formenti, Adam Frankish, Yan Gao, Nanibaa’A. Garrison, Carlos Garcia Giron, Richard E. Green, Leanne Haggerty, Kendra Hoekzema, Thibaut Hourlier, Hanlee P. Ji, Eimear E. Kenny, Barbara A. Koenig, Alexey Kolesnikov, Jan O. Korb, Jennifer Kordosky, Sergey Koren, HoJoon Lee, Alexandra P. Lewis, Hugo Magalhães, Santiago Marco-Sola, Pierre Marijon, Ann McCartney, Jennifer McDaniel, Jacquelyn Mountcastle, Maria Nattestad, Sergey Nurk, Nathan D. Olson, Alice B. Popejoy, Daniela Puiu, Mikko Rautiainen, Allison A. Regier, Arang Rhie, Samuel Sacco, Ashley D. Sanders, Valerie A. Schneider, Baergen I. Schultz, Kishwar Shafin, Michael W. Smith, Heidi J. Sofia, Ahmad N. Abou Tayoun, Françoise Thibaud-Nissen, Francesca Floriana Tricomi, Justin Wagner, Brian Walenz, Jonathan M. D. Wood, Aleksey V. Zimin, Guillaume Bourque, Mark J. P. Chaisson, Paul Flicek, Adam M. Phillippy, Justin M. 1025

1st Revision - Authors' Response to Reviewers: April 25, 2025

- Zook, Evan E. Eichler, David Haussler, Ting Wang, Erich D. Jarvis, Karen H. Miga, Erik Garrison, Tobias Marschall, Ira M. Hall, Heng Li, and Benedict Paten. A draft human pangenome reference. *Nature*, 617(7960):312–324, 2023. 1044–1046
- [24] Vivian Link, Joshua G Schraiber, Caoqi Fan, Bryan Dinh, Nicholas Mancuso, Charleston WK Chiang, and Michael D Edge. Tree-based QTL mapping with expected local genetic relatedness matrices. *The American Journal of Human Genetics*, 110(12):2077–2091, 2023. 1047–1049
- [25] Jiazheng Zhu, Georgios Kalantzis, Ali Pazokitoroudi, Árni Freyr Gunnarsson, Hrushikesh Loya, Han Chen, Sriram Sankararaman, and Pier Francesco Palamara. Fast variance component analysis using large-scale ancestral recombination graphs. *bioRxiv*, pages 2024–08, 2024. www.biorxiv.org/content/10.1101/2024.08.31.610262. 1050–1053
- [26] Ruoyi Cai and Sharon R Browning. Identity-by-descent mapping using multi-individual ibd with genome-wide multiple testing adjustment. *bioRxiv*, pages 2025–01, 2025. 1054–1055
- [27] Pierrick Wainschtein, Deepti Jain, Zhili Zheng, L Adrienne Cupples, Aladdin H Shadyab, Barbara McKnight, Benjamin M Shoemaker, Braxton D Mitchell, Bruce M Psaty, Charles Kooperberg, et al. Assessing the contribution of rare variants to complex trait heritability from whole-genome sequence data. *Nature Genetics*, 54(3):263–273, 2022. 1056–1059
- [28] Matthew T Maurano, Richard Humbert, Eric Rynes, Robert E Thurman, Eric Haugen, Hao Wang, Alex P Reynolds, Richard Sandstrom, Hongzhu Qu, Jennifer Brody, et al. Systematic localization of common disease-associated variation in regulatory DNA. *Science*, 337(6099):1190–1195, 2012. 1060–1062
- [29] Hakhamanesh Mostafavi, Jeffrey P. Spence, Sahin Naqvi, and Jonathan K. Pritchard. Limited overlap of eQTLs and GWAS hits due to systematic differences in discovery. *bioRxiv*, 2022. 1063–1064
- [30] Ian Barnett, Rajarshi Mukherjee, and Xihong Lin. The generalized higher criticism for testing SNP-set effects in genetic association studies. *Journal of the American Statistical Association*, 112(517):64–76, 2017. 1065–1067
- [31] R Core Team. *R: A Language and Environment for Statistical Computing*. R Foundation for Statistical Computing, Vienna, Austria, 2024. 1068–1069
- [32] Yun Deng, Rasmus Nielsen, and Yun S. Song. Robust and accurate Bayesian inference of genome-wide genealogies for large samples. *bioRxiv*, 2024. <https://www.biorxiv.org/content/early/2024/03/16/2024.03.16.585351>. 1070–1072
- [33] Regev Schweiger and Richard Durbin. Ultrafast genome-wide inference of pairwise coalescence times. *Genome Research*, 33(7):1023–1031, 2023. 1073–1074

1st Revision - Authors' Response to Reviewers: April 25, 2025

- [34] Jotun Hein, Mikkel H. Schierup, and Carsten Wiuf. *Gene Genealogies, Variation and Evolution: A Primer in Coalescent Theory*. Oxford University Press, 2004. 1075
1076
- [35] Na Li and Matthew Stephens. Modeling linkage disequilibrium and identifying recombination hotspots using single-nucleotide polymorphism data. *Genetics*, 165(4):2213–2233, 2003. 1077
1078
- [36] Yun S. Song. Na Li and Matthew Stephens on modeling linkage disequilibrium. *Genetics*, 203(3):1005–1006, 2016. 1079
1080
- [37] Lawrence R Rabiner. A tutorial on hidden Markov models and selected applications in speech recognition. *Proceedings of the IEEE*, 77(2):257–286, 1989. 1081
1082
- [38] Caoqi Fan, Nicholas Mancuso, and Charleston WK Chiang. A genealogical estimate of genetic relationships. *The American Journal of Human Genetics*, 109(5):812–824, 2022. 1083
1084
- [39] Franklin E Satterthwaite. An approximate distribution of estimates of variance components. *Biometrics Bulletin*, 2(6):110–114, 1946. 1085
1086
- [40] Thomas Lumley, Jennifer Brody, Gina Peloso, Alanna Morrison, and Kenneth Rice. FastSKAT: Sequence kernel association tests for very large sets of markers. *Genetic Epidemiology*, 42(6):516–527, 2018. 1087
1088
- [41] Bernie Devlin and Kathryn Roeder. Genomic control for association studies. *Biometrics*, 55(4):997–1004, 1999. 1089
1090
- [42] Ryan R. Christ and Louis J. M. Aslett. *QForm: Fast, safe CDF and PDF estimation and bounding for generalized chi-square random variables and quadratic forms*, 2024. <https://ryanchrist.r-universe.dev/QForm>. 1091
1092
1093
- [43] Jerome Kelleher, Alison M Etheridge, and Gilean McVean. Efficient coalescent simulation and genealogical analysis for large sample sizes. *PLoS Computational Biology*, 12(5):e1004842, 2016. 1094
1095
- [44] Frederick Mosteller and R.A. Fisher. Questions and answers. # 14. *The American Statistician*, 2(5):30–31, 1948. 1096
1097
- [45] August A Balkema and Laurens De Haan. Residual life time at great age. *The Annals of probability*, 2(5):792–804, 1974. 1098
1099
- [46] Andrew C Berry. The accuracy of the gaussian approximation to the sum of independent variates. *Transactions of the American Mathematical Society*, 49(1):122–136, 1941. 1100
1101
- [47] Carl-Gustav Esseen. On the liapounoff limit of error in the theory of probability. *Arkiv för Matematik, Astronomi och Fysik.*, A28:1–19, 1942. 1102
1103

1st Revision - Authors' Response to Reviewers: April 25, 2025

- [48] tskit Developers. Demography. <https://msprime.readthedocs.io/en/stable/tutorial.html#demography>, 1104
2020. Accessed: 2020-07-15. 1105
- [49] Ryan N Gutenkunst, Ryan D Hernandez, Scott H Williamson, and Carlos D Bustamante. Inferring the 1106
joint demographic history of multiple populations from multidimensional SNP frequency data. *PLoS* 1107
Genetics, 5(10):e1000695, 2009. 1108
- [50] Dominic Nelson, Jerome Kelleher, Aaron P Ragsdale, Claudia Moreau, Gil McVean, and Simon Gravel. 1109
Accounting for long-range correlations in genome-wide simulations of large cohorts. *PLoS Genetics*, 1110
16(5):e1008619, 2020. 1111
- [51] Jeffrey P Spence and Yun S Song. Inference and analysis of population-specific fine-scale recombination 1112
maps across 26 diverse human populations. *Science Advances*, 5(10):eaaw9206, 2019. 1113

5 Supplementary Figures

1114

5.1 Checkpointing

1115

Figure 5: **Computational Cost Using Optimal Checkpointing.** ~~Here we~~ We consider the total number of variants the forward algorithm needs to iterate over (the total distance D it has to cover) in order to visit all loci on a chromosome of length L using our optimal checkpointing strategy. ~~Here we~~ We plot the $\log_{10}(D/L)$ as a function of the number of checkpoints, C , available for chromosomes of three potential lengths indicated by the legend: $L \in \{10^4, 10^5, 10^6\}$. The dotted gray horizontal lines highlight that we are able to reduce D to under $100L$ with only 3 checkpoints and nearly $10L$ with only 10 checkpoints in all cases.

5.2 Null Simulation Q-Q Plots

Figure 6: **SMT Subtest Q-Q plot.** Q-Q plot of samples of $-\log_{10} \left(p_{\text{SMT}}^{(\ell)} \right)$ based on 1 million independent phenotype vectors simulated under the null hypothesis.

Figure 7: **SD Subtest Q-Q plot.** Q-Q plot of samples of $-\log_{10} \left(p_{\text{SD}}^{(\ell)} \right)$ based on 1 million independent phenotype vectors simulated under the null hypothesis.

Figure 8: **QForm Subtest Q-Q plot.** Q-Q plot of samples of $-\log_{10}(p_Q^{(\ell)})$ based on 1 million independent phenotype vectors simulated under the null hypothesis.

Figure 9: **LOCATER Combined Q-Q plot.** Q-Q plot of samples of $-\log_{10}(p_C^{(\ell)})$ based on 1 million independent phenotype vectors simulated under the null hypothesis.

5.3 Power, Source, & Localization Curves

Figure 10: Power comparison between SMT and LOCATER in simulations where there are 3 causal variants, all observed.

Figure 11: Average proportion of signal driven by each sub-test within LOCATER in simulations where there are 3 causal variants, all observed. For every simulation, the proportion of signal attributed to the *SMT* part of LOCATER (solid lines) defined as $\log(p_{\text{SMT}}^{\ell^*}) / \log(p_{\text{SMT}}^{\ell^*} p_{\text{SD}}^{\ell^*} p_{\text{Q}}^{\ell^*})$ where ℓ^* is the locus ℓ that achieves the largest combined LOCATER p-value. The proportion of signal attributed to *SD* (dashed lines) or quadratic form testing, abbreviated as *Q* (dotted lines), is defined analogously.

Figure 12: Width of confidence intervals (CI), centered on the position with the largest signal, that cover the midpoint of the causal region in 80% of simulations where there are 3 causal variants, all observed. In simulations where multiple variants are tied to have the largest association signal, we take the distance to the midpoint of the causal region to be the average distance from each of the tied variants.

Figure 13: Power comparison between SMT and LOCATER in simulations where there are 3 causal variants, all hidden.

Figure 14: Average proportion of signal driven by each sub-test within LOCATER in simulations where there are 3 causal variants, all hidden. For every simulation, the proportion of signal attributed to the *SMT* part of LOCATER (solid lines) defined as $\log(p_{SMT}^{\ell^*}) / \log(p_{SMT}^{\ell^*} p_{SD}^{\ell^*} p_Q^{\ell^*})$ where ℓ^* is the locus ℓ that achieves the largest combined LOCATER p-value. The proportion of signal attributed to *SD* (dashed lines) or quadratic form testing, abbreviated as *Q* (dotted lines), is defined analogously.

Figure 15: Width of confidence intervals (CI), centered on the position with the largest signal, that cover the midpoint of the causal region in 80% of simulations where there are 3 causal variants, all hidden. In simulations where multiple variants are tied to have the largest association signal, we take the distance to the midpoint of the causal region to be the average distance from each of the tied variants.

Figure 16: Power comparison between SMT and LOCATER in simulations where there are 9 causal variants, all observed.

Figure 17: Average proportion of signal driven by each sub-test within LOCATER in simulations where there are 9 causal variants, all observed. For every simulation, the proportion of signal attributed to the *SMT* part of LOCATER (solid lines) defined as $\log(p_{\text{SMT}}^{\ell^*}) / \log(p_{\text{SMT}}^{\ell^*} p_{\text{SD}}^{\ell^*} p_{\text{Q}}^{\ell^*})$ where ℓ^* is the locus ℓ that achieves the largest combined LOCATER p-value. The proportion of signal attributed to *SD* (dashed lines) or quadratic form testing, abbreviated as *Q* (dotted lines), is defined analogously.

Figure 18: Width of confidence intervals (CI), centered on the position with the largest signal, that cover the midpoint of the causal region in 80% of simulations where there are 9 causal variants, all observed. In simulations where multiple variants are tied to have the largest association signal, we take the distance to the midpoint of the causal region to be the average distance from each of the tied variants.

Figure 19: Power comparison between SMT and LOCATER in simulations where there are 9 causal variants, all hidden.

Figure 20: Average proportion of signal driven by each sub-test within LOCATER in simulations where there are 9 causal variants, all hidden. For every simulation, the proportion of signal attributed to the *SMT* part of LOCATER (solid lines) defined as $\log(p_{SMT}^{\ell^*}) / \log(p_{SMT}^{\ell^*} p_{SD}^{\ell^*} p_Q^{\ell^*})$ where ℓ^* is the locus ℓ that achieves the largest combined LOCATER p-value. The proportion of signal attributed to *SD* (dashed lines) or quadratic form testing, abbreviated as *Q* (dotted lines), is defined analogously.

Figure 21: Width of confidence intervals (CI), centered on the position with the largest signal, that cover the midpoint of the causal region in 80% of simulations where there are 9 causal variants, all hidden. In simulations where multiple variants are tied to have the largest association signal, we take the distance to the midpoint of the causal region to be the average distance from each of the tied variants.

Figure 22: Power comparison between SMT and LOCATER in simulations where there are 15 causal variants, all observed.

Figure 23: Average proportion of signal driven by each sub-test within LOCATER in simulations where there are 15 causal variants, all observed. For every simulation, the proportion of signal attributed to the *SMT* part of LOCATER (solid lines) defined as $\log(p_{\text{SMT}}^{\ell^*}) / \log(p_{\text{SMT}}^{\ell^*} p_{\text{SD}}^{\ell^*} p_{\text{Q}}^{\ell^*})$ where ℓ^* is the locus ℓ that achieves the largest combined LOCATER p-value. The proportion of signal attributed to *SD* (dashed lines) or quadratic form testing, abbreviated as *Q* (dotted lines), is defined analogously.

Figure 24: Width of confidence intervals (CI), centered on the position with the largest signal, that cover the midpoint of the causal region in 80% of simulations where there are 15 causal variants, all observed. In simulations where multiple variants are tied to have the largest association signal, we take the distance to the midpoint of the causal region to be the average distance from each of the tied variants.

Figure 25: Power comparison between SMT and LOCATER in simulations where there are 15 causal variants, all hidden.

Figure 26: Average proportion of signal driven by each sub-test within LOCATER in simulations where there are 15 causal variants, all hidden. For every simulation, the proportion of signal attributed to the *SMT* part of LOCATER (solid lines) defined as $\log(p_{\text{SMT}}^{\ell^*}) / \log(p_{\text{SMT}}^{\ell^*} p_{\text{SD}}^{\ell^*} p_{\text{Q}}^{\ell^*})$ where ℓ^* is the locus ℓ that achieves the largest combined LOCATER p-value. The proportion of signal attributed to *SD* (dashed lines) or quadratic form testing, abbreviated as *Q* (dotted lines), is defined analogously.

Figure 27: Width of confidence intervals (CI), centered on the position with the largest signal, that cover the midpoint of the causal region in 80% of simulations where there are 15 causal variants, all hidden. In simulations where multiple variants are tied to have the largest association signal, we take the distance to the midpoint of the causal region to be the average distance from each of the tied variants.

5.4 Power Curves: 100 kb Causal Window

Figure 28: Power comparison between SMT and LOCATER in simulations where there are 9 causal variants within a 100 kb causal window, all observed. Compare to Figure 16.

Figure 29: Power comparison between SMT and LOCATER in simulations where there are 9 causal variants within a 100 kb causal window, all hidden. Compare to Figure 19.

6 Supplementary Methods 1119

6.1 Running `kalis` 1120

As described in [35, 2, 8], the LS model requires a pre-defined recombination map and two model parameters: μ governing the probability of mutations and N_e recombination events. Throughout this paper, we provided `kalis` with the true recombination map under which haplotypes were simulated. Throughout we set $\mu = 10^{-4}$ and $N_e = 10^{-16}$.

6.2 Connecting our Generalized eGRM to the standard eGRM 1125

Here we explain the connection between the generalized eGRM matrix defined in Equation (6) to the standard eGRM, as presented in Equation 1 in [24]. For clarity, throughout this section we will assume that we are considering a particular target locus ℓ and suppress indexing objects by ℓ .

Let n be the number of samples in a genetic dataset. Let

$$\mathcal{H}_i = \{j \in \mathbb{N} : \text{haplotype } j \text{ belongs to sample } i\} \quad (14)$$

and $\Delta_i = |\mathcal{H}_i|$ be the ploidy of sample i so that $N = \sum_i \Delta_i$ is the total number of haplotypes in the sample. Let $H \in \{0, 1\}^{N \times p}$ be the haplotype matrix encoding the carriers of p in a genomic region centered on locus ℓ or on an inferred local ancestral tree at ℓ as in [24]. Let $f_k = \sum_{i=1}^n H_{ik}$ be the allele frequency of the k^{th} variant. Given a set of weights $w \in \mathbb{R}_{\geq 0}^p$ which may be based on inferred clade probabilities or otherwise, we start with a weighted haplotype relatedness matrix

$$\Psi' = \sum_{k=1}^p \frac{w_k}{f_k} H_{\cdot k} H_{\cdot k}^\top. \quad (15)$$

We can write this more compactly as $\Psi' = \tilde{H} \tilde{H}^\top$ where $\tilde{H} = H \text{diag} \left(\sqrt{w_k / f_k} \right)$. Assuming an additive model, we can collapse this weighted haplotype relatedness matrix to a weighted genotype relatedness matrix

$$\Omega'_{ij} = \sum_{k \in \mathcal{H}_i} \sum_{l \in \mathcal{H}_j} \Psi'_{kl}. \quad (16)$$

For simplicity, from here forward we will assume that (1) the ploidy of each sample Δ_i is the same constant ploidy Δ for all samples i and (2) the rows and columns of Ψ' are permuted such that haplotypes from the same sample are grouped together. This allows us to more succinctly write $\Omega' = B_{n,\Delta}^\top \Psi' B_{n,\Delta}$ where

$B_{n,\Delta} = I_{n \times n} \otimes \mathbf{1}_\Delta$ and \otimes is the Kronecker product. This gives us

1141

$$\begin{aligned} \Omega' &= B_{n,\Delta}^\top \left(\sum_{k=1}^p \frac{w_k}{f_k} H_{\cdot k} H_{\cdot k}^\top \right) B_{n,\Delta} \\ &= \sum_{k=1}^p \frac{w_k}{f_k} B_{n,\Delta}^\top H_{\cdot k} H_{\cdot k}^\top B_{n,\Delta} \\ &= \sum_{k=1}^p \frac{w_k}{f_k} G_{\cdot k} G_{\cdot k}^\top \\ &= G \text{diag}(w_k/f_k) G^\top \end{aligned}$$

where $G \{0, 1, 2\}^{n \times p}$ is the genotype matrix corresponding to the p variants on the local tree. Now if we consider a model without any background covariates besides an intercept, we have $P = I - \frac{1}{n} \mathbf{1}\mathbf{1}^\top$.

1142

1143

$$\begin{aligned} P\Omega'P &= PG \text{diag}(w_k/f_k) G^\top P \\ &= \sum_{k=1}^p \frac{w_k}{f_k} PG_{\cdot k} (PG_{\cdot k})^\top \end{aligned} \tag{17}$$

The standard GRM assumes that each variant is in Hardy–Weinberg equilibrium. So in order to compare Equation (17) to the standard GRM, let us also assume that each of our p variants is in perfect Hardy–Weinberg equilibrium. That means each centered genotype vector $PG_{\cdot k} = G_{\cdot k} - 2f_k \mathbf{1}$, allowing us to write

1144

1145

1146

1147

$$\begin{aligned} (P\Omega'P)_{ij} \Big|_{HW} &= \sum_{k=1}^p w_k \frac{(G_{ik} - 2f_k)(G_{jk} - 2f_k)}{f_k} \\ &= \frac{1}{p} \sum_{k=1}^p w_k \eta(f_k) \frac{(G_{ik} - 2f_k)(G_{jk} - 2f_k)}{2f_k(1 - f_k)} \end{aligned} \tag{18}$$

where $\eta(f_k) = 2p(1 - f_k)$. The standard GRM defined in Equation 1 of [24] is just a simple special case of Equation (18) where $\eta(f_k) = 1$ for all arguments f_k and $w_k = 1$ for all variants. The eGRM allows for non-equal weights w_k . Some simple algebra shows that our choice of a $\frac{1}{k}$ in our definition of ψ in Equation 5 corresponds to $\eta(f_k) = 2p(1 - f_k)$. We chose this weighting in order to upweight variants that have low derived allele frequency. Under mild selection pressure, we expect very common derived alleles to have smaller phenotypic effects. With very minimal modification of our current implementation, any weighting function of the derived allele frequency η may be used. We plan to make such a general weighting function available in future versions of LOCATER.

1148

1149

1150

1151

1152

1153

1154

1155

Having rewritten the eGRM in terms of Ψ' , let us connect Ψ' to Ψ from Equation (5), which we use

1156

in our generalized eGRM construction. Consider the special case where we have a chromosome without recombination so that our genetic distance matrix $d^{(\ell)}$ reflects a single marginal tree, making all clade calls are aligned. If we also set each g_k to the identity function and assume consistency across columns of our distance matrix so that $d_{(k+1)j}^{(\ell)} - d_{kj}^{(\ell)} = w_k$ for all haplotypes j carrying a variant of frequency k , then Ψ as defined in Equation (5) would be symmetric and proportional to the the weighted haplotype matrix Ψ' as defined in Equation (15). As a consequence, the similarity matrix we would obtain $B_{n,d}^\top \Psi B_{n,d}$ would equal $\Omega' = B_{n,d}^\top \Psi' B_{n,d}$. However, due to recombination, the distance matrix Ψ is not symmetric and the distances across columns may not be consistent with a single tree structure. Rather than attempting to synchronize the distances by clustering them into a tree structure, we trust the LS model, taking the distances as representing a weighted convolution of nearby tree structures.

6.3 Derivation of Robust Tail Approximation for Quadratic Forms: Accounting for Unmeasured Confounders Optimal Checkpointing Scheduler

Consider the presence of some unobserved variable $B \in \mathbb{R}^n$. Without loss of generality, assume $\|B\|_2 = \sqrt{n-q}$ and $PB = B$. For some scalar γ , we have $Y = A\alpha + \gamma B + \sigma\epsilon$. Then our estimator $\hat{\sigma}^2 = Y^\top PY / (n-q) \xrightarrow{P} \gamma^2 + \sigma^2$. This gives us the null distribution

$$\begin{aligned} \frac{(\gamma^2 + \sigma^2)^{-1} Y^\top PMPY}{\gamma^2 + \sigma^2} &= \frac{(\gamma^2 + \sigma^2)^{-1} \sum_{j=1}^{n-q} \lambda_j (\sigma V_j^\top P\epsilon + \gamma V_j^\top B)^2}{\gamma^2 + \sigma^2} \\ &\sim \frac{\sigma^2}{\gamma^2 + \sigma^2} \sum_{j=1}^{n-q} \lambda_j \left(Z_j + \frac{\gamma}{\sigma} V_j^\top B \right)^2 \\ &= \nu \sum_{j=1}^{n-q} \lambda_j (Z_j + \delta_j)^2 \\ &\sim \nu \sum_{j=1}^{n-q} \lambda_j W_j \end{aligned}$$

Let j index the positions of our L target loci from smallest to largest. We will solve for an optimal checkpointing schedule for the forward algorithm to propagate to target loci sequentially in reverse order, from $j = L$ to $j = 1$. We will use $j = 0$ to index the HMM prior hidden state probabilities used to initialize the forward algorithm. Informally, for each subsequent target locus, a checkpointing schedule states the checkpoint at which the forward algorithm should start and the intervening target loci where it should stop to store any new checkpoints on its way to reach that target locus. To optimize this schedule, we do not need to consider storing checkpoints at variants that are not target loci since any such checkpointing strategy

could be improved by moving those checkpoints to target loci. Thus, our checkpointing schedule will only need to render integers in $1, \dots, L$.

Optimizing the checkpointing schedule is based on recycling the memory used to store checkpoints that are no longer being used. We solve for an optimal checkpointing schedule where the computational cost to propagate the forward algorithm from any position to any other may be arbitrary. The checkpointing routine that minimizes the total computational cost in this setting may be obtained by solving nested optimization problems. While tractable, solving this system of nested optimization problems can be rather time consuming to solve. Therefore, we subsequently consider the special case where the cost of propagating the forward algorithm between consecutive target positions is constant. For example, in a genomics context this may be true if we have equally spaced target loci. In that special case, we explain how the optimal checkpointing schedule can be solved rapidly via dynamic programming.

We start with the general case. Define $g: \mathbb{Z}_{\geq 0}^2 \rightarrow \mathbb{R}_{\geq 0}$ such that $g(s, t)$ is the cost of propagating the forward algorithm directly from target locus s to target locus t , defined to be zero whenever $s = t$. Define $f: \mathbb{Z}_{\geq 0}^3 \rightarrow \mathbb{R}_{\geq 0}$ such that $f(s, t, c)$ is the minimal cost required for the forward algorithm to sequentially visit target loci $t, t-1, t-2, \dots, s$ given c available checkpoints and a known set of forward probabilities (i.e. a checkpoint, or the HMM prior) at s . That is, $f(s, t, c)$ corresponds to the cost of the (unknown) optimal schedule given memory for c checkpoints.

We aim to obtain a checkpoint schedule which enables us to achieve the cost $f(0, T, C)$. This objective is defined by the three following equations. First, if there are no checkpoints available, simply propagating to each target locus sequentially from the initial position is our only option, which comes with cost $f(x, y, 0) = \sum_{j=x}^y g(x, j)$. Second, if we already know the forward probabilities at a given position, there is no computational cost to obtaining them, so $f(x, x, c) = 0$ for any c . Third, we have the following recursive relationship

$$f(s, t, c) = \min_{j \in \mathbb{Z}_{\geq 0}: j \in (s, t]} g(s, j) + f(s, j-1, c) + f(j, t, c-1). \quad (19)$$

where $\nu = \frac{\sigma^2}{\gamma^2 + \sigma^2}$, $\delta_j = \frac{\gamma}{\sigma} V_j^\top B$,

Intuitively, this recursion expresses the idea that solving for an optimal checkpointing schedule with c checkpoints over an interval can be thought of as placing one of the checkpoints at some optimal index to divide the interval so that the upper part is solved with $c-1$ checkpoints; the lower part, c checkpoints. We will write that optimal index, the argument that achieves the minimum of Equation (19) as

$$h(s, t, c) = \underset{j \in \mathbb{Z}_{\geq 0}: j \in (s, t]}{\operatorname{argmin}} g(s, j) + f(s, j - 1, c) + f(j, t, c - 1). \quad (20)$$

Together, these three equations allow us to solve for an optimal checkpoint schedule by transversing a bifurcating tree from right to left, from the leaves toward the root, where each node can be identified with an optimal index h .

While solving this recursion is tractable, it can be rapidly accelerated if we assume that the computational cost of propagating the forward probabilities between adjacent target loci is fixed. In other words, we assume $g(x - 1, x)$ is constant for all target indices $x \in \{1, 2, \dots, L\}$. This makes the computational cost a simple linear function of the distance between indices, yielding the following simplified system of equations.

Analogous to above, we have $\tilde{f}(i, 0) = i(i + 1)/2$ and $\tilde{f}(0, c) = 0$ for all c . Our recursions simplify to

$$\tilde{f}(i, c) = \min_{j \in \mathbb{Z}_{\geq 0}: j \leq i} j + \tilde{f}(j - 1, c) + \tilde{f}(i - j, c - 1), \quad (21)$$

$$\tilde{h}(i, c) = \underset{j \in \mathbb{Z}_{\geq 0}: j \leq i}{\operatorname{argmin}} j + \tilde{f}(j - 1, c) + \tilde{f}(i - j, c - 1). \quad (22)$$

For some pre-specified maximum number of target loci L and maximum number of available checkpoints C , let us define two matrices that we will use as look-up tables for our dynamic program. Let $F \in \mathbb{R}_{\geq 0}^{(L+1) \times (C+1)}$ such that $F_{ij} = \tilde{f}(i - 1, j - 1)$ and each $W_j \sim \chi_1^2(\delta_j^2)$, independent $H \in \mathbb{Z}_{\geq 0}^{(L+1) \times (C+1)}$ such that $H_{ij} = \tilde{h}(i - 1, j - 1)$. Given the nested nature of 21 and 22, we can rapidly solve for each entry of F and H by synchronously iterating over the entries of both matrices in column-major order. Once we have a completed F and H , we can use 22 to read off the appropriate entries in table H to obtain the optimal checkpointing schedule for any problem with up to L target variants and C available checkpoints. Methods for constructing H and reading it to construct a checkpoint schedule are available in `kalis v2`. This implementation covers the slightly more general case than (21) where the cost depends on distance but is independent of locus. This corresponds to a translation invariance assumption that does not cover the most general case in (19).

6.4 Robust Tail Approximation

E18► The rate of tail decay under Satterthwaite approximation, $(2\alpha)^{-1}$, tends to chronically over-estimate the true rate of decay in the tails of R_k , which is $\left(2 \max_{j > k} \lambda_j\right)^{-1}$. In fact, in the positive semi-definite case,

the Hölder Inequality (with $p = 1, q = \infty$) shows that

$$(2\alpha)^{-1} = \left(\frac{2 \sum_{j>k} \lambda_j^2}{\sum_{j>k} \lambda_j} \right)^{-1} = \left(\frac{2 \sum_{j>k} |\lambda_j \lambda_j|}{\sum_{j>k} |\lambda_j|} \right)^{-1} \geq \left(\frac{2 \left(\sum_{j>k} |\lambda_j| \right) \max_{j>k} |\lambda_j|}{\sum_{j>k} |\lambda_j|} \right)^{-1} = \left(2 \max_{j>k} \lambda_j \right)^{-1}. \quad (23)$$

Our proposed approximation \ddot{R}_k addresses this problem by using tails that decay at least as slowly as R_k . However, this alone does not guarantee accurate p-value estimation across the range of quantiles required for LOCATER. Our goal is to estimate $-\log$ p-values given by $-\log(1 - F_{R_k}(x))$ where F_{R_k} is the CDF of R_k . Since R_k has exponential tails, $-\log(1 - F_{R_k}(x))$ is a linear function of x for large x , with a slope determined the rate of decay in the tail and an intercept that is determined by the body of the distribution of R_k . Thus, estimating the body of the distribution of R_k is required for accurately estimating p-values in the tail of the distribution of R_k . The parameter $\nu \in [0, 1]$ captures the proportion of the variance of Y that is not explained by the unobserved variable B and each $(\delta_j^2)_{j=1}^{n-q}$ captures how correlated the clade structure encoded in M is with the unobserved variable B .

~~A typical clade matrix typically~~ We will make use of the observation that the spectrum of a typical $P\Omega^{(\ell)}P$ observed in our genomics application has the following three-part structure. First the spectrum decays over the ~~first leading~~ ten or so eigenvalues ~~and~~. This segment of the spectrum corresponds to large-scale population structure, orthogonal to the columns of A , encoded by $\Omega^{(\ell)}$. Then the spectrum levels off to a plateau ~~of eigenvalues that~~ where the eigenvalues are all roughly of the same magnitude. ~~The initial eigenvalues also tend to explain a small proportion of the variance of the matrix. This behavior is expected given the relative~~ This second segment of the spectrum corresponds to the fine-scale population structure encoded by $\Omega^{(\ell)}$. Most eigenvalues in this second segment of the spectrum are positive with occasional negative eigenvalues interspersed. Eventually this plateau in the spectrum gives way to the third and final segment of the spectrum where the magnitude of the eigenvalues decays to zero.

Our strategy for estimating tail probabilities for $Y^\top P\Omega^{(\ell)}PY$ is to explicitly evaluate the eigenvalues in the first segment of the spectrum, placing them into T_k , so that we can treat R_k as the sum of terms arising from the second segment and third segment.

In regions with high local recombination rates, within hotspots, there are relatively few proximal mutations that are informative about the local genealogy. Thus, the spectrum of $P\Omega^{(\ell)}P$ decays rapidly in these regions. In these cases we can easily evaluate enough eigenvalues so that R_k explains little of the variance of $T_k + R_k$. At these loci, approximating R_k with \ddot{R}_k is more than sufficient to calculate p-values for $T_k + R_k = Y^\top P\Omega^{(\ell)}PY$. Almost any unimodal distribution matching the first and second moments would work for approximating R_k in this high-recombination context, which is achieved by our moment matching

of \ddot{R}_k .

However, at the vast majority of loci along the genome are in between recombination hotspots. There the spectrum of a typical $P\Omega^{(\ell)}P$ tends to decay over the leading ten or so eigenvalues and then level off to a very long plateau of eigenvalues, the vast majority of which are positive. This plateau arises from the abundance of rare variants (small clades) which correspond to small eigenvalues encoding small clades in $\Omega^{(\ell)}$. While eventually this plateau decays to zero, the eigenvalues within the plateau segment of the spectrum make up the vast majority of the total sum of squared eigenvalues. As a consequence, the contribution of R_k to the overall variance of the test statistic $T_k + R_k$ is much larger than the contribution of T_k for any tractable k . Thus, to estimate accurate tail probabilities for $T_k + R_k$, it is imperative that we do not underestimate the tail probabilities of R_k . The role of T_k is primarily to define the rate of decay in the tail. Let k^* denote the transition point in the spectrum to a plateau, defined as follows

$$k_* = \inf \left\{ k \in [n - q]: \left(\sum_{j=1}^k \lambda_j^2 / \sum_{j=1}^{n-q} \lambda_j^2 \right) \geq 0.95 \text{ or } (\lambda_k / \lambda_{k-1})^2 \geq 0.95 \right\}.$$

To do this, we must ensure that we have control of the body of the distribution. When we top- k eigendecompose $P\Omega^{(\ell)}P$ in LOCATER, we increase the number of eigenvalues k we evaluate until we reach the following stopping criteria.

$$k^* = \inf \left\{ k \in [n - q]: \left(\sum_{j=1}^k \lambda_j^2 / \sum_{j=1}^{n-q} \lambda_j^2 \right) \geq 0.95 \text{ or } (\lambda_k / \lambda_{k-1})^2 \geq 0.95 \right\}. \quad (24)$$

In order to ensure reliable estimation of the null distribution, assume that we compute the top $k \geq k^*$ eigenvalues of PMP , so we have

$$T_k = \nu \left(\sum_{j=1}^{k^*} \lambda_j W_j + \sum_{j=k^*+1}^k \lambda_j W_j \right).$$

$k > k^*$ eigenvalues of PMP so that

$$T_k = \sum_{j=1}^{k^*} \lambda_j Z_j^2 + \sum_{j=k^*+1}^k \lambda_j Z_j^2. \quad (25)$$

The first term in our stopping criteria catches the high-recombination case where R_k makes only a small contribution to the distribution of $T_k + R_k$. The second term in our stopping criteria catches the more typical case where the spectrum decays to a long plateau and ensures that we eigendecompose until that

plateau is reached. Once the plateau is reached, all of leading terms with relatively large eigenvalues have been moved into T_k . This leaves R_k consisting of many terms with nearly equal eigenvalues. Since these terms have nearly equal variance, by the Berry–Esseen Theorem [46, 47], the body of R_k will be very close to a Gaussian. Since the Gaussian distribution is determined by its the mean and variance, this means that we can reliably estimate the body of the distribution with an approximate distribution that is also Gaussian in the body (with the appropriate mean and variance). Given the preponderance of positive eigenvalues in the plateau, minimizing $|\mu|$ when selecting (a, b, μ) (Section 6.5), ensures a large value of a for \tilde{R}_k . This guarantees that the body of the distribution of \tilde{R}_k will also be very close to Gaussian. Thus \tilde{R}_k reliably matches the body of the target null distribution. In combination with the robust rate of decay in the right tail \tilde{R}_k , this ensures reliable p-value estimation even for very small tail probabilities. ◀E18 Reply

6.5 Accounting for Inflation Driven by Unmeasured Confounders & Polygenicity

Consider the presence of some unobserved variable $B \in \mathbb{R}^n$. Without loss of generality, assume $\|B\|_2 = \sqrt{n-q}$ and $PB = B$. For some scalar γ , we have $Y = A\alpha + \gamma B + \sigma\epsilon$. Then our estimator $\hat{\sigma}^2 = Y^\top PY / (n-q) \xrightarrow{P} \gamma^2 + \sigma^2$. This gives us the null distribution

$$\begin{aligned} (\gamma^2 + \sigma^2)^{-1} Y^\top PMPY &= (\gamma^2 + \sigma^2)^{-1} \sum_{j=1}^{n-q} \lambda_j (\sigma V_j^\top P\epsilon + \gamma V_j^\top B)^2 \\ &\sim \frac{\sigma^2}{\gamma^2 + \sigma^2} \sum_{j=1}^{n-q} \lambda_j \left(Z_j + \frac{\gamma}{\sigma} V_j^\top B \right)^2 \\ &= \nu \sum_{j=1}^{n-q} \lambda_j (Z_j + \delta_j)^2 \\ &\sim \nu \sum_{j=1}^{n-q} \lambda_j W_j \end{aligned}$$

where $\nu = \frac{\sigma^2}{\gamma^2 + \sigma^2}$, $\delta_j = \frac{\gamma}{\sigma} V_j^\top B$, and each $W_j \sim \chi_1^2(\delta_j^2)$, independent. The parameter $\nu \in [0, 1]$ captures the proportion of the variance of Y that is not explained by the unobserved variable B and each $(\delta_j^2)_{j=1}^{n-q}$ captures how correlated the clade structure encoded in $\Omega^{(\ell)}$ is with the unobserved variable B .

We can gather information about ν by looking at the distribution of this quadratic form test statistic at a grid of loci spaced far apart along the genome. Unfortunately, we cannot attempt to estimate each individual δ_j in a similar way since each specific δ_j depends on the precise j^{th} eigenvector V_j calculated at a given position. However, the spherical symmetry of the multivariate normal distribution helps us tackle this problem.

If all of the eigenvalues, λ_j were equal, our null distribution would have a scaled non-central chi-square

distribution with $n - q$ degrees of freedom and a single non-centrality parameter $\frac{\sum_{j=1}^{n-q} \delta_j^2}{\sigma^2} = \frac{\gamma^2}{\sigma^2} \|V^\top B\|^2 \sum_{j=1}^{n-q} \delta_j^2$.
 This collapse into a single non-centrality parameter means that we would not need to model each individual δ_j^2 , but rather just the average δ_j^2 .

To interpret this average, again, assuming all of the eigenvalues were equal,

$$\sum_{j=1}^{n-q} \delta_j^2 = \frac{\gamma^2}{\sigma^2} \sum_{j=1}^{n-q} (V_j^\top B)^2 \propto B^\top PMPB = B^\top MB. \quad (26)$$

Thus $\sum_{j=1}^{n-q} \delta_j^2$ can be thought as proportional to the quadratic form test statistic we would obtain taking B as our phenotype vector, which is a measure of how correlated B is with the structure encoded in $\Omega^{(\ell)}$. This suggests a second, related interpretation. Recall B and all V_j s are scaled and that the inclusion of an intercept in A ensures that B and each V_j are centered. This means that $\sum_{j=1}^{n-q} \delta_j^2$ is proportional to average correlation between B and each V_j .

$$\sum_{j=1}^{n-q} \delta_j^2 = \frac{\gamma^2}{\sigma^2} \sum_{j=1}^{n-q} (V_j^\top B)^2 = \frac{n\gamma^2}{\sigma^2} \frac{1}{n} \sum_{j=1}^{n-q} \text{Corr}(V_j, B)^2 = \frac{n\gamma^2}{\sigma^2} \overline{\text{Corr}(V_j, B)^2}. \quad (27)$$

This is proportional to the coefficient of multiple correlation (R^2) one would obtain by regressing B onto the eigenvectors V under the ordinary least squares model.

Since we expect that the typical angle correlation between B and each V_j may be different for V_j s capturing common variant structure versus rare variant structure, we replace each δ_j^2 such that

$$W_j \sim \begin{cases} \chi_1^2(\delta_\star^2) & j \leq k^\star \\ \chi_1^2(\delta_\dagger^2) & j > k^\star \end{cases}$$

$$W_j \sim \begin{cases} \chi_1^2(\delta_\star^2) & j \leq k^\star \\ \chi_1^2(\delta_\dagger^2) & j > k^\star. \end{cases}$$

This leaves us with a null distribution with three scalar inflation parameters, $\nu, \delta_\star^2, \delta_\dagger^2 \in \mathbb{R}_{\geq 0}$.

E26► In summary, we have the following interpretations.

- ν is the proportion of variance in Y not explained by unobserved confounders
- γ_\star^2 is proportional to the average correlation between unobserved confounders and the large-scale haplotype structure encoded in $\Omega^{(\ell)}$ (that is not included in A)

- γ_{\dagger}^2 is proportional to the average correlation between unobserved confounders and the fine scale haplotype structure encoded in $\Omega^{(\ell)}$

◀E26 Reply Now we turn to estimating these parameters. We can use $\text{tr}(PMP)$ and $\|PMP\|_{HS}^2$ to calculate the mean and variance of R_k . Using those moments, we approximate R_k with a shifted difference of chi-square random variables. Let

$$\ddot{R}_k = \nu |\lambda_k| \left(X_{a, \delta_{\dagger}^2} - X'_{b, \delta_{\dagger}^2} + \mu (1 + \delta_{\dagger}^2) \right)$$

where $X_{a, \delta_{\dagger}^2} \sim \chi_a^2(a\delta_{\dagger}^2)$ is distributed chi-squared with a degrees of freedom and non-centrality parameter $a\delta_{\dagger}^2$ and independent of $X'_{b, \delta_{\dagger}^2} \sim \chi_b^2(b\delta_{\dagger}^2)$.

We will select parameters a, b, μ so that \ddot{R}_k matches R_k on the first two moments for all values of ν and δ_{\dagger} . We can do this using matrix traces to exactly calculate $\sum_{j=k+1}^{n-q} \lambda_j$ and $\sum_{j=k+1}^{n-q} \lambda_j^2$. If we then set the first moments equal we get

$$\mathbb{E}[R_k] = \nu (1 + \delta_{\dagger}^2) \sum_{j=k+1}^{n-q} \lambda_j = \nu |\lambda_k| (1 + \delta_{\dagger}^2) (a - b + \mu) = \mathbb{E}[\ddot{R}_k],$$

yielding the constraint

$$a - b + \mu = |\lambda_k|^{-1} \sum_{j=k+1}^{n-q} \lambda_j =: C_1.$$

Similarly, setting the variances equal we have

$$\text{Var}[R_k] = 2\nu^2 (1 + 2\delta_{\dagger}^2) \sum_{j=k+1}^{n-q} \lambda_j^2 = 2\nu^2 \lambda_k^2 (1 + 2\delta_{\dagger}^2) (a + b) = \text{Var}[\ddot{R}_k],$$

which yields the constraint,

$$a + b = \lambda_k^{-2} \sum_{j=k+1}^{n-q} \lambda_j^2 =: C_2.$$

With these constraints, we select a, b, μ as follows

1. If $C_1 \in [-C_2, C_2]$, we set $a = (C_2 + C_1)/2$ and then set $b = C_2 - a$ and $\mu = 0$.
2. If $C_1 > C_2$, we set $a = C_2$, $b = 0$ and $\mu = C_1 - C_2$.
3. If $C_1 < -C_2$, we set $b = C_2$, $a = 0$ and $\mu = C_1 + C_2$.

This specification deliberately minimizes the contribution of μ to the mean relative to a and b while respecting our constraints. ~~This yields a~~ As discussed above in Section 6.4, this has the important consequence of ensuring large values of a at loci where the spectrum reaches a plateau (in between recombination hotspots). This large a in turn ensures a reliable p-value estimation at these loci.

Bringing everything together, our final null distribution is

1334

$$\nu \left(\sum_{j=1}^{k^*} \lambda_j W_j (\delta_{\star}^2) + \sum_{j=k^*+1}^k \lambda_j W_j (\delta_{\dagger}^2) + |\lambda_k| \left(X_{a, \delta_{\dagger}^2} - X'_{b, \delta_{\dagger}^2} + \mu (1 + \delta_{\dagger}^2) \right) \right). \quad (28)$$

Critically, this parameterization of the null distribution is expressed in two sets of orthogonal parameters. 1335
 The first set are our spectral parameters $\{\lambda_j\}_{j=1}^k$, a , b , and μ corresponding to the spectrum of PMP . The 1336
 second are our mis-specification parameters ν , δ_{\star}^2 , δ_{\dagger}^2 corresponding to features of the confounding process. 1337
 This means that, given the phenotype vector Y and the spectral parameters at a given locus, we can readily 1338
 recompute the observed p-value for that locus for any set of misspecification parameters. Thus, if we see 1339
 inflation in the Q-Q plot of observed quadratic form p-values for a given phenotype, we can readily adjust 1340
 the mis-specification parameters and recalculate our p-values in order to correct and calibrate the Q-Q plot. 1341

6.6 Fast Trace Calculation

1342

Here we present our trace calculation approach for a *symmetric* matrix $M \in \mathbb{R}^{n \times n}$. Let $B = I_n - QZ$ where 1343
 $Q \in \mathbb{R}^{n \times m}$ and $Z \in \mathbb{R}^{m \times n}$. Our goal is to calculate $\|BMB\|_{HS}^2$ and $\text{diag}(BMB)$ efficiently using a set of 1344
 worker nodes. First we compute $J = ZM \in \mathbb{R}^{m \times n}$ (mn^2 FLOPs). Second we compute $X = Q(JQ) - MQ \in$ 1345
 $\mathbb{R}^{n \times m}$ ($mn^2 + 2nm^2 + nm$ FLOPs). Then map out the relevant sub-blocks of Q, Z, J, X to each worker as 1346
 follows: 1347

$$\begin{aligned} (I - QZ) M (I - QZ) &= M - QZM - MQZ + QZMQZ \\ &= M + XZ - QJ \end{aligned}$$

So,

1348

$$\begin{aligned} \|(I - QZ) M (I - QZ)\|_{HS}^2 &= \sum_{i,j} (M_{ij} + X_i^T Z_{.j} - Q_i^T J_{.j})^2 \\ &= \sum_{i,j} \left(M_{ij} + \sum_{l=1}^m (X_{il} Z_{lj} - Q_{il} J_{lj}) \right)^2 \end{aligned}$$

This accumulation requires a total of $(4m + 1)n^2$ FLOPs. Since $\text{diag}(BMB)_i = M_{ii} + \sum_{l=1}^m (X_{il} Z_{li} - Q_{il} J_{li})$, 1349
 $\text{diag}(BMB)$ is easily obtained as a by-product of this calculation. Taking the sum of those diagonal 1350
 elements to obtain $\text{tr}(BMB)$ requires $n - 1$ FLOPs. Adding the FLOPs together, we obtain a total of 1351
 $(6m + 1)n^2 + 2nm^2 + nm + n - 1$ FLOPs. Since $m \geq 1$, this is upper bounded by $7mn^2 + 3nm^2$. Connecting 1352
 this result back to our notation in Section ??6.5, this means we can calculate our desired traces η_1 and η_2 1353
 with fewer than $7(q + 1)n^2 + 3n(q + 1)^2$ FLOPs where q is the number of columns in our background covariate 1354

matrix A .

If M is distributed in columns across nodes, we can accelerate the above calculation by exporting $W = \begin{bmatrix} X & -Q \end{bmatrix} \in \mathbb{R}^{n \times 2m}$ to every node and the appropriate columns of $\begin{bmatrix} Z \\ J \end{bmatrix} \in \mathbb{R}^{2m \times n}$ to their corresponding nodes.

6.7 Haplotype Simulation

We simulated haplotypes from three 1000 Genomes Populations – Yoruba, Han Chinese, and Central European – using a demographic model adapted from the `msprime` [43] demography tutorial [48], which itself was based on the population parameters related those three populations presented in [49]. We only made one modification to the demographic model specified in [48]: we use the Discrete Time Wright–Fisher model for the first 100 generations into the past before reverting back to the classic Hudson model as proposed in [50]. The mutation rate was set to 1.2×10^{-8} .

Each 1 Mb region of simulated haplotypes was generated using a population-specific human recombination map estimated by `pyrho` [51]. Explicitly, in each simulation, we randomly selected one of our three 1000 Genomes populations (YRI, CHB, or CEU) and 1 Mb segment from genome (excluding heterochromatic regions), then we loaded the recombination map estimated for that population and region by `pyrho`.

Finally, all singleton variants (those with a derived allele count of 1) were removed from the simulated haplotypes before being passed to any association testing method.

6.8 Running ARG-Needle

E12► We ran ARG-Needle using the `arg_association` command line utility. This utility is automatically installed when installing `arg-needle-lib` with `pip` and documented in the ARG-Needle manual available at <https://palamaralab.github.io/software/argneedle/manual/>. As of March 20, 2025, we found that the python logging module (available at <https://pypi.org/project/logging/>) imported by `arg_association` has not been updated since 2013. We could not run `arg_association` because the logging module was not compatible with the python installation (version 3.12.3) required for our other simulation utilities. Upon inspecting the source code for the `arg_association` utility available on GitHub at https://github.com/PalamaraLab/arg-needle-scripts/blob/main/arg_needle_lib/arg_mlma/scripts/association.py, we found that the logging module was only used to print progress statements for the user. Commenting out all lines invoking the logging module allowed us to successfully run `arg_association`. We saved this modified script as `arg_needle_association_without_logging.py` and used it to perform association testing with the ARG-Needle library.

Since singletons were excluded from all of our simulations, we ran ARG-Needle using `arg_needle_association_without_logging.py` with optional argument `-min-mac 2` so that it would not lose power by attempting to test singletons. We also set the optional argument `-sampling_rate 1e-3`. Otherwise ARG-Needle was run with all default parameters. Filtering the results table returned by ARG-Needle down to rows with $MU_STAR \leq 10^{-5}$, we were able to obtain results as if we had invoked ARG-Needle with `-sampling_rate 1e-5`. Similarly, filtering down to rows with $MU_STAR \leq 10^{-7}$, we were able to obtain results as if we had invoked ARG-Needle with `-sampling_rate 1e-7`. ◀E12 Reply

6.9 Running ACAT-O

ACAT-O was run using the STAAR function from the STAAR R package available at <https://github.com/xihaoli/STAAR>. For each simulation, ACAT-O was run on all genotypes within the causal region. The ACAT-O was run testing rare variants with $MAF < 0.01$ (by setting `maf_cutoff=0.01`, the default) and all variants (by setting `maf_cutoff=1`). Otherwise, ACAT-O was run with all default settings.

6.10 Calculating Discovery Thresholds

Taking the same approach we used in our power simulations, we ran null simulations to estimate a discovery threshold to control the genome-wide family-wise error rate (FWER) at 0.05 for LOCATER, SMT, AN3, AN5, and AN7. Under the haplotype simulation framework described in Section 4.1, we simulated 900 independent genomic datasets, each consisting of a 1Mb segment observed for 30,000 individuals. For each of these segments, we sampled 10 independent null simulations of the phenotype vector Y (Section 4.2). We applied every testing method to each of the resulting 9,000 phenotype vectors and corresponding 1Mb genomic segments. We stored the smallest p-value obtained the entire segment, yielding 9,000 p-values per method. Since, by default, LOCATER conserves compute by only testing variants with single-marker test p-values smaller than 10^{-4} , sometimes LOCATER did not return any p-value for a given simulation. This occurred in 3,363 out of our 9,000 simulations. For these simulations, we simply assigned a p-value of 1 to LOCATER.

Since the genome is 3,000 Mb long and each of the resulting 9,000 p-values was estimated by taking the smallest p-value over a 1 Mb segment, we estimated our 0.05 genome-wide FWER threshold by multiplying the 5th percentile of the resulting 9,000 p-values by 3,000.

We used the same 9,000 null simulations to estimate a 0.05 genome-wide FWER threshold for ACAT-O (all) and ACAT-O (rare). However, we only ran ACAT-O on the central 10 kb causal window in each simulation rather than over the entire 1Mb region. If we consider a genome-wide screen using a 10kb sliding window with 5kb overlap between adjacent windows, that's a total of 600,000 genome-wide tests. Thus we multiplied the 5th percentile of the resulting 9,000 p-values obtain for ACAT-O (all) and ACAT-O (rare) by

600,000 to obtain their corresponding discovery thresholds.

1417

6.11 Phenotype Rank Matching

1418

Before we run LOCATER, as has become standard practice in testing quantitative traits, we require that the phenotype residuals be rank normalized after fitting background covariates. The theory underlying SD requires that the phenotypes be independent under H_0^C . Standard inverse-rank-normalization maps phenotype residuals to a fixed grid of values based on the Gaussian CDF, which in turn induces dependency between phenotypes. In order to avoid inducing that dependency, we substitute each phenotype residual with a rank-matched Gaussian random variable. This achieves a rank normalization that is very similar to inverse-rank-normalization but with a small amount of noise added in order to make the normalized phenotypes truly independent under the null hypothesis.

1419

1420

1421

1422

1423

1424

1425

1426

Explicitly, given some raw phenotype vector $\tilde{Y} \in \mathbb{R}^n$ and a matrix of background covariates $A \in \mathbb{R}^{n \times q}$ (including an intercept), we begin by calculating the residuals $\tilde{Y}^\perp = \tilde{Y} - A(A^\top A)^{-1}A^\top \tilde{Y}$ of the standard ordinary least squares model

1427

1428

1429

$$\tilde{Y} \sim A\alpha + \sigma\epsilon. \quad (29)$$

Then we find the permutation $\pi : [n] \rightarrow [n]$ that orders the residuals \tilde{Y}^\perp such that $\tilde{Y}_{\pi(1)}^\perp \leq \tilde{Y}_{\pi(2)}^\perp \leq \dots \leq \tilde{Y}_{\pi(n)}^\perp$. We simulate a new vector of Gaussian phenotypes, $Z \sim N(0, I_n)$ and likewise find the permutation $\pi' : [n] \rightarrow [n]$ that orders the entries of Z such that $Z_{\pi'(1)} \leq Z_{\pi'(2)} \leq \dots \leq Z_{\pi'(n)}$. Then we obtain our normalized vector of rank-matched phenotypes, $Y \in \mathbb{R}^n$, by assigning $Y_j = Z_{\pi'(\pi^{-1}(j))}$.

1430

1431

1432

1433

June 14, 2025

GENETICS-2025-308111

Clade Distillation for Genome-wide Association Studies

Dear Dr. Christ:

Two experts in the field have reviewed your manuscript, and I have read it as well. I am pleased to inform you that, with minor revisions, it is potentially suitable for publication in GENETICS. The reviewers have comments and concerns that need to be addressed in a revised manuscript. You can read their reviews at the end of this email.

As you will see, the reviewers agree that the manuscript is much improved (and indeed, thanks very much for the hard work on that!), and they have relatively minor but still (in my view) important comments to address - mostly, clarifying definitions and concepts (R1's suggestion about confidence intervals and an additional less abstract figure than Figs 2-4 seems good). R1 also raises the point that the question of application to real data is left relatively unaddressed because of population structure (in the current draft addressed by a reference to the medRxiv paper and comments in several parts of the paper, e.g., "open and challenging problem"). I defer to the author's assertion that sorting out this difficult problem is beyond the scope of the current paper - certainly, the current paper does a lot of good work - however, given the importance of the problem, a bit more might be usefully said to communicate the authors' view on this? In particular: Does the medRxiv paper demonstrate the degree to which this procedure is hampered by population structure? Is there reason to expect that ARG-Needle deals with population structure better? (Or, worse?)

We look forward to receiving your revised manuscript. Please let the editorial office know approximately how long you expect to need for revisions.

Upon resubmission, please include:

1. A clean version of your manuscript;
2. A marked version of your manuscript in which you highlight significant revisions carried out in response to the major points raised by the editor/reviewers (track changes is acceptable if preferred);
3. A detailed response to the editor's/reviewers' comments and to the concerns listed above. Please reference line numbers in this response to aid the editors.

Additionally, please ensure that your resubmission is formatted for GENETICS.

<https://academic.oup.com/genetics/pages/general-instructions>

Follow this link to submit the revised manuscript: Link Not Available

Sincerely,

Peter Ralph
Associate Editor
GENETICS

Approved by:
Konrad Lohse
Senior Editor
GENETICS

Reviewer #1 :

See attached file.

Reviewer #2 :

Summary

The authors have developed a new version of their ancestry-based methodology kalis to accommodate large-scale phenotype-

genotype associations when there are multiple (possibly hidden) causal variants (possibly with allelic heterogeneity). This paper provides another useful complement to SMT GWAS, which will help the field make further use of biobank resources. Following the first round of reviews, the authors have compared their method to another competing method and demonstrated the favorable performance of their method. The relegation of some technical sections to Methods and Supplementary Material has improved the narrative of the main text. The authors have provided extensive detail and mathematical motivations for their work, which appear correct. My few major comments, which are focused on the Figures, concern the reader better grappling with "total association signal strength" and what that would correspond to in terms of effect sizes.

I appreciate that the authors were comprehensive in responding to the reviews. They have addressed my initial major comments, including a detailed description of their now available and documented software implementation. (While I prefer to combine Figures when they are similar, I respect their decision, and it would add some tedious additional work.)

I will provide my line number (L) for the Response to Reviewers document. (It seems that the Genetics manuscript style compiled to restart the line numbers on each page.)

Major comments

Many Figures: I'm still confused about the definition of "Total association signal strength". The authors define it in terms of an oracle ANOVA model. What would that oracle be for LOCATER, i.e., what are the active predictors? The causal sprigs?

Main Figures: could you show more x-axis ticks? It seems like kalis results are typically between 20 and 50. It would be nice to have better discernment on what that value is.

- Please consider additional plot(s) of total association signal to achieve 20, 40, or 50% power (something smaller than 80). The total association signal strength to reach 80% power seems very large, which could correspond to a huge effect size of the causal variants.

x-axis labels of many Supplementary Figures: Could you define Signal Strength in the caption?

- For signal strength s , could you provide ballpark numbers of what the beta coefficients would be (at least for some simulations)? This detail could provide the audience with a notion of how large these effect sizes are.

Minor comments

Sometimes I find it confusing what is kalis versus kalis v2. Is it necessary to say kalis v2? You already say at various points that this paper provides enhancements to kalis. This is not a large point to consider.

L69-72: Which approaches are you specifically referring to? Surely, the SKAT-like rare SMT methods and IBD mapping are not testing all (or any) clades.

L153-155: Are "increase statistical power" and "identify loci missed" the right terms to use? I haven't carefully read the real data paper. But, how do you know that these are true causal signals?

L599: Do you have a benchmark on your Finnish analysis in the medRxiv paper, which is published and citable work? This is not an important point to address. Providing a time benchmark from unpublished work feels questionable to me. Or maybe the problem is that it is unpublished data. Could you describe the essence/core features of that dataset?

L613-620: Do you have a recommended less stringent threshold for the user if we do not perform null simulations? If we perform null simulations, how much runtime and expertise would this require (ballpark)?

Very minor line comments

Some of your citations in the compiled Genetics manuscript should have `()` around them, with some command like `\citep`? This is something to work with the proofs writer on, and be watchful when you're reviewing yours!

- Or, something weird is happening in the citations compilation of Genetics. For example, on page 2 line 34 Gunnarsson et. al. is written out twice.

The manuscript seems to compile such that sometimes the paragraph does not start on a new line after the Section name.

There is some Section number compilation issue on page 6 line 111 of Genetics manuscript. Also, there are some on line 96 and 104. Be watchful during proofs!

Sometimes you use font style `\texttt` for software and sometimes you do not.

L40: Browning and Thompson is manually written out, not with the citation link.

L201-203: Do you have a citation showing that more distant relationships are harder to infer?

L488, L1403: Put a space in between "1Mb"

6.10 Calculating Discovery Thresholds: You may want to write "3000" and "9000" for cases with only 3 zeros, but I could be wrong in terms of academic conventions. Maybe the proofs editor will address this to the journal's standard! (This also happens in other sections of the paper.)

Overall, there are not many typos or grammatical issues. There are a few spots where sentences are missing commas. For example, on L569 you use "Overall, LOCATER ..." and then on L576 "Overall our results ...". Another case: you say on L592 "In parallel work, we ..." and then on L598 "In other preliminary work we have ...". Just make a final pass before publication through Grammarly or some other grammar checker.

Associate Editor Comments:

The authors have done a good job of addressing my previous comments. I have a couple of additional minor comments about the content that was added in revision:

- I think readers will be surprised to see in Figures 2 & 3 that SMT outperforms ARG-Needle the majority of cases, even when the latter is given the true ARG. Is this expected and if so why? It's not the authors' job to debug ARG-Needle; if these findings are indeed correct, fine, but I think they should be discussed in the text, as readers may be confused.

- I agree with R2 that a genealogical based method that can correct for population structure would be a major advance. I don't find the authors' response to this comment very compelling since a) all real datasets will have population structure, b) the current crop of ARG inference procedures don't have any way to correct for this, e.g. by incorporating prior information, and c) confounding is likely worse when conditioning on inferred ARGs because of the illusion of a stronger signal (the authors' paper on running LOCATOR on the Finnish cohort makes this point too, I think.) I leave it up to the AE to decide whether to press hard on this point or not.

- After reading the manuscript multiple times, I find the term "confidence interval" in the localization section very helpful as a statistics-trained person. Identical to the usual definition, the unknown true parameter is the location of the true causal signal (the midpoint of the region) and the interval is a set surrounding the estimate (the lead variant). For several repeated trials of the same experiment, the proposed 80% confidence interval will include the true midpoint 80% for the times. This point has already been clearly made in P8L58-64. Nonetheless, this frequentist notion of confidence intervals is a commonly un/misunderstood in the scientific community. As an alternative to my previous suggestion to pull up one of the localization figures to the main section, how about adding an illustrative figure that explains the suggested confidence interval? Figure 1 emphasizes LOCATOR's functionality to generate P-values, so adding a second row that demonstrates localization will be helpful. A quick sketch of the idea could be something like below in which I appropriated from Wikipedia.

Author Replies to Reviewer Comments for Manuscript

GENETICS-2025-308111

The authors would like to sincerely thank the editor and reviewers again for their very helpful comments and suggestions regarding our manuscript titled *Clade Distillation for Genome-wide Association Studies*. Please see our replies below.

For the reviewers' convenience, the line numbers at the end of each of our replies are links that jump to the corresponding edits in the main text. They may not appear highlighted as links after this pdf has been processed by the journal platform but they will hopefully still work if you try clicking on them. At the end of each section of edits in the main text, we have also included reverse links which appear as Reply, that should allow reviewers to easily jump back to the corresponding reply. Figure and Section numbers also link to their locations in the text.

Associate Editor Comments

Associate Editor Comment — As you will see, the reviewers agree that the manuscript is much improved (and indeed, thanks very much for the hard work on that!), and they have relatively minor but still (in my view) important comments to address - mostly, clarifying definitions and concepts (R1's suggestion about confidence intervals and an additional less abstract figure than Figs 2-4 seems good). R1 also raises the point that the question of application to real data is left relatively unaddressed because of population structure (in the current draft addressed by a reference to the medRxiv paper and comments in several parts of the paper, e.g., "open and challenging problem"). I defer to the author's assertion that sorting out this difficult problem is beyond the scope of the current paper - certainly, the current paper does a lot of good work - however, given the importance of the problem, a bit more might be usefully said to communicate the authors' view on this? In particular: Does the medRxiv paper demonstrate the degree to which this procedure is hampered by population structure? Is there reason to expect that ARG-Needle deals with population structure better? (Or, worse?)

Reply:

Thank you, we really appreciate the thoughtful feedback here. We've added content to help clarify various definitions and concepts (please see specific replies below to Reviewer 1 and Reviewer 2). We completely agree that population structure is a very important problem, and in our view, there is some confusion about the nature of the challenge in the field. At the hazard of introducing too much conjecture into the paper, we attempted to

address your concern by adding a deeper dive to the Discussion, hoping that it will be useful to readers and further investigations. We are very happy to condense or cut it. See **E23** on Line 567

Reviewer 1

Reviewer Comment 1.1 — I think readers will be surprised to see in Figures 2 & 3 that SMT outperforms ARG-Needle in the majority of cases, even when the latter is given the true ARG. Is this expected and if so why? It's not the authors' job to debug ARG-Needle; if these findings are indeed correct, fine, but I think the should be discussed in the text, as readers may be confused.

Reply: This is a good point! We have added a couple of paragraphs to the Discussion addressing this. See **E20** on Line 528

Reviewer Comment 1.2 — I agree with R2 that a genealogical based method that can correct for population structure would be a major advance. I don't find the authors' response to this comment very compelling since a) all real datasets will have population structure, b) the current crop of ARG inference procedures don't have any way to correct for this, e.g. by incorporating prior information, and c) confounding is likely to be worse when conditioning on inferred ARGs because of the illusion of a stronger signal (the authors' paper on running LOCATOR on the Finnish cohort makes this point too, I think.) I leave it up to the AE to decide whether to press hard on this point or not.

Reply: We totally agree about the importance of the population structure question for the analysis of real datasets. Please see our reply to the Associate Editor (AE) above to see a more thorough discussion of this issue. Here we briefly address points (b) and (c) made in Reviewer Comment 1.2.

As delineated in our AE response, the concern around population structure in GWAS is false discoveries. If we find an associated locus, we would like to try to ensure that it is not driven by some unobserved confounder. Thus the problem arises when trying to associate some phenotype Y with the genetic variation at some locus ℓ , not during ARG inference. In other words, we view ARG inference as separable from the problem of population structure in GWAS. Any ARG-focused method should be focused on inferring the ARG as precisely as possible. Regarding point (b), it's not clear to us what it would mean for an ARG inference method itself to correct for population structure.

Regarding point (c), while it is natural and interesting to try to incorporate uncertainty in ARG estimation into association studies, we are not concerned that conditioning on some estimated ARG will yield false discoveries. From the point of view of association testing, the part of ARG inference that matters is mainly getting the membership of each inferred clade right. In the vast majority of cases, we expect that any error in clade membership inference will tend to downward bias rather than upward bias association signals.

Reviewer Comment 1.3 — After reading the manuscript multiple times, I find the term “confidence interval” in the localization section very helpful as a statistics-trained person. Identical to the usual definition, the unknown true parameter is the klocation of the true causal signal (the midpoint of the region) and interval is a set surround the estimate (the lead variant). For several repeated trials of the same experiment, the proposed 80% confidence interval will include the true midpoint 80% of the times. This point has already been clearly made in P8L58-64. Nonetheless, this frequentist notion of confidence intervals is a commonly un/misunderstood in the scientific community. As an alternative to my previous suggestion to pull up one of the localization figures to the main section, how about adding an illustrative figure that explains the suggested confidence interval? Figure 1 emphasizes LOCATOR’s functionality to generate P-values, so adding a second row that demonstrates localization will be helpful. A quick sketch of the idea could be something like below in which I appropriate from Wikipedia

Reply: We completely agree that the community can often be confused regarding confidence sets. To help address this, we have added a paragraph to the results providing further intuition behind our 80% confidence intervals. See **E21** on Line 466

We also carefully considered adding a graphic explainer of confidence intervals to Figure 1, but in the end decided against it. Figure 1 is designed to provide an overview of the inputs and outputs of LOCATER. If we added a confidence interval graphic to that figure, it might suggest that LOCATER could localization generate such confidence intervals on a particular real dataset. That is not the case: there is no central limit theorem or other result we can use to construct such an interval if the true causal region is not known. The confidence intervals we report in our localization section are empirically calculated from our simulations where we know the location of the true causal region. Thus, these intervals are not an inherent part of the method, and we believe it would be misleading to add them to Figure 1.

We also considered adding a supplementary figure explaining confidence intervals, but found that we could not meaningfully improve upon what an ill-informed reader could otherwise find on Wikipedia.

Reviewer 2

Reviewer Comment 2.1 — The authors have developed a new version of their ancestry-based methodology kalis to accommodate large-scale phenotype-genotype associations when there are multiple (possibly hidden) causal variants (possibly with allelic heterogeneity). This paper provides another useful complement to SMT GWAS, which will help the field make further use of biobank resources. Following the first round of reviews, the authors have compared their method to another competing method and demonstrated the favorable performance of their method. The relegation of some technical sections to Methods and Supplementary Material has improved the narrative of the main text. The authors have provided extensive detail and

mathematical motivations for their work, which appear correct. My few major comments, which are focused on the Figures, concern the reader better grappling with "total association signal strength" and what that would correspond to in terms of effect sizes.

I appreciate that the authors were comprehensive in responding to the reviews. They have addressed my initial major comments, including a detailed description of their now available and documented software implementation. (While I prefer to combine Figures when they are similar, I respect their decision, and it would add some tedious additional work.)

I will provide my line number (L) for the Response to Reviewers document. (It seems that the Genetics manuscript style compiled to restart the line numbers on each page.)

Reply: Thank you for your thoughtful comments and thoroughness, we believe the manuscript is much improved as a result.

Reviewer Comment 2.2 — Many Figures: I'm still confused about the definition of "Total association signal strength". The authors define it in terms of an oracle ANOVA model. What would that oracle be for LOCATER, i.e., what are the active predictors? The causal sprigs?

Reply: We apologize for the confusion here. Yes, the causal variants are the active predictors. We've gone through and replaced "active predictors" with "causal variants" to try to avoid any ambiguity. See **E1** on Line 393 We've also added a little more detail to clarify the ANOVA oracle model in the corresponding part of the Supplement. See **E2** on Line 625 In case this might be causing some confusion, note our definition of the oracle model stands alone: it is the best model one could use to test for allelic heterogeneity if the causal variants were known. There is no oracle model "for" LOCATER.

Reviewer Comment 2.3 — Main Figures: could you show more x-axis ticks? It seems like kalis results are typically between 20 and 50. It would be nice to have better discernment on what that value is.

Reply: We've added a finer grid of tick marks to the main text figures and all corresponding power simulation figures in the Supplement. See **E3** on Line 413 See **E4** on Line 1047

Reviewer Comment 2.4 — Please consider additional plot(s) of total association signal to achieve 20, 40, or 50% power (something smaller than 80). The total association signal strength to reach 80% power seems very large, which could correspond to a huge effect size of the causal variants.

Reply: This is a fair consideration, but we think that this is already addressed by the power curves in the Supplement 5.4 which simultaneously show the total association signal required to achieve power at every level. As mentioned in our reply to Reviewer Comment 2.3, we have added finer tick marks to all of those plots in order to make them easier to interpret.

The goal of our main text figures is to summarize these many complex curves with a measure of “how much signal does a method need to reliably detect an association.” We believe that a method that only provides a 50:50 shot at finding a signal is not really reliable. In our experience, grant applications require studies to have a reasonable chance of detecting the proposed signal, typically at least 80% power, in order to be considered for funding. Hence our choice of a 80% power threshold for summarizing our power curves.

Reviewer Comment 2.5 — x-axis labels of many Supplementary Figures: Could you define Signal Strength in the caption?

Reply: We've written it out as "Total Association Signal Strength" in all x-axis labels and added a definition to all of the captions.

Reviewer Comment 2.6 — For signal strength s , could you provide ballpark numbers of what the beta coefficients would be (at least for some simulations)? This detail could provide the audience with a notion of how large these effect sizes are.

Reply: This is a really good suggestion. We have added a new section to the supplement delineating the effect sizes induced in our simulations. See **E6** on Line 1004 We have added a reference to this new section in the main text. See **E5** on Line 639

Reviewer Comment 2.7 — Sometimes I find it confusing what is kalis versus kalis v2. Is it necessary to say kalis v2? You already say at various points that this paper provides enhancements to kalis. This is not a large point to consider.

Reply: This is a good point. We have decided to switch to the more canonical way of referring to new software versions: we've gone from using "kalis v2" to using "kalis 2.0." We've also been more deliberate in our usage. Throughout the text, when we refer to specific advancements introduced in kalis 2.0, we use "kalis 2.0." Otherwise, we simply say "kalis." We hope this avoids any unnecessary confusion.

Reviewer Comment 2.8 — L69-72: Which approaches are you specifically referring to? Surely, the SKAT-like rare SMT methods and IBD mapping are not testing all (or any) clades.

Reply: We've clarified the language there. See **E7** on Line 67

Reviewer Comment 2.9 — L153-155: Are "increase statistical power" and "identify loci missed" the right terms to use? I haven't carefully read the real data paper. But, how do you know that these are true causal signals?

Reply: We are confident that the majority of the loci identified by LOCATER in that paper [1] correspond to true causal signals because they have been replicated in much larger cohorts and mechanistically validated via *in vivo* experiments. For example, in that study, LOCATER identified a HDL association with lipase G (*LIPG*), “a well-known member of the triglyceride lipase family of proteins [that] is primarily involved in the metabolism of HDL [1]”. Please see [1] for more details.

Reviewer Comment 2.10 — L599: Do you have a benchmark on your Finnish analysis in the medRxiv paper, which is published and citable work? This is not an important point to address. Providing a time benchmark from unpublished work feels questionable to me. Or maybe the problem is that it is unpublished data. Could you describe the essence/core features of that dataset?

Reply: The time benchmarking discussion on real data that we provide at the end of the Results (just above the Discussion 3) reports the observed time requirements needed for the study of 6,795 genomes as detailed in the medRxiv paper [1]. We could remove the subsequent remark describing the time requirements on a larger dataset of 12,964 genomes (the results of which are not yet published) but we think reporting those additional time requirements provides a helpful sense of scalability to readers. We've tried to clarify this in the main text. See **E8** on Line 500

Reviewer Comment 2.11 — L613-620: Do you have a recommended less stringent threshold for the user if we do not perform null simulations? If we perform null simulations, how much runtime and expertise would this require (ballpark)?

Reply: Well, as we already say there in the text, our results suggest “that it will be safe for investigators to apply their SMT discovery threshold for a given dataset to their LOCATER results.” For example, if an investigator is happy to use the canonical $10^{-8.5}$ threshold for their SMT analysis, they should feel comfortable using that threshold for their LOCATER analysis. We've added some a couple of lines in the main text to try to address the questions around runtime and expertise. See **E9** on Line 518

Reviewer Comment 2.12 — Very minor line comments

Reply: Thank you very much for noting these errors/issues! We've addressed most of them and have left some for final editing if/when this manuscript goes to proofs.

Clade Distillation for Genome-wide Association Studies

Ryan Christ ^{1*}, Xinxin Wang ^{1,2}, Louis J.M. Aslett ³, David Steinsaltz ⁴, Ira Hall ^{1*},

1 Department of Genetics, Yale University School of Medicine, New Haven, CT, USA

2 Department of Genetics, Washington University School of Medicine, Saint Louis, MO, USA

3 Department of Mathematical Sciences, Durham University, Durham, UK

4 Department of Statistics, University of Oxford, Oxford, UK

✉ Current Address: Department of Genetics, Yale University School of Medicine, New Haven, CT, USA

* ryanchrist@yale.edu

Abstract

Testing inferred haplotype genealogies for association with phenotypes has been a longstanding goal in human genetics given their potential to detect association signals driven by allelic heterogeneity — when multiple causal variants modulate a phenotype — in both coding and noncoding regions. Recent scalable methods for inferring locus-specific genealogical trees along the genome, or representations thereof, have made substantial progress towards this goal; however, the problem of testing these trees for association with phenotypes has remained unsolved due to the growth in the number of clades with increasing sample size. To address this issue, we introduce several practical improvements to the kalis ancestry inference engine, including a general optimal checkpointing algorithm for decoding hidden Markov models, thereby enabling efficient genome-wide analyses. We then propose ‘LOCATER’, a powerful new procedure based on the recently proposed Stable Distillation framework, to test local tree representations for trait association. Although LOCATER is demonstrated here in conjunction with kalis, it may be used for testing output from any ancestry inference engine, regardless of whether such engines return discrete tree structures, relatedness matrices, or some combination of the two at each locus. Using simulated quantitative phenotypes, our results indicate that LOCATER achieves substantial power gains over traditional single marker testing, ARG-Needle, and window-based testing in cases of allelic heterogeneity, while also improving causal region localization. These findings suggest that genealogy-based association testing will be a fruitful approach for gene discovery, especially for signals driven by multiple ultra-rare variants.

1 Introduction

Recent heritability estimates predict that rare variants in regions with low linkage disequilibrium account for a substantial fraction of the unexplained (missing) heritability of common traits and diseases [2]. Since the statistical power to detect effects driven by rare variants is inherently limited by their low frequency, methods for identifying rare variant associations leverage allelic heterogeneity: the presence of multiple independent causal mutations affecting the trait of interest. These methods merge association signals from nearby rare variants under the premise that rare causal variants may be proximal to other causal variants [3, 4].

Mounting evidence suggests that allelic heterogeneity is quite common for human traits [5, 6]. Notably, a large-scale *in vitro* study following up on identified associations estimated that between 10% and 20% of expression quantitative trait loci (eQTLs) have multiple causal regulatory variants circulating in human populations [7]. Such results underscore the importance of these methods for defining new alleles and genes contributing to disease risk [8, 9, 2]. The opportunity for methods that can leverage allelic heterogeneity to identify overlooked associations will only increase with the size, population diversity, and sequencing depth of

emerging genomic datasets. 32

Early groundbreaking association methods designed to harness allelic heterogeneity focused on testing 33
inferred locus-specific genealogies, which provide a natural way of collecting independent association signals 34
driven by nearby variants and imputing any unobserved variants [10, 11, 12]. These approaches have the added 35
benefit of implicitly imputing unobserved variants, making it particularly advantageous for analyzing datasets 36
with partial variant calling: SNP array data, low-coverage sequencing data, or data from understudied species. 37
The uncertainty and poor scalability of early local genealogy inference methods hamstrung the adoption 38
of these early testing approaches. To partly address these challenges, Browning and Thompson proposed 39
identity-by-descent mapping to test only very recent, locus-specific relationships, which could be rapidly 40
and confidently inferred from the observed haplotypes [13]. While very elegant, all of these early methods 41
suffer from technical limitations related to statistical testing, such as requiring resampling or permutation to 42
generate p-values, and ultimately have not been used much. 43

Recent advances in ancestry inference algorithms have made it possible to revisit genealogy-based trait 44
association. Algorithms such as ARG-Needle [14], Relate [15], tsinfer-tsinfer [16], and kalis-kalis [17] 45
have made it possible to perform local ancestry inference across the entire genome in modern datasets 46
with hundreds of thousands or millions of samples. Given their accuracy in resolving recent genealogical 47
relationships, inferred local ancestries are expected to be especially useful for detecting loci with multiple 48
causal ultra-rare variants, which are signals that standard single marker testing (SMT) will struggle to identify. 49
This may be a particularly effective strategy for traits under strong purifying selection and cases where some 50
of these ultra-rare causal variants correspond to complex hidden structural variations that are only observed 51
in a given sequencing dataset via ultra-rare tagging variants that are far upstream or downstream. However, 52
recent work aimed at applying these algorithms to improve disease mapping, most notably ARG-Needle 53
[14], has focused on imputing hidden variants not explicitly observed in the original dataset and testing 54
the inferred genotypes via SMT. Given the plummeting cost of high-coverage sequencing data and recent 55
initiatives to improve structural variant detection and imputation [18], the number of missing variants is 56
shrinking in modern datasets, limiting the gains available from testing hidden variation via local ancestries. 57

Link et al. evidenced a renewed interest in using genealogies to map genes by leveraging allelic heterogeneity 58
[19]. Building on earlier efforts like that of Zöllner & Pritchard [10], their approach targets loci with allelic 59
heterogeneity by using local ancestry inference methods to build a local genetic relatedness matrix for 60
pre-specified windows along the genome or gene regions. These matrices are then tested for association with 61
the phenotype of interest using a quadratic form test statistic. Using effectively the same default test statistic, 62
very recent work by Zhu et al. [20] and Gunnarsson et al. [21] provides a much more scalable implementation 63
of this approach. Building on the idea of identity-by-descent mapping, Cai and Browning [22] very recently 64
proposed a distinct, scalable approach that also relies on a quadratic form test statistic. All of these methods 65

mirror SKAT and more recent approaches in rare-variant gene-based testing that also aim to harness allelic
heterogeneity to gain statistical power [3, 23]. **E7**► However, due to the inherent sensitivity of quadratic form
test statistics to the presence of many non-causal (null) variants [24], these approaches struggle to maintain
statistical power under the enormous multiple testing burden incurred when testing ~~all of the clades in a
local tree~~ many components of the local tree structure, which will often reflect but may not always directly
correspond to inferred clades in that tree structure, for trait association. ◀**E7 Reply**

Existing rare variant association methods that aim to leverage allelic heterogeneity by testing collections
of variants, such as STAAR, limit their multiple testing burden by using functional information, such as
gene coding sequences, to define restricted sets of variants or more flexibly down-weight certain variants
[3, 23]. This approach has been applied to many different sequencing based studies, and has proven to be a
fruitful approach for identifying new gene-phenotype associations across a multitude of traits [25]. Despite
their success in coding regions, it has proven difficult to extend rare variant association tests beyond coding
regions where the majority of biologically critical signals are found. Ninety percent of GWAS hits for common
diseases lie in non-coding regions, at a median distance of 36 kilobase pair (kb) from the nearest transcription
start site [26, 27]. It is unclear how one should define collections of variants in non-coding regions; sliding
windows are the standard approach [23].

Outside of a gene's coding region, which in humans has a median length ≈ 3 kb, there is a much larger
regulatory region over which causal variants may be dispersed, complicating the use of sliding windows.
Ideally, one would try to incorporate many variants over a genomic region in order to maximize the chance
of aggregating signals from more than one causal variant. However, including too many non-causal (null)
variants diminishes statistical power, and variant impact prediction, which could be used to narrow down
variants to test in a given sliding window, remains extremely challenging in noncoding regions.

There are "sparse-signal" statistical methods, often deployed alongside quadratic forms, that aim to
improve power in the presence of many null variants. Notable examples include the Cauchy Combination
Test (CCT), which underlies the Aggregated Cauchy Association Test (ACAT) routine in STAAR [23], and
Generalized Higher Criticism [28]. However, these sparse-signal methods do not distinguish between a variant
set where two highly-linked variants are observed to be associated with the phenotype and a variant set
where two unlinked variants are observed to be associated with the phenotype. In the highly-linked case, one
variant is essentially a proxy for the other and we have only one association signal. In the unlinked case, the
signals coming from the two independent variants serve as independent pieces of evidence against the null
hypothesis and should be combined. In order to control the type-I error in the highly-linked case, the CCT
underlying ACAT cannot combine signals across variants with high efficiency, which yields a loss of power in
the unlinked case. This simple two-variant argument extends to the case where we may be attempting to
combine association signals across several variants. Section 1 of [24] provides further discussion of this point.

We recently proposed a general statistical approach, Stable Distillation (SD), which can distinguish 100
between the highly-linked and unlinked case [24]. There, in a gene-testing example using simulated data, we 101
used SD to explicitly model the dependence structure between variants and achieved increased power over 102
ACAT and related methods as a result. Building on SD, we present a general framework, LOCATER, for trait 103
association based on inferred local genealogies in both coding and non-coding regions. We focus exclusively 104
on testing quantitative traits, although we plan to extend LOCATER to binary traits in future work. 105

Modern ancestry inference methods typically represent local ancestries as discrete trees (perhaps with 106
probabilistic weights on the edges), local relatedness matrices, or some combination of the two. Examples of 107
discrete tree inference methods include `tsinfer`. These clades may have probabilistic weights, as provided by 108
recent probabilistic ARG inference methods such as SINGER [29]. On the other hand, ancestry may also 109
be represented in terms of local pairwise relatedness, typically summarized as a local relatedness matrix, as 110
produced by `kalis`, `Relate`, and *Gamma-SMC* [30]. A set of observed haplotypes can typically be explained 111
by an enormous number of underlying tree topologies, especially once we look beyond the recent past; 112
pairwise methods account for this topological uncertainty. As described in Section 2.1, LOCATER provides a 113
framework for boosting SMT results with independent association signals based on local ancestry represented 114
in either, or both, of these two forms produced by any ancestry inference engine. In order to highlight this 115
feature, in this paper we apply LOCATER to complementary discrete clade and matrix-based representations 116
of local ancestry obtained via the local ancestry inference engine `kalis`. 117

LOCATER is designed to work in conjunction with any ancestry inference engine of the user's choosing 118
with an easy-to-use API available through our `locater` package for the R language [31]. Since our `locater` 119
package exposes all of our testing subroutines as documented R functions, if a set of clade calls or a local 120
relatedness matrix produced by some ancestry inference engine can be coerced into base or sparse matrices in 121
R, then `locater` can be directly used to test those structures for association with a given phenotype. Please 122
see the Data and Code Availability section for details. 123

Although `kalis` does not scale as well as alternatives like `tsinfer`, a probabilistic model allows us to 124
limit statistical testing to clades that have substantial evidence of existing at a locus of interest, thereby 125
conserving statistical power. SINGER may provide a strong probabilistic alternative in future studies. The 126
algorithmic improvements to `kalis` that we present in this paper, including an optimal checkpointing routine 127
for discrete-time hidden Markov models (HMMs) and linear-time clustering algorithm, may be useful for 128
accelerating alternative models. 129

Our focus on using genealogies to boost SMT signals rather than testing gene windows or sliding windows 130
along the genome is another key point of departure of LOCATER from existing work. This approach leverages 131
the statistical efficiency of SMT against sparse signals. Testing the inferred genealogy at a locus also removes 132
questions of window size and step length. By returning “genealogy-boosted” SMT signals, we find LOCATER 133

generally improves the localization of causal variants relative to SMT in the presence of allelic heterogeneity (Section 2.6). The precise variants aggregated at a given locus will depend on the structure of the local ancestral recombination graph (ARG) and the parameters of the ancestry inference engine used [32, 33, 15]. At a high level, older edges in a local genealogy tend to persist over much shorter stretches of the genome than recent edges due to recombination [32]. Accordingly, any procedure that tests a local genealogy for association with a phenotype will tend to aggregate association signals from rare variants over a wider region than common variants.

In this paper we characterize LOCATER's performance in simulated datasets. Dealing with the challenges of trait mapping in real datasets, such as rigorously adjusting for population structure and cryptic relatedness, is a difficult and open problem for genealogy-based trait mapping methods. We take on these problems in a subsequent work [1]. There we demonstrate the ability of LOCATER to substantially increase statistical power at loci with allelic heterogeneity and identify loci missed by SMT in a dataset of 6,795 Finnish genomes with extensive quantitative trait data.

2 Results

LOCATER assumes that genome-wide SMT has already been performed. This is done simply to avoid the computational burden of inferring ancestries and running LOCATER at every locus. We focus on the subset of variants with putatively significant SMT results (e.g., $p < 10^{-4}$) and compute the local ancestry at each of those variants. At each target variant, LOCATER then takes the residuals from the SMT and tests any inferred discrete clade structure with Stable Distillation (SD) [24]. As discussed in Section 2.3, SD returns a new set of residuals which are guaranteed to be independent of the original SMT p-value and the p-value returned by SD under the null hypothesis. We then pass this set of residuals to any quadratic-form based method that tests the pairwise-relatedness structure inferred at the variant of interest. The resulting three independent p-values may then be combined to obtain a potentially boosted signal at a locus with allelic heterogeneity. This approach makes it straightforward to integrate LOCATER into the analysis of genome-wide association results. Of course, the resulting p-values must still be compared against a genome-wide multiple testing threshold as if LOCATER was run at every candidate variant. LOCATER can easily be applied in special cases where the inferred ancestry at a locus only comes in the form of discrete clades (eg: a tree) or pairwise-relatedness (eg: a local relatedness matrix). In both our SD procedure and quadratic form testing procedure, we have developed scalable methods to adjust for population structure and background covariates (Methods).

Below we introduce the LOCATER model. We then proceed to describe our methodological contributions in three parts: generating the ancestry representations required by LOCATER using the new routines we

have introduced in an update to `kalis` [17], making the association testing procedures in LOCATER fast and robust, and efficiently combining the p-values returned by LOCATER. Finally, we demonstrate the calibration and power of LOCATER via simulation.

2.1 The LOCATER Model

Consider a genomic dataset with n participants phased in segments along the genome, each segment consisting of $N = 2n$ phased haplotypes along an entire or subsection of a chromosome with a total of V variants. Although we only address the diploid case in this paper, our approach may be readily extended to non-diploid organisms. Below we will consider testing each variant within a given segment for association with some quantitative phenotype of interest $Y \in \mathbb{R}^n$. When determining genome-wide significance thresholds, the total number of candidate variants across all genomic segments must be accounted for. However, in order to conserve computational resources, as depicted in Figure 1, we may only be interested in a subset of candidate variants within each segment based on preliminary SMT results or other genomic annotations. We call this set of target loci $\mathcal{L} \subseteq [V]$ and index them by their position along a given segment sequentially from $\ell = 1, \dots, L = |\mathcal{L}|$.

Let $A \in \mathbb{R}^{n \times q}$ be a matrix of background covariates and $G^{(\ell)} \in \{0, 1, 2\}^n$ be the genotype vector observed at locus ℓ . Depending on the type of inference engine used to infer the local ancestry structure at locus ℓ , we may have inferred clade genotypes $X^{(\ell)} \in \{0, 1, 2\}^{n \times p}$ corresponding to edges in a tree inferred at ℓ , a local relatedness matrix $\Omega^{(\ell)} \in \mathbb{R}^{n \times n}$ inferred at ℓ , or both. In other words, for each of p inferred clades at locus ℓ , $X_{ij}^{(\ell)}$ is the number of haplotypes in sample i that have been assigned to an inferred clade j at locus ℓ . We tackle the general case assuming that our ancestry inference engine has returned both $X^{(\ell)}$ and $\Omega^{(\ell)}$, each capturing different parts of the ancestral structure at locus ℓ . Our approach can easily be applied to the special cases where only $X^{(\ell)}$ or $\Omega^{(\ell)}$ are available.

As further described in Section 2.2.4, by allowing genealogical relationships to be expressed in terms of pairwise similarity rather than explicitly called clades, $\Omega^{(\ell)}$ accommodates more uncertainty about the precise topology of the underlying tree than $X^{(\ell)}$. However, this flexibility comes with a cost to power: as our simulations demonstrate, it is generally preferable to encode clades in $X^{(\ell)}$ rather than $\Omega^{(\ell)}$, at least when their membership is known with high confidence (end of Section 2.5). Due to recombination, more distant genealogical relationships, corresponding to larger clades, at a given locus ℓ are more difficult to accurately estimate than more recent genealogical relationships, corresponding to small clades in a local genealogy. Thus, in this paper, we will demonstrate LOCATER by encoding small clades (typically each with at most 10 haplotypes under them), which we will refer to as “sprigs,” in $X^{(\ell)}$ and encoding larger clades in $\Omega^{(\ell)}$. Section 2.2.3 further details how we call sprigs.

For a set of fixed effects $\alpha \in \mathbb{R}^q$, $\gamma \in \mathbb{R}$, and $\beta \in \mathbb{R}^p$, and a variance component parameter $\tau \in \mathbb{R}_{\geq 0}$,

LOCATER assumes the following model for a quantitative phenotype vector $Y \in \mathbb{R}^n$.

$$Y = A\alpha + G^{(\ell)}\gamma + X^{(\ell)}\beta + \epsilon \quad \text{where} \quad \epsilon \sim N\left(0, \exp\left(\tau\Omega^{(\ell)}\right)\right) \quad (1)$$

Here, \exp denotes the matrix exponential. Under this model we test whether genetic variation at locus ℓ affects phenotype Y by testing the null hypothesis $H_0^C : \{\gamma = 0, \beta = 0, \tau = 0\}$. To be clear, H_0^C represents the union, not the intersection, of these statements about the parameters. In other words, in what follows, we will reject H_0^C if there is evidence that $\gamma \neq 0$, $\beta \neq 0$, or $\tau \neq 0$. We will assume that Y has been obtained using the rank-matching procedure described in Section 6.11. This normalization ensures that the residuals of Y have unit variance under H_0^C , justifying the absence of a variance scale parameter (typically denoted as σ^2) in Equation (1).

LOCATER tests H_0^C by decomposing it into three sub-hypotheses. First, we use the standard SMT to test $H_0^{\text{SMT}} : \{\gamma = 0 | \beta = 0, \tau = 0\}$, yielding a p-value p_{SMT} . Then we test whether any of the locally inferred clades predict the phenotype $H_0^{\text{SD}} : \{\beta = 0 | \tau = 0\}$ using SD, yielding a p-value p_{SD} . Finally, we test whether any remaining local ancestry structure encoded in $\Omega^{(\ell)}$ affects the phenotype by testing $H_0^{\text{Q}} : \{\tau = 0\}$ with a quadratic form test statistic, yielding a p-value p_{Q} . We further describe the explicit routines LOCATER uses to test H_0^{SD} and H_0^{Q} in Section 2.3. As further explained in Section 2.3, when these three sub-hypotheses are tested in this order, the independence guarantees of SD ensure that the resulting p-values ($p_{\text{SMT}}, p_{\text{SD}}, p_{\text{Q}}$) are mutually independent under the null hypothesis H_0^C . Thus, after running LOCATER, the user may combine these three p-values using any valid method for aggregating independent p-values. We propose a variant of Fisher's method that we call Maximizing over Subsets of Summed Exponentials (MSSE) which yields more power in this setting (see Section 4.8). Figure 1 overviews the role of LOCATER in the context of an ancestry-based association testing pipeline. Next we delineate how the algorithms we have **implemented in** introduced in our new release of `kalis-v2`, `kalis 2.0`, allow us to rapidly obtain $X^{(\ell)}$ and $\Omega^{(\ell)}$ across target loci in our present study.

2.2 Algorithmic Advances in ~~kalis-v2~~ kalis 2.0

2.2.1 Local Genealogy Inference with `kalis`

`kalis` [17] provides a high-performance implementation of various versions of the Li & Stephens (LS) haplotype copying model which have become ubiquitous in modern genomic analysis [33, 34]. We outline our novel algorithmic contributions ~~to a new release, `kalis-v2`, in `kalis 2.0`~~, which together allow us to efficiently calculate $X^{(\ell)}$ and $\Omega^{(\ell)}$ sequentially at a given set of target loci $\ell = 1, \dots, L$ so that they can be tested downstream using LOCATER. In order to explain these contributions, we begin with a brief overview of local ancestry inference using `kalis`.

Figure 1: **The LOCATER Pipeline.** We begin ancestry-based association testing with a set of putatively interesting target loci, typically identified via single marker testing, indexed $\{1, \dots, L\}$. At each target locus ℓ , we extract the genotype vector $G^{(\ell)} \in \{0, 1, 2\}^n$ and use an ancestry inference engine to infer local clade genotypes $X^{(\ell)} \in \{0, 1, 2\}^{n \times p}$ and/or a local relatedness matrix $\Omega^{(\ell)} \in \mathbb{R}^{n \times n}$. We then use LOCATER to calculate three p-values testing whether $G^{(\ell)}$, $X^{(\ell)}$, or $\Omega^{(\ell)}$ predict the phenotype respectively. These three p-values are guaranteed to be independent under the null hypothesis, so they may be easily combined with many methods, in this paper we propose and use MSSE 4.8, to obtain a combined ancestry-association p-value $p_C^{(\ell)}$ at each target locus.

As with all ancestry inference engines, the ancestry at a given target locus ℓ is learned based on the observed genomic variation upstream and downstream of ℓ . Since the LS model is a special case of an HMM, ancestry information provided by variants upstream of ℓ can be summarized by the forward probabilities at ℓ ; and variants downstream by the backward probabilities at ℓ [35]. Given a set of N haplotypes and a single target locus ℓ , **kalis** implements the forward algorithm to iterate over variants upstream of ℓ , starting at the left end of the genomic segment, to obtain a matrix of forward probabilities $f^{(\ell)} \in \mathbb{R}^{N \times N}$ at ℓ . Similarly, **kalis** implements the backward algorithm to iterate over variants downstream of ℓ , starting at the right end of the genomic segment, to obtain a matrix of backward probabilities $b^{(\ell)} \in \mathbb{R}^{N \times N}$ at ℓ . Each column $f_{\cdot j}^{(\ell)}$ and column $b_{\cdot j}^{(\ell)}$ corresponds to a separate LS HMM where we model recipient haplotype j as a mosaic of the other $N - 1$ haplotypes in the sample. This separation allows **kalis** to compute the columns of $f^{(\ell)}$ and $b^{(\ell)}$ in parallel and exploit modern compute architectures. See Aslett & Christ [17] for further details. The product $f_{ij}^{(\ell)} b_{ij}^{(\ell)}$ can be interpreted as proportional to the probability that recipient haplotype j “copies” from donor haplotype i at locus ℓ under the LS model. By definition, $f_{ii}^{(\ell)} = b_{ii}^{(\ell)} = 0$ for all haplotypes i . **kalis** makes $f^{(\ell)}$ and $b^{(\ell)}$ easily and rapidly accessible in the R language [31] for downstream computation, with all time-critical code written in high performance C.

Along the lines of Speidel et al. [15] and Aslett & Christ [17], we define the distance from haplotype j to haplotype i as

$$d_{ij}^{(\ell)} = -\log \left(\max \left(\frac{f_{ij}^{(\ell)} b_{ij}^{(\ell)}}{\sum_{k=1}^N f_{kj}^{(\ell)} b_{kj}^{(\ell)}}, v \right) \right) \quad (2)$$

where $v \approx 4.94 \times 10^{-324}$ to guard against underflow to zero with double precision floating point arithmetic.

For efficiency the distance matrix $d^{(\ell)} = (d_{ij}^{(\ell)}) \in \mathbb{R}_{\geq 0}^{N \times N}$ is never explicitly constructed, but it is implicitly used to construct $X^{(\ell)}$ and $\Omega^{(\ell)}$ for testing with LOCATER, as further delineated below.

Throughout this paper, we use `kalis` to run the modified LS model used in `Relate` [15]. See Section 6.1 for further details. This modified model leverages ancestral allele information to improve local genealogy inference. In this paper we only simulate phased genomic datasets where the ancestral allele of each variant is known; this is a feature of our chosen ancestry inference engine and not a general requirement of LOCATER. Under this modified LS model, Speidel et al. showed that the distance $d_{ij}^{(\ell)}$ will be proportional to the number of proximal variants that differ between haplotype i and haplotype j in non-recombining segments and that the full matrix $d^{(\ell)}$ yields consistent local ancestry inference.

2.2.2 Optimal Checkpointing

Especially when processing many phenotypes in parallel, the number of target variants along a given genomic segment, L , may be very large. Since the amount of memory required to store a local relatedness matrix $\Omega^{(\ell)}$ at a given target locus scales $\mathcal{O}(n^2)$, storing these matrices at any appreciable number of variants quickly becomes untenable: even in the case where $n = 30,000$ samples (a scale we will consider in our simulations), 28.8 GB of memory is required to store a single $\Omega^{(\ell)}$. Offloading from memory also incurs a considerable time cost from writing and reading $\Omega^{(\ell)}$ to and from disk. To avoid storing any $\Omega^{(\ell)}$, we will take a “test-it-and-forget-it” approach: we will obtain $X^{(\ell)}$ and $\Omega^{(\ell)}$ at one target variant at a time and test both $X^{(\ell)}$ and $\Omega^{(\ell)}$ with LOCATER before moving on to the next target variant in \mathcal{L} . This “test-it-and-forget-it” approach is only computationally tractable due to the checkpointing algorithm for discrete-time HMMs that we have introduced in `kalis` v22.0. Checkpointing involves repeatedly updating a cache of forward matrices $f^{(\ell)}$ that are used to seed subsequent iterations of the forward algorithm. Each stored forward matrix is a “checkpoint,” and we assume that the user has a fixed memory budget sufficient to store C checkpoints. Our checkpointing algorithm schedules where and when to overwrite each of the C checkpoints in order minimize the computational cost required to sequentially propagate the forward algorithm to consecutive target loci $\ell = L, L - 1, L - 2, \dots, 1$. While maintaining a cache of checkpoints is far from a new idea in HMM inference – the idea is used in several implementations of the LS model [15] – our checkpointing algorithm achieves the lower bound on computational cost given memory for C checkpoints and can be applied to any discrete time HMM where sequential posterior decodings are required at consecutive times. Our approach yields massive reductions in computational cost compared to more naive checkpointing approaches (Section 4.3).

2.2.3 Calculating Inferred Clade Genotypes from the LS Model

The LOCATER model (Equation (1)) admits a matrix of genotypes $X^{(\ell)}$ encoding any clades (marginal tree edges) inferred at locus ℓ . In principle, given the distance matrix $d^{(\ell)}$ obtained via `kalis` at locus ℓ

(Equation (2)), any number of clustering algorithms could be used to infer a marginal tree topology from $d^{(\ell)}$. For example, `Relate` clusters a normalized version of $d^{(\ell)}$ with average linkage (UPGMA) [15]. From the resulting tree topology, one could then encode each of the inferred clades (or some subset of them) via $X^{(\ell)}$. While this is a promising approach for future work, in this paper, we only stored clade genotypes corresponding to very small inferred clades (each typically including 2 to 10 haplotypes) in $X^{(\ell)}$ and encode all larger-scale relatedness structure in $\Omega^{(\ell)}$. Focusing on just these rare clades, which we will refer to as “sprigs,” rather than all of the clades in the tree, allows us to showcase the flexibility of LOCATER — the ability of LOCATER to incorporate hard-called clades via $X^{(\ell)}$ and remaining relatedness structure via $\Omega^{(\ell)}$.

We identify sprigs at a given locus ℓ based on the neighborhood — i.e., the set of tied nearest-neighbors — of each haplotype j :

$$\eta_j^\ell = \left\{ i \in [N] : d_{ij}^{(\ell)} \leq \min_{i \neq j} d_{ij}^{(\ell)} \right\}. \quad (3)$$

Note that, by this definition, haplotype j is always a nearest neighbor of itself. In practice we obtain these neighborhoods as a by-product of the clustering algorithm we use to construct $\Omega^{(\ell)}$ (Section 2.2.4). We implicitly use the collection of neighborhoods $\{\eta_j\}_{j=1}^N$ to construct an undirected graph where each haplotype is a vertex, and edges connect haplotypes that agree on being in each other’s neighborhood. We use a greedy clique-finding procedure over the nearest neighborhoods to rapidly identify maximal cliques within this implicit graph. Haplotypes within each clique are assumed to belong to the same sprig, yielding sprig genotypes that we encode in $X^{(\ell)}$. Having encoded the locally inferred sprig genotypes in $X^{(\ell)}$, we summarize all of the remaining genealogical structure in $d^{(\ell)}$ via $\Omega^{(\ell)}$.

2.2.4 Calculating Relatedness Matrices from the LS Model

While LOCATER can accept any real symmetric matrix $\Omega^{(\ell)}$, in order to optimize power we model our choice of $\Omega^{(\ell)}$ in this paper after the expected genetic relatedness matrix (eGRM) proposed by Fan, Mancuso, and Chiang [36]. We cannot directly use their definition of the eGRM because the construction there requires a set of discrete clade calls. Constructing a local relatedness matrix from the distances $d^{(\ell)}$ is more complicated because, as described in Section 2.2.1, each column is calculated using an independent LS HMM. Thus, different columns of $d^{(\ell)}$ may disagree on the exact boundaries of particular clades in the underlying genealogy. This is a general feature of ancestry inference methods that work in a parallel or pairwise fashion across haplotypes. Rather than overriding the LS model and using hierarchical clustering or some other approach to try to align these clade calls, we generalize the eGRM to allow for this asynchrony. Our generalization, presented in Section 4.4, expresses an eGRM in terms of asymmetric distances like those provided by $d^{(\ell)}$ while allowing for such unaligned probabilistic clade calls.

This generalization of the eGRM requires us to use the distances within each column of $d^{(\ell)}$ to call a set

of nested neighborhoods around the corresponding haplotype. Calling these nested neighborhoods amounts to clustering the distances in each column of $d^{(\ell)}$. In order to do this efficiently for large n datasets, we developed a general, multithreaded, single-pass algorithm based on doubly linked lists to cluster real numbers on a closed interval when clusters must be separated by some fixed minimum distance. This approach allows us to cluster each column of $d^{(\ell)}$ in $\mathcal{O}(N)$ time. In experiments on simulated haplotype data, we achieve roughly an order of magnitude speedup over merge sort. In order to conserve memory, our implementation does not explicitly store the clustering results for each column of $d^{(\ell)}$. Rather, we use these clusters to directly construct columns of an asymmetric version of $\Omega^{(\ell)}$ on the fly, directly collapsing haplotype level relatedness down to sample level relatedness as we go. Taking the symmetric part of the resulting matrix gives us $\Omega^{(\ell)}$. Section 4.5 further details our construction of $\Omega^{(\ell)}$.

As a by-product of the clustering used to construct $\Omega^{(\ell)}$, we also return the nearest-neighbor set of each haplotype, which is then used to call sprigs and construct $X^{(\ell)}$ (Section 2.2.3). After calling sprigs using these neighborhoods, in order to avoid testing these sprigs in both $X^{(\ell)}$ and $\Omega^{(\ell)}$, we efficiently remove the structure associated with those sprigs from $\Omega^{(\ell)}$ using some additional statistics reported by our clustering algorithm before passing $X^{(\ell)}$ and $\Omega^{(\ell)}$ on to LOCATER for testing. All of these methods are available in `kalis v22.0`.

2.3 LOCATER Testing Routines

All of LOCATER's routines have been written in terms of matrix operations, allowing multiple quantitative traits to be tested in parallel with minimal additional computational cost. This includes the first implementation of a parallelized SD algorithm. For a given phenotype, this SD algorithm yields decoupled estimators for the effect β_j of each inferred clade genotype $X_j^{(\ell)}$. We then combine the independent two-sided p-values corresponding to these independent estimators via the Rényi Outlier Test [37] to obtain $p_{\text{SD}}^{(\ell)}$ at each locus. As demonstrated in [24], this approach yields considerable gains in power over alternative methods when very few (but more than one) of the β_j are non-zero; in other words, when more than one of the inferred genotype clades is associated with the phenotype. See Section 4.6 for details about the specific SD procedure we use.

As explained in Section 3 and shown in Figure 6 of [24], SD returns an updated version of the data, there denoted as $Y^{(L+1)}$, which is independent of the information extracted to calculate $p_{\text{SD}}^{(\ell)}$. It is this $Y^{(L+1)}$ that LOCATER passes on to calculate the quadratic form testing procedure to calculate $p_{\text{Q}}^{(\ell)}$, guaranteeing the independence of $p_{\text{SD}}^{(\ell)}$ and $p_{\text{Q}}^{(\ell)}$.

LOCATER also deploys several statistical and algorithmic innovations to efficiently calculate $p_{\text{Q}}^{(\ell)}$. Under the LOCATER model (Equation (1)), the score statistic against the null $\tau = 0$ is a quadratic form,

$$Y^\top P \Omega^{(\ell)} P Y \quad (4)$$

where $P = I - QQ^\top$ and $(A, G^{(\ell)}) = QR$ is the QR decomposition adjusting for the background covariates and the tested genotype $G^{(\ell)}$. In order to avoid launching unnecessary and expensive partial eigendecomposition routines at every target variant, we use a series of approximations to first assess whether the combined LOCATER p-value $p_C^{(\ell)}$ is sufficiently small to be interesting across any of the phenotypes. When it is, further eigendecomposition of $P\Omega^{(\ell)}P$ is deployed in order to obtain precise estimates.

We found that the Satterthwaite approximation [38], which is commonly used for testing quadratic forms [39], did not yield robust tail probability estimates for LOCATER. This may be because in this setting the matrices $P\Omega^{(\ell)}P$ are typically close to, but not quite, positive semi-definite — a key assumption of the Satterthwaite approximation. We overcame this obstacle with a new, robust tail approximation method for quadratic forms based on a shifted difference of chi-square random variables. In combination with our approximation stopping criteria, this tail approximation provides a basis for emerging genealogy-based association methods to reliably test local pairwise relatedness matrices, which may often not be positive semi-definite. Our tail approximation method has the added advantage that it admits three parameters — ν , δ_\star^2 , and δ_\dagger^2 — to help control inflation of the null distribution due to population structure and polygenicity. In effect, these parameters generalize genomic control to quadratic forms [40]. Importantly, these three parameters were chosen to be orthogonal to the spectral parameters governing the distribution of Equation (4). If any inflation is observed in the Q-Q plot of $p_Q^{(\ell)}$ p-values after running LOCATER, this orthogonal parameterization allows us to adjust $(\nu, \delta_\star^2, \delta_\dagger^2)$ and rapidly calculate new $p_Q^{(\ell)}$ p-values without requiring the re-estimation of local ancestries at any target locus (Section 4.7). An interpretation of our parameters $(\nu, \delta_\star^2, \delta_\dagger^2)$ is provided in Section 6.5.

We use a novel multi-threaded algorithm for efficiently projecting out background covariates when calculating the matrix traces needed for these tail approximations (Section 6.6). All final p-values p_Q involving eigenvalue terms are calculated using the fast Fourier transform implemented in the R package `QForm` [41]. Finally, we combine our three p-values, $(p_{\text{SMT}}^{(\ell)}, p_{\text{SD}}^{(\ell)}, p_Q^{(\ell)})$, using a modified version of Fisher's combination test we call MSSE (Section 4.8).

2.4 Type-I Error Control

In order to confirm the calibration of LOCATER empirically, we simulated 1,000 independent genomic datasets, each consisting of 30,000 samples of a 1 Mb chromosome. See Section 4.1 for more details. For each dataset, we simulated 1,000 independent phenotype vectors assuming no causal variants. See Section 4.2 for more details of our phenotype simulation approach. This yielded a total of 1 million independent phenotype vectors. We tested each phenotype vector for association with the ancestry inferred at the mid-point of the corresponding chromosome using LOCATER. We display a Q-Q plot for $-\log_{10}$ of those p-values, as well as for each LOCATER sub-test — SMT, SD, and QForm — in Section 5.2 (Supplementary Figures 6,7,8,9).

These Q-Q plots confirm that the p-values returned by each sub-test and the combined LOCATER p-value are all well calibrated under the null hypothesis.

2.5 Power

We compared LOCATER to ARG-Needle and standard SMT across a variety of genetic architectures. Following Zhang et al. [14], we ran ARG-Needle with mutation rates $\mu = 10^{-3}$ and $\mu = 10^{-5}$. For an additional comparison, we also ran ARG-Needle with $\mu = 10^{-7}$ (Section 6.8). We refer to these three variations of ARG-Needle as AN3, AN5, and AN7 respectively. In order to give ARG-Needle the best possible advantage, in each simulation, we gave ARG-Needle the true underlying ARG generated by `msprime` rather than asking ARG-Needle to infer that ARG from the observed haplotypes. Thus our results represent an upper bound on the performance of ARG-Needle. We assessed every possible combination of the following causal variant assumptions. We considered 3, 9, or 15 causal variants based on the number of independent causal alleles that were observed in the large follow-up study of GWAS hits by Abell et al. (see Figure 4b of [7]). We also considered causal variants with any derived allele count, derived allele count of 2 (doubletons only), or intermediate variants with derived allele count in [150,750). That is equivalent to a derived allele frequency (DAF) in [0.0025,0.0125). Lastly, we considered the case where all causal variants are observed or all causal variants are hidden. This yielded a total of 18 genetic architectures. Note that by ‘observed’, we mean that the causal variants were included in the dataset passed to each association method; by ‘hidden’, that they were not included in the dataset passed to each association method and thus could only be inferred via LD. In each simulation, causal variants were randomly assigned from among those fulfilling the required allele count requirements within a 10 kb window in the center of each simulated 1 Mb segment. **E1**► Under each genetic architecture, we estimated power as a function of the underlying total association signal strength: the $-\log_{10}$ p-value that one would obtain by testing the simulated phenotype Y with an oracle ANOVA model that “knows” ~~which are the active predictors~~ the causal variants and targets only those for testing. See Section 4.2 for a more precise definition. ◀**E1 Reply** To improve the interpretability of our power curves, following [24], we used the QR -decomposition to ensure that the total association signal was evenly split among the causal variants in every simulation. In other words, we ensured that the *observed* contribution of each causal variant to the total association signal was essentially equal for each simulated Y . See Section 4.2 for more details of our phenotype simulation approach.

For each method, we used 9,000 independent null simulations to estimate a genome-wide discovery threshold to maintain a family-wise error rate (FWER) below 0.05 (Section 6.10). This yielded a $-\log_{10}$ genome-wide discovery threshold 8.40 for LOCATER, 8.79 for SMT, 9.36 for AN7, 9.73 for AN5, and 9.78 for AN3. In calculating power, we count our causal region as “discovered” if a testing method has a $-\log_{10}$ p-value greater than their discovery threshold *anywhere* along the entire ~~1Mb~~ 1 Mb region. This definition reflects

how new associations are discovered in practice and provides a relatively strict benchmark. Each point of the resulting power curves was estimated via 1k independent samples: we simulated 10 independent phenotype vectors for each of 100 independent genomic datasets, each consisting of a 1 Mb chromosome sampled for 30k individuals. These power curves are available in Section 5.4 (Supplementary Figures 12,18,24,15,21,27). We summarize each of these curves with the estimated minimum signal strength required to achieve 80% power (lower is better). Figure 2 displays those estimates for LOCATER and SMT across all simulations where the underlying causal variants were observed; Figure 3, for those hidden. **E3**

Figure 2: Dotplot of total association signal strength required to achieve 80% power (lower is better) under various simulation conditions where all causal variants were observed. Total association signal strength is the $-\log_{10}$ p-value that one would obtain by testing the simulated phenotype Y with an oracle ANOVA model that “knows” ~~which are the active predictors~~ causal variants and targets only those for testing. Causal variant # denotes the number of simulated causal variants. Causal variant type “any” means any variant could be causal; “doubletons” means only doubletons could be causal; “DAC [150,750]” means only variants with a derived allele count in [150, 750], corresponding to a derived allele frequency in [0.0025, 0.0125], could be causal.

E3 Reply From both Figure 2 and Figure 3 we see that LOCATER ties or improves upon the statistical power of SMT and ARG-Needle across all settings. LOCATER can detect substantially weaker association signals than SMT and ARG-Needle when there are 9 or 15 causal variants. The power gains achieved by LOCATER over SMT in the observed causal variants case (Figure 2) are impressive given the fact that SMT has been shown to be surprisingly powerful in this context. Across all analogous power simulations with full

variant ascertainment and allelic heterogeneity, Link et al. found that SMT (which they refer to as "GWAS") had the same or more power than their ancestry-based quadratic form (eGRM) approach (see [19] Figure S2). To be clear, Link et al. did observe that the eGRM had more power than SMT in simulated array data with very incomplete variant ascertainment.

Comparing Figure 3 to Figure 2, we see that the relative power gains available from LOCATER are typically less in the case of hidden causal variants compared to the case of observed causal variants, but still substantial, across settings. The performance of ARG-Needle is the same in both figures because we provided ARG-Needle with the true underlying ARG in each simulation, making its performance unaffected by whether the simulated causal variants were observed or hidden. Except for the case of 3 doubletons, the power results reported in Figure 3 for SMT are remarkably similar to those reported in Figure 2 despite all of the causal variants being hidden. For the case of 3 doubletons, we see that the power of SMT is markedly reduced when the causal variants are hidden, making the relative power gain from LOCATER markedly large.

Figure 3: Dotplot of total association signal strength required to achieve 80% power (lower is better) under various simulation conditions where all causal variants were hidden. Total association signal strength is the $-\log_{10}$ p-value that one would obtain by testing the simulated phenotype Y with an oracle ANOVA model that “knows” ~~which are the active predictors~~ causal variants and targets only those for testing. Causal variant # denotes the number of simulated causal variants. Causal variant type “any” means any variant could be causal; “doubletons” means only doubletons could be causal; “DAC [150,750]” means only variants with a derived allele count in [150, 750], corresponding to a derived allele frequency in [0.0025, 0.0125], could be causal.

In order to confirm that these power results are robust to our choice of 10 kb as the size of the causal region, we replicated all of our experiments involving 9 causal variants assuming a 100 kb causal region. As can be seen in Section 5.5 (Supplementary Figures 30,31), the resulting power curves are very similar.

In order to compare LOCATER to the results one might obtain using sliding windows, we ran ACAT-O (STAAR without variant annotations) on our observed variant simulations from Figure 2, where any variant could be causal. Rather than testing all sliding windows for every simulation and effect size, we gave ACAT-O the precise location and width of the 10 kb causal window for each simulated dataset. This is an upper bound on the performance of ACAT-O in real-world settings where the location and size of the causal window are unknown. We ran ACAT-O in two different ways: one in which we restricted the variants considered to rare variants (MAF < 0.01) and another where all variants are tested regardless of frequency (Section 6.9). Similar to the other methods, the genome-wide discovery threshold for ACAT-O was determined via null simulations (Section 6.10). As can be seen from Figure 4, the performance of both oracle ACAT-O approaches is roughly the same as SMT. LOCATER maintains its power advantage.

Figure 4: Dotplot of total association signal strength required to achieve 80% power (lower is better) under various simulation conditions where all causal variants were observed, including comparison to oracle ACAT-O methods that are given the causal variant window. ACAT-O (rare) only tests variants with MAF < 0.01 whereas ACAT-O (all) tests all variants within the causal window. Total association signal strength is the $-\log_{10}$ p-value that one would obtain by testing the simulated phenotype Y with an oracle ANOVA model that “knows” ~~which are the active predictors~~ causal variants and targets only those for testing.

In Section 5.4, we pair each plot of power curves with a companion plot showing which LOCATER sub-test is driving the gain in power (Supplementary Figures 13,19,25,16,22,28). These results show that SD is typically the source of LOCATER power gains, not QForm, reflecting the statistical advantage of SD-based methods over quadratic form based procedures in the case of sparse signals [24]. While these simulations appear to imply that SD is only effective at capturing signals driven by very rare variants, this is expected since we only encoded very rare variants in the clade genotype matrix $X^{(\ell)}$ passed to SD. If SD was used to test the entire local genealogy at every locus, we may see increased statistical efficiency in incorporating common variant associations. We return to this point in the Discussion section. Substantial power gains are

possible via LOCATER in the case of multiple rare causal variants.

2.6 Localization

Alongside power, an important factor for real-world utility is the ability of an association method to accurately localize causal variant(s) within a relatively narrow genomic interval. In Section 5.4, we also pair each plot of power curves with a companion localization plot (Supplementary Figures 14,20,26,17,23,29). To measure the ability of a given method to localize the causal region, we calculated the distance between the most significant marker (lead variant) reported by a method and the midpoint of the causal region in every simulation, taking the average distance in the case of tied lead variants. We used these distances to estimate the width of an 80% confidence interval. This width represents the answer to the question: how large of a search window centered on the lead variant would an investigator need in order for that window to capture the midpoint of the causal region 80% of the time? In our plots reporting these confidence interval widths, we only report confidence interval widths at signal strengths where the corresponding method had at least 80% power to detect the causal region. As in our power curves, every point is estimated based on 1k independent simulated samples.

E21► For intuition, in this setting, 50% confidence intervals would be equivalent to 2 times the median distance from the lead variant to the causal region; 100% confidence intervals, 2 times the max distance from the lead variant to the causal region. Our 80% confidence intervals are a compromise between these two extremes, providing a estimate of how far out from a lead variant an investigator would have to look before feeling reasonably confident that they've reached the causal region. ◀ **E21 Reply**

As expected, both LOCATER and SMT struggled to localize the causal region more when the causal variants were hidden. The localization performance of ARG-Needle was unaffected by whether the simulated causal variants were observed or hidden because we provided ARG-Needle with the true underlying ARG in every simulation.

Across simulations where all variants in the causal region were potential causal variants, LOCATER was substantially more accurate in localizing the causal region than SMT or ARG-Needle, regardless of whether the causal variants were observed or hidden. In simulations with intermediate causal variants (derived allele count in [150, 750)), ARG-Needle and SMT slightly outperformed LOCATER for some signal strengths. Overall, LOCATER improved or effectively tied the localization accuracy of SMT and ARG-Needle in the intermediate causal variant case. With the exception of a few signal strengths when there were 9 causal variants, LOCATER also outperformed SMT and ARG-Needle in localizing the causal region when doubleton causal variants were observed. In all simulations where the doubleton causal variants were hidden, regardless of the number of causal variants, all methods performed very poorly in localization: all methods reported confidence intervals wider than 500 kb across all signal strengths where they achieved at least 80% power.

Overall, our results suggest that LOCATER can leverage allelic heterogeneity to improve the localization of trait mapping compared to standard SMT and ARG-Needle.

2.7 Scalability

The ability to scale to large modern datasets with hundreds of thousands of samples is essential for the success of any trait mapping approach and the size of local genealogies presents a significant challenge. Based on our simulations involving 30k samples (60k haplotypes), LOCATER took an average of 19.14 seconds (sd: 2.88 seconds) to perform sprig testing and an average of 3.07 minutes (sd: 0.98 minutes) to perform quadratic form testing at each variant. These simulations were run on a shared-time university HPC cluster with heterogeneous nodes hosting a mix of CPU architectures. All jobs requested 8 cores and 160 GB of memory. When combined with the computational overhead required to form the clade genotype matrix $X^{(\ell)}$ and $\Omega^{(\ell)}$ provided to LOCATER, our simulations required an average of 4.00 minutes (SD: 0.99 minutes) to test each target locus. When this is combined with the additional cost of performing ancestry inference with kaliskalis, our simulations required an overall average of 6.42 minutes per target locus. These results suggest that future applications of LOCATER that only use inferred clades $X^{(\ell)}$ and avoid use of $\Omega^{(\ell)}$ will achieve substantial computational savings.

E8► In parallel work, we ran LOCATER in combination with kalisv2 on a real sequencing dataset including ~~6795~~6,795 individuals and 101 correlated quantitative traits [1]. We divided the genome into ~~4580~~4,580 (partially overlapping) segments; each segment had an average of 13,000 variants. We allocated 12 cores and 60 GB of memory per segment. This allowed kalisv2 to store two checkpoints in memory. The average time required for kalisv2 and LOCATER to screen each segment was 32 minutes. That is equivalent to 3.35 years of single-core compute time. While substantial, this equates to 1.22 days using a cluster of 1000 CPUs. In other preliminary work, we have run LOCATER on ~~as many as~~ 12,964 genomes ~~on commodity hardware, which using commodity hardware. This experiment~~ required 8,257 CPU-days to ~~analyze screen~~ 4 traits quantitative traits for associations genome-wide (unpublished data). These results show that it is feasible to run LOCATER on a moderately large genomic dataset using an academic compute cluster. ◀**E8**

Reply

3 Discussion

We have presented a general framework for using inferred local ancestries to boost SMT association signals in the presence of allelic heterogeneity. To our knowledge, this is the first demonstration of any ancestry-testing approach that yields significant power gains over SMT in a genome-wide screen that includes non-coding regions. More importantly, our approach can be applied in conjunction with any ancestry inference engine,

thus providing a flexible association testing framework that can adapt to rapidly improving ancestry inference methods. As mentioned in the Introduction, we have demonstrated the real-world power gains attainable via LOCATER in a dataset of 6,795 Finnish genomes with extensive quantitative trait measurements [1].

E9► There, as in our simulations in Section 2.5, we find that LOCATER requires a less stringent genome wide discovery threshold than SMT. We believe this is due to the increased dependence between proximal tests induced by the dependence between proximal local genealogies, which suggests that it will be safe for investigators to apply their SMT discovery threshold for a given dataset (e.g. $10^{-8.5}$) to their LOCATER results. However, these results also suggest that running null simulations, analogous to those described in Section 6.10, to estimate a LOCATER-specific genome-wide discovery threshold for a given dataset will improve statistical power. Since LOCATER is parallelized across phenotypes, performing these null simulations is straightforward once a user can run LOCATER on a given dataset, and in our experience, requires computational resources comparable to a genome-wide association screen. ◀**E9 Reply**

E20► Our power simulations demonstrate that SMT typically outperforms in the context of high variant ascertainment (e.g. high coverage sequencing data). The fact that the performance of SMT is essentially unchanged when the causal variants are hidden (Figure 3) rather than observed (Figure 2), besides the case of 3 causal doubletons, reflects the fact that SMT typically detects associated loci by finding a lead variant that serves as a proxy for one or more of the causal variants. This is precisely the same strategy used by ARG-Needle: it infers an ARG from observed variants, discards the observed variants, and then places new hypothetical mutations on edges of the inferred ARG for testing. In short, in the case of high variant ascertainment, ARG-Needle effectively is just exchanging one set of potential proxy variants for another. Thus, it is not surprising that the set of observed variants does about as well as a resampled set of hypothetical variants in most cases.

Our hidden causal variant simulations expose the multiple testing tradeoff inherent to the ARG-Needle approach. In our simulations, running ARG-Needle with a mutation rate of 10^{-3} (AN3) or 10^{-5} (AN5) places a hypothetical variant on effectively every edge in the ARG. Since we provide ARG-Needle with the true ARG, the question becomes whether imputing the hidden causal variants was worth the additional testing burden of testing all of the clades in the ARG. It turns out this is true in the case of 3 hidden doubletons. There's about a 1/3 chance that there will be no strong proxy among the observed variants for any of the 3 causal doubletons when they are hidden, as evidenced by the abrupt kink in the SMT power curve where it reaches Power ≈ 0.66 (Supplementary Figure 15). However, this turns out to be false in all of our other simulations: there is virtually always a strong proxy variant somewhere along each simulated 1 Mb chromosome (Figure 3). ◀**E20 Reply**

The largest power gains demonstrated in this paper were seen in the case of multiple rare causal variants evaluated using the SD sub-test. This suggests that if we had tested more of the underlying tree structure at

each locus with SD we may have achieved even greater power. In other words, as mentioned above, we may have clustered each distance matrix and tested more common clades than simply the sprigs with SD. This approach would present more of the underlying ancestral tree to LOCATER via $X^{(\ell)}$ rather than via $\Omega^{(\ell)}$. Exploring the power of LOCATER at different points along the continuum between testing all of the ancestral structure with $X^{(\ell)}$ and testing all of the ancestral structure with $\Omega^{(\ell)}$, will be a focus of future research.

In conjunction with the new features added to `kalis` [17], LOCATER provides an efficient method for genome-wide testing that is ready for use on real-world datasets now. These new features involve several algorithms — a general HMM checkpointing algorithm, a fast clustering algorithm, and a fast trace calculation method — that will likely prove helpful for the acceleration of other ancestry inference and association methods. Our quadratic form tail approximation approach, based on a shifted difference of chi-square random variables, see Equation (11), provides a basis for emerging association methods to reliably test local relatedness matrices that may not be positive semi-definite.

Adequately adjusting for population structure when testing inferred local ancestries is an open and challenging problem. In this initial version of LOCATER, we allow principal components (PCs) to be included in A . As mentioned in Section 2.3, we also parameterized our quadratic form tail approximation in a way that accommodates genomic-control-like inflation adjustments without requiring recalculating the genealogy at any LOCATER target loci. See Section 6.5 for details and theoretical motivation. ~~However, future work~~

E23▶ While future analyses applying LOCATER or any local ancestry testing method to real genomic data will need to take special care when examining Q-Q plots for inflation, we believe that these methods have promise to avoid false discoveries due to residual population structure that would be mistakenly found by classic SMT. In parallel work applying LOCATER and SMT to real quantitative traits, adjusting both models for the same principal components, we observed that LOCATER p-values typically exhibit substantial genome-wide inflation even when SMT p-values appear well-calibrated [1]. This can be explained by the presence of cryptic confounding population structure: residual medium-to-fine scale population structure that is correlated with our trait(s) of interest but orthogonal to our PCs. Current SMT methods struggle to correct for this cryptic structure [42]. Since the vast majority of individual variants genome-wide are not correlated with these cryptic population features, their presence is hidden in SMT Q-Q plots.

However, by testing large sets of variants (clades) in every test rather than individual variants, LOCATER makes it much more likely that a given test statistic will be affected by some confounding aspect of residual population structure. Hence the inflation in the body of the unadjusted LOCATER Q-Q plots. We expect this sensitivity is not unique to LOCATER or even ARG-focused methods, but rather a feature of any method targeting allelic heterogeneity: testing sliding windows with ACAT-O. ~~We further address adjusting for population and cryptic structure in parallel work [1].~~

We expect that the observed inflation in LOCATER will prove to be a feature rather than a bug. First,

this inflation reveals the presence of residual population structure that is likely to still drive false positive discoveries in SMT results that appear to be calibrated. Since ARG-Needle is effectively performing SMT on inferred clades, such signals are also likely to beguile ARG-Needle results. Second, the observed inflation provides a means to adjust for the cryptic structure. As we show in [1], the inflation in LOCATER can be removed via a generalized version of genomic control to obtain calibrated p-values. A similar solution is not available to SMT. Since the vast majority of variants are not correlated with the confounding process, the SMT p-values already appear calibrated. In other words, genomic control implicitly assumes that the confounding effects impacting the p-values in the body of the Q-Q plot are exchangeable with those impacting the p-values in the tail of the Q-Q plot. By incorporating residual confounding signals into the majority of test statistics genome-wide, LOCATER makes this crucial assumption safer. In contrast, this assumption fails for SMT when relatively few of the individual variant test statistics are confounded. We leave further discussion and investigation of population structure adjustment for ancestry-based methods to [1] and future work. ◀E23 Reply

LOCATER makes a number of critical methodological advances towards powerful ancestry-based association testing. We expect that further work building on these advances alongside the application of LOCATER to more diverse datasets will yield new functional discoveries.

4 Methods

4.1 Haplotype Data Simulation

In order to assess the calibration and power of LOCATER, we simulated 100 genomic datasets, each consisting of a 1 Mb chromosome for 30k human samples (60k haplotypes). Each dataset was simulated using `msprime` [43]. In order to model the diversity of arising genomic datasets, 10k samples in each dataset were drawn from each of three 1000 Genomes populations – Yoruba, Han Chinese, and Central European. See Section 6.7 for further details.

4.2 Phenotype Simulation

To simulate each phenotype vector, following [23], we first simulated two background covariate vectors: a_1 a vector of independent standard Gaussian random variables and a_2 a vector of independent Rademacher random variables. We tested each phenotype vector assuming that these two background covariates were observed and included in A from Equation (1). For our null simulations – without any causal variants – this amounted to sampling each phenotype vector from $Y \sim N(a_1 + a_2, I)$. For our power simulations, in the middle of each 1 Mb region, we selected causal variants within a 10 kb causal window. We fully replicated these simulations under all 18 possible combinations of 3 parameters: the number of causal variants, the allele

frequency constraint imposed on those causal variants, and whether the causal variants were assumed to have been observed (called during sequencing) or hidden. More explicitly, we considered the case of 3, 9, or 15 causal variants. These causal variants were selected uniformly at random from among variants within the 10 kb causal window meeting the given allele frequency constraint. As our primary focus, we considered the case of no allele frequency restraint, in which case every variant in the 10 kb causal window had an equal chance of being selected as a causal variant. We also considered the case where all causal variants were constrained to be doubletons (present in two copies) and the case where all of the causal variants were constrained to have derived allele frequency in the half open interval $[0.0025, 0.0125)$. If the simulated chromosomes did not include the requisite number of causal variants within the 10 kb causal window, we rejected that simulated dataset and simulated a new set of chromosomes.

~~Given an active~~ **E2** ▶ Given a set of causal variants (variants with non-zero effects), we simulated Y while distributing the observed effects across the causal variants as evenly as possible by manipulating the QR -decomposition as done by [24]. Following their approach, let \mathcal{A} denote this selected set of causal variants and $X_{\mathcal{A}}$ denote the genotype matrix encoding those causal variants. We define the total association signal strength as the \$-\log_{10}\$ p-value that one would obtain by testing the resulting \$Y\$ with an oracle ANOVA model that “knows” the causal variants. More explicitly, it is the \$-\log_{10}\$ p-value that one would obtain from the likelihood ratio test comparing the oracle regression model \$Y \sim A\alpha + X_{\mathcal{A}}\beta + \epsilon\$ to its nested null model \$Y \sim A\alpha + \epsilon\$ where the variance of the noise term is known to be one: \$\epsilon \sim N(0, I_n)\$. Consider the QR -decomposition $QR = \tilde{X}_{\mathcal{A}}$, which we define as $P_{\mathcal{A}}^{\perp} X_{\mathcal{A}}$ with length-normalized columns. The sufficient statistic for ~~the oracle model testing oracle model against its nested null model~~ is $\|Q^{\top} Y\|_2^2$ with expected value $\|R\beta\|_2^2$. For a desired total association signal strength s , we solve $R\beta = \sqrt{F_{\chi_a^2}^{-1}(1 - 10^{-s})} \mathbf{1}$ to make the magnitude of each entry of β as similar as possible. Then, we simulate $Y = A\alpha + (\tilde{X}_{\mathcal{A}}\beta + P_{\mathcal{A}}^{\perp}\epsilon)$ where $\epsilon \sim N(0, I_n)$. This ensures that the observed $\hat{\beta} = R^{-1}Q^{\top}Y = \beta$ is stable across simulations and that the ~~$-\log_{10}$ p-value that one would obtain by testing the resulting Y with an oracle ANOVA model that “knows” the active predictors~~ total association signal strength will be approximately s in every simulation. **E5** ▶ Section 5.3 delineates the effect sizes, \$\beta\$ s, induced by this procedure as a function of \$s\$ and minor allele count. ◀E5 Reply ◀E2 Reply

4.3 Checkpointing Approach

To understand the need for checkpointing, as mentioned above and further detailed below, recall that we implicitly need the pairwise distance matrix $d^{(\ell)}$ from Equation (2) returned by `kalis` in order to obtain $X^{(\ell)}$ and $\Omega^{(\ell)}$ at each target locus ℓ . While the original `kalis` v1.0 release can efficiently propagate the forward and backward recursions to obtain the forward probability matrix $f^{(\ell)}$ and backward probability matrix $b^{(\ell)}$ needed to calculate $d^{(\ell)}$ at a single locus ℓ , obtaining the pair $(f^{(\ell)}, b^{(\ell)})$ at sequential positions ℓ is challenging

for HMMs due to the uni-directionality of the forward and backward recursions. The compute-minimizing approach would involve running a single pass of the forward algorithm – iterating the forward recursion to target locus 1, then locus 2, and so on until locus L – and a single pass of the backward recursion from target locus L to target locus 1 while storing $f^{(\ell)}$ and $b^{(\ell)}$ at every $\ell = 1, \dots, L$. Since each $f^{(\ell)}$ and $b^{(\ell)}$ consumes $8N^2$ bytes of memory (e.g.: 80 GB for $n = N/2 = 50,000$ haplotypes), this approach requires far too much storage for most genomic datasets. On the other hand, we have the memory-minimizing approach, where we restart the forward and backward recursions from the respective ends of the genomic segment for every target locus. While this approach only requires storing a single $f^{(\ell)}$ and a single $b^{(\ell)}$ at any given time, it demands far too much compute time for most genomic datasets, requiring $\mathcal{O}(L^2N^2)$ floating point operations (FLOPs) — a prohibitive cost. An attempt to rescue this approach by splitting the genome into smaller segments (running in chunks) would still require $\mathcal{O}(L^2N^2)$ compute time.

We provide a checkpointing algorithm that finds an optimal balance in this memory-compute trade-off, minimizing the compute time required given a fixed memory budget. The overall idea is to occasionally stop the forward recursion and store $f^{(\ell)}$ at its current position as a checkpoint (typically overwriting an old checkpoint) in order to avoid repeatedly restarting the forward recursion from the beginning of the genomic segment. We start with a user-specified memory budget capable of holding C checkpoints, each storing a $N \times N$ matrix of forward probabilities $f^{(\ell)}$. We run the backward recursion once across the entire chromosome or genomic segment, stopping at each consecutive target locus sequentially from the target locus with the largest position ($\ell = L$) to the target locus with the smallest position ($\ell = 1$). When the backward recursion stops at a given target locus, we run the forward recursion from the nearest checkpoint to meet the backward recursion and so obtain $X^{(\ell)}$ and $\Omega^{(\ell)}$ at that target locus. Note it is natural for us to perform this backwards along the genomic segment, since there is a slightly higher computational cost for the backward recursion and hence we favor repetitive restarts of the forward recursion.

Iterating from locus L down to locus 1 makes minimizing the compute required for the backward recursion trivial: we simply visit each locus sequentially in a single pass. The challenge is determining where and when to overwrite existing checkpoints to minimize the total distance (number of variants) that the forward algorithm needs to iterate over in order to provide forward matrices in reverse order $f^{(L)} \rightarrow f^{(L-1)} \rightarrow \dots$. In Section 6.3 we show how to solve for a schedule of checkpoints that achieves this minimum for any discrete time HMM, given storage for a fixed number of checkpoints C . We call this solution the optimal checkpointing schedule. After a forward matrix $f^{(\ell)}$ is obtained at a given target locus ℓ , this schedule instructs `kalis` which checkpoint to use to restart the forward recursion to obtain the next forward matrix at locus $\ell - 1$, and where to lay down new checkpoints (if any) as the forward recursion proceeds to that locus. The checkpointing schedule also dictates where to initialize the C checkpoints as we iterate the forward recursion to the first target locus L . Supplementary Figure 5 shows how the efficiency of our checkpointing algorithm scales in

both L and C . In the case where $L = 10^5$ equally-spaced target variants, Supplementary Figure 5 shows that the forward algorithm would be required to propagate over a total distance $D > 10^4 L$ target variants without checkpoints. Sufficient memory to store $C = 2$ checkpoints with our approach brings D under $100L$ target variants; $C = 8$ checkpoints brings D under $10L$ target variants. Even in the $L = 10^6$ case, which is larger than we would expect over any phased genomic segments since we only test target variants with moderately significant SMT p-values, $C = 10$ checkpoints brings D nearly down to $10L$ target variants.

Solving for the optimal checkpointing schedule can be computationally intensive for any given set of target loci. The version of the checkpointing schedule solver currently implemented in LOCATER assumes that target loci are evenly spaced. This simplification does not qualitatively change performance but allows us to solve for the optimal checkpointing strategy for a given L via a dynamic program, making the solution readily available (Section 6.3). Our checkpointing implementation is available via the `ForwardIterator` function and associated helper functions now provided in `kalis v22.0`.

Of course, for datasets with a large number of samples, there may not be sufficient capacity to store many checkpoints in memory. At a minimum, running `kalis` on n samples ($2n$ phased haplotypes) to obtain each $\Omega^{(\ell)}$ requires $32n^2$ bytes to store the forward and backward probabilities and another $8n^2$ bytes to store $\Omega^{(\ell)}$. Storing each additional checkpoint of forward probabilities requires $16n^2$ bytes. Given the nested nature of our checkpointing algorithm, most checkpoints can be stored on disk rather than memory, which comes at minimal computational cost as long as one or two of the checkpoints (the ones that are closest to the current target loci) are always kept in memory. We plan to add native support for storing file-backed checkpoints to `kalis` in the near future. Looking further ahead, `kalis` can already be distributed across machines, each running the LS model on a different subset of recipient haplotypes [17], but running LOCATER across distributed machines would require substantial network communication. Reducing this communication is a direction of future work.

4.4 Defining our Generalized Relatedness Matrix

Here we build up to our definition of a generalized eGRM, which we will pass to LOCATER as $\Omega^{(\ell)}$. We will construct $\Omega^{(\ell)}$ based on an asymmetric genetic distance matrix $d^{(\ell)} \in \mathbb{R}_{\geq 0}^{N \times N}$, such as the one provided by `kalis` (see Equation (2)), and a set of monotonic regularization functions g_1, \dots, g_N which we will introduce shortly. Recall that $d_{ij}^{(\ell)}$ measures the distance to haplotype i from haplotype j . The distance from any given haplotype to itself $d_{ij}^{(\ell)} = 0$. Let $\pi_j : [N] \rightarrow [N]$ be the permutation that sorts $d_{\cdot j}^{(\ell)}$ such that $d_{\pi_j(1)j}^{(\ell)} \leq d_{\pi_j(2)j}^{(\ell)} \leq \dots \leq d_{\pi_j(N)j}^{(\ell)}$. By convention, $\pi_j(1) = j$ and $\pi_j(N+1) = \pi_j(N)$. Using $d^{(\ell)}$, we define a local haplotype relatedness matrix $\Psi^{(\ell)} \in \mathbb{R}_{\geq 0}^{N \times N}$ with elements

$$\Psi_{ij}^{(\ell)} = \sum_{k=\pi_j^{-1}(i)}^N \psi_j(k) \quad \text{where} \quad \psi_j(k) = \frac{1}{k} g_k \left(d_{(k+1)j}^{(\ell)} - d_{kj}^{(\ell)} \right) \quad (5)$$

and each $g_k : \mathbb{R}_{\geq 0} \rightarrow \mathbb{R}_{\geq 0}$ is a monotonic function of x such that $g_k(0) = 0$. Note, given these definitions, $\psi_j(N) = 0$.

Our construction does not require the assumption of diploid samples but we will assume that here for ease of exposition. We will assume that the rows and columns of $\Psi^{(\ell)}$ are permuted such that haplotypes from the same sample are grouped together. This allows us to succinctly write our generalized eGRM in terms of Equation (5) as

$$\Omega^{(\ell)} = \text{sym} \left(B_{n,2}^\top \Psi B_{n,2} \right) = \frac{1}{2} \left(B_{n,2}^\top \Psi B_{n,2} + \left(B_{n,2}^\top \Psi B_{n,2} \right)^\top \right), \quad (6)$$

where $B_{n,2} = I_{n \times n} \otimes \mathbf{1}_2$ and \otimes is the Kronecker product.

In Section 6.2 we explicitly show how this construction of $\Omega^{(\ell)}$ generalizes the standard eGRM. In short, there we show that the eGRM can be expressed in terms of a haplotype similarity matrix assuming Hardy-Weinberg equilibrium and specific choices for the background covariates and allele frequency weights. Then we connect that haplotype similarity matrix representation to our choice of $\Psi^{(\ell)}$ in Equation (5) above.

4.5 Efficiently Constructing Clade Genotypes and our Generalized eGRM in LOCATER

Building on the notation above in Section 4.4, currently in LOCATER and this paper, we set

$$g_k(x) = \begin{cases} 0 & x < c \\ \min(x, 1) & x \geq c \end{cases} \quad (7)$$

for all k where our threshold $c = -0.2 \log(\mu)$ and μ is the mutation probability parameter provided to the LS model. This choice of regularization function(s) tends to filter out many low evidence clades. This function also restricts the clade matrix representation so that at most one mutation can be present on a given branch.

Given such a regularization, $\Psi_{\cdot j}$ has a series of nested neighborhoods of donor haplotypes along $i = \pi_j(1), \pi_j(2), \dots, \pi_j(N)$ where there are distances of at least c between adjacent level sets of distances. This allows us to represent the level sets of Ψ_{ij} as the solution to a clustering problem on the real interval from 0 to the maximum possible distance (D) where we require unique clusters to be at least distance c apart. Each cluster corresponds to a level set of donor haplotypes. We use a single-pass partial sorting algorithm based on doubly-linked-lists to solve this clustering problem in $\mathcal{O}(N)$ time. In our experiments on simulated haplotype data, our partial sorting algorithm achieves roughly an order of magnitude speed up over merge sort. Given

the definition of v in Equation (2), the maximum possible distance is $D = -\log(v) \approx 744.44$. This maximum is helpful in accelerating our implementation because the number of possible level sets (clusters) d is bounded above $d \leq \lceil D/c + 1 \rceil$, allowing us to efficiently pre-allocate sufficient memory to store the clustering solution. Since this partial sorting algorithm can be run in parallel for each recipient haplotype (for each column $\Psi_{\cdot j}$), we use a multi-threaded implementation in `kalis v22.0` that processes columns of Ψ in pairs. This allows us to directly compute our sample by sample matrix $B_{n,d}^\top \Psi B_{n,d}$ in a single pass. Symmetrizing this matrix yields $\Omega^{(\ell)}$ as defined in Equation (6). Future work will focus on reducing the computational and memory requirements of this symmetrization step.

4.6 Parallelized Stable Distillation Procedure

Given a matrix of inferred, clade-based genotypes $X^{(\ell)}$, we use the one-predictor-at-a-time SD procedure described in Equation 4 of [24] equipped with the simple quantile filter presented in Algorithm 1 of [24]. In this SD procedure, we take $(A, G^{(\ell)})$ as the background covariates when testing H_0^{SD} at a particular target locus ℓ . In short, using this approach, we “distill” one β_j for $j = 1, \dots, p$ at a time, obtaining an independent p-value for each. These p-values are then tested using the Rényi Outlier Test [37]. To run this procedure, LOCATER requires an estimated upper bound c on the number of independent causal clades: $c \geq |\{j : \beta_j \neq 0\}|$. By default and throughout this paper, we set $c = 16$. This bound c is used to set the simple quantile filtering threshold used during distillation. Explicitly, LOCATER sets the quantile filtering threshold $t = F_{c,p-c+1}^{-1}(0.01)$ where $F_{a,b}$ is the CDF of the Beta distribution with expectation $\frac{a}{a+b}$. By default, the maximum number of outliers considered by the Rényi Outlier Test is set to c .

4.7 Quadratic Form Testing & Tail Approximation

Let $QR = (A, G_j)$ be the QR-decomposition of the $n \times (q+1)$ matrix (A, G_j) and let $P = I - QQ^\top$ project onto the subspace orthogonal to the columns of (A, G_j) . Differentiating the Gaussian likelihood corresponding to Equation (1) with respect to τ yields $Y^\top P\Omega^{(\ell)}PY$ as the score statistic. Under H_0^Q ,

$$Y^\top P\Omega^{(\ell)}PY \sim \sum_{j=1}^n \lambda_j Z_j^2 \quad (8)$$

where each $Z_j \stackrel{iid}{\sim} N(0, 1)$. Following the approach of FastSKAT [39], we use partial eigendecomposition to obtain a computationally tractable approximation to this null distribution. Given, a top- k eigendecomposition in which we explicitly calculate the leading k eigenvalues, we have

$$Y^\top P\Omega^{(\ell)}PY \sim T_k + R_k \quad (9)$$

where $T_k = \sum_{j=1}^k \lambda_j Z_j^2$ and $R_k = \sum_{j=k+1}^n \lambda_j Z_j^2$. Since $\lambda_{k+1}, \dots, \lambda_n$ are unknown, FastSKAT proposes approximating the distribution of R_k using a single chi-square random variable $\tilde{R}_k \sim \alpha \chi_\nu^2$. The scale parameter α and degrees of freedom parameter ν are set to match the mean and variance of R_k , as initially proposed by Satterthwaite [38]:

$$\alpha = \frac{\eta_2}{\eta_1} \quad \nu = \frac{\eta_1^2}{\eta_2} \quad (10)$$

where $\eta_1 = \text{tr}(P\Omega^{(\ell)}P) - \sum_{j=1}^k \lambda_j$ and $\eta_2 = \|P\Omega^{(\ell)}P\|_{HS}^2 - \sum_{j=1}^k \lambda_j^2$. Substituting \tilde{R}_k in for R_k , FastSKAT uses $T_k + \tilde{R}_k$ as an approximate null distribution. This is still a linear combination of chi-square random variables, but since all of its parameters are now known, its distribution is readily available via the fast Fourier transform (FFT).

Unfortunately, in the context of LOCATER, \tilde{R}_k does not provide an adequate approximation to the distribution of R_k . First, the Satterthwaite approximation assumes that $P\Omega^{(\ell)}P$ is positive semi-definite (PSD). This is not at all guaranteed in our application. The closeness of any $P\Omega^{(\ell)}P$ to PSD will depend on the ancestry at ℓ as well as the user's choice of ancestry inference engine and clade-encoding method. The $P\Omega^{(\ell)}P$ matrices we have observed in the development of LOCATER are typically close to, but not exactly, PSD.

Second, we found that the tails of \tilde{R}_k tended to decay much faster than those of R_k , yielding anti-conservative estimates of more extreme p-values. This is a consequence of the fact that, \tilde{R}_k , as a gamma random variable, has an exponential right tail with a rate of decay $(2\alpha)^{-1}$. Since R_k is itself a linear combination of gamma random variables, the term with the largest coefficient governs the behavior in the right tail. More precisely, as long as some $\lambda_j > 0$ for $j > k$, R_k has an exponential right tail with rate $\left(2 \max_{j>k} \lambda_j\right)^{-1}$. Intuitively, this is a consequence of the fact that extreme observations of R_k in the far right tail are much more likely to arise from the term with the largest scale. We have analogous behavior in the left tail: as long as some $\lambda_j < 0$ for $j > k$, R_k has an exponential left tail with rate $\left(2 \min_{j>k} \lambda_j\right)^{-1}$. We found that the Satterthwaite approximation consistently over-estimates the rate of decay because $(2\alpha)^{-1}$ tends to be much larger than $\left(2 \max_{j>k} \lambda_j\right)^{-1}$. In fact, in the PSD case, $(2\alpha)^{-1} \geq \left(2 \max_{j>k} \lambda_j\right)^{-1}$ (Section 6.4).

In order to make the tails of our approximating distribution more robust (heavier) and obviate the PSD requirement, we propose approximating R_k with a difference of independent chi-square random variables:

$$\ddot{R}_k \sim |\lambda_k| (W' - W'' + \mu) \quad (11)$$

where $W' \sim \chi_a^2$ is independent of $W'' \sim \chi_b^2$. A derivation and explicit equations for the three scalar parameters a , b , and μ are given in Section 6.5 in terms of η_1 and η_2 . In short, these parameters are set so that our approximation matches the mean and variance of R_k while simultaneously minimizing $|\mu|$. Like $T_k + \tilde{R}_k$, $T_k + \ddot{R}_k$ is also a convolution of weighted chi-square random variables, making it just as easy to obtain

p-values for using the FFT. 793

Since the left and right exponential tails of \check{R}_k decay with rate $|2\lambda_k|^{-1} \leq \left(2 \max_{j>k} |\lambda_j|\right)^{-1}$, the tails of \check{R}_k at least as heavy as those of R_k . Of course, this alone does not guarantee accurate p-value estimation across the range of quantiles required for LOCATER. We must also accurately estimate of the body of the distribution across the full range of $P\Omega^{(\ell)}P$ we observe. Section 6.4 explains how the stopping criteria we use for top- k eigendecomposition in LOCATER addresses this issue and combines with \check{R}_k to yield reliable p_Q p-values. As shown in Section 6.5, a key advantage of our specification of \check{R}_k is that it can be easily generalized to accommodate three inflation parameters — ν , δ_{\star}^2 , and δ_{\dagger}^2 — that are orthogonal to a , b , and μ . There we show how these parameters arise when considering the presence of some unobserved confounder although, in practice, inflation of the null distribution may also be driven by polygenicity. The orthogonality allows us to tune ν , δ_{\star}^2 , and δ_{\dagger}^2 to help reduce any inflation observed in an initial Q-Q plot of p-values $p_Q^{(\ell)}$ without having to recalculate a , b , or μ . In other words, we can adjust ν , δ_{\star}^2 , and δ_{\dagger}^2 without having to re-calculate the genealogy at any target variant. While we do not make use of that capability in this paper, we expect it will play an important role in future analyses. 794
795
796
797
798
799
800
801
802
803
804
805
806

Critical to the scalability of this approach is the availability of η_1 and η_2 via fast trace calculations. FastSKAT [39] uses stochastic estimator of η_2 . Starting with a symmetric $\Omega^{(\ell)}$, LOCATER employs a fully parallelizable and distributable trace calculation routine that obtains both η_1 and η_2 with fewer than $7(q+1)n^2 + 3n(q+1)^2$ FLOPs (Section 6.6). Recall that q is the number of columns in our background covariate matrix A . 807
808
809
810
811

4.8 Combining p-values: MSSE 812

The classic Fisher method [44] of combining a vector of p-values $U = (U_1, \dots, U_p)^\top$ is based on comparing the test statistic 813
814

$$\mathcal{F}(U) = \sum_{i=1}^p -\log(U_i) \quad (12)$$

to its null distribution: a gamma distribution with scale parameter 1 and shape parameter p , whose survival function we denote by G_p . Since it takes the form of simple sum, \mathcal{F} implicitly considers all possible alternatives: any combination of the p-values may be non-null. However, in applications such as LOCATER, where we are only interested in certain subsets of alternatives, this can lead to an unnecessary loss in power. 815
816
817
818

The aim of LOCATER is to enhance SMT signals by leveraging allelic heterogeneity. In a genomic region with allelic heterogeneity, we expect the most significant LOCATER signal to arise at a target locus where the core marker at that locus tags one of the causal alleles. Thus, we are only interested in testing combinations of p-values where p_{SMT} is non-null. This in effect reduces the number of combinations of non-null p-value 819
820
821
822

alternatives we want to test, which we can achieve by using

823

$$\min (G_1^{-1}(\mathcal{F}(p_{\text{SMT}})), G_2^{-1}(\mathcal{F}(p_{\text{SMT}}, p_{\text{SD}})), G_2^{-1}(\mathcal{F}(p_{\text{SMT}}, p_{\text{Q}})), G_1^{-3}(\mathcal{F}(p_{\text{SMT}}, p_{\text{SD}}, p_{\text{Q}}))) \quad (13)$$

as our test statistic. The null distribution of this test statistic is easily pre-calculated via simulation. Based on 100 million samples, we verified that the null distribution was very smooth and rapidly converged to an exponential tail [45]. Hence we summarized the body of the null distribution via a monotone cubic spline and fit the exponential tail to obtain a rapidly computable test.

824

825

826

827

Data and Code Availability

828

Our `locator` R package, alongside installation instructions and the source code, is available on GitHub at <https://github.com/ryanchrist/locator>. Package documentation and a simple example vignette/article is available on the package website <https://ryanchrist.github.io/locator/>. Functions documented in the `locator` package provide an API exposing all of our testing subroutines. These functions accept generic matrices Y , A , $G^{(\ell)}$, $X^{(\ell)}$ and $\Omega^{(\ell)}$. Porting these matrices into R, as base matrices or sparse matrices, is sufficient to run the testing routines implemented in `locator`. While this requires some work, it allows users hoping to deploy `locator` in conjunction with other ancestry inference engines the ability to do so without re-implementing our testing procedures. The Simple LOCATER Example vignette on the `locator` package website provides further guidance on how to do this: https://ryanchrist.github.io/locator/articles/simple_gwas_example.html. Scripts required for replicating our results are available here: https://github.com/ryanchrist/locator_paper_scripts. They may be run under our public Docker image, which can be found here <https://hub.docker.com/r/rchrist7/mini-shark>. This image contains an installation of `msprime` and other python utilities used to run our haplotype simulations.

829

830

831

832

833

834

835

836

837

838

839

840

841

Acknowledgments

842

The authors would like to thank Nathan Stitzel, Chris Holmes, and Chris Spencer for helpful advice and discussions.

843

844

Funding

RC and XW were supported by NIH grants R01HG013371-01 and UM1HG008853 to IH. LJMA was partially supported by the EPSRC research grant "PINCODE", reference EP/X028100/1, and UKRI grant, "OCEAN", reference EP/Y014650/1. DS was supported by BBSRC research grant BB/S001824/1. For the purpose of Open Access, the authors have applied a CC BY public copyright license to any Author Accepted Manuscript version arising from this submission.

Declaration of Interests

The authors declare no competing interests.

References

- [1] Xinxin Wang, Ryan Christ, Erica Young, Chul Joo Kang, Indrani Das, Edward A Belter Jr, Markku Laakso, Louis JM Aslett, David Steinsaltz, Nathan O Stitzel, et al. Genealogy based trait association with locater boosts power at loci with allelic heterogeneity. *medRxiv*, pages 2024–11, 2024.
- [2] Quanli Wang, Ryan S Dhindsa, Keren Carss, Andrew R Harper, Abhishek Nag, Ioanna Tachmazidou, Dimitrios Vitsios, Sri VV Deevi, Alex Mackay, Daniel Muthas, et al. Rare variant contribution to human disease in 281,104 UK Biobank exomes. *Nature*, 597(7877):527–532, 2021.
- [3] Michael C Wu, Seunggeun Lee, Tianxi Cai, Yun Li, Michael Boehnke, and Xihong Lin. Rare-variant association testing for sequencing data with the sequence kernel association test. *The American Journal of Human Genetics*, 89(1):82–93, 2011.
- [4] Seunggeun Lee, Gonçalo R Abecasis, Michael Boehnke, and Xihong Lin. Rare-variant association analysis: study designs and statistical tests. *The American Journal of Human Genetics*, 95(1):5–23, 2014.
- [5] Farhad Hormozdiari, Anthony Zhu, Gleb Kichaev, Chelsea J-T Ju, Ayellet V Segrè, Jong Wha Joo, Hyejung Won, Sriram Sankararaman, Bogdan Pasaniuc, Sagiv Shifman, et al. Widespread allelic heterogeneity in complex traits. *The American Journal of Human Genetics*, 100(5):789–802, 2017.
- [6] GTEx Consortium. The GTEx consortium atlas of genetic regulatory effects across human tissues. *Science*, 369(6509):1318–1330, 2020.

- [7] Nathan S Abell, Marianne K DeGorter, Michael J Gloude-mans, Emily Greenwald, Kevin S Smith, Zihuai He, and Stephen B Montgomery. Multiple causal variants underlie genetic associations in humans. *Science*, 375(6586):1247–1254, 2022. 871
872
873
- [8] Yukihide Momozawa and Keijiro Mizukami. Unique roles of rare variants in the genetics of complex diseases in humans. *Journal of Human Genetics*, 66(1):11–23, 2021. 874
875
- [9] Elizabeth T Cirulli, Simon White, Robert W Read, Gai Elhanan, William J Metcalf, Francisco Tanudjaja, Donna M Fath, Efren Sandoval, Magnus Isaksson, Karen A Schlauch, et al. Genome-wide rare variant analysis for thousands of phenotypes in over 70,000 exomes from two cohorts. *Nature Communications*, 11(1):542, 2020. 876
877
878
879
- [10] Sebastian Zöllner and Jonathan K Pritchard. Coalescent-based association mapping and fine mapping of complex trait loci. *Genetics*, 169(2):1071–1092, 2005. 880
881
- [11] Mark J Minichiello and Richard Durbin. Mapping trait loci by use of inferred ancestral recombination graphs. *The American Journal of Human Genetics*, 79(5):910–922, 2006. 882
883
- [12] Katherine L Thompson and Laura S Kubatko. Using ancestral information to detect and localize quantitative trait loci in genome-wide association studies. *BMC Bioinformatics*, 14:1–10, 2013. 884
885
- [13] Sharon R Browning and Elizabeth A Thompson. Detecting rare variant associations by identity-by-descent mapping in case-control studies. *Genetics*, 190(4):1521–1531, 2012. 886
887
- [14] Brian C Zhang, Arjun Biddanda, Árni Freyr Gunnarsson, Fergus Cooper, and Pier Francesco Palamara. Biobank-scale inference of ancestral recombination graphs enables genealogical analysis of complex traits. *Nature Genetics*, 55(5):768–776, 2023. 888
889
890
- [15] Leo Speidel, Marie Forest, Sinan Shi, and Simon R Myers. A method for genome-wide genealogy estimation for thousands of samples. *Nature Genetics*, 51(9):1321–1329, 2019. 891
892
- [16] Jerome Kelleher, Yan Wong, Anthony W. Wohms, Chaimaa Fadil, Patrick K. Albers, and Gil McVean. Inferring whole-genome histories in large population datasets. *Nature Genetics*, 51(9):1330–1338, September 2019. 893
894
895
- [17] Louis J.M. Aslett and Ryan R. Christ. kalis: a modern implementation of the Li & Stephens model for local ancestry inference in R. *BMC Bioinformatics*, 25(86), 2024. 896
897
- [18] Wen-Wei Liao, Mobin Asri, Jana Ebler, Daniel Doerr, Marina Haukness, Glenn Hickey, Shuangjia Lu, Julian K. Lucas, Jean Monlong, Haley J. Abel, Silvia Buonaiuto, Xian H. Chang, Haoyu Cheng, Justin Chu, Vincenza Colonna, Jordan M. Eizenga, Xiaowen Feng, Christian Fischer, Robert S. Fulton, 898
899
900

- Shilpa Garg, Cristian Groza, Andrea Guarracino, William T. Harvey, Simon Heumos, Kerstin Howe, 901
Miten Jain, Tsung-Yu Lu, Charles Markello, Fergal J. Martin, Matthew W. Mitchell, Katherine M. 902
Munson, Moses Njagi Mwaniki, Adam M. Novak, Hugh E. Olsen, Trevor Pesout, David Porubsky, Pjotr 903
Prins, Jonas A. Sibbesen, Jouni Sirén, Chad Tomlinson, Flavia Villani, Mitchell R. Vollger, Lucinda L. 904
Antonacci-Fulton, Gunjan Baid, Carl A. Baker, Anastasiya Belyaeva, Konstantinos Billis, Andrew 905
Carroll, Pi-Chuan Chang, Sarah Cody, Daniel E. Cook, Robert M. Cook-Deegan, Omar E. Cornejo, 906
Mark Diekhans, Peter Ebert, Susan Fairley, Olivier Fedrigo, Adam L. Felsenfeld, Giulio Formenti, 907
Adam Frankish, Yan Gao, Nanibaa'A. Garrison, Carlos Garcia Giron, Richard E. Green, Leanne 908
Haggerty, Kendra Hoekzema, Thibaut Hourlier, Hanlee P. Ji, Eimear E. Kenny, Barbara A. Koenig, 909
Alexey Kolesnikov, Jan O. Korb, Jennifer Kordosky, Sergey Koren, HoJoon Lee, Alexandra P. Lewis, 910
Hugo Magalhães, Santiago Marco-Sola, Pierre Marijon, Ann McCartney, Jennifer McDaniel, Jacquelyn 911
Mountcastle, Maria Nattestad, Sergey Nurk, Nathan D. Olson, Alice B. Popejoy, Daniela Puiu, Mikko 912
Rautiainen, Allison A. Regier, Arang Rhie, Samuel Sacco, Ashley D. Sanders, Valerie A. Schneider, 913
Baergen I. Schultz, Kishwar Shafin, Michael W. Smith, Heidi J. Sofia, Ahmad N. Abou Tayoun, Françoise 914
Thibaud-Nissen, Francesca Floriana Tricomi, Justin Wagner, Brian Walenz, Jonathan M. D. Wood, 915
Aleksey V. Zimin, Guillaume Bourque, Mark J. P. Chaisson, Paul Flicek, Adam M. Phillippy, Justin M. 916
Zook, Evan E. Eichler, David Haussler, Ting Wang, Erich D. Jarvis, Karen H. Miga, Erik Garrison, 917
Tobias Marschall, Ira M. Hall, Heng Li, and Benedict Paten. A draft human pangenome reference. 918
Nature, 617(7960):312–324, 2023. 919
- [19] Vivian Link, Joshua G Schraiber, Caoqi Fan, Bryan Dinh, Nicholas Mancuso, Charleston WK Chiang, 920
and Michael D Edge. Tree-based QTL mapping with expected local genetic relatedness matrices. *The* 921
American Journal of Human Genetics, 110(12):2077–2091, 2023. 922
- [20] Jiazheng Zhu, Georgios Kalantzis, Ali Pazokitoroudi, Árni Freyr Gunnarsson, Hrushikesh Loya, Han 923
Chen, Sriram Sankararaman, and Pier Francesco Palamara. Fast variance component analysis using 924
large-scale ancestral recombination graphs. *bioRxiv*, pages 2024–08, 2024. [www.biorxiv.org/content/](http://www.biorxiv.org/content/10.1101/2024.08.31.610262) 925
[10.1101/2024.08.31.610262](http://www.biorxiv.org/content/10.1101/2024.08.31.610262). 926
- [21] Árni Freyr Gunnarsson, Jiazheng Zhu, Brian C Zhang, Zoi Tsangalidou, Alex Allmont, and Pier Francesco 927
Palamara. A scalable approach for genome-wide inference of ancestral recombination graphs. *bioRxiv*, 928
pages 2024–08, 2024. <https://www.biorxiv.org/content/10.1101/2024.08.31.610248>. 929
- [22] Ruoyi Cai and Sharon R Browning. Identity-by-descent mapping using multi-individual ibd with 930
genome-wide multiple testing adjustment. *bioRxiv*, pages 2025–01, 2025. 931

- [23] Xihao Li, Zilin Li, Hufeng Zhou, Sheila M Gaynor, Yaowu Liu, Han Chen, Ryan Sun, Rounak Dey, Donna K Arnett, Stella Aslibekyan, et al. Dynamic incorporation of multiple in silico functional annotations empowers rare variant association analysis of large whole-genome sequencing studies at scale. *Nature Genetics*, 52(9):969–983, 2020.
- [24] Ryan Christ, Ira Hall, and David Steinsaltz. Stable distillation and high-dimensional hypothesis testing. *arXiv:2212.12539v3*, 2024.
- [25] Pierrick Wainschtein, Deepti Jain, Zhili Zheng, L Adrienne Cupples, Aladdin H Shadyab, Barbara McKnight, Benjamin M Shoemaker, Braxton D Mitchell, Bruce M Psaty, Charles Kooperberg, et al. Assessing the contribution of rare variants to complex trait heritability from whole-genome sequence data. *Nature Genetics*, 54(3):263–273, 2022.
- [26] Matthew T Maurano, Richard Humbert, Eric Rynes, Robert E Thurman, Eric Haugen, Hao Wang, Alex P Reynolds, Richard Sandstrom, Hongzhu Qu, Jennifer Brody, et al. Systematic localization of common disease-associated variation in regulatory DNA. *Science*, 337(6099):1190–1195, 2012.
- [27] Hakhamanesh Mostafavi, Jeffrey P. Spence, Sahin Naqvi, and Jonathan K. Pritchard. Limited overlap of eQTLs and GWAS hits due to systematic differences in discovery. *bioRxiv*, 2022.
- [28] Ian Barnett, Rajarshi Mukherjee, and Xihong Lin. The generalized higher criticism for testing SNP-set effects in genetic association studies. *Journal of the American Statistical Association*, 112(517):64–76, 2017.
- [29] Yun Deng, Rasmus Nielsen, and Yun S. Song. Robust and accurate Bayesian inference of genome-wide genealogies for large samples. *bioRxiv*, 2024. <https://www.biorxiv.org/content/early/2024/03/16/2024.03.16.585351>.
- [30] Regev Schweiger and Richard Durbin. Ultrafast genome-wide inference of pairwise coalescence times. *Genome Research*, 33(7):1023–1031, 2023.
- [31] R Core Team. *R: A Language and Environment for Statistical Computing*. R Foundation for Statistical Computing, Vienna, Austria, 2024.
- [32] Jotun Hein, Mikkel H. Schierup, and Carsten Wiuf. *Gene Genealogies, Variation and Evolution: A Primer in Coalescent Theory*. Oxford University Press, 2004.
- [33] Na Li and Matthew Stephens. Modeling linkage disequilibrium and identifying recombination hotspots using single-nucleotide polymorphism data. *Genetics*, 165(4):2213–2233, 2003.

- [34] Yun S. Song. Na Li and Matthew Stephens on modeling linkage disequilibrium. *Genetics*, 203(3):1005–1006, 2016. 961
- [35] Lawrence R Rabiner. A tutorial on hidden Markov models and selected applications in speech recognition. *Proceedings of the IEEE*, 77(2):257–286, 1989. 962
- [36] Caoqi Fan, Nicholas Mancuso, and Charleston WK Chiang. A genealogical estimate of genetic relationships. *The American Journal of Human Genetics*, 109(5):812–824, 2022. 963
- [37] Ryan Christ, Ira Hall, and David Steinsaltz. The Rényi outlier test, 2024. <https://arxiv.org/abs/2411.13542>. 964
- [38] Franklin E Satterthwaite. An approximate distribution of estimates of variance components. *Biometrics Bulletin*, 2(6):110–114, 1946. 965
- [39] Thomas Lumley, Jennifer Brody, Gina Peloso, Alanna Morrison, and Kenneth Rice. FastSKAT: Sequence kernel association tests for very large sets of markers. *Genetic Epidemiology*, 42(6):516–527, 2018. 966
- [40] Bernie Devlin and Kathryn Roeder. Genomic control for association studies. *Biometrics*, 55(4):997–1004, 1999. 967
- [41] Ryan R. Christ and Louis J. M. Aslett. *QForm: Fast, safe CDF and PDF estimation and bounding for generalized chi-square random variables and quadratic forms*, 2024. <https://ryanchrist.r-universe.dev/QForm>. 968
- [42] Jennifer Blanc and Jeremy J Berg. Testing for differences in polygenic scores in the presence of confounding. *Genetics*, 230(2):iyaf071, 04 2025. 969
- [43] Jerome Kelleher, Alison M Etheridge, and Gilean McVean. Efficient coalescent simulation and genealogical analysis for large sample sizes. *PLoS Computational Biology*, 12(5):e1004842, 2016. 970
- [44] Frederick Mosteller and R.A. Fisher. Questions and answers. # 14. *The American Statistician*, 2(5):30–31, 1948. 971
- [45] August A Balkema and Laurens De Haan. Residual life time at great age. *The Annals of probability*, 2(5):792–804, 1974. 972
- [46] Jacob Cohen. *Statistical power analysis for the behavioral sciences*. routledge, 2013. 973
- [47] Andrew C Berry. The accuracy of the gaussian approximation to the sum of independent variates. *Transactions of the American Mathematical Society*, 49(1):122–136, 1941. 974

- [48] Carl-Gustav Esseen. On the liapounoff limit of error in the theory of probability. *Arkiv för Matematik, Astronomi och Fysik.*, A28:1–19, 1942. 989
990
- [49] tskit Developers. Demography. <https://msprime.readthedocs.io/en/stable/tutorial.html#demography>, 2020. Accessed: 2020-07-15. 991
992
- [50] Ryan N Gutenkunst, Ryan D Hernandez, Scott H Williamson, and Carlos D Bustamante. Inferring the joint demographic history of multiple populations from multidimensional SNP frequency data. *PLoS Genetics*, 5(10):e1000695, 2009. 993
994
995
- [51] Dominic Nelson, Jerome Kelleher, Aaron P Ragsdale, Claudia Moreau, Gil McVean, and Simon Gravel. Accounting for long-range correlations in genome-wide simulations of large cohorts. *PLoS Genetics*, 16(5):e1008619, 2020. 996
997
998
- [52] Jeffrey P Spence and Yun S Song. Inference and analysis of population-specific fine-scale recombination maps across 26 diverse human populations. *Science Advances*, 5(10):eaaw9206, 2019. 999
1000

5 Supplementary Figures

1001

5.1 Checkpointing

1002

Figure 5: **Computational Cost Using Optimal Checkpointing.** We consider the total number of variants the forward algorithm needs to iterate over (the total distance D it has to cover) in order to visit all loci on a chromosome of length L using our optimal checkpointing strategy. We plot the $\log_{10}(D/L)$ as a function of the number of checkpoints, C , available for chromosomes of three potential lengths indicated by the legend: $L \in \{10^4, 10^5, 10^6\}$. The dotted gray horizontal lines highlight that we are able to reduce D to under $100L$ with only 3 checkpoints and nearly $10L$ with only 10 checkpoints in all cases.

5.2 Null Simulation Q-Q Plots

Figure 6: **SMT Subtest Q-Q plot.** Q-Q plot of samples of $-\log_{10} \left(p_{\text{SMT}}^{(\ell)} \right)$ based on 1 million independent phenotype vectors simulated under the null hypothesis.

Figure 7: **SD Subtest Q-Q plot.** Q-Q plot of samples of $-\log_{10} \left(p_{\text{SD}}^{(\ell)} \right)$ based on 1 million independent phenotype vectors simulated under the null hypothesis.

Figure 8: **QForm Subtest Q-Q plot.** Q-Q plot of samples of $-\log_{10}(p_Q^{(\ell)})$ based on 1 million independent phenotype vectors simulated under the null hypothesis.

Figure 9: **LOCATER Combined Q-Q plot.** Q-Q plot of samples of $-\log_{10}(p_C^{(\ell)})$ based on 1 million independent phenotype vectors simulated under the null hypothesis.

5.3 Power Simulation Effect Sizes

As outlined in Section 4.2, we simulate phenotypes under the model

$$Y \sim A\alpha + X_{\mathcal{A}}\beta + \epsilon \quad (14)$$

where $X_{\mathcal{A}} \in \{0, 1, 2\}^{n \times a}$ is a matrix of genotypes corresponding to the $a = |\mathcal{A}|$ causal variants and $\epsilon \sim N(0, I_n)$.

The precise effect sizes induced by our simulation scheme for any specific $X_{\mathcal{A}}$ will be affected by the LD structure among those variants and the precise genotypes counts for each variant. Here we provide a rough estimate of the magnitude of the causal variant effect sizes β_j s induced as a function of the total association signal strength (s) and the minor allele count, which we denote by m in this section, by making a few simplifying assumptions. For a given causal variant with genotype vector X_j , the contribution of that genotype to $\mathbb{E}[Y]$ is

$$\beta X_j = \beta \|X_j\| \frac{X_j}{\|X_j\|} = \beta^* \frac{X_j}{\|X_j\|}. \quad (15)$$

We will call $|\beta_j|$ the absolute effect size and $\beta^* = |\beta_j| \|X_j\|$ the standardized absolute effect size.

We make the following assumptions. First, we ignore LD among the causal variants. More precisely, assuming that $X_{\mathcal{A}}^{\top} X_{\mathcal{A}}$ to be a diagonal matrix. Since causal variants are chosen at random within a window in each simulation, the expected pairwise LD between variants is close to zero, especially among rare variants. Second, we ignore migration and assume Hardy-Weinberg equilibrium (HWE). Recall that under the three population demographic model we use in our main power simulations, we draw 10,000 samples from each of three 1000 Genomes populations (YRI, CHB, and CEU). For each causal variant, we assume that minor alleles are only found in one of those three populations (ignoring migration) and that the genotypes for each causal variant are observed at HWE proportions within that population.

Under these assumptions, we will use $\tilde{\beta}(s, a, m)$ denote our estimate of the absolute value effect size $|\beta_j|$ assigned to a causal variant j with minor allele count m in a simulation with a causal variants and a desired total association signal strength s . Recall we sample $N^* = 20,000$ haplotypes from each of our three simulated sub-populations. Under our assumption of HWE and ignoring migration, after projecting out the population membership indicators in A , a genotype with minor allele count m has a genotype vector length $\|X_j\|$ of

$$l(m) = N^* \left(\frac{m}{N^*}\right) \left(1 - \left(\frac{m}{N^*}\right)\right) \left(1 - \frac{1}{N^*} - \frac{m}{N^*(N^*-1)}\right). \quad (16)$$

The last term arises as a finite sample correction in the following variance calculation, assuming HWE. Let $p^* = m/N^*$ and $X_{ij} = G_1 + G_2$ where $G_1, G_2 \sim \text{Bern}(p^*)$.

$$\mathbf{E} \left[\|X_j\|^2 \right] = \frac{N^*}{2} \text{Var} [X_{ij}] = \frac{N^*}{2} (\text{Var} [G_1] + \text{Var} [G_2] + 2\text{Cov} [G_1, G_2]) \quad (17)$$

$$= N^* (p^* (1 - p^*) + \mathbf{E} [\mathbf{E} [(G_1 - p^*) (G_2 - p^*) | G_1]]) \quad (18)$$

$$= N^* \left(p^* (1 - p^*) + (1 - p^*) \left(\frac{m-1}{N^*} - p^* \right) p^* + (-p^*) \left(\frac{m}{N^*-1} - p^* \right) (1 - p^*) \right) \quad (19)$$

$$= N^* p^* (1 - p^*) \left(1 - \frac{1}{N^*} - \frac{p^*}{(N^*-1)} \right) \quad (20)$$

Plugging this estimate of $\|X_j\|$ into Equation (15) and rearranging, we have

1031

$$\tilde{\beta}(s, a, m) = \beta^*(s, a) (l(m))^{-1}, \quad (21)$$

where $\beta^*(s, a)$ is an estimate of the standardized absolute effect size. Recall that we define the total association signal strength s as the $-\log_{10}$ p-value that one would obtain by testing the resulting Y with an oracle ANOVA model that “knows” the causal variants. Ignoring LD, the oracle ANOVA model p-value is given by the right tail probability $1 - F_{\chi^2_a} \left(\sum_{i=1}^a (\beta^*(s, a))^2 \right)$ where $F_{\chi^2_a}$ is the CDF of a χ^2 distribution with a degrees of freedom. Inverting this formula yields

1032

1033

1034

1035

1036

$$\beta^*(s, a) = \sqrt{\frac{1}{a} F_{\chi^2_a}^{-1} (1 - 10^{-s})}. \quad (22)$$

In Figure 10 we plot our estimate of the inverse genotype vector length, $(l(m))^{-1}$, as a function of minor allele count m . In Figure 11, we plot $\beta^*(s, a)$ as a function of s for the three values of a used in our simulations: 3, 9, and 15. As shown by Equation 21, our estimated absolute effect size $\tilde{\beta}(s, a, m)$ can be written as a product of $\beta^*(s, a)$ and $(l(m))^{-1}$. Table 1 provides $\tilde{\beta}(s, a, m)$ for various combinations of s , a , and m relevant to our simulations. To get a sense of the signal-to-noise ratio of these effects, recall that these effect sizes are applied in the context of a noise term that is Gaussian with scale 1. Thus entries in Table 1 can be interpreted as estimates of Cohen’s d : effect size divided by the standard deviation of the phenotype [46].

1037

1038

1039

1040

1041

1042

1043

1044

Figure 10: Estimated Inverse Length $(l(m))^{-1}$ as a function of minor allele count (m). Dots are displayed for all values of $m \in \{1, 2, \dots, 200\}$. Dots corresponding to values of m explicitly calculated in Table 1 are highlighted in orange.

Figure 11: Standardized Effect Size, $\beta^*(s)$, as a function of Total Association Signal Strength s .

MAC (m)	# Causal (a)	Total Association Signal Strength (s)												
		10	20	30	40	50	60	70	80	90	100	110	120	130
2	3	2.87	4.01	4.88	5.61	6.26	6.85	7.39	7.89	8.37	8.82	9.24	9.65	10.04
2	9	1.91	2.54	3.02	3.43	3.80	4.13	4.43	4.72	4.99	5.24	5.48	5.71	5.94
2	15	1.62	2.10	2.46	2.78	3.05	3.30	3.54	3.76	3.96	4.15	4.34	4.52	4.69
20	3	0.91	1.27	1.54	1.78	1.98	2.17	2.34	2.50	2.65	2.79	2.92	3.05	3.18
20	9	0.61	0.80	0.96	1.09	1.20	1.31	1.40	1.49	1.58	1.66	1.73	1.81	1.88
20	15	0.51	0.66	0.78	0.88	0.97	1.05	1.12	1.19	1.25	1.31	1.37	1.43	1.48
200	3	0.29	0.40	0.49	0.56	0.63	0.69	0.74	0.79	0.84	0.89	0.93	0.97	1.01
200	9	0.19	0.25	0.30	0.34	0.38	0.41	0.45	0.47	0.50	0.53	0.55	0.57	0.60
200	15	0.16	0.21	0.25	0.28	0.31	0.33	0.36	0.38	0.40	0.42	0.44	0.45	0.47

Table 1: Estimates of the absolute value of effect sizes used in our simulations, $\tilde{\beta}(s, a, m)$ as a function of m , a , and s . Minor allele counts should be interpreted as relative to a 20,000 alleles in a population; thus the estimates in this table correspond to variants with minor allele frequency 0.0001 ($m = 2$), 0.01 ($m = 20$), and 0.001 ($m = 200$).

◀E6 Reply

1045

5.4 Power, Source, & Localization Curves

1046

E4▶

Figure 12: Power comparison between SMT, ARG-Needle, and LOCATER in simulations where there are 3 causal variants, all observed. Power is displayed as a function of total association signal strength s : the $-\log_{10}$ p-value that one would obtain by testing with an oracle ANOVA model that “knows” the causal variants.

1047

Figure 13: Average proportion of signal driven by each sub-test within LOCATER in simulations where there are 3 causal variants, all observed. For every simulation, the proportion of signal attributed to the *SMT* part of LOCATER (solid lines) defined as $\log(p_{SMT}^{\ell^*}) / \log(p_{SMT}^{\ell^*} p_{SD}^{\ell^*} p_Q^{\ell^*})$ where ℓ^* is the locus ℓ that achieves the largest combined LOCATER p-value. The proportion of signal attributed to *SD* (dashed lines) or quadratic form testing, abbreviated as *Q* (dotted lines), is defined analogously. Proportions are displayed as a function of total association signal strength s : the $-\log_{10}$ p-value that one would obtain by testing with an oracle ANOVA model that “knows” the causal variants.

Figure 14: Width of confidence intervals (CI), centered on the position with the largest signal, that cover the midpoint of the causal region in 80% of simulations where there are 3 causal variants, all observed. In simulations where multiple variants are tied to have the largest association signal, we take the distance to the midpoint of the causal region to be the average distance from each of the tied variants. CI width is displayed as a function of total association signal strength s : the $-\log_{10}$ p-value that one would obtain by testing with an oracle ANOVA model that “knows” the causal variants.

Figure 15: Power comparison between SMT, ARG-Needle, and LOCATER in simulations where there are 3 causal variants, all hidden. Power is displayed as a function of total association signal strength s : the $-\log_{10}$ p-value that one would obtain by testing with an oracle ANOVA model that “knows” the causal variants.

Figure 16: Average proportion of signal driven by each sub-test within LOCATER in simulations where there are 3 causal variants, all hidden. For every simulation, the proportion of signal attributed to the SMT part of LOCATER (solid lines) defined as $\log(p_{SMT}^{\ell^*}) / \log(p_{SMT}^{\ell^*} p_{SD}^{\ell^*} p_Q^{\ell^*})$ where ℓ^* is the locus ℓ that achieves the largest combined LOCATER p-value. The proportion of signal attributed to SD (dashed lines) or quadratic form testing, abbreviated as Q (dotted lines), is defined analogously. Proportions are displayed as a function of total association signal strength s : the $-\log_{10}$ p-value that one would obtain by testing with an oracle ANOVA model that “knows” the causal variants.

Figure 17: Width of confidence intervals (CI), centered on the position with the largest signal, that cover the midpoint of the causal region in 80% of simulations where there are 3 causal variants, all hidden. In simulations where multiple variants are tied to have the largest association signal, we take the distance to the midpoint of the causal region to be the average distance from each of the tied variants. CI width is displayed as a function of total association signal strength s : the $-\log_{10}$ p-value that one would obtain by testing with an oracle ANOVA model that “knows” the causal variants.

Figure 18: Power comparison between SMT, ARG-Needle, and LOCATER in simulations where there are 9 causal variants, all observed. Power is displayed as a function of total association signal strength s : the $-\log_{10}$ p-value that one would obtain by testing with an oracle ANOVA model that “knows” the causal variants.

Figure 19: Average proportion of signal driven by each sub-test within LOCATER in simulations where there are 9 causal variants, all observed. For every simulation, the proportion of signal attributed to the *SMT* part of LOCATER (solid lines) defined as $\log(p_{SMT}^{\ell^*}) / \log(p_{SMT}^{\ell^*} p_{SD}^{\ell^*} p_Q^{\ell^*})$ where ℓ^* is the locus ℓ that achieves the largest combined LOCATER p-value. The proportion of signal attributed to *SD* (dashed lines) or quadratic form testing, abbreviated as *Q* (dotted lines), is defined analogously. Proportions are displayed as a function of total association signal strength s : the $-\log_{10}$ p-value that one would obtain by testing with an oracle ANOVA model that “knows” the causal variants.

Figure 20: Width of confidence intervals (CI), centered on the position with the largest signal, that cover the midpoint of the causal region in 80% of simulations where there are 9 causal variants, all observed. In simulations where multiple variants are tied to have the largest association signal, we take the distance to the midpoint of the causal region to be the average distance from each of the tied variants. CI width is displayed as a function of total association signal strength s : the $-\log_{10}$ p-value that one would obtain by testing with an oracle ANOVA model that “knows” the causal variants.

Figure 21: Power comparison between SMT, ARG-Needle, and LOCATER in simulations where there are 9 causal variants, all hidden. Power is displayed as a function of total association signal strength s : the $-\log_{10}$ p-value that one would obtain by testing with an oracle ANOVA model that “knows” the causal variants.

Figure 22: Average proportion of signal driven by each sub-test within LOCATER in simulations where there are 9 causal variants, all hidden. For every simulation, the proportion of signal attributed to the *SMT* part of LOCATER (solid lines) defined as $\log(p_{\text{SMT}}^{\ell^*}) / \log(p_{\text{SMT}}^{\ell^*} p_{\text{SD}}^{\ell^*} p_{\text{Q}}^{\ell^*})$ where ℓ^* is the locus ℓ that achieves the largest combined LOCATER p-value. The proportion of signal attributed to *SD* (dashed lines) or quadratic form testing, abbreviated as *Q* (dotted lines), is defined analogously. Proportions are displayed as a function of total association signal strength s : the $-\log_{10}$ p-value that one would obtain by testing with an oracle ANOVA model that “knows” the causal variants.

Figure 23: Width of confidence intervals (CI), centered on the position with the largest signal, that cover the midpoint of the causal region in 80% of simulations where there are 9 causal variants, all hidden. In simulations where multiple variants are tied to have the largest association signal, we take the distance to the midpoint of the causal region to be the average distance from each of the tied variants. CI width is displayed as a function of total association signal strength s : the $-\log_{10}$ p-value that one would obtain by testing with an oracle ANOVA model that “knows” the causal variants.

Figure 24: Power comparison between SMT, ARG-Needle, and LOCATER in simulations where there are 15 causal variants, all observed. Power is displayed as a function of total association signal strength s : the $-\log_{10}$ p-value that one would obtain by testing with an oracle ANOVA model that “knows” the causal variants.

Figure 25: Average proportion of signal driven by each sub-test within LOCATER in simulations where there are 15 causal variants, all observed. For every simulation, the proportion of signal attributed to the *SMT* part of LOCATER (solid lines) defined as $\log(p_{\text{SMT}}^{\ell^*}) / \log(p_{\text{SMT}}^{\ell^*} p_{\text{SD}}^{\ell^*} p_{\text{Q}}^{\ell^*})$ where ℓ^* is the locus ℓ that achieves the largest combined LOCATER p-value. The proportion of signal attributed to *SD* (dashed lines) or quadratic form testing, abbreviated as *Q* (dotted lines), is defined analogously. Proportions are displayed as a function of total association signal strength s : the $-\log_{10}$ p-value that one would obtain by testing with an oracle ANOVA model that “knows” the causal variants.

Figure 26: Width of confidence intervals (CI), centered on the position with the largest signal, that cover the midpoint of the causal region in 80% of simulations where there are 15 causal variants, all observed. In simulations where multiple variants are tied to have the largest association signal, we take the distance to the midpoint of the causal region to be the average distance from each of the tied variants. CI width is displayed as a function of total association signal strength s : the $-\log_{10}$ p-value that one would obtain by testing with an oracle ANOVA model that “knows” the causal variants.

Figure 27: Power comparison between SMT, ARG-Needle, and LOCATER in simulations where there are 15 causal variants, all hidden. Power is displayed as a function of total association signal strength s : the $-\log_{10}$ p-value that one would obtain by testing with an oracle ANOVA model that “knows” the causal variants.

Figure 28: Average proportion of signal driven by each sub-test within LOCATER in simulations where there are 15 causal variants, all hidden. For every simulation, the proportion of signal attributed to the *SMT* part of LOCATER (solid lines) defined as $\log(p_{SMT}^{\ell^*}) / \log(p_{SMT}^{\ell^*} p_{SD}^{\ell^*} p_Q^{\ell^*})$ where ℓ^* is the locus ℓ that achieves the largest combined LOCATER p-value. The proportion of signal attributed to *SD* (dashed lines) or quadratic form testing, abbreviated as *Q* (dotted lines), is defined analogously. Proportions are displayed as a function of total association signal strength s : the $-\log_{10}$ p-value that one would obtain by testing with an oracle ANOVA model that “knows” the causal variants.

Figure 29: Width of confidence intervals (CI), centered on the position with the largest signal, that cover the midpoint of the causal region in 80% of simulations where there are 15 causal variants, all hidden. In simulations where multiple variants are tied to have the largest association signal, we take the distance to the midpoint of the causal region to be the average distance from each of the tied variants. CI width is displayed as a function of total association signal strength s : the $-\log_{10}$ p-value that one would obtain by testing with an oracle ANOVA model that “knows” the causal variants.

5.5 Power Curves: 100 kb Causal Window

Figure 30: Power comparison between SMT and LOCATER in simulations where there are 9 causal variants within a 100 kb causal window, all observed. Compare to Figure 18. Power is displayed as a function of total association signal strength s : the $-\log_{10}$ p-value that one would obtain by testing with an oracle ANOVA model that “knows” the causal variants.

Figure 31: Power comparison between SMT and LOCATER in simulations where there are 9 causal variants within a 100 kb causal window, all hidden. Compare to Figure 21. Power is displayed as a function of total association signal strength s : the $-\log_{10}$ p-value that one would obtain by testing with an oracle ANOVA model that “knows” the causal variants.

◀E4 Reply

6 Supplementary Methods

1050

6.1 Running `kalis`

1051

As described in [33, 15, 17], the LS model requires a pre-defined recombination map and two model parameters: μ governing the probability of mutations and N_e recombination events. Throughout this paper, we provided `kalis` with the true recombination map under which haplotypes were simulated. Throughout we set $\mu = 10^{-4}$ and $N_e = 10^{-16}$.

1052

1053

1054

1055

6.2 Connecting our Generalized eGRM to the standard eGRM

1056

1057

Here we explain the connection between the generalized eGRM matrix defined in Equation (6) to the standard eGRM, as presented in Equation 1 in [19]. For clarity, throughout this section we will assume that we are considering a particular target locus ℓ and suppress indexing objects by ℓ .

1058

1059

1060

Let n be the number of samples in a genetic dataset. Let

1061

$$\mathcal{H}_i = \{j \in \mathbb{N} : \text{haplotype } j \text{ belongs to sample } i\} \quad (23)$$

and $\Delta_i = |\mathcal{H}_i|$ be the ploidy of sample i so that $N = \sum_i \Delta_i$ is the total number of haplotypes in the sample.

1062

Let $H \in \{0, 1\}^{N \times p}$ be the haplotype matrix encoding the carriers of p in a genomic region centered on locus

1063

ℓ or on an inferred local ancestral tree at ℓ as in [19]. Let $f_k = \sum_{i=1}^n H_{ik}$ be the allele frequency of the k^{th}

1064

variant. Given a set of weights $w \in \mathbb{R}_{\geq 0}^p$ which may be based on inferred clade probabilities or otherwise, we

1065

start with a weighted haplotype relatedness matrix

1066

$$\Psi' = \sum_{k=1}^p \frac{w_k}{f_k} H_{\cdot k} H_{\cdot k}^\top. \quad (24)$$

We can write this more compactly as $\Psi' = \tilde{H} \tilde{H}^\top$ where $\tilde{H} = H \text{diag} \left(\sqrt{w_k / f_k} \right)$. Assuming an additive model,

1067

we can collapse this weighted haplotype relatedness matrix to a weighted genotype relatedness matrix

1068

$$\Omega'_{ij} = \sum_{k \in \mathcal{H}_i} \sum_{l \in \mathcal{H}_j} \Psi'_{kl}. \quad (25)$$

For simplicity, from here forward we will assume that (1) the ploidy of each sample Δ_i is the same constant

1069

ploidy Δ for all samples i and (2) the rows and columns of Ψ' are permuted such that haplotypes from

1070

the same sample are grouped together. This allows us to more succinctly write $\Omega' = B_{n,\Delta}^\top \Psi' B_{n,\Delta}$ where

1071

$B_{n,\Delta} = I_{n \times n} \otimes \mathbf{1}_\Delta$ and \otimes is the Kronecker product. This gives us

1072

$$\begin{aligned}\Omega' &= B_{n,\Delta}^\top \left(\sum_{k=1}^p \frac{w_k}{f_k} H_{\cdot k} H_{\cdot k}^\top \right) B_{n,\Delta} \\ &= \sum_{k=1}^p \frac{w_k}{f_k} B_{n,\Delta}^\top H_{\cdot k} H_{\cdot k}^\top B_{n,\Delta} \\ &= \sum_{k=1}^p \frac{w_k}{f_k} G_{\cdot k} G_{\cdot k}^\top \\ &= G \text{diag}(w_k/f_k) G^\top\end{aligned}$$

where $G \{0, 1, 2\}^{n \times p}$ is the genotype matrix corresponding to the p variants on the local tree. Now if we consider a model without any background covariates besides an intercept, we have $P = I - \frac{1}{n} \mathbf{1}\mathbf{1}^\top$.

1073

1074

$$\begin{aligned}P\Omega'P &= PG \text{diag}(w_k/f_k) G^\top P \\ &= \sum_{k=1}^p \frac{w_k}{f_k} PG_{\cdot k} (PG_{\cdot k})^\top\end{aligned}\tag{26}$$

The standard GRM assumes that each variant is in Hardy–Weinberg equilibrium. So in order to compare Equation (26) to the standard GRM, let us also assume that each of our p variants is in perfect Hardy–Weinberg equilibrium. That means each centered genotype vector $PG_{\cdot k} = G_{\cdot k} - 2f_k \mathbf{1}$, allowing us to write

1075

1076

1077

1078

$$\begin{aligned}(P\Omega'P)_{ij} \big|_{HW} &= \sum_{k=1}^p w_k \frac{(G_{ik} - 2f_k)(G_{jk} - 2f_k)}{f_k} \\ &= \frac{1}{p} \sum_{k=1}^p w_k \eta(f_k) \frac{(G_{ik} - 2f_k)(G_{jk} - 2f_k)}{2f_k(1 - f_k)}\end{aligned}\tag{27}$$

where $\eta(f_k) = 2p(1 - f_k)$. The standard GRM defined in Equation 1 of [19] is just a simple special case of Equation (27) where $\eta(f_k) = 1$ for all arguments f_k and $w_k = 1$ for all variants. The eGRM allows for non-equal weights w_k . Some simple algebra shows that our choice of a $\frac{1}{k}$ in our definition of ψ in Equation 5 corresponds to $\eta(f_k) = 2p(1 - f_k)$. We chose this weighting in order to upweight variants that have low derived allele frequency. Under mild selection pressure, we expect very common derived alleles to have smaller phenotypic effects. With very minimal modification of our current implementation, any weighting function of the derived allele frequency η may be used. We plan to make such a general weighting function available in future versions of LOCATER.

1079

1080

1081

1082

1083

1084

1085

1086

Having rewritten the eGRM in terms of Ψ' , let us connect Ψ' to Ψ from Equation (5), which we use

1087

in our generalized eGRM construction. Consider the special case where we have a chromosome without recombination so that our genetic distance matrix $d^{(\ell)}$ reflects a single marginal tree, making all clade calls are aligned. If we also set each g_k to the identity function and assume consistency across columns of our distance matrix so that $d_{(k+1)j}^{(\ell)} - d_{kj}^{(\ell)} = w_k$ for all haplotypes j carrying a variant of frequency k , then Ψ as defined in Equation (5) would be symmetric and proportional to the the weighted haplotype matrix Ψ' as defined in Equation (24). As a consequence, the similarity matrix we would obtain $B_{n,d}^\top \Psi B_{n,d}$ would equal $\Omega' = B_{n,d}^\top \Psi' B_{n,d}$. However, due to recombination, the distance matrix Ψ is not symmetric and the distances across columns may not be consistent with a single tree structure. Rather than attempting to synchronize the distances by clustering them into a tree structure, we trust the LS model, taking the distances as representing a weighted convolution of nearby tree structures.

6.3 Optimal Checkpointing Scheduler

Let j index the positions of our L target loci from smallest to largest. We will solve for an optimal checkpointing schedule for the forward algorithm to propagate to target loci sequentially in reverse order, from $j = L$ to $j = 1$. We will use $j = 0$ to index the HMM prior hidden state probabilities used to initialize the forward algorithm. Informally, for each subsequent target locus, a checkpointing schedule states the checkpoint at which the forward algorithm should start and the intervening target loci where it should stop to store any new checkpoints on its way to reach that target locus. To optimize this schedule, we do not need to consider storing checkpoints at variants that are not target loci since any such checkpointing strategy could be improved by moving those checkpoints to target loci. Thus, our checkpointing schedule will only need to render integers in $1, \dots, L$.

Optimizing the checkpointing schedule is based on recycling the memory used to store checkpoints that are no longer being used. We solve for an optimal checkpointing schedule where the computational cost to propagate the forward algorithm from any position to any other may be arbitrary. The checkpointing routine that minimizes the total computational cost in this setting may be obtained by solving nested optimization problems. While tractable, solving this system of nested optimization problems can be rather time consuming to solve. Therefore, we subsequently consider the special case where the cost of propagating the forward algorithm between consecutive target positions is constant. For example, in a genomics context this may be true if we have equally spaced target loci. In that special case, we explain how the optimal checkpointing schedule can be solved rapidly via dynamic programming.

We start with the general case. Define $g : \mathbb{Z}_{\geq 0}^2 \rightarrow \mathbb{R}_{\geq 0}$ such that $g(s, t)$ is the cost of propagating the forward algorithm directly from target locus s to target locus t , defined to be zero whenever $s = t$. Define $f : \mathbb{Z}_{\geq 0}^3 \rightarrow \mathbb{R}_{\geq 0}$ such that $f(s, t, c)$ is the minimal cost required for the forward algorithm to sequentially visit target loci $t, t - 1, t - 2, \dots, s$ given c available checkpoints and a known set of forward probabilities (i.e. a

checkpoint, or the HMM prior) at s . That is, $f(s, t, c)$ corresponds to the cost of the (unknown) optimal schedule given memory for c checkpoints. 1121
1122

We aim to obtain a checkpoint schedule which enables us to achieve the cost $f(0, T, C)$. This objective is defined by the three following equations. First, if there are no checkpoints available, simply propagating to each target locus sequentially from the initial position is our only option, which comes with cost $f(x, y, 0) = \sum_{j=x}^y g(x, j)$. Second, if we already know the forward probabilities at a given position, there is no computational cost to obtaining them, so $f(x, x, c) = 0$ for any c . Third, we have the following recursive relationship 1123
1124
1125
1126
1127

$$f(s, t, c) = \min_{j \in \mathbb{Z}_{\geq 0}: j \in (s, t]} g(s, j) + f(s, j - 1, c) + f(j, t, c - 1). \quad (28)$$

Intuitively, this recursion expresses the idea that solving for an optimal checkpointing schedule with c checkpoints over an interval can be thought of as placing one of the checkpoints at some optimal index to divide the interval so that the upper part is solved with $c - 1$ checkpoints; the lower part, c checkpoints. We will write that optimal index, the argument that achieves the minimum of Equation (28) as 1128
1129
1130
1131

$$h(s, t, c) = \operatorname{argmin}_{j \in \mathbb{Z}_{\geq 0}: j \in (s, t]} g(s, j) + f(s, j - 1, c) + f(j, t, c - 1). \quad (29)$$

Together, these three equations allow us to solve for an optimal checkpoint schedule by transversing a bifurcating tree from right to left, from the leaves toward the root, where each node can be identified with an optimal index h . 1132
1133
1134

While solving this recursion is tractable, it can be rapidly accelerated if we assume that the computational cost of propagating the forward probabilities between adjacent target loci is fixed. In other words, we assume $g(x - 1, x)$ is constant for all target indices $x \in \{1, 2, \dots, L\}$. This makes the computational cost a simple linear function of the distance between indices, yielding the following simplified system of equations. 1135
1136
1137
1138

Analogous to above, we have $\tilde{f}(i, 0) = i(i + 1)/2$ and $\tilde{f}(0, c) = 0$ for all c . Our recursions simplify to 1139

$$\tilde{f}(i, c) = \min_{j \in \mathbb{Z}_{\geq 0}: j \leq i} j + \tilde{f}(j - 1, c) + \tilde{f}(i - j, c - 1), \quad (30)$$

$$\tilde{h}(i, c) = \operatorname{argmin}_{j \in \mathbb{Z}_{\geq 0}: j \leq i} j + \tilde{f}(j - 1, c) + \tilde{f}(i - j, c - 1). \quad (31)$$

For some pre-specified maximum number of target loci L and maximum number of available checkpoints C ,
 let us define two matrices that we will use as look-up tables for our dynamic program. Let $F \in \mathbb{R}_{\geq 0}^{(L+1) \times (C+1)}$
 such that $F_{ij} = \tilde{f}(i-1, j-1)$ and $H \in \mathbb{Z}_{\geq 0}^{(L+1) \times (C+1)}$ such that $H_{ij} = \tilde{h}(i-1, j-1)$. Given the nested
 nature of 30 and 31, we can rapidly solve for each entry of F and H by synchronously iterating over the
 entries of both matrices in column-major order. Once we have a completed F and H , we can use 31 to read
 off the appropriate entries in table H to obtain the optimal checkpointing schedule for any problem with up
 to L target variants and C available checkpoints. Methods for constructing H and reading it to construct a
 checkpoint schedule are available in `kalis v22.0`. This implementation covers the slightly more general case
 than (30) where the cost depends on distance but is independent of locus. This corresponds to a translation
 invariance assumption that does not cover the most general case in (28).

6.4 Robust Tail Approximation

The rate of tail decay under Satterthwaite approximation, $(2\alpha)^{-1}$, tends to chronically over-estimate the true
 rate of decay in the tails of R_k , which is $\left(2 \max_{j>k} \lambda_j\right)^{-1}$. In fact, in the positive semi-definite case, the Hölder
 Inequality (with $p = 1, q = \infty$) shows that

$$(2\alpha)^{-1} = \left(\frac{2 \sum_{j>k} \lambda_j^2}{\sum_{j>k} \lambda_j} \right)^{-1} = \left(\frac{2 \sum_{j>k} |\lambda_j \lambda_j|}{\sum_{j>k} |\lambda_j|} \right)^{-1} \geq \left(\frac{2 \left(\sum_{j>k} |\lambda_j| \right) \max_{j>k} |\lambda_j|}{\sum_{j>k} |\lambda_j|} \right)^{-1} = \left(2 \max_{j>k} \lambda_j \right)^{-1}. \quad (32)$$

Our proposed approximation \tilde{R}_k addresses this problem by using tails that decay at least as slowly as R_k .
 However, this alone does not guarantee accurate p-value estimation across the range of quantiles required
 for LOCATER. Our goal is to estimate $-\log$ p-values given by $-\log(1 - F_{R_k}(x))$ where F_{R_k} is the CDF
 of R_k . Since R_k has exponential tails, $-\log(1 - F_{R_k}(x))$ is a linear function of x for large x , with a slope
 determined the rate of decay in the tail and an intercept that is determined by the body of the distribution
 of R_k . Thus, estimating the body of the distribution of R_k is required for accurately estimating p-values in
 the tail of the distribution of R_k .

We will make use of the observation that the spectrum of a typical $P\Omega^{(\ell)}P$ observed in our genomics
 application has the following three-part structure. First the spectrum decays over the leading ten or so
 eigenvalues. This segment of the spectrum corresponds to large-scale population structure, orthogonal to
 the columns of A , encoded by $\Omega^{(\ell)}$. Then the spectrum levels off to a plateau where the eigenvalues are
 all roughly of the same magnitude. This second segment of the spectrum corresponds to the fine-scale
 population structure encoded by $\Omega^{(\ell)}$. Most eigenvalues in this second segment of the spectrum are positive
 with occasional negative eigenvalues interspersed. Eventually this plateau in the spectrum gives way to the

third and final segment of the spectrum where the magnitude of the eigenvalues decays to zero. 1168

Our strategy for estimating tail probabilities for $Y^\top P\Omega^{(\ell)}PY$ is to explicitly evaluate the eigenvalues in 1169
the first segment of the spectrum, placing them into T_k , so that we can treat R_k as the sum of terms arising 1170
from the second segment and third segment. 1171

In regions with high local recombination rates, within hotspots, there are relatively few proximal mutations 1172
that are informative about the local genealogy. Thus, the spectrum of $P\Omega^{(\ell)}P$ decays rapidly in these regions. 1173
In these cases we can easily evaluate enough eigenvalues so that R_k explains little of the variance of $T_k + R_k$. At 1174
these loci, approximating R_k with \ddot{R}_k is more than sufficient to calculate p-values for $T_k + R_k = Y^\top P\Omega^{(\ell)}PY$. 1175
Almost any unimodal distribution matching the first and second moments would work for approximating R_k 1176
in this high-recombination context, which is achieved by our moment matching of \ddot{R}_k . 1177

However, at the vast majority of loci along the genome are in between recombination hotspots. There 1178
the spectrum of a typical $P\Omega^{(\ell)}P$ tends to decay over the leading ten or so eigenvalues and then level off 1179
to a very long plateau of eigenvalues, the vast majority of which are positive. This plateau arises from the 1180
abundance of rare variants encoding small clades in $\Omega^{(\ell)}$. While eventually this plateau decays to zero, the 1181
eigenvalues within the plateau segment of the spectrum make up the vast majority of the total sum of squared 1182
eigenvalues. As a consequence, the contribution of R_k to the overall variance of $T_k + R_k$ is much larger than 1183
the contribution of T_k for any tractable k . Thus, to estimate accurate tail probabilities for $T_k + R_k$, it is 1184
imperative that we do not underestimate the tail probabilities of R_k . 1185

To do this, we must ensure that we have control of the body of the distribution. When we top- k 1186
eigendecompose $P\Omega^{(\ell)}P$ in LOCATER, we increase the number of eigenvalues k we evaluate until we reach 1187
the following stopping criteria. 1188

$$k^* = \inf \left\{ k \in [n - q] : \left(\frac{\sum_{j=1}^k \lambda_j^2}{\sum_{j=1}^{n-q} \lambda_j^2} \right) \geq 0.95 \text{ or } (\lambda_k / \lambda_{k-1})^2 \geq 0.95 \right\}. \quad (33)$$

In other words, when we want precise p-values, we compute the top $k \geq k^*$ eigenvalues of PMP so that 1189

$$T_k = \sum_{j=1}^{k^*} \lambda_j Z_j^2 + \sum_{j=k^*+1}^k \lambda_j Z_j^2. \quad (34)$$

The first term in our stopping criteria catches the high-recombination case where R_k makes only a small 1190
contribution to the distribution of $T_k + R_k$. The second term in our stopping criteria catches the more typical 1191
case where the spectrum decays to a long plateau and ensures that we eigendecompose until that plateau is 1192
reached. Once the plateau is reached, all of leading terms with relatively large eigenvalues have been moved 1193
into T_k . This leaves R_k consisting of many terms with nearly equal eigenvalues. Since these terms have 1194
nearly equal variance, by the Berry–Esseen Theorem [47, 48], the body of R_k will be very close to a Gaussian. 1195

Since the Gaussian distribution is determined by its the mean and variance, this means that we can reliably estimate the body of the distribution with an approximate distribution that is also Gaussian in the body (with the appropriate mean and variance). Given the preponderance of positive eigenvalues in the plateau, minimizing $|\mu|$ when selecting (a, b, μ) (Section 6.5), ensures a large value of a for \ddot{R}_k . This guarantees that the body of the distribution of \ddot{R}_k will also be very close to Gaussian. Thus \ddot{R}_k reliably matches the body of the target null distribution. In combination with the robust rate of decay in the right tail \ddot{R}_k , this ensures reliable p-value estimation even for very small tail probabilities.

6.5 Accounting for Inflation Driven by Unmeasured Confounders & Polygenicity

Consider the presence of some unobserved variable $B \in \mathbb{R}^n$. Without loss of generality, assume $\|B\|_2 = \sqrt{n-q}$ and $PB = B$. For some scalar γ , we have $Y = A\alpha + \gamma B + \sigma\epsilon$. Then our estimator $\hat{\sigma}^2 = Y^\top PY / (n-q) \xrightarrow{P} \gamma^2 + \sigma^2$. This gives us the null distribution

$$\begin{aligned} (\gamma^2 + \sigma^2)^{-1} Y^\top PMPY &= (\gamma^2 + \sigma^2)^{-1} \sum_{j=1}^{n-q} \lambda_j (\sigma V_j^\top P\epsilon + \gamma V_j^\top B)^2 \\ &\sim \frac{\sigma^2}{\gamma^2 + \sigma^2} \sum_{j=1}^{n-q} \lambda_j \left(Z_j + \frac{\gamma}{\sigma} V_j^\top B \right)^2 \\ &= \nu \sum_{j=1}^{n-q} \lambda_j (Z_j + \delta_j)^2 \\ &\sim \nu \sum_{j=1}^{n-q} \lambda_j W_j \end{aligned}$$

where $\nu = \frac{\sigma^2}{\gamma^2 + \sigma^2}$, $\delta_j = \frac{\gamma}{\sigma} V_j^\top B$, and each $W_j \sim \chi_1^2(\delta_j^2)$, independent. The parameter $\nu \in [0, 1]$ captures the proportion of the variance of Y that is not explained by the unobserved variable B and each $(\delta_j^2)_{j=1}^{n-q}$ captures how correlated the clade structure encoded in $\Omega^{(\ell)}$ is with the unobserved variable B .

We can gather information about ν by looking at the distribution of this quadratic form test statistic at a grid of loci spaced far apart along the genome. Unfortunately, we cannot attempt to estimate each individual δ_j in a similar way since each specific δ_j depends on the precise j^{th} eigenvector V_j calculated at a given position. However, the spherical symmetry of the multivariate normal distribution helps us tackle this problem.

If all of the eigenvalues, λ_j were equal, our null distribution would have a scaled non-central chi-square distribution with $n - q$ degrees of freedom and a single non-centrality parameter $\sum_{j=1}^{n-q} \delta_j^2$. This collapse into a single non-centrality parameter means that we would not need to model each individual δ_j^2 , but rather just the average δ_j^2 .

To interpret this average, again, assuming all of the eigenvalues were equal,

1219

$$\sum_{j=1}^{n-q} \delta_j^2 = \frac{\gamma^2}{\sigma^2} \sum_{j=1}^{n-q} (V_j^\top B)^2 \propto B^\top PMPB = B^\top MB. \quad (35)$$

Thus $\sum_{j=1}^{n-q} \delta_j^2$ can be thought as proportional to the quadratic form test statistic we would obtain taking B as our phenotype vector, which is a measure of how correlated B is with the structure encoded in $\Omega^{(\ell)}$. This suggests a second, related interpretation. Recall B and all V_j s are scaled and that the inclusion of an intercept in A ensures that B and each V_j are centered. This means that $\sum_{j=1}^{n-q} \delta_j^2$ is proportional to average correlation between B and each V_j .

1220

1221

1222

1223

1224

$$\sum_{j=1}^{n-q} \delta_j^2 = \frac{\gamma^2}{\sigma^2} \sum_{j=1}^{n-q} (V_j^\top B)^2 = \frac{n\gamma^2}{\sigma^2} \frac{1}{n} \sum_{j=1}^{n-q} \text{Corr}(V_j, B)^2 = \frac{n\gamma^2}{\sigma^2} \overline{\text{Corr}(V_j, B)^2}. \quad (36)$$

This is proportional to the coefficient of multiple correlation (R^2) one would obtain by regressing B onto the eigenvectors V under the ordinary least squares model.

1225

1226

Since we expect that the typical correlation between B and each V_j may be different for V_j s capturing common variant structure versus rare variant structure, we replace each δ_j^2 such that

1227

1228

$$W_j \sim \begin{cases} \chi_1^2(\delta_\star^2) & j \leq k^\star \\ \chi_1^2(\delta_\dagger^2) & j > k^\star. \end{cases}$$

This leaves us with a null distribution with three scalar inflation parameters, $\nu, \delta_\star^2, \delta_\dagger^2 \in \mathbb{R}_{\geq 0}$. In summary, we have the following interpretations.

1229

1230

- ν is the proportion of variance in Y not explained by unobserved confounders
- γ_\star^2 is proportional to the average correlation between unobserved confounders and the large-scale haplotype structure encoded in $\Omega^{(\ell)}$ (that is not included in A)
- γ_\dagger^2 is proportional to the average correlation between unobserved confounders and the fine scale haplotype structure encoded in $\Omega^{(\ell)}$

1231

1232

1233

1234

1235

Now we turn to estimating these parameters. We can use $\text{tr}(PMP)$ and $\|PMP\|_{HS}^2$ to calculate the mean and variance of R_k . Using those moments, we approximate R_k with a shifted difference of chi-square random variables. Let

1236

1237

1238

$$\ddot{R}_k = \nu |\lambda_k| \left(X_{a, \delta_\dagger^2} - X'_{b, \delta_\star^2} + \mu (1 + \delta_\dagger^2) \right)$$

where $X_{a,\delta_{\dagger}^2} \sim \chi_a^2(a\delta_{\dagger}^2)$ is distributed chi-squared with a degrees of freedom and non-centrality parameter $a\delta_{\dagger}^2$ and independent of $X'_{b,\delta_{\dagger}^2} \sim \chi_b^2(b\delta_{\dagger}^2)$. 1239
1240

We will select parameters a, b, μ so that \ddot{R}_k matches R_k on the first two moments for all values of ν and δ_{\dagger} . We can do this using matrix traces to exactly calculate $\sum_{j=k+1}^{n-q} \lambda_j$ and $\sum_{j=k+1}^{n-q} \lambda_j^2$. If we then set the first moments equal we get

$$\mathbb{E}[R_k] = \nu(1 + \delta_{\dagger}^2) \sum_{j=k+1}^{n-q} \lambda_j = \nu |\lambda_k| (1 + \delta_{\dagger}^2) (a - b + \mu) = \mathbb{E}[\ddot{R}_k],$$

yielding the constraint

$$a - b + \mu = |\lambda_k|^{-1} \sum_{j=k+1}^{n-q} \lambda_j =: C_1.$$

Similarly, setting the variances equal we have

$$\text{Var}[R_k] = 2\nu^2(1 + 2\delta_{\dagger}^2) \sum_{j=k+1}^{n-q} \lambda_j^2 = 2\nu^2 \lambda_k^2 (1 + 2\delta_{\dagger}^2) (a + b) = \text{Var}[\ddot{R}_k],$$

which yields the constraint,

$$a + b = \lambda_k^{-2} \sum_{j=k+1}^{n-q} \lambda_j^2 =: C_2.$$

With these constraints, we select a, b, μ as follows 1241

1. If $C_1 \in [-C_2, C_2]$, we set $a = (C_2 + C_1)/2$ and then set $b = C_2 - a$ and $\mu = 0$. 1242
2. If $C_1 > C_2$, we set $a = C_2, b = 0$ and $\mu = C_1 - C_2$. 1243
3. If $C_1 < -C_2$, we set $b = C_2, a = 0$ and $\mu = C_1 + C_2$. 1244

This specification deliberately minimizes the contribution of μ to the mean relative to a and b while respecting our constraints. As discussed above in Section 6.4, this has the important consequence of ensuring large values of a at loci where the spectrum reaches a plateau (in between recombination hotspots). This large a in turn ensures a reliable p-value estimation at these loci. 1245
1246
1247
1248

Bringing everything together, our final null distribution is 1249

$$\nu \left(\sum_{j=1}^{k^*} \lambda_j W_j(\delta_{\star}^2) + \sum_{j=k^*+1}^k \lambda_j W_j(\delta_{\dagger}^2) + |\lambda_k| \left(X_{a,\delta_{\dagger}^2} - X'_{b,\delta_{\dagger}^2} + \mu(1 + \delta_{\dagger}^2) \right) \right). \quad (37)$$

Critically, this parameterization of the null distribution is expressed in two sets of orthogonal parameters. The first set are our spectral parameters $\{\lambda_j\}_{j=1}^k, a, b,$ and μ corresponding to the spectrum of PMP . The second are our mis-specification parameters $\nu, \delta_{\star}^2, \delta_{\dagger}^2$ corresponding to features of the confounding process. This means that, given the phenotype vector Y and the spectral parameters at a given locus, we can readily 1250
1251
1252
1253

recompute the observed p-value for that locus for any set of misspecification parameters. Thus, if we see 1254
inflation in the Q-Q plot of observed quadratic form p-values for a given phenotype, we can readily adjust 1255
the mis-specification parameters and recalculate our p-values in order to correct and calibrate the Q-Q plot. 1256

6.6 Fast Trace Calculation 1257

Here we present our trace calculation approach for a *symmetric* matrix $M \in \mathbb{R}^{n \times n}$. Let $B = I_n - QZ$ where 1258
 $Q \in \mathbb{R}^{n \times m}$ and $Z \in \mathbb{R}^{m \times n}$. Our goal is to calculate $\|BMB\|_{HS}^2$ and $\text{diag}(BMB)$ efficiently using a set of 1259
worker nodes. First we compute $J = ZM \in \mathbb{R}^{m \times n}$ (mn^2 FLOPs). Second we compute $X = Q(JQ) - MQ \in$ 1260
 $\mathbb{R}^{n \times m}$ ($mn^2 + 2nm^2 + nm$ FLOPs). Then map out the relevant sub-blocks of Q, Z, J, X to each worker as 1261
follows: 1262

$$\begin{aligned} (I - QZ)M(I - QZ) &= M - QZM - MQZ + QZMQZ \\ &= M + XZ - QJ \end{aligned}$$

So, 1263

$$\begin{aligned} \|(I - QZ)M(I - QZ)\|_{HS}^2 &= \sum_{i,j} (M_{ij} + X_i^T Z_{.j} - Q_i^T J_{.j})^2 \\ &= \sum_{i,j} \left(M_{ij} + \sum_{l=1}^m (X_{il}Z_{lj} - Q_{il}J_{lj}) \right)^2 \end{aligned}$$

This accumulation requires a total of $(4m + 1)n^2$ FLOPs. Since $\text{diag}(BMB)_i = M_{ii} + \sum_{l=1}^m (X_{il}Z_{li} - Q_{il}J_{li})$, 1264
 $\text{diag}(BMB)$ is easily obtained as a by-product of this calculation. Taking the sum of those diagonal 1265
elements to obtain $\text{tr}(BMB)$ requires $n - 1$ FLOPs. Adding the FLOPs together, we obtain a total of 1266
 $(6m + 1)n^2 + 2nm^2 + nm + n - 1$ FLOPs. Since $m \geq 1$, this is upper bounded by $7mn^2 + 3nm^2$. Connecting 1267
this result back to our notation in Section 6.5, this means we can calculate our desired traces η_1 and η_2 with 1268
fewer than $7(q + 1)n^2 + 3n(q + 1)^2$ FLOPs where q is the number of columns in our background covariate 1269
matrix A . 1270

If M is distributed in columns across nodes, we can accelerate the above calculation by exporting 1271
 $W = \begin{bmatrix} X & -Q \end{bmatrix} \in \mathbb{R}^{n \times 2m}$ to every node and the appropriate columns of $\begin{bmatrix} Z \\ J \end{bmatrix} \in \mathbb{R}^{2m \times n}$ to their 1272
corresponding nodes. 1273

6.7 Haplotype Simulation

1274

We simulated haplotypes from three 1000 Genomes Populations – Yoruba, Han Chinese, and Central European – using a demographic model adapted from the `msprime` [43] demography tutorial [49], which itself was based on the population parameters related those three populations presented in [50]. We only made one modification to the demographic model specified in [49]: we use the Discrete Time Wright–Fisher model for the first 100 generations into the past before reverting back to the classic Hudson model as proposed in [51]. The mutation rate was set to 1.2×10^{-8} .

1275

1276

1277

1278

1279

1280

Each 1 Mb region of simulated haplotypes was generated using a population-specific human recombination map estimated by `pyrho` [52]. Explicitly, in each simulation, we randomly selected one of our three 1000 Genomes populations (YRI, CHB, or CEU) and 1 Mb segment from genome (excluding heterochromatic regions), then we loaded the recombination map estimated for that population and region by `pyrho`.

1281

1282

1283

1284

Finally, all singleton variants (those with a derived allele count of 1) were removed from the simulated haplotypes before being passed to any association testing method.

1285

1286

6.8 Running ARG-Needle

1287

We ran ARG-Needle using the `arg_association` command line utility. This utility is automatically installed when installing `arg-needle-lib` with `pip` and documented in the ARG-Needle manual available at <https://palamaralab.github.io/software/argneedle/manual/>. As of March 20, 2025, we found that the python `logging` module (available at <https://pypi.org/project/logging/>) imported by `arg_association` has not been updated since 2013. We could not run `arg_association` because the `logging` module was not compatible with the python installation (version 3.12.3) required for our other simulation utilities. Upon inspecting the source code for the `arg_association` utility available on GitHub at https://github.com/PalamaraLab/arg-needle-scripts/blob/main/arg_needle_lib/arg_mlma/scripts/association.py, we found that the `logging` module was only used to print progress statements for the user. Commenting out all lines invoking the `logging` module allowed us to successfully run `arg_association`. We saved this modified script as `arg_needle_association_without_logging.py` and used it to perform association testing with the ARG-Needle library.

1288

1289

1290

1291

1292

1293

1294

1295

1296

1297

1298

1299

Since singletons were excluded from all of our simulations, we ran ARG-Needle using `arg_needle_association_without_logging.py` with optional argument `-min-mac 2` so that it would not lose power by attempting to test singletons. We also set the optional argument `-sampling_rate 1e-3`. Otherwise ARG-Needle was run with all default parameters. Filtering the results table returned by ARG-Needle down to rows with $MU_STAR \leq 10^{-5}$, we were able to obtain results as if we had invoked ARG-Needle with `-sampling_rate 1e-5`. Similarly, filtering down to rows with $MU_STAR \leq 10^{-7}$, we were able to obtain

1300

1301

1302

1303

1304

1305

results as if we had invoked ARG-Needle with `-sampling_rate 1e-7`.

1306

6.9 Running ACAT-O

1307

ACAT-O was run using the `STAAR` function from the STAAR R package available at <https://github.com/xihaoli/STAAR>. For each simulation, ACAT-O was run on all genotypes within the causal region. The ACAT-O was run testing rare variants with $MAF < 0.01$ (by setting `maf_cutoff=0.01`, the default) and all variants (by setting `maf_cutoff=1`). Otherwise, ACAT-O was run with all default settings.

1308

1309

1310

1311

6.10 Calculating Discovery Thresholds

1312

Taking the same approach we used in our power simulations, we ran null simulations to estimate a discovery threshold to control the genome-wide family-wise error rate (FWER) at 0.05 for LOCATER, SMT, AN3, AN5, and AN7. Under the haplotype simulation framework described in Section 4.1, we simulated 900 independent genomic datasets, each consisting of a ~~1Mb~~1 Mb segment observed for 30,000 individuals. For each of these segments, we sampled 10 independent null simulations of the phenotype vector Y (Section 4.2). We applied every testing method to each of the resulting 9,000 phenotype vectors and corresponding ~~1Mb~~1 Mb genomic segments. We stored the smallest p-value obtained the entire segment, yielding 9,000 p-values per method. Since, by default, LOCATER conserves compute by only testing variants with single-marker test p-values smaller than 10^{-4} , sometimes LOCATER did not return any p-value for a given simulation. This occurred in 3,363 out of our 9,000 simulations. For these simulations, we simply assigned a p-value of 1 to LOCATER.

1313

1314

1315

1316

1317

1318

1319

1320

1321

1322

Since the genome is 3,000 Mb long and each of the resulting 9,000 p-values was estimated by taking the smallest p-value over a 1 Mb segment, we estimated our 0.05 genome-wide FWER threshold by multiplying the 5th percentile of the resulting 9,000 p-values by 3,000.

1323

1324

1325

We used the same 9,000 null simulations to estimate a 0.05 genome-wide FWER threshold for ACAT-O (all) and ACAT-O (rare). However, we only ran ACAT-O on the central 10 kb causal window in each simulation rather than over the entire ~~1Mb~~1 Mb region. If we consider a genome-wide screen using a ~~10kb~~10 kb sliding window with 5kb overlap between adjacent windows, that's a total of 600,000 genome-wide tests. Thus we multiplied the 5th percentile of the resulting 9,000 p-values obtain for ACAT-O (all) and ACAT-O (rare) by 600,000 to obtain their corresponding discovery thresholds.

1326

1327

1328

1329

1330

1331

6.11 Phenotype Rank Matching

1332

Before we run LOCATER, as has become standard practice in testing quantitative traits, we require that the phenotype residuals be rank normalized after fitting background covariates. The theory underlying SD requires that the phenotypes be independent under H_0^C . Standard inverse-rank-normalization maps

1333

1334

1335

phenotype residuals to a fixed grid of values based on the Gaussian CDF, which in turn induces dependency 1336
 between phenotypes. In order to avoid inducing that dependency, we substitute each phenotype residual 1337
 with a rank-matched Gaussian random variable. This achieves a rank normalization that is very similar 1338
 to inverse-rank-normalization but with a small amount of noise added in order to make the normalized 1339
 phenotypes truly independent under the null hypothesis. 1340

Explicitly, given some raw phenotype vector $\tilde{Y} \in \mathbb{R}^n$ and a matrix of background covariates $A \in \mathbb{R}^{n \times q}$ 1341
 (including an intercept), we begin by calculating the residuals $\tilde{Y}^\perp = \tilde{Y} - A(A^\top A)^{-1}A^\top \tilde{Y}$ of the standard 1342
 ordinary least squares model 1343

$$\tilde{Y} \sim A\alpha + \sigma\epsilon. \tag{38}$$

Then we find the permutation $\pi : [n] \rightarrow [n]$ that orders the residuals \tilde{Y}^\perp such that $\tilde{Y}_{\pi(1)}^\perp \leq \tilde{Y}_{\pi(2)}^\perp \leq \dots \leq \tilde{Y}_{\pi(n)}^\perp$. 1344
 We simulate a new vector of Gaussian phenotypes, $Z \sim N(0, I_n)$ and likewise find the permutation $\pi' : [n] \rightarrow [n]$ 1345
 that orders the entries of Z such that $Z_{\pi'(1)} \leq Z_{\pi'(2)} \leq \dots \leq Z_{\pi'(n)}$. Then we obtain our normalized vector 1346
 of rank-matched phenotypes, $Y \in \mathbb{R}^n$, by assigning $Y_j = Z_{\pi'(\pi^{-1}(j))}$. 1347

July 10, 2025

RE: GENETICS-2025-308355

Dr. Ryan Christ
Yale University School of Medicine
Genetics
Yale University School of Medicine
333 Cedar St.
New Haven, Connecticut 06510

Dear Dr. Christ:

Congratulations, your manuscript titled "Clade Distillation for Genome-wide Association Studies" is accepted for publication in GENETICS! Many thanks for submitting your research to the journal, and thanks for the thoughtful attention to the remaining (tricky) points of discussion.

To Proceed to Publication:

1. Format your article according to GENETICS style: <https://academic.oup.com/genetics/pages/general-instructions>
2. Ensure that you comply with data and community resource citation guidelines: <https://academic.oup.com/genetics/pages/general-instructions#Data-Policy>
3. Upload your final files at <https://genetics.msubmit.net>
4. Add oupsupport@scipris.com and genetics.oup@novatechset.com (or the domains @scipris.com and @novatechset.com) to your email program's "safe senders" list. You will be contacted by both at various points during the production process.

Notes:

- Your currently-accepted manuscript (unedited, as submitted, reviewed, and accepted) will be published at GENETICS and deposited into PubMed as an Advance Access article. Notify sourcefiles@thegsajournals.org before signing your license if you do not wish to publish your article via Advance Access.
- We invite you to submit an original color figure related to your paper for consideration as cover art. Please email your submission to the editorial office or upload it with your final files. You can submit a small-sized image for evaluation, and if selected, the final image must be a TIFF file 2513px wide by 3263px high (8.375 by 10.875 inches; resolution of 600ppi). Please avoid graphs and small type.
- After files are sent to Oxford University Press we use SciPris to manage article licensing and payment. If you do not have a SciPris account, you will receive an email from no-reply@scipris.com to sign up to use Oxford University Press' author portal. After logging in, follow the online instructions to sign your license and arrange any payment due.

If you have any questions or encounter any problems while uploading your accepted manuscript files, please email the editorial office at sourcefiles@thegsajournals.org.

Sincerely,

Peter Ralph
Associate Editor
GENETICS

Approved by:
Konrad Lohse
Senior Editor
GENETICS

Review comments (if applicable):